# Optimistic Value Iteration with Representation Learning for Low-Rank POMGs

## Abstract

Partially Observable Markov Games (POMGs) pose significant challenges for multi-agent reinforcement learning due to the combination of partial observability and strategic interactions. Recent advances explore the inherent structure of the POMG dynamics and develop efficient representation methods to facilitate planning in the latent space rather than directly operating on the history trajectory. In this paper, we focus on the low-rank POMGs and propose a unified optimistic value iteration (OVI) framework that accommodates different low-rank representation learning methods. With a given representation, OVI constructs an optimistic bonus and integrates it into the value function to inspire exploration and mitigate the bias caused by the representation approximation error. When the exact value function oracle is unavailable, OVI instead utilizes the low-rank representation to construct optimistic/pessimistic estimators of the value functions via the Bellman recursion, and selects the final solution based on the optimistic-pessimistic gap. Our theoretical analysis shows that, once the representation approximation error is bounded, the OVI converges to an approximate equilibrium. We instantiate the framework with two provable representation learning methods: an MLE-based approach and a spectral decomposition representation method. Furthermore, we develop a novel representation method, $L$-step Latent Variable Representation (LLVR), for POMGs with infinite-dimensional latent spaces, i.e., infinite rank, and prove that OVI with LLVR also achieves approximate equilibria, with an extra $L$-decodability assumption. Collectively, these results establish the first systematic representation learning perspective for POMGs.

## 1 Introduction

Markov games (MGs) have emerged as a foundational framework for multi-agent reinforcement learning (MARL), enabling rigorous analysis of agents' performance in strategic interactions (1; 2; 3; 4). Recently, Partially Observable Markov Game (POMG) has been proposed as an extension of MG under partial observability (5; 6). Specifically, each agent in POMGs only observes its own actions and local signals, leading to incomplete information about the true state. The non-Markovian nature of the observations forces the agent to maintain memory and reason about beliefs of the system state, all while exploring to collect information about the environment. Consequently, even in cooperative settings, POMGs have been shown to be NEXP-complete (7), implying that solving them in the worst case requires super-exponential complexity.

Recent advances have sought to explore specific structured subclasses of POMGs that admit tractable solutions. For instance, (8; 9) investigated POMGs with $\gamma$-observability where observations probabilistically reveal state information, enabling hierarchical state estimation via information-sharing mechanisms. (10) studied weakly revealing POMGs where observations are sufficiently informative to infer state properties. More recently, other structured subclasses, such as POMGs with low generalized eluder coefficient (11) have also been investigated. However, these methods either focus on tabular state spaces or are restricted to two-player zero-sum games, which limits their applicability to general POMGs. To the best of our knowledge, the only prior work addressing general POMGs is (12). However, their approach employs a computationally inefficient representation learning method and relies on access to an exact value function oracle.

Table 1: Sample complexity of different representation learning methods under our OVI framework. For the same representation method, value function oracle-free setting(OVI-OF) require more samples than oracle-based setting(OVI-OB). The additional complexity is highlighted in blue.

| Representation Method | OVI-OB | OVI-OF |
|---|---|---|
| MLE | $\tilde{\mathcal{O}}(\varepsilon^{-2}\log(H|\mathcal{M}|/\delta\varepsilon))$ | $\tilde{\mathcal{O}}(H^6d^4|\mathcal{A}|^2 \cdot \varepsilon^{-2}\log(d|\mathcal{A}|H|\mathcal{M}|/\delta\varepsilon))$ |
| SDR | $\tilde{\mathcal{O}}(d^2\varepsilon^{-2}\log(H|\mathcal{M}|/\delta\varepsilon))$ | $\tilde{\mathcal{O}}(H^6d^2|\mathcal{A}|^2 \cdot d^2\varepsilon^{-2}\log(d|\mathcal{A}|H|\mathcal{M}|/\delta\varepsilon))$ |
| LLVR | $\tilde{\mathcal{O}}(\varepsilon^{-2}\log(H|\mathcal{M}|/\delta\varepsilon))$ | $\tilde{\mathcal{O}}\left(H^6|\mathcal{A}|^LC\log(H|\mathcal{A}|^{L/2}|\mathcal{M}|/\delta\varepsilon)\right.$ $\left.\cdot\varepsilon^{-2}\log(|\mathcal{A}|^{L/2}H|\mathcal{M}|/\delta\varepsilon)\right)$ |

In this work, we propose a unified optimistic value iteration (OVI) framework that accommodates different low-rank representation learning method. With a given representation, it constructs an optimistic bonus to encourage exploration, and then performs value iteration based on the representation and bonus. The framework is compatible with two distinct settings; one that assumes access to an exact value function oracle and another that operates without it. In the oracle-free setting, OVI bypasses the need of the value function oracle by using the representation and bonus to construct optimistic and pessimistic value function estimators via Bellman recursion. We show that if the representation error is bounded, the framework provides sample-efficient guarantees for learning approximate equilibria. We instantiate OVI with two concrete representation learning algorithms, one based on Maximum Likelihood Estimation (MLE) and another on Spectral Decomposition Representation (SDR), and demonstrate that both converge to approximate equilibria. Furthermore, we develop a novel $L$-step latent variable representation (LLVR) method for POMGs with infinite-dimensional latent space, i.e., infinite rank. Specifically, LLVR utilizes a computationally tractable ELBO to learn an effective representation with only the recent $L$-step trajectory. We make the following contributions:

- We propose a unified OVI framework for low-rank POMGs that accommodates various low-rank representation learning methods. Given a representation, OVI augments rewards with an optimistic bonus to both encourage exploration and compensate for approximation error. Notably, this framework is compatible with two distinct settings, supporting scenarios that assumes access to an exact value function oracle and those that operates without it. In particular, in the oracle-free setting, OVI constructs optimistic/pessimistic estimators of the value function based on the representation and bonus, then performs the value iteration with the constructed estimators, and selects the final solution by minimizing the gap between these estimators. For both settings of OVI, we show that they converge to an approximate Nash, Correlated, or Coarse Correlated Equilibrium if the representation error is bounded.

- We instantiate the framework with two concrete representation learning algorithms: MLE and SDR. We characterize the corresponding approximation error of both representations and derive the sample complexities of OVI-MLE and OVI-SDR to reach approximate equilibria. While MLE achieves tighter approximation guarantees, SDR offers a more computationally efficient alternative by reparameterizing an $l_2$ norm objective.

- We develop a novel representation method, LLVR, for POMGs with infinite-dimensional latent spaces, i.e., infinite rank. LLVR learns the latent representation of the transition kernel by optimizing a computationally friendly ELBO, and yields an exact and tractable linear form of the value function over the latent space. Our theoretical results establish that OVI-LLVR retains provable convergence to approximate equilibria under an extra $L$-decodability assumption. Note that LLVR only needs to access the recent $L$-step history instead of the full history information. Our empirical study in Appendix I also shows the efficiency of LLVR.

## 1.1 RELATED WORK

**Theoretical guaranteed methods in POMGs.** Structural information has been extensively leveraged to develop theoretically guaranteed methods for POMGs. A rich body of work has investigated structured subclasses of single-agent POMGs, i.e. POMDPs, such as $L$-decodable POMDPs (13; 14; 15), weakly revealing POMDPs (16; 17), observable POMDPs (18) and low-rank POMDPs

(19; 20). In the multi-agent setting, solving POMGs is significant challenging: it is NEXP-complete even under cooperative objectives (7). As a result, while there has been substantial progress on fully observable MGs (21; 22; 23; 24; 25; 26; 27; 28; 29; 30), research on POMGs remains relatively scarce. Motivated by the POMDP literature, recent works have explored structural subclasses of POMGs that admit tractable solutions. For instance, (8; 9) investigated information-sharing mechanisms for POMGs with proposed $\gamma$-observability, where agents' observations contain probabilistic information about the underlying state, enabling hierarchical state estimation and efficient coordination. (10) proposed a sample-efficient approach for weakly revealing POMGs by assuming informative observations, while (31) studied two-player competitive and tree-structured transition POMGs that permit game-theoretic planning via backward induction. More recently, (11) developed posterior sampling methods for two-player zero-sum games with low generalized eluder coefficients, extending applicability to continuous state spaces. However, these methods either focus on tabular state spaces (8; 9; 10; 31) or are restricted to two-player zero-sum games (31; 11), which limits their applicability to general POMGs. To the best of our knowledge, the only prior work addressing general POMGs is (12), which analyzed POMGs under information rank structure assumptions, characterizing how partial observability interacts with agent interactions. However, their approach employs a computationally inefficient representation learning method and relies on access to an exact value function oracle.

**Representation learning in RL.** A growing body of research has focused on representation learning in RL, i.e. learning latent representations to capture the underlying dynamics. For instance, (13; 32; 33) investigated representation learning in block MDPs, which is a special case of low-rank MDPs. (34) studied representation for MDPs with the structure of Gaussian noise. Several recent papers studied low-rank MDPs via MLE and facilitating sample-efficient RL (35; 36; 37). Model-free representation learning methods in Low-Rank MDPs have also been studied (38; 39). Meanwhile, several methods extracted computationally efficient spectral representations from the low-rank MDPs (40; 41). Recently, (42; 17) considered POMDPs and constructed the MLE confidence set for low-rank structured models. (16) explored POMDPs within an spectral estimation set. (15) studied latent variable spectral representation for $L$-decodable POMDPs. In the multi-agent setting, (21) represented the environment linearly for two-player zero-sum MGs with the structure of Gaussian noise. (43) and (30) explored representation learning in low-rank fully observable MGs via contrastive self-supervised learning and MLE, respectively. For the more difficult POMG tasks, (12) constructed a generalized PSR representation under $\gamma$-well-conditioned assumption. Note that there is still a lack of research on comprehensive representation learning for general low-rank POMGs.

## 2 BACKGROUND

In a POMG, each player does not have complete information about the current state of the game. Instead, players only have access to partial observations of the state. This partial observability introduces additional challenges to the design and analysis of policies, as players must make decisions based on these noisy or incomplete observations.

A POMG is defined by a tuple: $(\mathcal{S}, \{\mathcal{A}_i\}_{i=1}^M, \mathcal{P}, \{r_i\}_{i=1}^M, H, \mu_0, \{\mathcal{O}_i\}_{i=1}^M, \mathbb{O})$, where $H$ denotes the length of each episode, $\mathcal{S}$ is the state space with $|\mathcal{S}| = S$, $\mathcal{A}_i$ denotes the action space for the $i^{\text{th}}$ player with $|\mathcal{A}_i| = A_i$, $\mathcal{P} = \{\mathcal{P}_h\}_{h=0}^H$ is the collection of transition probabilities, $\mu_0$ is the initial state distribution, $\mathcal{O}_i = \{\mathcal{O}_{h,i}\}_{h=0}^H$ denotes the observation space for the $i^{\text{th}}$ player with $|\mathcal{O}_i| = O_i$, $r_i = \{r_{h,i} : \mathcal{O}_i \times \mathcal{A}_i \to [0,1]\}_{h=0}^H$ is the reward function for player $i$. We denote by $\boldsymbol{o} := (o_1, \ldots, o_M) \in \mathcal{O} := \mathcal{O}_1 \times \cdots \times \mathcal{O}_M$ the joint observations of all $m$ players and $\boldsymbol{a} := (a_1, \ldots, a_M) \in \mathcal{A} := \mathcal{A}_1 \times \cdots \times \mathcal{A}_M$ the joint observations of all $M$ players, respectively. $\mathbb{O}(\cdot|s) : \mathcal{S} \to \Delta(\mathcal{O})$ is the emission kernel so that $\mathbb{O}_h(\boldsymbol{o}|s)$ is the probability of having a partial observation $\boldsymbol{o} \in \mathcal{O}$ at state $s$.

At each time step $h$, each player $i$ receives an observation $o_{h,i}$ and a reward $r_{h,i}$ based on the true state $s_h \in \mathcal{S}$. A key feature of POMGs is that the observation does not fully reveal the true state. Observing $o_i$ instead of the true state $s$ leads to a non-Markovian transition between observations, which means each player needs to consider policies $\pi_i := \{\pi_{h,i} : ((\mathcal{O}_i \times \mathcal{A}_i)^{h-1} \times \mathcal{O}_i) \to \Delta_{\mathcal{A}_i}\}_{h \in [H]}$ that depend on the entire history, denoted by $\tau_h = \{\boldsymbol{o}_0, \boldsymbol{a}_0 \ldots, \boldsymbol{o}_h\}$. We denote the joint policy of all players as $\pi := \pi_1 \times \ldots \times \pi_M$, the action of each player is sampled independently according to their own policy. We denote the space of $\tau_h$ as $\mathcal{T}_h$ and the policy of all the players except player $i$ as $\pi_{-i}$.

For a given joint policy $\pi$, we can define the state value function $V_{h,i}^{\pi}(s_h) = \mathbb{E}\left[\sum_{t=h}^{H} r_{t,i}(o_t, \boldsymbol{a}_t)|s_h\right]$ and state-action value function $Q_{h,i}^{\pi}(s_h, \boldsymbol{a}_h) = \mathbb{E}\left[\sum_{t=h}^{H} r_{t,i}(o_t, \boldsymbol{a}_t)|s_h, \boldsymbol{a}_h\right]$ for each player $i$ at step $h$, respectively, for the POMG. Therefore, the Bellman equation can be expressed as $V_{h,i}^{\pi}(s_h) = \mathbb{E}_{\pi}\left[Q_{h,i}^{\pi}(s_h, \boldsymbol{a}_h)\right]$, $Q_{h,i}^{\pi}(s_h, \boldsymbol{a}_h) = r_{h,i}(o_h, \boldsymbol{a}_h) + \mathbb{E}_{\mathbb{P}}\left[V_{h+1,i}^{\pi}(s_{h+1})\right]$. For the convenience of notation, we denote $v_i^{\pi} := \mathbb{E}_{s\sim\mu_0}\left[V_{0,i}^{\pi}(s)\right]$.

For any policy $\pi_{-i}$, there exists a best response policy of player $i$, which is a policy $\mu^{\dagger}(\pi_{-i})$ satisfying $V_{h,i}^{\mu^{\dagger}(\pi_{-i}),\pi_{-i}}(s) = \max_{\pi_i} V_{h,i}^{\pi_i,\pi_{-i}}(s)$ for any $(s,h) \in \mathcal{S} \times [H]$. We denote $V_{h,i}^{\dagger,\pi_{-i}} := V_{h,i}^{\mu^{\dagger}(\pi_{-i}),\pi_{-i}}$ and let $v_i^{\dagger,\pi_{-i}} := \mathbb{E}_{s\sim\mu_0}\left[V_{0,i}^{\dagger,\pi_{-i}}(s)\right]$.

We focus on three classic equilibrium concepts in game theory—Nash Equilibrium, Correlated Equilibrium (CE) and Coarse Correlated Equilibrium (CCE) (30). First, a NE is defined as a product policy in which no player can increase her value by changing only her own policy. Formally,

**Definition 1** (NE). *A joint policy $\pi$ is a Nash equilibrium (NE) if $v_i^{\pi} = v_i^{\dagger,\pi_{-i}}, \forall i \in [M]$. And we call $\pi$ an $\varepsilon$-approximate NE if $\max_{i\in[M]}\{v_i^{\dagger,\pi_{-i}} - v_i^{\pi}\} < \varepsilon$.*

Second, a CCE is a relaxed version of Nash equilibrium in which we consider general correlated policies instead of joint policies.

**Definition 2** (CCE). *A correlated policy $\pi$ is a CCE if $V_{h,i}^{\dagger,\pi_{-i}}(s) \leqslant V_{h,i}^{\pi}(s)$ for all $s \in \mathcal{S}, h \in [H], i \in [M]$. And we call $\pi$ an $\varepsilon$-approximate CCE if $\max_{i\in[M]}\{v_i^{\dagger,\pi_{-i}} - v_i^{\pi}\} < \varepsilon$.*

Finally, a CE is defined as a joint policy where no player can increase her value by unilaterally applying any strategy modification. To define CE, we first introduce the concept of policy modification: A policy modification $\omega_i := \{\omega_{h,i}\}_{h\in[H]}$ for player $i$ is a set of $H$ functions from $\mathcal{S} \times \mathcal{A}_i$ to $\mathcal{A}_i$. Let $\Omega_i := \{\Omega_{h,i}\}_{h\in[H]}$ denote the set of all possible policy modifications for player $i$. One can compose a policy modification $\omega_i$ with any Markov policy $\pi$ and obtain a new policy $\omega_i \circ \pi$ such that when policy $\pi$ chooses to play $\boldsymbol{a} := (a_1, \ldots, a_M)$ at state $s$ and step $h$, policy $\omega_i \circ \pi$ will play $(a_1, \ldots, a_{i-1}, \omega_{h,i}(s, a_i), a_{i+1}, \ldots, a_M)$ instead.

**Definition 3** (CE). *A correlated policy $\pi$ is a CE if $\max_{\omega_i\in\Omega_i} V_{h,i}^{\omega_i\circ\pi}(s) \leqslant V_{h,i}^{\pi}(s)$ for all $(s,h) \in \mathcal{S} \times [H], i \in [M]$. And we call $\pi$ an $\varepsilon$-approximate CE if $\max_{i\in[M]}\{\max_{\omega_i\in\Omega_i} v_i^{\omega_i\circ\pi} - v_i^{\pi}\} < \varepsilon$.*

Solving general POMGs is notoriously hard due to their inherent complexity, which is known to be NEXP-complete. To overcome this, prior works has focused on structured subclasses that allow for more tractable solutions. Examples include weakly revealing POMGs (10), $\gamma$-observable POMGs (8; 9). In this paper, we focus on POMGs with low-rank dynamics, a structure widely used in the RL literature (36; 37; 40). A POMG is called low-rank if its transition function factorizes through a pair of low-dimensional embeddings. This structural assumption applies only to valid history triplets $(\tau, \boldsymbol{a}, \tau')$, so the rank $d$ does not scale with $(|\mathcal{O}||\mathcal{A}|)^h$ and can remain small in practice.

**Definition 4** (low-rank POMG). *A POMG is low-rank if, there exist embeddings: $\phi_h : \mathcal{T}_h \times \mathcal{A} \to \mathbb{R}^d$ and $\mu_h : \mathcal{T}_{h+1} \to \mathbb{R}^d$ for all $h \in [H]$ such that*

$$\forall \tau \in \mathcal{T}_h, \tau' \in \mathcal{T}_{h+1}, \boldsymbol{a} \in \mathcal{A} : \mathcal{P}_h(\tau'|\tau, \boldsymbol{a}) = \langle \phi_h(\tau, \boldsymbol{a}), \mu_h(\tau')\rangle \text{ if } (\tau, \boldsymbol{a}) \text{ forms the history of } \tau'.$$

*For normalization, we assume that $\|\phi(\tau_h, \boldsymbol{a})\| \leqslant 1$ for all $\tau_h, \boldsymbol{a}$ and for any function $g : \mathcal{T}_h \to [0,1]$, $\|\int \mu(\tau_h)g(\tau_h)d\tau_h\| \leqslant \sqrt{d}$.*

## 3 REPRESENTATION LEARNING BASED OPTIMISTIC VALUE ITERATION

We propose a unified optimistic value iteration (OVI) framework that accommodates various low-rank representation learning method. The central idea is to use a given latent representation of the dynamics to construct an optimistic bonus, and then perform value iteration based on the representation and bonus. Notably, this framework is compatible with two distinct settings, supporting scenarios both with and without an exact value function oracle. We prove that when the representation error is upper bounded by $\zeta$, our framework provides sample-efficient guarantees for learning approximate equilibria, with detailed proof provided in Appendix B.

### 3.1 Representation Learning for POMGs

If the transition and reward functions of a POMG are known, planning-based approaches like heuristic search value iteration (44) and linear programming (45) can be used to compute the solution.

However, in practice the transition function is typically unknown. In such cases, it is common to employ representation learning to construct a latent representation of the transition dynamics. Formally, this is captured by considering a model class $\mathcal{M} = \{(\hat{\phi}_h, \hat{\mu}_h) : \hat{\phi}_h \in \Phi_h, \hat{\mu}_h \in \Phi_h\}_{h=0}^H$, with the assumptions that $|\mathcal{M}| < \infty$ and the true transition is included in the class, i.e., $\phi_h \in \Phi_h, \mu_h \in \Psi_h, \forall h \in [H]$. The representation quality can be measured by

$$\mathbb{E}_{(\tau_h, \boldsymbol{a}_h) \sim \mathcal{D}_{h,n}} \left\| \mathbb{P}_h^{\mathcal{P}}(\cdot|\tau_h, \boldsymbol{a}_h) - \mathbb{P}_h^{\widehat{\mathcal{P}}_n}(\cdot|\tau_h, \boldsymbol{a}_h) \right\|^2 \leqslant \zeta_n,$$

where $\mathcal{D}_{h,n}$ denotes the empirical dataset. A smaller representation error $\zeta_n$ indicates higher-quality representations, which in turn yield stronger performance guarantees for subsequent value-iteration based planning procedure.

### 3.2 OVI based on Representation

With a learned representation, oracle-based and oracle-free value-iteration based planning procedure can be conducted. Suppose we have access to an oracle that computes the exact value function $V_{P,r}^{\pi,i}$ for a policy $\pi$ and player $i$ with transition $P$ and reward $r$ in a POMG for all $i \in [M]$. In this setting, we can apply OVI by augmenting rewards with a confidence bonus $\hat{b}_n$ derived from the representation. Specifically, The oracle-based OVI, denoted as OVI-OB, maintains an estimate $\hat{\mathcal{P}}_n$ of the transition and defines an augmented reward $r_n^+ = r + \hat{b}_n$. The policy is then updated via $\pi = \arg\max_{\pi'} V_{\hat{\mathcal{P}}_n, r_n^+}^{\pi'}$.

In most realistic POMGs, value-function oracles are often unavailable (46; 18). To handle this, we propose oracle-free OVI, denoted as OVI-OF, that maintains both optimistic and pessimistic estimates of the value function, updated recursively with confidence intervals derived from the bonus $\hat{b}_{n,h}$. Based on the bonus term, we construct both optimistic and pessimistic estimators $(\overline{V}_{h,i}^n, \underline{V}_{h,i}^n, \overline{Q}_{h,i}^n, \underline{Q}_{h,i}^n)$ according to the Bellman recursion (Line 10 of Algorithm 1). Depending on the problem's solution requirement, the policy $\pi_h^n$ can be updated as follows,

$$NE: \quad \pi_{h,i}^n(\cdot|\tau_h) = \arg\max_{\pi_{h,i}} \left( \mathbb{D}_{\pi_{h,i}, \pi_{h,-i}^n} \overline{Q}_{h,i}^n \right)(\tau_h), \quad \forall \tau_h \in \mathcal{T}_h, i \in [M]. \tag{1}$$

$$CCE: \quad \max_{\pi_{h,i}} \left( \mathbb{D}_{\pi_{h,i}, \pi_{h,-i}^n} \overline{Q}_{h,i}^n \right)(\tau_h) \leqslant \left( \mathbb{D}_{\pi^n} \overline{Q}_{h,i}^n \right)(\tau_h), \quad \forall \tau_h \in \mathcal{T}_h, i \in [M]. \tag{2}$$

$$CE: \quad \max_{\omega_{h,i} \in \Omega_{h,i}} \left( \mathbb{D}_{\omega_{h,i} \circ \pi_h^n} \overline{Q}_{h,i}^n \right)(\tau_h) \leqslant \left( \mathbb{D}_{\pi^n} \overline{Q}_{h,i}^n \right)(\tau_h), \quad \forall \tau_h \in \mathcal{T}_h, i \in [M]. \tag{3}$$

Here, $(\mathbb{D}_\pi f)(\tau) := \mathbb{E}_{\boldsymbol{a} \sim \pi(\cdot|\tau)} [f(\tau, \boldsymbol{a})], \forall f : \mathcal{T} \times \mathcal{A} \to \mathbb{R}$. Without loss of generality, we assume that the solution to each formulation is unique; if not, a deterministic selection rule can be applied so that the same input yields the same policy. Note that although the policy update relies only on the optimistic estimator, we still maintain the pessimistic estimator to compute the gap $\Delta^n = \max_{i \in [M]} \{\bar{v}_i^n - \underline{v}_i^n\}$, with $\bar{v}_i^n = \int_\mathcal{T} \overline{V}_{0,i}^n(\tau)\mu_0(\tau)\,d\tau$ and $\underline{v}_i^n = \int_\mathcal{T} \underline{V}_{0,i}^n(\tau)\mu_0(\tau)\,d\tau$. Finally, the algorithm selects the policy $\hat{\pi}$ that achieves the smallest estimated gap.

Algorithm 1 summarizes OVI-OB and OVI-OF. Under bounded representation error, the output policy is guaranteed to be an approximate NE/CE/CCE.

**Lemma 5.** *Assume that the representation error of RepLearn in Alg.1 is bounded as*

$$\mathbb{E}_{(\tau_h, \boldsymbol{a}_h) \sim \mathcal{D}_{h,n}} \left\| \mathbb{P}_h^{\mathcal{P}}(\cdot|\tau_h, \boldsymbol{a}_h) - \mathbb{P}_h^{\widehat{\mathcal{P}}_n}(\cdot|\tau_h, \boldsymbol{a}_h) \right\|^2 \leqslant \zeta_n,$$

*and set $\lambda = \Theta(d\log(nH|\mathcal{M}|/\delta))$ and suitable bonus. With probability $1 - \delta$, the output policy $\hat{\pi}$ of OVI-OB and OVI-OF is an $\varepsilon$-approximate $\{\text{NE}, \text{CCE}, \text{CE}\}$ with $\varepsilon = \tilde{O}(\|V^\pi\|\sqrt{\zeta_n})$ and $\varepsilon = \tilde{O}\left(H^2 d\sqrt{N|\mathcal{A}|\alpha_N^2 \log(1 + \frac{N}{d\lambda})}\right)$, respectively, where $\|V^\pi\|$ is the upper bound of the norm of value function for any policy $\pi$, $i \in [N], h \in [H]$.*

---

**Algorithm 1 O**ptimistic **V**alue **I**teration framework with **O**racle **B**ased/**O**racle **F**ree value functions (OVI-OB/OF)

---

1: **Input:** Regularizer $\lambda$, iteration $N$, parameter $\{\alpha_n\}_{n=1}^N$, value function oracle $V$.
2: Initialize $\pi^0$ to be uniform; set dataset $\mathcal{D}_h^0 = \emptyset, \hat{\mathcal{D}}_h^0 = \emptyset, \forall h \in [H]$.
3: **for** episode $n = 1, 2, \cdots, N$ **do**
4:     **if** oracle-free: Set $\overline{V}_{H+1,i}^n = 0, \underline{V}_{H+1,i}^n = 0$ for all $i \in [M]$.
5:     Sample data with policy $\pi_n$: $\tau_H = (\boldsymbol{o}_0, \boldsymbol{a}_0, \ldots, \boldsymbol{o}_{H-1}, \boldsymbol{a}_{H-1}, \boldsymbol{o}_H)$.
6:     Update dataset: $\mathcal{D}_h^n = \mathcal{D}_h^{n-1} \cup \{\tau_h, \boldsymbol{a}_h, \boldsymbol{o}_{h+1}\}, \hat{\mathcal{D}}_h^n = \hat{\mathcal{D}}_h^{n-1} \cup \{\tau_{h+1}, \boldsymbol{a}_{h+1}, \boldsymbol{o}_{h+2}\}$ for all $h \in [H]$.

7:     **for** step $h = H, H-1 \ldots, 1$ **do**
8:         Learning representation and compute bonus: $(\hat{\phi}_n, \hat{\mathcal{P}}_n, \hat{b}_{n,h}) = RepLearn(\mathcal{D}_h^n \bigcup \hat{\mathcal{D}}_h^n)$.
9:         **if** oracle free **then**
10:         Update $\overline{Q}, \underline{Q}$ as following:

$$\overline{Q}_{h,i}^n(\tau_h, \boldsymbol{a}) = r_{h,i}(\tau_h, \boldsymbol{a}) + \mathbb{E}_{\hat{\mathcal{P}}_h^n}\left[\overline{V}_{h+1,i}^n(\tau_{h+1}) | \tau_h, \boldsymbol{a}\right] + \hat{b}_{n,h}(\tau_h, \boldsymbol{a})$$

$$\underline{Q}_{h,i}^n(\tau_h, \boldsymbol{a}) = r_{h,i}(\tau_h, \boldsymbol{a}) + \mathbb{E}_{\hat{\mathcal{P}}_h^n}\left[\underline{V}_{h+1,i}^n(\tau_{h+1}) | \tau_h, \boldsymbol{a}\right] - \hat{b}_{n,h}(\tau_h, \boldsymbol{a})$$

11:         Compute the NE/CE/CCE solution $\pi_h^n$ according to equation equation 1/equation 2/equation 3 and update value function as following:

$$\overline{V}_{h,i}^n(\tau_h) = \mathbb{E}_{\boldsymbol{a} \sim \pi_h^n(\cdot|\tau_h)}[\overline{Q}_{h,i}^n(\tau_h, \boldsymbol{a})], \quad \underline{V}_{h,i}^n(\tau_h) = \mathbb{E}_{\boldsymbol{a} \sim \pi_h^n(\cdot|\tau_h)}[\underline{Q}_{h,i}^n(\tau_h, \boldsymbol{a})].$$

12:         **end if**
13:     **end for**
14:     **if** oracle based: Compute $\pi^n = \arg\max_\pi V_{\hat{\mathcal{P}}_n, r+\hat{b}_n}^\pi(\tau)$.
15:     **if** oracle free: Compute $\Delta^n = \max_{i \in [M]}\{\overline{v}_i^n - \underline{v}_i^n\}$ with $\overline{v}_i^n = \int_{\mathcal{T}} \overline{V}_{0,i}^n(\tau)\mu_0(\tau)d\tau, \underline{v}_i^n = \int_{\mathcal{T}} \underline{V}_{0,i}^n(\tau)\mu_0(\tau)d\tau$.
16: **end for**
17: **if** oracle based: **Return** $\hat{\pi} = $ NE/CE/CCE solution with $V_{\hat{\mathcal{P}}_N, r}^{\hat{\pi}, i}$ for all $i \in [M]$.
18: **if** oracle free: **Return** $\hat{\pi} = \pi^{n^\star}$ where $n^\star = \arg\min_{n \in [N]} \Delta^n$.

---

**Algorithm 2** MLE Representation Learning and Bonus Computation

---

1: **Input:** Dataset $\mathcal{D}_h^n \bigcup \hat{\mathcal{D}}_h^n$.
2: Compute $(\hat{\phi}_{n,h}, \hat{\mu}_{n,h}) = \arg\max_{(\phi,\mu) \in \mathcal{M}} \mathbb{E}_{\mathcal{D}_h^n \bigcup \hat{\mathcal{D}}_h^n}[\log \mu(\tau_h)^\top \phi(\tau_{h-1}, \boldsymbol{a}_{h-1})]$ for all $h \in [H]$, and obtain $\hat{\mathcal{P}}_n = \{\hat{\mu}_{n,h}(\tau_h)^\top \hat{\phi}_{n,h}(\tau_{h-1}, \boldsymbol{a}_{h-1})\}_{h \in [H]}$.
3: Compute $\hat{b}_{n,h}$ from equation 4 for all $h \in [H]$.
4: **Return** $\hat{\phi}_n = \{\hat{\phi}_{n,h}\}_{h \in [H]}, \hat{\mathcal{P}}_n$ and $\hat{b}_n = \{\hat{b}_{n,h}\}_{h \in [H]}$.

---

# 4 INSTANTIATIONS OF REPRESENTATION LEARNING FOR LOW-RANK POMGS

In this section, we instantiate the general framework from Section 3 with concrete representation learning algorithms, i.e. Maximum Likelihood Estimation (MLE), Spectral Decomposition Representation (SDR). We then analyze their representation errors and resulting sample complexities when combined with OVI. The full proof is provided in Appendix B.

## 4.1 MLE-BASED REPRESENTATION LEARNING

As shown in Alg. 2, with the latent low-rank structure, MLE estimates the transition kernel by maximizing the log-likelihood of observed trajectories. The bonus term is computed as

$$\hat{b}_{n,h} = \min\left\{\alpha_n \|\hat{\phi}(\tau_{h-1}, \boldsymbol{a}_{h-1})\|_{L_2(\mu), \hat{\Sigma}_{n,h,\hat{\phi}_n}^{-1}}, H\right\}, \tag{4}$$

---

**Algorithm 3** SDR Representation Learning and Bonus Computation

---

1: **Input:** Dataset $\mathcal{D}_h^n \bigcup \hat{\mathcal{D}}_h^n$.
2: Learning $(\hat{\phi}_{n,h}(\tau_{h-1}, \boldsymbol{a}_{h-1}), \mu'_{n,h}(\tau_h))$ with $\mathcal{D}_h^n \bigcup \hat{\mathcal{D}}_h^n$ via Equation 5, and obtain $\hat{\mathcal{P}}_n =$
$\{(p'(\tau_h)\mu'_{n,h}(\tau_h))^\top \hat{\phi}_{n,h}(\tau_{h-1}, \boldsymbol{a}_{h-1})\}_{h \in [H]}$.
3: Compute $\hat{b}_{n,h}$ from equation 4 for all $h \in [H]$.
4: **Return** $\hat{\phi}_n = \{\hat{\phi}_{n,h}\}_{h \in [H]}$, $\hat{\mathcal{P}}_n$ and $\hat{b}_n = \{\hat{b}_{n,h}\}_{h \in [H]}$.

---

where $\hat{\Sigma}_{n,h,\phi} := \sum_{(\tau_{h-1}, \boldsymbol{a}_{h-1}) \in \mathcal{D}_h^n} \phi_h(\tau_{h-1}, \boldsymbol{a}_{h-1}) \phi_h(\tau_{h-1}, \boldsymbol{a}_{h-1})^\top + \lambda I_d$.

The representation error of Algorithm 2 can be characterized by Lemma 51. We refer to OVI-OB and OVI-OF with Alg. 2 as OVI-OB-MLE and OVI-OF-MLE, respectively. Based on the representation error, we can derive a PAC guarantee of OVI-OB-MLE and OVI-OF-MLE, which exploit the low-dimensional latent representation learned by MLE.

**Theorem 6** (PAC guarantee of OVI-OB-MLE and OVI-OF-MLE). *Assume OVI-OB-MLE and OVI-OF-MLE are applied with parameters $\zeta_n = \Theta\left(\log(Hn|\mathcal{M}|/\delta)/n\right)$, $\lambda = \Theta(d \log(NH|\mathcal{M}|/\delta))$, and $\alpha_n = \Theta(\sqrt{\lambda d + n|\mathcal{A}|\zeta_n})$. By setting the number of episodes $N$ to be $N = \tilde{O}(\varepsilon^{-2} \log(H|\mathcal{M}|/\delta\varepsilon))$ and $N = \tilde{\mathcal{O}}(H^6 d^4 |\mathcal{A}|^2 \epsilon^{-2} \log(Hd|\mathcal{A}||\mathcal{M}|/\delta\varepsilon))$, respectively, the output policy $\hat{\pi}$ is an $\varepsilon$-approximate $\{\mathrm{NE}, \mathrm{CCE}, \mathrm{CE}\}$ with probability $1 - \delta$.*

Thus, we obtain sample complexities that are independent of $|\mathcal{S}|$, while exhibit polynomial dependency $|\mathcal{A}|, H, d, \varepsilon$ and $\log|\mathcal{M}|$. Notably, OVI-OF-MLE incurs an additional $H^6 d^4 |\mathcal{A}|^2$ complexity to circumvent the need for a value oracle.

## 4.2 SDR Representation Learning

While the MLE oracle offers strong theoretical guarantees, computing it is computationally difficult (40). Inspired by spectral representation in MDPs (40), we adopt the Spectral Decomposition Representation (SDR) approach for low-rank POMGs. As outlined in Algorithm 3, SDR reparameterizes an $l_2$ norm objective, leading to the computationally friendly objective in Equation 5.

$$\min_{\phi, \mu'} -\mathbb{E}_{(\tau, \boldsymbol{a}, \tau') \sim d_0 \times \mathcal{P}} \left[ \phi(\tau, \boldsymbol{a})^\top \mu'(\tau')p(\tau') \right] + (\mathbb{E}_{p(\tau')} \left[ p(\tau')\mu'(\tau')^\top \mu'(\tau') \right])/(2d) \quad (5)$$

$$s.t. \qquad \mathbb{E}_{(\tau, a) \sim d_0} \left[ \phi(\tau, a)\phi(\tau, a)^\top \right] = I_d/d,$$

where we use reparameterization $\mu(\tau') = p(\tau')\mu'(\tau')$. Generally, solving SDR is easier than solving MLE since SDR bypasses the difficult integral calculation in MLE with an easy-to-compute expectation (40).

Similarly, we refer to OVI-OB and OVI-OF with Alg. 3 as OVI-OB-SDR and OVI-OF-SDR, respectively. We have the representation error of Algorithm 3 in Lemma 52 and we can derive a PAC guarantee of OVI-OB-SDR and OVI-OF-SDR, which exploit the low-dimensional latent representation learned by SDR.

**Theorem 7** (PAC guarantee of OVI-OB-SDR and OVI-OF-SDR). *Assume Assumption 1 in Appendix B holds. Consider running OVI-OB-SDR and OVI-OF-SDR with parameters $\zeta_n = \Theta\left(\frac{\log(Hn|\mathcal{M}|/\delta)}{n}\right)$, $\lambda = \Theta\left(d \log\left(NH|\mathcal{M}|/\delta\right)\right)$, and $\alpha_n = \Theta\left(Hd\sqrt{\lambda d + n|\mathcal{A}|\zeta_n}\right)$. If the number of episodes is set to $N = \tilde{O}\left(\varepsilon^{-2} d^2 \log\left(H|\mathcal{M}|/(\delta\varepsilon)\right)\right)$ and $N = \tilde{O}\left(H^6 d^4 |\mathcal{A}|^2 \varepsilon^{-2} \log\left(Hd|\mathcal{A}||\mathcal{M}|/(\delta\varepsilon)\right)\right)$, respectively, then the output policy $\hat{\pi}$ is an $\varepsilon$-approximate $\{\mathrm{NE}, \mathrm{CCE}, \mathrm{CE}\}$ with probability at least $1 - \delta$.*

Compared to MLE, SDR generally yields larger representation error but remains polynomially bounded, and provides a more computationally tractable approach. When combined with OVI, SDR achieves efficient sample complexity guarantees while being implementable at scale.

**Remark 8** (Comparison with (12)). *(12) construct a generalized PSR representation for $\gamma$-well-conditioned POMGs with an exact value function oracle. Note that if the rank of the core test set is uniform across all time steps $h$, i.e. $d_h = d$ for all $h$, the POMG satisfies the additional assumptions*

*in (12) is a special subclass of low-rank POMGs and this representation can be integrated into our framework. We extend their method to the oracle-free setting in Appendix G.*

## 5 LLVR INFINITE-DIMENSIONAL LATENT SPACE

In this section, we develop a novel representation method, $L$-step Latent Variable Representation (LLVR), for POMGs with infinite-dimensional latent spaces, i.e., infinite rank, with an extra $L$-decodability assumption. When incorporated into the OVI framework, it yields provable convergence to approximate equilibria. Notably, LLVR requires access only to the most recent $2L$ steps of history rather than the entire trajectory.

### 5.1 $L$-DECODABILITY

We now introduce the $L$-decodability assumption, which relies on the belief function. $f_{belief}(\cdot) : \mathcal{O} \times (\mathcal{A} \times \mathcal{O})^h \to \Delta(\mathcal{S})$. This function represents the distribution over the underlying state given the history of observations and actions. It is initialized as $f_{belief}(s_0|\boldsymbol{o}_0) = \mathbb{P}(s_0|\boldsymbol{o}_0)$ and updated recursively as: $f_{belief}(s_{h+1}|\tau_{h+1}) \propto \int_{\mathcal{S}} f_{belief}(s_h|\tau_h)\mathcal{P}(s_{h+1}|s_h, \boldsymbol{a}_h)\mathbb{O}(\boldsymbol{o}_{h+1}|s_{h+1})\, ds_h$. See Appendix C for a detailed explanation of the belief function and the $L$-decodability assumption.

**Definition 9** ($L$-decodability (11)). $\forall h \in [H]$, define $\tau_h^L \in \mathcal{T}^L := (\mathcal{O} \times \mathcal{A})^{L-1} \times \mathcal{O}$, $\tau_h^L = (\boldsymbol{o}_{h-L+1}, \boldsymbol{a}_{h-L+1}, \cdots, \boldsymbol{o}_h)$. A POMG is $L$-decodable if there exists a decoder $p^\star : \mathcal{T}^L \to \Delta(\mathcal{S})$ such that $p^\star(\tau_h^L) = f_{belief}(\tau_h)$.

Note that under the $L$-decodability assumption, there exists an $L$-step joint policy that constitutes a Nash equilibrium. Therefore, it suffices to restrict our analysis to $L$-step policy in the discussion under the $L$-decodability assumption.

### 5.2 LLVR REPRESENTATION LEARNING

We now propose LLVR under the $L$-decodability assumption. LLVR leverages the underlying $L$-decodability structure to enable an exact and tractable linear representation of the value functions over the latent space. Due to space limitations, we have deferred the detailed derivation to Appendix C. The ultimate objective of the LLVR is to provide a computationally tractable ELBO objective:

$$\max_{q \in \Delta(\mathcal{Z})} \mathbb{E}_{q(\cdot|\tau_h^L, \boldsymbol{a}_h, \boldsymbol{o}_{h+1:h+l})}[\log \mathbb{P}^{\chi_\pi}(\boldsymbol{o}_{h+1:h+l}|z_h)] - KL(q(\cdot|\tau_h^L, \boldsymbol{a}_h, \boldsymbol{o}_{h+1:h+l})||p(z_h|\tau_h^L, \boldsymbol{a}_h)). \quad (6)$$

where $\chi_\pi$ is the moment-matching policy for $\pi$ (defined in Appendix F), $\mathbb{P}^\pi(\cdot)$ denotes the probability distribution under policy $\pi$, $z_h \in \mathcal{Z}$ is the latent variable, and $l$ is a fixed constant with $l < L$. Note that LLVR only requires sampling the past $L$ steps and the future $l$ steps, where $L + l < 2L$, rather than the entire trajectory.

The solution of ELBO can be parameterized with a variational distribution class $\mathcal{Q} = \{\{q_h(z|\tau_h^L, \boldsymbol{a}_h, \boldsymbol{o}_{h+1:h+l})\}_{h \in [H]}\}$ and model class $\mathcal{M} = \{\{(p_h(z|\tau_h^L, \boldsymbol{a}_h), p_h(\boldsymbol{o}_{h+1:h+l}|z))\}_{h \in [H]}\}$. Practically, both $\mathcal{Q}$ and $\mathcal{M}$ can be implemented as neural networks, yielding approximate solutions $\hat{q}(z|\tau_h^L, \boldsymbol{a}_h, \boldsymbol{o}_{h+1:h+l})$, $\hat{p}_{h,n}(\boldsymbol{o}_{h+1:h+l}|z_h)$ and $\hat{p}_{n,h}(z_h|\tau_h^L, \boldsymbol{a}_h)$ and approximated transition $\hat{\mathcal{P}}_n = \{(\hat{p}_{h,n}(z_h|\tau_h^L, \boldsymbol{a}_h), \hat{p}_{h,n}(\boldsymbol{o}_{h+1}|z_h))\}_{h \in [H]}$.

Once $\hat{p}_{n,h}(z|\tau_h^L, \boldsymbol{a}_h)$ is obtained, the Q-function can be approximated as $Q_h^\pi(\tau_h^L, \boldsymbol{a}_h) = \langle \hat{p}(z|\tau_h^L, \boldsymbol{a}), \omega(z) \rangle$ and can be obtained by a least square regression (18). However, if $z$ is continuous, then $\omega(z)$ is infinite-dimensional. To deal with the infinite-dimensional $\omega(z)$, we follow the trick in (41) that forms $Q^\pi(\tau_h^L, \boldsymbol{a}_h)$ as an expectation $Q^\pi(\tau_h^L, \boldsymbol{a}_h) = \langle p(z|\tau_h^L, \boldsymbol{a}_h), w^\pi(z) \rangle = \mathbb{E}_{p(z|\tau_h^L, \boldsymbol{a}_h)}[w^\pi(z)]$ and then approximate it with random feature quadrature. Specifically, we consider $\omega(z)$ lying in certain RKHS with $\varphi$ as its random feature basis, i.e., $\omega(z) = \mathbb{E}_{P(\xi)}[\varphi(\xi, z)]$. As a result, $Q^\pi(\tau_h^L, \boldsymbol{a}_h) \approx \frac{1}{K}\sum_{i=1}^{K} \omega^\pi(\xi_i)\varphi(z_i, \xi_i)$ where the latent variables $z_i \sim \hat{p}(z|\tau_h^L, \boldsymbol{a}_h)$ and random features $\xi_i \sim P(\xi)$. If the random feature $\varphi$ is specified, then $\omega$ can be implemented by a neural network $\omega_\theta$. Due to space limitation, we defer the detailed derivation to Appendix E.1.

### 5.3 OVI BASED ON LLVR

Based on the learned representation, we construct the following ellipsoid bonus term $\hat{b}_h^n$ to get both optimistic and pessimistic estimation of the value function,

$$\hat{b}_{n,h}(\tau^L, \boldsymbol{a}) = \min\{\alpha_n \left\|\hat{\psi}_{n,h}(\cdot|\tau_{h-L}^L, \boldsymbol{a}_{h-L})\right\|_{\sum_{n,h}^{-1}}, H\}, \quad (7)$$

---

**Algorithm 4** LLVR Representation Learning and Bonus Computation

---

1: **Input:** Dataset $\mathcal{D}_h^n \bigcup \hat{\mathcal{D}}_h^n$.

2: Learn $\hat{p}_n(z|\tau_h^L, \boldsymbol{a}_h)$ with $\mathcal{D}_h^n \bigcup \hat{\mathcal{D}}_h^n$ via maximizing the ELBO objective, i.e. equation 6, and obtain $\hat{\mathcal{P}}_n = \{(\hat{p}_{n,h}(z|\tau_h^L, \boldsymbol{a}_h), \hat{p}_{n,h}(\boldsymbol{o}_{h+1}|z))\}_{h \in [H]}$.

3: Compute $\hat{b}_{n,h}$ from equation 7 for all $h \in [H]$.

4: **Return** $\hat{p}_n = \{\hat{p}_{n,h}\}_{h \in [H]}$, $\hat{\mathcal{P}}_n$ and $\hat{b}_n = \{\hat{b}_{n,h}\}_{h \in [H]}$.

---

where $\hat{\psi}_{n,h}(\tau_h^L, \boldsymbol{a}_h) = [\varphi(z_1; \xi_1), \cdots, \varphi(z_K; \xi_K)]$ denotes the random feature sampled from the RKHS and the covariance matrix is defined as $\sum_{n,h}^{-1} = \sum_{(\tau_i^L, \boldsymbol{a}_i) \in \mathcal{D}_h^n} \hat{\psi}_{n,h}(\tau_h^L, \boldsymbol{a}_h)\hat{\psi}_{n,h}(\tau_h^L, \boldsymbol{a}_h)^\top + \lambda I$.

Optimistic and pessimistic estimators are constructed using the bonus as in Alg. 1. These estimators can also be approximated by neural networks and computed using least-squares regression. Crucially, as shown in Appendix C, since all terms in the least-squares formulation are derived from the feature space spanned by $\hat{p}(z|\tau_h^L, \boldsymbol{a}_h)$, parameterizing them enables highly efficient computation.

The theoretical guarantees for OVI-OB-LLVR and OVI-OF-LLVR are proven in Appendix E.

**Theorem 10** (PAC guarantee of OVI-OB-LLVR and OVI-OF-LLVR). *Assume Assumption 4,5 in Appendix E.4 hold and the kernel $K$ satisfies the regularity conditions in Appendix E.2. Consider running OVI-OB-LLVR and OVI-OF-LLVR with proper parameters $\zeta_n$, $\hat{b}_{n,h}$, $\alpha_n$ and $\lambda$. By setting the number of episodes $N$ to be $N = \tilde{O}(\varepsilon^{-2} \log(H|\mathcal{M}|/\delta\varepsilon))$ and $N = \text{poly}(C, H, |\mathcal{A}|^L, \epsilon, \log \frac{H|\mathcal{M}|}{\delta})$, respectively, the output policy $\hat{\pi}$ is an $\varepsilon$-approximate $\{NE, CCE, CE\}$ with probability $1 - \delta$.*

Notably, the complexity is also independent of $|\mathcal{Z}|$, implying that $z$ can be a continuous variable.

**Remark 11** (Comparison of MLE, SDR, and LLVR). *The three instantiations of our framework exhibit complementary strengths. MLE achieves the tightest theoretical guarantees, but solving the MLE problem is computationally demanding in practice. SDR yields looser bounds due to a larger representation error, yet it offers a more computationally tractable approach. Finally, LLVR extends our framework to POMGs with infinite-dimensional latent spaces. Under the additional $L$-decodability assumption, LLVR preserves approximate equilibrium guarantees while only requiring access to short $2L$-step histories, thereby broadening the applicability of our framework.*

### 5.3.1 OFFLINE POLICY OPTIMIZATION.

We also propose an offline OVI-OF-LLVR algorithm for sample-efficient policy optimization using only a static dataset of size $n$, which is assumed to be drawn from the stationary distribution $\rho$ of a fixed behavior policy $\pi_b$. Consequently, unlike the online setting where new data can be collected to explore unseen state–action pairs, the offline scenario precludes further exploration beyond what is in the static dataset. Despite this limitation, our offline algorithm retains the core structure of its online counterpart, differing only in the absence of new data from the environment. A detailed description of the offline algorithm and PAC analysis for it is provided in Appendix E.5.

## 6 CONCLUSION

In this paper, we present a unified optimistic value iteration (OVI) framework for POMGs. OVI integrates an optimism bonus derived from suitable representations into the value function and provably converges to approximate equilibria under bounded representation error, in both oracle-based and oracle-free settings. We instantiate OVI with two concrete representation learners: an MLE-based method offering the tightest guarantees but higher computational cost, and an SDR-based method yielding looser bounds but better tractability. We further proposed a novel LLVR representation that extends OVI to infinite-dimensional latent spaces under an additional $L$-decodability assumption and show that OVI with LLVR also achieves approximate equilibria while relying only on short histories. Overall, our results establish the first systematic representation learning view for low-rank POMGs.

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

## A   ADDITIONAL NOTATIONS

This section collects additional notations and technical definitions used throughout our analysis.

Given a (possibly not normalized) transition probability $P$ and a policy $\pi$, we define the density function of $(x, \boldsymbol{a})$ at step $h$ under $P$ and $\pi$ by

$$d^\pi_{P,0}(x, \boldsymbol{a}) := \mu_0(x)\pi_0(\boldsymbol{a}|x), \quad d^\pi_{P,h+1}(x, \boldsymbol{a}) := \sum_{\tilde{x} \in \mathcal{X}, \tilde{\boldsymbol{a}} \in \mathcal{A}} d^\pi_{P,h}(\tilde{x}, \tilde{\boldsymbol{a}}) P_h(x|\tilde{x}, \tilde{\boldsymbol{a}})\pi_{h+1}(\boldsymbol{a}|x), \quad \forall h \geqslant 0.$$

We abuse the notations a bit and denote $d^\pi_{P,h}(x)$ as the marginalized state distribution, i.e., $d^\pi_{P,h}(x) = \sum_{\boldsymbol{a} \in \mathcal{A}} d^\pi_{P,h}(x, \boldsymbol{a})$.

We then define

$$\rho_{n,h}(x, \boldsymbol{a}) = \frac{1}{n} \sum_{i \in [n]} d^{\pi_i}_{P,h}(x, \boldsymbol{a}),$$

$$\hat{\rho}_{n,h}(x, \boldsymbol{a}) = \frac{1}{n} \sum d^{\pi_i}_{P,h}(x) u_\mathcal{A}(\boldsymbol{a}),$$

$$\tilde{\rho}_{n,h}(x, \boldsymbol{a}) = \frac{1}{n} \sum \mathbb{E}_{\tilde{x} \sim d^{\pi_i}_{P,h-1}, \tilde{\boldsymbol{a}} \sim U(\mathcal{A})} \left[ P(x|\tilde{x}, \tilde{\boldsymbol{a}}) u_\mathcal{A}(\boldsymbol{a}) \right],$$

and $\circ^L \mathcal{U}(\mathcal{A})$ means uniformly taking actions in the consecutive $L$ steps.

When we use the expectation $\mathbb{E}_{(x,\boldsymbol{a})\sim\rho}[f(x, \boldsymbol{a})]$ (or $\mathbb{E}_{x\sim\rho}[f(x)]$) for some (possibly not normalized) distribution $\rho$ and function $f$, we simply mean $\sum_{x \in \mathcal{X}, \boldsymbol{a} \in \mathcal{A}} \rho(x, \boldsymbol{a}) f(x, \boldsymbol{a})$ (or $\sum_{x \in \mathcal{X}} \rho(x) f(x)$) so that the expectation can be naturally extended to the unnormalized distributions. For an iteration $n$, a distribution $\rho$ and a feature $\phi$, we denote the expected feature covariance as

$$\Sigma_{n,\rho,\phi} = n\mathbb{E}_{(x,\boldsymbol{a})\sim\rho} \left[ \phi(x, \boldsymbol{a})\phi(x, \boldsymbol{a})^\top \right] + \lambda I_d.$$

Meanwhile, define the empirical covariance by

$$\hat{\Sigma}_{n,h,\phi} := \sum_{(x,\boldsymbol{a}) \in \mathcal{D}^n_h} \phi(x, \boldsymbol{a})\phi(x, \boldsymbol{a})^\top + \lambda I_d.$$

Finally, we define the following operators in the space of $L_2(\mu) \to L_2(\mu)$:

$$\Sigma_{\rho_n \times \mathcal{U}(\mathcal{A}), \phi} = n\mathbb{E}_{x\sim\rho_n, \boldsymbol{a}\sim\mathcal{U}(\mathcal{A})} \left[ \phi(x, \boldsymbol{a})\phi^\top(x, \boldsymbol{a}) \right] + \lambda T^{-1}_n$$

$$\Sigma_{\rho_n, \phi} = n\mathbb{E}_{(x,\boldsymbol{a})\sim\rho_n} \left[ \phi(x, \boldsymbol{a})\phi^\top(x, \boldsymbol{a}) \right] + \lambda T^{-1}_n$$

## B   THEORETICAL ANALYSIS FOR METHODS FOR LOW-RANK POMGS

This section presents the theoretical guarantees for our algorithms for low-rank POMGs.

### B.1   PROOF OF SEC. 3

We will provide the proof of Section 3 in this subsection.

**Lemma 12.** *If the representation error in Alg. 1 is bounded*

$$\mathbb{E}_{(\tau_h, \boldsymbol{a}_h)\sim\mathcal{D}_{h,n}} \left\| \mathbb{P}^\mathcal{P}_h(\cdot|\tau_h, \boldsymbol{a}_h) - \mathbb{P}^{\widehat{\mathcal{P}}_n}_h(\cdot|\tau_h, \boldsymbol{a}_h) \right\|^2 \leqslant \zeta_n,$$

*with probability $1 - \delta$, the output policy $\hat{\pi}$ of OBOVI is an $\varepsilon$-approximate $\{\mathrm{NE}, \mathrm{CCE}, \mathrm{CE}\}$ with $\varepsilon = \tilde{O}(\|V^\pi\|\sqrt{\zeta_n})$, where $\|V^\pi\|$ is the upper bound of the norm of value function for any policy $\pi$, $i \in [N], h \in [H]$.*

*Proof.* Denote by $V_P^i(\pi)$ the value function of player $i$ under policy $\pi$ and transition $P$. Since the returned policy $\hat{\pi}$ is an equilibrium with respect to $\hat{\mathcal{P}}$, we have for all $i \in [N]$: $V_{\hat{\mathcal{P}}}^i(\hat{\pi}) = \max_{\tilde{\pi}^i} V_{\hat{\mathcal{P}}}^i(\tilde{\pi}^i, \hat{\pi}^{-i}) := V_{\hat{\mathcal{P}}}^{i,\dagger}(\hat{\pi}^i)$.

Note that

$$|V_{\hat{\mathcal{P}}}^{i,\dagger}(\hat{\pi}^i) - V_{\mathcal{P}}^{i,\dagger}(\hat{\pi}^i)| = |\max_{\tilde{\pi}^i} V_{\hat{\mathcal{P}}}^i(\tilde{\pi}^i, \pi^{-i}) - \max_{\tilde{\pi}^i} V_{\mathcal{P}}^i(\tilde{\pi}^i, \pi^{-i})|$$

$$\leqslant \max_{\tilde{\pi}^i} |V_{\hat{\mathcal{P}}}^i(\tilde{\pi}^i, \pi^{-i}) - V_{\mathcal{P}}^i(\tilde{\pi}^i, \pi^{-i})|$$

$$\leqslant \|V^\pi\| \sqrt{\zeta_n}$$

Thus, we have

$$V_{\mathcal{P}}^i(\hat{\pi}) \geqslant V_{\hat{\mathcal{P}}}^i(\hat{\pi}) - \|V^\pi\| \sqrt{\zeta_n}$$

$$= V_{\hat{\mathcal{P}}}^{i,\dagger}(\hat{\pi}^{-i}) - \|V^\pi\| \sqrt{\zeta_n}$$

$$\geqslant V_{\mathcal{P}}^{i,\dagger}(\hat{\pi}^{-i}) - 2\|V^\pi\| \sqrt{\zeta_n}$$

Hence, $\hat{\pi}$ is an $2\|V^\pi\|\sqrt{\zeta_n}$-approximate equilibrium. $\qquad\square$

**Lemma 13.** *If the representation error in Alg. 1 is bounded as*

$$\mathbb{E}_{(\tau_h, \boldsymbol{a}_h) \sim \mathcal{D}_{h,n}} \left\| \mathbb{P}_h^{\mathcal{P}}(\cdot|\tau_h, \boldsymbol{a}_h) - \mathbb{P}_h^{\hat{\mathcal{P}}_n}(\cdot|\tau_h, \boldsymbol{a}_h) \right\|^2 \leqslant \zeta_n,$$

*with $\lambda = \Theta(d\log(nH|\mathcal{M}|/\delta))$ and properly chosen bonus, with probability $1 - \delta$, the output policy $\hat{\pi}$ of OVI-OF is an $\varepsilon$-approximate $\{NE, CCE, CE\}$ with $\varepsilon = \tilde{O}\left(H^2 d\sqrt{N|\mathcal{A}|\alpha_N^2 \log(1 + \frac{N}{d\lambda})}\right)$, where $\alpha_N = \Theta(\sqrt{\lambda d + N|\mathcal{A}|\zeta_N})$ and $\|V^\pi\|$ is the upper bound of the norm of value function for any policy $\pi$, $i \in [N], h \in [H]$.*

The proof of 13 is included in the proof of Theorem 21.

### B.2 PROOF OF SEC. 4.1

We will provide the proof of Theorem 6 in this subsection.

**Theorem 14** (PAC guarantee of OBOVI-MLE)**.** *When OBOVI-MLE is applied with parameters $\zeta^n = \Theta\left(\log(Hn|\mathcal{M}|/\delta)/n\right)$, $\lambda = \Theta(d\log(NH|\mathcal{M}|/\delta))$, and $\alpha_n = \Theta(\sqrt{\lambda d + n|\mathcal{A}|\zeta_n})$ by setting the number of episodes $N$ to be at most $N = \tilde{O}(\varepsilon^{-2}\log(H|\mathcal{M}|/\delta\varepsilon))$ with probability $1 - \delta$, the output policy $\hat{\pi}$ is an $\varepsilon$-approximate $\{NE, CCE, CE\}$.*

*Proof.* Recall that the estimated transition satisfies

$$\mathbb{E}_{(\tau_h, \boldsymbol{a}_h) \sim \mathcal{D}_{h,n}} \left\| \mathbb{P}_h^{\mathcal{P}}(\cdot|\tau_h, \boldsymbol{a}_h) - \mathbb{P}_h^{\hat{\mathcal{P}}_n}(\cdot|\tau_h, \boldsymbol{a}_h) \right\|_1^2 \leqslant \zeta_n.$$

Denote by $V_P^i(\pi)$ the value function of player $i$ under policy $\pi$ and transition $P$. Since the returned policy $\hat{\pi}$ is an equilibrium with respect to $\hat{\mathcal{P}}$, we have for all $i \in [N]$: $V_{\hat{\mathcal{P}}}^i(\hat{\pi}) = \max_{\tilde{\pi}^i} V_{\hat{\mathcal{P}}}^i(\tilde{\pi}^i, \hat{\pi}^{-i}) := V_{\hat{\mathcal{P}}}^{i,\dagger}(\hat{\pi}^i)$.

Note that

$$|V_{\hat{\mathcal{P}}}^{i,\dagger}(\hat{\pi}^i) - V_{\mathcal{P}}^{i,\dagger}(\hat{\pi}^i)| = |\max_{\tilde{\pi}^i} V_{\hat{\mathcal{P}}}^i(\tilde{\pi}^i, \pi^{-i}) - \max_{\tilde{\pi}^i} V_{\mathcal{P}}^i(\tilde{\pi}^i, \pi^{-i})|$$

$$\leqslant \max_{\tilde{\pi}^i} |V_{\hat{\mathcal{P}}}^i(\tilde{\pi}^i, \pi^{-i}) - V_{\mathcal{P}}^i(\tilde{\pi}^i, \pi^{-i})|$$

$$\leqslant \sqrt{\zeta_n}$$

Thus, we have

$$V_{\mathcal{P}}^i(\hat{\pi}) \geqslant V_{\hat{\mathcal{P}}}^i(\hat{\pi}) - \sqrt{\zeta_n}$$

$$= V_{\hat{\mathcal{P}}}^{i,\dagger}(\hat{\pi}^{-i}) - \sqrt{\zeta_n}$$

$$\geqslant V_{\mathcal{P}}^{i,\dagger}(\hat{\pi}^{-i}) - 2\sqrt{\zeta_n}$$

Hence, $\hat{\pi}$ is an $2\sqrt{\zeta_n}$-approximate equilibrium.

To guarantee an $\varepsilon$-approximate equilibrium, we require $2\sqrt{\zeta_n} \leqslant \varepsilon$, which leads to $N = \tilde{O}(\varepsilon^{-2} \log(H|\mathcal{M}|/\delta\varepsilon))$. $\qquad\square$

Then, we prove the PAC guarantee for OFOVI-MLE, establishing key technical lemmas that culminate in the finite-sample convergence theorem.

**Lemma 15** (one-step back inequality for the true model). *Given a set of functions $[g_h]_{h\in[H]}$, where $g_h : \mathcal{T}_h \times \mathcal{A} \to \mathbb{R}$, $\|g_h\|_\infty \leqslant B$, $\forall h \in [H]$, we have that $\forall\pi$,*

$$
\sum_{h\in[H]} \mathbb{E}_{(\tau_h,\boldsymbol{a}_h)\sim\rho_{n,h}}[g_h(\tau_h,\boldsymbol{a}_h)] \leqslant \sum_{h\in[H]} \mathbb{E}_{(\tau_{h-1},\boldsymbol{a}_{h-1})\sim\rho_{n,h-1}} \left[ \|\phi_{h-1}(\tau_{h-1},\boldsymbol{a}_{h-1})\|_{L_2(\mu),\Sigma_{\rho_{n,h-1},\mathcal{P}}^{-1}} \right]
$$
$$
\cdot \sqrt{n|\mathcal{A}| \cdot \mathbb{E}_{(\tilde{\tau}_h,\tilde{\boldsymbol{a}}_h)\sim\rho_{n,h-1}\circ\mathcal{U}(\mathcal{A})}\left[g_h(\tilde{\tau}_h,\tilde{\boldsymbol{a}}_h)^2\right] + \lambda B^2 d}
$$

*Proof.* The proof can be adapted from the proof of Lemma B.4 in (30), and we include it for the completeness. We observe the following one-step-back decomposition:

$$
\mathbb{E}_{(\tau_h,\boldsymbol{a}_h)\sim\rho_{n,h}}[g_h(\tau_h,\boldsymbol{a}_h)]
$$
$$
=\mathbb{E}_{(\tau_{h-1},\boldsymbol{a}_{h-1})\sim\rho_{n,h-1}} \left[ \int_{o_h} \langle\phi_{h-1}(\tau_{h-1},\boldsymbol{a}_{h-1}), \mu_{h-1}(\tau_h)\rangle_{L_2(\mu)} \cdot \mathbb{E}_{\boldsymbol{a}_h\sim\pi_h(\cdot|\tau_h)}[g_h(\tau_h,\boldsymbol{a}_h)]do_h \right]
$$
$$
\leqslant \mathbb{E}_{(\tau_{h-1},\boldsymbol{a}_{h-1})\sim\rho_{n,h-1}} \|\phi_{h-1}(\tau_{h-1},\boldsymbol{a}_{h-1})\|_{L_2(\mu),\Sigma_{\rho_{n,h-1},\phi}^{-1}}
$$
$$
\cdot \mathbb{E}_{(\tau_{h-1},\boldsymbol{a}_{h-1})\sim\rho_{n,h-1}} \left\| \int_{o_h} \mu_{h-1}(\tau_h)\mathbb{E}_{\boldsymbol{a}_h\sim\pi_h(\cdot|\tau_h)}[g_h(\tau_h,\boldsymbol{a}_h)]do_h \right\|_{L_2(\mu),\Sigma_{\rho_{n,h-1},\phi}} .
$$

Direct computation shows that for all

$$
\mathbb{E}_{(\tau_{h-1},\boldsymbol{a}_{h-1})\sim\rho_{n,h-1}} \left\| \int_{o_h} \mu_{h-1}(\tau_h)\mathbb{E}_{\boldsymbol{a}_h\sim\pi_h(\cdot|\tau_h)}[g_h(\tau_h,\boldsymbol{a}_h)]do_h \right\|_{L_2(\mu),\Sigma_{\rho_{n,h-1},\phi}}^2
$$
$$
=n\mathbb{E}_{(\tilde{\tau}_{h-1},\tilde{\boldsymbol{a}}_{h-1})\sim\rho_{n,h-1}} \left[ \mathbb{E}_{\tau_h\sim\mathbb{P}^\pi(\cdot|\tau_{h-1},\boldsymbol{a}_{h-1}),\boldsymbol{a}_h\sim\pi_h(\cdot|\tau_h)}[g_h(\tau_h,\boldsymbol{a}_h)] \right]^2
$$
$$
+ \lambda\mathbb{E}_{(\tau_{h-1},\boldsymbol{a}_{h-1})\sim\rho_{n,h-1}} \left\| \int_{o_h} \mu_{h-1}(\tau_h) \cdot \mathbb{E}_{\boldsymbol{a}_h\sim\pi_h(\cdot|\tau_h)}[g_h(\tau_h,\boldsymbol{a}_h)]do_h \right\|_{\mathcal{H}}^2
$$
$$
\leqslant n\mathbb{E}_{(\tilde{\tau}_{h-1},\tilde{\boldsymbol{a}}_{h-1})\sim\rho_{n,h-1}}\mathbb{E}_{\tau_h\sim\mathbb{P}^\pi(\cdot|\tau_{h-1},\boldsymbol{a}_{h-1}),\boldsymbol{a}_h\sim\pi_h(\cdot|\tau_h)}[g_h(\tau_h,\boldsymbol{a}_h)]^2 + \lambda B^2 d
$$
$$
\leqslant n|\mathcal{A}|\mathbb{E}_{(\tilde{\tau}_h,\tilde{\boldsymbol{a}}_h)\sim\rho_{n,h-1}\circ\mathcal{U}(\mathcal{A})}[g_h(\tilde{\tau}_h,\tilde{\boldsymbol{a}}_h)]^2 + \lambda B^2 d,
$$

which finishes the proof. $\qquad\square$

**Lemma 16** (one-step back inequality for the learned model). *Assume we have a set of functions $[g_h]_{h\in[H]}$, where $g_h : \mathcal{X} \times \mathcal{A} \to \mathbb{R}$, $\|g_h\|_\infty \leqslant B$, $\forall h \in [H]$. Given Lemma 51, we have that $\forall\pi$,*

$$
\sum_{h\in[H]} \mathbb{E}_{(\tau_h,\boldsymbol{a}_h)\sim\rho_{n,h}}[g_h(\tau_h,\boldsymbol{a}_h)]
$$
$$
\leqslant \sum_{h\in[H]} \mathbb{E}_{(\tau_{h-1},\boldsymbol{a}_{h-1})\sim\rho_{n,h-1}} \left[ \left\|\hat{\phi}_{h-1}(\tau_{h-1},\boldsymbol{a}_{h-1})\right\|_{L_2(\mu),\Sigma_{\rho_{n,h-1},\hat{\phi}}^{-1}} \right]
$$
$$
\cdot \sqrt{n|\mathcal{A}|\mathbb{E}_{(\tilde{\tau}_h,\tilde{\boldsymbol{a}}_h)\sim\rho_{n,h-1}\circ\mathcal{U}(\mathcal{A})}[g_h(\tilde{\tau}_h,\tilde{\boldsymbol{a}}_h)]^2 + nB^2\zeta_n + B^2\lambda d}
$$

*Proof.* The proof can be adapted from the proof of Lemma B.3 in (30), and we include it for the completeness. We make the following one-step-back decomposition:

$$\mathbb{E}_{(\tau_h, \boldsymbol{a}_h)\sim\rho_{n,h}}[g_h(\tau_h, \boldsymbol{a}_h)]$$

$$=\mathbb{E}_{(\tau_{h-1}, \boldsymbol{a}_{h-1})\sim\rho_{n,h-1}}\left[\int_{o_h}\langle\hat{\phi}_{h-1}(\tau_{h-1}, \boldsymbol{a}_{h-1}), \hat{\mu}_{h-1}(\tau_h)\rangle_{L_2(\mu)}\cdot\mathbb{E}_{\boldsymbol{a}_h\sim\pi_h(\cdot|\tau_h)}[g_h(\tau_h, \boldsymbol{a}_h)]do_h\right]$$

$$\leqslant\mathbb{E}_{(\tau_{h-1}, \boldsymbol{a}_{h-1})\sim\rho_{n,h-1}}\left\|\hat{\phi}_{h-1}(\tau_{h-1}, \boldsymbol{a}_{h-1})\right\|_{L_2(\mu),\Sigma_{\rho_{n,h-1},\hat{\phi}}^{-1}}$$

$$\cdot\mathbb{E}_{(\tau_{h-1}, \boldsymbol{a}_{h-1})\sim\rho_{n,h-1}}\left\|\int_{o_h}\hat{\mu}_{h-1}(\tau_h)\mathbb{E}_{\boldsymbol{a}_h\sim\pi_h(\cdot|\tau_h)}[g_h(\tau_h, \boldsymbol{a}_h)]do_h\right\|_{L_2(\mu),\Sigma_{\rho_{n,h-1},\hat{\phi}}}.$$

Direct computation shows that

$$\mathbb{E}_{(\tau_{h-1}, \boldsymbol{a}_{h-1})\sim\rho_{n,h-1}}\left\|\int_{o_h}\hat{\mu}_{h-1}(\tau_h)\mathbb{E}_{\boldsymbol{a}_h\sim\pi_h(\cdot|\tau_h)}[g_h(\tau_h, \boldsymbol{a}_h)]do_h\right\|_{L_2(\mu),\Sigma_{\rho_{n,h-1},\hat{\phi}}}^2$$

$$=n\mathbb{E}_{(\tilde{\phi}_{h-1}, \tilde{\boldsymbol{a}}_{h-1})\sim\rho_{n,h-1}}\left[\mathbb{E}_{\tau_h\sim\widehat{\mathcal{P}}_n(\cdot|\tilde{\tau}_{h-1}, \tilde{\boldsymbol{a}}_{h-1}), \boldsymbol{a}_h\sim\pi_h(\cdot|\tau_h)}[g_h(\tau_h, \boldsymbol{a}_h)]\right]^2$$

$$+\lambda\mathbb{E}_{(\tau_{h-1}, \boldsymbol{a}_{h-1})\sim\rho_{n,h-1}}\left\|\int_{o_h}\hat{\mu}_{h-1}(\tau_h|\cdot)\mathbb{E}_{\boldsymbol{a}_h\sim\pi_h(\cdot|\tau_h)}[g_h(\tau_h, \boldsymbol{a}_h)]do_h\right\|_{\mathcal{H}}^2$$

$$\leqslant n\mathbb{E}_{(\tilde{\tau}_{h-1}, \tilde{\boldsymbol{a}}_{h-1})\sim\rho_{n,h-1}}\mathbb{E}_{\tau_h\sim\widehat{\mathcal{P}}_n(\cdot|\tilde{\tau}_{h-1}, \tilde{\boldsymbol{a}}_{h-1}), \boldsymbol{a}_h\sim\pi_h(\cdot|\tau_h)}[g_h(\tau_h, \boldsymbol{a}_h)]^2 + B^2\lambda d$$

$$\leqslant n|\mathcal{A}|\mathbb{E}_{(\tilde{\tau}_h, \tilde{\boldsymbol{a}}_h)\sim\rho_{n,h-1}\circ\mathcal{U}(\mathcal{A})}[g_h(\tilde{\tau}_h, \tilde{\boldsymbol{a}}_h)]^2 + nB^2\zeta_n + B^2\lambda d,$$

where we use the MLE guarantee for each individual step to obtain the last inequality. This finishes the proof. $\square$

**Lemma 17** (Optimism for NE and CCE). *For episode* $n \in [N]$, *set*

$$\hat{b}_{n,h} = \min\left\{\alpha_n\|\hat{\phi}_{n,h-1}(\tau_{h-1}, \boldsymbol{a}_{h-1})\|_{L_2(\mu),\hat{\Sigma}_{n,h,\hat{\phi}_n}^{-1}}, H\right\},$$

*with* $\alpha_n = \Theta(H\sqrt{\lambda d + nA\zeta_n})$, $\lambda = \Theta(d\log(nH|\mathcal{M}|/\delta))$,

$$\hat{\Sigma}_{n,h,\hat{\phi}_n} : L_2(\mu) \to L_2(\mu), \quad \hat{\Sigma}_{n,h,\hat{\phi}_n} := \sum_{(\tau_h, \boldsymbol{a}_h)\in\mathcal{D}_{n,h}}\left[\hat{\phi}_{n,h}(\tau_h, \boldsymbol{a}_h)\hat{\phi}_{n,h}(\tau_h, \boldsymbol{a}_h)^\top\right] + \lambda I_d.$$

$\pi^n$ *is computed by solving NE or CCE. Then with probability at least* $1 - \delta$, $\forall n \in [N], i \in [M]$ *we have*

$$\overline{v}_i^n - v_i^{\dagger, \pi_{-i}^n} \geqslant 0.$$

*Proof.* Define $\tilde{\mu}_{h,i}^n(\cdot|\tau) := \arg\max_\mu\left(\mathbb{D}_{\mu, \pi_{h,-i}^n}Q_{h,i}^{\dagger, \pi_{-i}^n}\right)(\tau)$ as the best response policy for player $i$ at step $h$, and let $\tilde{\pi}_h^n = \tilde{\mu}_{h,i}^n \times \pi_{h,-i}^n$. Let $f_h^n(\tau, \boldsymbol{a}) = \left\|\widehat{\mathcal{P}}_{n,h}(\cdot|\tau, \boldsymbol{a}) - P_h(\cdot|\tau, \boldsymbol{a})\right\|_1$, then according to lemma 51 and lemma 56, we have that using the chosen $\lambda$, with probability at least $1 - \delta$, $\forall n \in [N], h \in [H], \widehat{\mathcal{P}} \in \mathcal{M}$,

$$\mathbb{E}_{(\tau, \boldsymbol{a})\sim\hat{\rho}_{n,h}}\left[(f_h^n(\tau, \boldsymbol{a}))^2\right] \leqslant \zeta_n, \quad \mathbb{E}_{(\tau, \boldsymbol{a})\sim\tilde{\rho}_{n,h}}\left[(f_h^n(\tau, \boldsymbol{a}))^2\right] \leqslant \zeta_n,$$

$$\|\phi_{h-1}(\tau_{h-1}, \boldsymbol{a}_{h-1})\|_{\hat{\Sigma}_{n,h-1,\phi}^{-1}} = \Theta\left(\|\phi_{h-1}(\tau_{h-1}, \boldsymbol{a}_{h-1})\|_{\hat{\Sigma}_{\rho_{n,h-1},\phi}^{-1}}\right).$$

A direct conclusion is we can find an absolute constant $c$, such that

$$\hat{b}_{n,h}(\tau_h, \boldsymbol{a}_h) = \min\left\{\alpha_n\left\|\hat{\phi}_{n,h}(\tau_{h-1}, \boldsymbol{a}_{h-1})\right\|_{\Sigma_{n,h-1,\hat{\phi}}^{-1}}, H\right\}$$

$$\geqslant \min\left\{c\alpha_n\left\|\hat{\phi}_{n,h}(\tau_{h-1}, \boldsymbol{a}_{h-1})\right\|_{\Sigma_{n,h-1,\hat{\phi}}^{-1}}, H\right\}, \quad \forall n \in [N], h \in [H].$$

Next, we prove by induction that $\forall h \in [H]$,

$$
\mathbb{E}_{\tau \sim d_{\mathcal{P}}^{\tilde{\pi}^n}} \left[ \overline{V}_{h,i}^n(\tau) - V_{h,i}^{\dagger, \pi_{-i}^n}(\tau) \right] \geqslant \sum_{h'=h}^{H} \mathbb{E}_{(\tau_{h'}, \boldsymbol{a}_{h'}) \sim d_{\mathcal{P}}^{\tilde{\pi}^n}} \left[ \hat{b}_{n,h'}(\tau_{h'}, \boldsymbol{a}_{h'}) - H \min\{ f_{h'}^n(\tau_{h'}, a_{h'}), 1 \} \right].
$$

(8)

First, notice that $\forall h \in [H]$,

$$
\begin{aligned}
\mathbb{E}_{x \sim d_{\mathcal{P}}^{\tilde{\pi}^n}} \left[ \overline{V}_{h,i}^n(\tau) - V_{h,i}^{\dagger, \pi_{-i}^n}(\tau) \right] &= \mathbb{E}_{\tau \sim d_{\mathcal{P}}^{\tilde{\pi}^n}} \left[ \left( \mathbb{D}_{\pi_h^n} \overline{Q}_{h,i}^n \right)(\tau) - \left( \mathbb{D}_{\tilde{\pi}_h^n} Q_{h,i}^{\dagger, \pi_{-i}^n} \right)(\tau) \right] \\
&\geqslant \mathbb{E}_{\tau \sim d_{\mathcal{P}}^{\tilde{\pi}^n}} \left[ \left( \mathbb{D}_{\tilde{\pi}_h^n} \overline{Q}_{h,i}^n \right)(\tau) - \left( \mathbb{D}_{\tilde{\pi}_h^n} Q_{h,i}^{\dagger, \pi_{-i}^n} \right)(\tau) \right] \\
&= \mathbb{E}_{(\tau, \boldsymbol{a}) \sim d_{\mathcal{P}}^{\tilde{\pi}^n}} \left[ \overline{Q}_{h,i}^n(\tau, \boldsymbol{a}) - Q_{h,i}^{\dagger, \pi_{-i}^n}(\tau, \boldsymbol{a}) \right],
\end{aligned}
$$

where the inequality uses the fact that $\pi_h^n$ is the NE (or CCE) solution for $\left\{ \overline{Q}_{h,i}^n \right\}_{i=1}^M$. Now we are ready to prove equation 8:

- When $h = H$, we have

$$
\begin{aligned}
\mathbb{E}_{\tau \sim d_{\mathcal{P}}^{\tilde{\pi}^n}} \left[ \overline{V}_{H,i}^n(\tau) - V_{H,i}^{\dagger, \pi_{-i}^n}(\tau) \right] &\geqslant \mathbb{E}_{(\tau, \boldsymbol{a}) \sim d_{\mathcal{P}}^{\tilde{\pi}^n}} \left[ \overline{Q}_{H,i}^n(\tau, \boldsymbol{a}) - Q_{H,i}^{\dagger, \pi_{-i}^n}(\tau, \boldsymbol{a}) \right] \\
&= \mathbb{E}_{(\tau, \boldsymbol{a}) \sim d_{\mathcal{P}}^{\tilde{\pi}^n}} \left[ \hat{b}_{n,H}(\tau, \boldsymbol{a}) - H \min\{ f_H^n(\tau, \boldsymbol{a}), 1 \} \right].
\end{aligned}
$$

- Suppose the statement is true for step $h + 1$, then for step $h$, we have

$$
\begin{aligned}
&\mathbb{E}_{\tau \sim d_{\mathcal{P}}^{\tilde{\pi}^n}} \left[ \overline{V}_{h,i}^n(\tau) - V_{h,i}^{\dagger, \pi_{-i}^n}(\tau) \right] \\
&\geqslant \mathbb{E}_{(\tau, \boldsymbol{a}) \sim d_{\mathcal{P}}^{\tilde{\pi}^n}} \left[ \overline{Q}_{h,i}^n(\tau, \boldsymbol{a}) - Q_{h,i}^{\dagger, \pi_{-i}^n}(\tau, \boldsymbol{a}) \right] \\
&= \mathbb{E}_{(\tau, \boldsymbol{a}) \sim d_{\mathcal{P}}^{\tilde{\pi}^n}} \left[ \hat{b}_{n,h}(\tau, \boldsymbol{a}) + \left( \hat{\mathcal{P}}_h \overline{V}_{h+1,i}^n \right)(\tau, \boldsymbol{a}) - \left( \mathcal{P}_h V_{h+1,i}^{\dagger, \pi_{-i}^n} \right)(\tau, \boldsymbol{a}) \right] \\
&= \mathbb{E}_{(\tau_h, \boldsymbol{a}_h) \sim d_{\hat{\mathcal{P}}_{n,h}}^{\tilde{\pi}^n}} \Big[ \hat{b}_{n,h}(\tau_h, \boldsymbol{a}_h) \\
&\qquad + \left( \hat{\mathcal{P}}_{n,h} \left( \overline{V}_{h+1,i}^n - V_{h+1,i}^{\dagger, \pi_{-i}^n} \right) \right)(\tau_h, \boldsymbol{a}_h) + \left( \left( \hat{\mathcal{P}}_{n,h} - \mathcal{P}_h \right) V_{h+1,i}^{\dagger, \pi_{-i}^n} \right)(\tau_h, \boldsymbol{a}_h) \Big] \\
&= \mathbb{E}_{(\tau_h, \boldsymbol{a}_h) \sim d_{\hat{\mathcal{P}}_{n,h}}^{\tilde{\pi}^n}} \left[ \hat{b}_{n,h}(\tau_h, \boldsymbol{a}_h) + \left( \left( \hat{\mathcal{P}}_{n,h} - \mathcal{P}_h \right) V_{h+1,i}^{\dagger, \pi_{-i}^n} \right)(\tau_h, \boldsymbol{a}_h) \right] \\
&\qquad + \mathbb{E}_{\tau_{h+1} \sim d_{\hat{\mathcal{P}}_{n,h+1}}^{\tilde{\pi}^n}} \left[ \overline{V}_{h+1,i}^n(\tau_{h+1}) - V_{h+1,i}^{\dagger, \pi_{-i}^n}(\tau_{h+1}) \right] \\
&\geqslant \mathbb{E}_{(\tau_h, \boldsymbol{a}_h) \sim d_{\hat{\mathcal{P}}_{n,h}}^{\tilde{\pi}^n}} \left[ \hat{b}_{n,h}(\tau_h, \boldsymbol{a}_h) - H \min \{ f_h^n(\tau_h, \boldsymbol{a}_h), 1 \} \right] \\
&\qquad + \mathbb{E}_{\tau_{h+1} \sim d_{\hat{\mathcal{P}}_{n,h+1}}^{\tilde{\pi}^n}} \left[ \overline{V}_{h+1,i}^n(\tau_{h+1}) - V_{h+1,i}^{\dagger, \pi_{-i}^n}(\tau_{h+1}) \right] \\
&\geqslant \sum_{h'=h}^{H} \mathbb{E}_{(\tau_h, \boldsymbol{a}_h) \sim d_{\hat{\mathcal{P}}_{n,h'}}^{\tilde{\pi}^n}} \left[ \hat{b}_{n,h'}(\tau_h, \boldsymbol{a}_h) - H \min \{ f_{h'}^n(\tau_h, \boldsymbol{a}_h), 1 \} \right],
\end{aligned}
$$

where we use the fact

$$
\begin{aligned}
\left| \left( \hat{\mathcal{P}}_{n,h} - \mathcal{P}_h \right) V_{h+1,i}^{\dagger, \pi_{-i}^n} \right| (\tau, \boldsymbol{a}) &\leqslant \min \left\{ H, \left\| \hat{\mathcal{P}}_{n,h}(\cdot | \tau, \boldsymbol{a}) - \mathcal{P}_h(\cdot | \tau, \boldsymbol{a}) \right\|_1 \left\| V_{h+1,i}^{\dagger, \pi_{-i}^n} \right\|_\infty \right\} \\
&\leqslant H \min \left\{ 1, \left\| \hat{\mathcal{P}}_{n,h}(\cdot | \tau, \boldsymbol{a}) - \mathcal{P}_h(\cdot | \tau, \boldsymbol{a}) \right\|_1 \right\} \\
&= H \min \left\{ 1, f_{h'}^n(\tau, \boldsymbol{a}) \right\}
\end{aligned}
$$

and the last row uses the induction assumption.

Therefore, we have proved equation 8. We then apply $h = 0$ to equation 8, and get

$$\mathbb{E}_{\tau \sim d_0} \left[ \overline{V}_{0,i}^n(\tau) - V_{0,i}^{\dagger, \pi_{-i}^n}(\tau) \right]$$

$$= \mathbb{E}_{\tau \sim d_{\hat{\mathcal{P}}_{n,0}}^{\tilde{\pi}^n}} \left[ \overline{V}_{0,i}^n(\tau) - V_{0,i}^{\dagger, \pi_{-i}^n}(\tau) \right]$$

$$\geqslant \sum_{h=0}^{H} \mathbb{E}_{(\tau, \boldsymbol{a}) \sim d_{\hat{\mathcal{P}}_{n,h}}^{\tilde{\pi}^n}} \left[ \hat{b}_{n,h}(\tau, \boldsymbol{a}) - H \min \left\{ f_h^n(\tau, \boldsymbol{a}), 1 \right\} \right]$$

$$= \sum_{h=0}^{H} \mathbb{E}_{(\tau, \boldsymbol{a}) \sim d_{\hat{\mathcal{P}}_{n,h}}^{\tilde{\pi}^n}} \left[ \hat{b}_{n,h}(\tau, \boldsymbol{a}) \right] - H \sum_{h=0}^{H} \mathbb{E}_{(\tau, \boldsymbol{a}) \sim d_{\hat{\mathcal{P}}_{n,h}}^{\tilde{\pi}^n}} \left[ \min \left\{ f_h^n(\tau, \boldsymbol{a}), 1 \right\} \right].$$

Next we are going to bound the second term. Applying Lemma 35 to $g_h(x, \boldsymbol{a}) = \min \left\{ f_h^n(x, \boldsymbol{a}), 1 \right\}$, we have

$$\sum_{k \in [n]} \sum_{h=0}^{H} \mathbb{E}_{(\tau_h, \boldsymbol{a}_h) \sim \rho_{k,h}} \left[ \min \left\{ f_h^k(\tau_h, \boldsymbol{a}_h), 1 \right\} \right]$$

$$\leqslant \sum_{k \in [n]} \sum_{h=0}^{H} \mathbb{E}_{(\tau_{h-1}, \boldsymbol{a}_{h-1}) \sim \rho_{k,h-1}} \left[ \left\| \hat{\phi}_{k,h-1}(\tau_{h-1}, \boldsymbol{a}_{h-1}) \right\|_{\Sigma_{\rho_{k,h-1}, \hat{\phi}}^{-1}} \right]$$

$$\cdot \sqrt{n |\mathcal{A}| \cdot \mathbb{E}_{(\tilde{\tau}_h, \tilde{\boldsymbol{a}}_h) \sim \rho_{k,h-1}} \left[ \min \left\{ f_h^k(\tilde{\tau}_h, \tilde{\boldsymbol{a}}_h), 1 \right\}^2 \right] + \lambda d + n \zeta_n}$$

$$\leqslant \sum_{k \in [n]} \sum_{h=0}^{H} \mathbb{E}_{(\tau_{h-1}, \boldsymbol{a}_{h-1}) \sim \rho_{k,h-1}} \left[ \left\| \alpha_k \hat{\phi}_{k,h-1}(\tau_{h-1}, \boldsymbol{a}_{h-1}) \right\|_{\Sigma_{\rho_{k,h-1}, \hat{\phi}}^{-1}} \right]$$

Note that we here use the fact $\min \left\{ f_h^n(\tau, \boldsymbol{a}), 1 \right\} \leqslant 1$, $\mathbb{E}_{(\tilde{\tau}_h, \tilde{\boldsymbol{a}}_h) \sim \rho_{n,h-1}} \left[ \min \left\{ f_h^n(\tilde{\tau}_h, \tilde{\boldsymbol{a}}_h), 1 \right\}^2 \right] \leqslant \zeta_n$ and our choice of $\alpha_n$.

Combining all things together,

$$\sum_{k \in [n]} \overline{v}_i^k - v_i^{\dagger, \pi_{-i}^k} = \sum_{k \in [n]} \mathbb{E}_{\tau \sim d_0} \left[ \overline{V}_{0,i}^k(\tau) - V_{0,i}^{\dagger, \pi_{-i}^k}(\tau) \right]$$

$$\geqslant \sum_{k \in [n]} \sum_{h=1}^{H} \mathbb{E}_{(\tau, \boldsymbol{a}) \sim \rho_{k,h}} \left[ \hat{b}_h^k(\tau, \boldsymbol{a}) \right] - H \sum_{k \in [n]} \sum_{h=1}^{H} \mathbb{E}_{(\tau, \boldsymbol{a}) \sim \rho_{k,h}} \left[ \min \left\{ f_h^k(\tau, \boldsymbol{a}), 1 \right\} \right]$$

$$\geqslant 0.$$

Since the inequality holds for all $n$, we have that $\overline{v}_i^n - v_i^{\dagger, \pi_{-i}^n}$ for all $n$.

$\square$

**Lemma 18** (Optimism for CE). *For episode $n \in [N]$, set*

$$\hat{b}_{n,h} = \min \left\{ \alpha_n \| \hat{\phi}_{n,h-1}(\tau_{h-1}, \boldsymbol{a}_{h-1}) \|_{L_2(\mu), \hat{\Sigma}_{n,h,\hat{\phi}_n}^{-1}}, H \right\},$$

*with $\alpha_n = \Theta(H \sqrt{\lambda d + n A \zeta_n})$, $\lambda = \Theta(d \log(n H |\mathcal{M}| / \delta))$,*

$$\hat{\Sigma}_{n,h,\hat{\phi}_n} : L_2(\mu) \to L_2(\mu), \quad \hat{\Sigma}_{n,h,\hat{\phi}_n} := \sum_{(\tau_h, \boldsymbol{a}_h) \in \mathcal{D}_{n,h}} \left[ \hat{\phi}_{n,h}(\tau_h, \boldsymbol{a}_h) \hat{\phi}_{n,h}(\tau_h, \boldsymbol{a}_h)^\top \right] + \lambda I_d.$$

*$\pi^n$ is computed by solving CE. Then with probability at least $1 - \delta$, $\forall n \in [N], i \in [M]$ we have*

$$\overline{v}_i^n - \max_{\omega \in \Omega_i} v_i^{\omega \circ \pi^n} \geqslant 0, \quad \forall n \in [N], i \in [M].$$

*Proof.* Denote $\tilde{\omega}_{h,i}^{(n)} = \arg\max_{\omega_h \in \Omega_{h,i}} \left( \mathbb{D}_{\omega_h \circ \pi_h^{(n)}} \max_{\omega \in \Omega_i} Q_{h,i}^{\omega \circ \pi^{(n)}} \right)(\tau)$ and let $\tilde{\pi}_h^{(n)} = \tilde{\omega}_{h,i} \circ \pi_h^{(n)}$. Let $f_h^n(\tau, \boldsymbol{a}) = \left\| \widehat{\mathcal{P}}_{n,h}(\cdot | \tau, \boldsymbol{a}) - P_h(\cdot | \tau, \boldsymbol{a}) \right\|_1$, then according to lemma 51 and lemma 56, we have that using the chosen $\lambda$, with probability at least $1 - \delta$, $\forall n \in [N], h \in [H], \widehat{\mathcal{P}} \in \mathcal{M}$,

$$\mathbb{E}_{(\tau, \boldsymbol{a}) \sim \hat{\rho}_{n,h}} \left[ (f_h^n(\tau, \boldsymbol{a}))^2 \right] \leqslant \zeta_n, \quad \mathbb{E}_{(\tau, \boldsymbol{a}) \sim \tilde{\rho}_{n,h}} \left[ (f_h^n(\tau, \boldsymbol{a}))^2 \right] \leqslant \zeta_n,$$

$$\|\phi_h(\tau_{h-1}, \boldsymbol{a}_{h-1})\|_{\hat{\Sigma}_{n,h-1,\phi}^{-1}} = \Theta \left( \|\phi_h(\tau_{h-1}, \boldsymbol{a}_{h-1})\|_{\hat{\Sigma}_{\rho_{n,h-1},\phi}^{-1}} \right).$$

A direct conclusion is we can find an absolute constant $c$, such that

$$\hat{b}_{n,h}(\tau_h, \boldsymbol{a}_h) = \min \left\{ \alpha_n \left\| \hat{\phi}_{n,h}(\tau_{h-1}, \boldsymbol{a}_{h-1}) \right\|_{\Sigma_{n,h-1,\hat{\phi}}^{-1}}, H \right\}$$

$$\geqslant \min \left\{ c\alpha_n \left\| \hat{\phi}_{n,h}(\tau_{h-1}, \boldsymbol{a}_{h-1}) \right\|_{\Sigma_{n,h-1,\hat{\phi}}^{-1}}, H \right\}, \quad \forall n \in [N], h \in [H].$$

Next, we prove by induction that $\forall h \in [H]$,

$$\mathbb{E}_{\tau \sim d_{\mathcal{P}}^{\tilde{\pi}^n}} \left[ \overline{V}_{h,i}^n(\tau) - \max_{\omega \in \Omega_i} V_{h,i}^{\omega \circ \pi^n}(\tau) \right] \geqslant \sum_{h'=h}^{H} \mathbb{E}_{(\tau_{h'}, \boldsymbol{a}_{h'}) \sim d_{\mathcal{P}}^{\tilde{\pi}^n}} \left[ \hat{b}_{n,h'}(\tau_{h'}, \boldsymbol{a}_{h'}) - H \min\{f_{h'}^n(\tau_{h'}, \boldsymbol{a}_{h'}), 1\} \right].$$

$$(9)$$

First, notice that $\forall h \in [H]$,

$$\mathbb{E}_{\tau \sim d_{\mathcal{P}}^{\tilde{\pi}^n}} \left[ \overline{V}_{h,i}^n(\tau) - \max_{\omega \in \Omega_i} V_{h,i}^{\omega \circ \pi^n}(\tau) \right] = \mathbb{E}_{\tau \sim d_{\mathcal{P}}^{\tilde{\pi}^n}} \left[ \left( \mathbb{D}_{\pi_h^n} \overline{Q}_{h,i}^n \right)(\tau) - \left( \mathbb{D}_{\tilde{\pi}_h^n} \max_{\omega \in \Omega_i} Q_{h,i}^{\omega \circ \pi^n} \right)(\tau) \right]$$

$$\geqslant \mathbb{E}_{\tau \sim d_{\mathcal{P}}^{\tilde{\pi}^n}} \left[ \left( \mathbb{D}_{\tilde{\pi}_h^n} \overline{Q}_{h,i}^n \right)(\tau) - \left( \mathbb{D}_{\tilde{\pi}_h^n} \max_{\omega \in \Omega_i} Q_{h,i}^{\omega \circ \pi^n} \right)(\tau) \right]$$

$$= \mathbb{E}_{(\tau, \boldsymbol{a}) \sim d_{\mathcal{P}}^{\tilde{\pi}^n}} \left[ \overline{Q}_{h,i}^n(\tau, \boldsymbol{a}) - \max_{\omega \in \Omega_i} Q_{h,i}^{\omega \circ \pi^n}(\tau, \boldsymbol{a}) \right],$$

where the inequality uses the fact that $\pi_h^n$ is the CE solution for $\left\{ \overline{Q}_{h,i}^n \right\}_{i=1}^{M}$. Now we are ready to prove equation 9:

- When $h = H$, we have

$$\mathbb{E}_{\tau \sim d_{\mathcal{P}}^{\tilde{\pi}^n}} \left[ \overline{V}_{H,i}^n(\tau) - \max_{\omega \in \Omega_i} V_{H,i}^{\omega \circ \pi^n}(\tau) \right]$$

$$\geqslant \mathbb{E}_{(\tau, \boldsymbol{a}) \sim d_{\mathcal{P}}^{\tilde{\pi}^n}} \left[ \overline{Q}_{H,i}^n(\tau, \boldsymbol{a}) - \max_{\omega \in \Omega_i} Q_{H,i}^{\omega \circ \pi^n}(\tau, \boldsymbol{a}) \right]$$

$$= \mathbb{E}_{(\tau, \boldsymbol{a}) \sim d_{\mathcal{P}}^{\tilde{\pi}^n}} \left[ \hat{b}_{n,H}(\tau, \boldsymbol{a}) - H \min\{f_H^n(\tau, \boldsymbol{a}), 1\} \right].$$

- Suppose the statement is true for step $h + 1$, then for step $h$, we have

$$\mathbb{E}_{\tau \sim d_{\mathcal{P}}^{\tilde{\pi}^n}} \left[ \overline{V}_{h,i}^n(\tau) - \max_{\omega \in \Omega_i} V_{h,i}^{\omega \circ \pi^n}(\tau) \right]$$

$$\geqslant \mathbb{E}_{(\tau, \boldsymbol{a}) \sim d_{\mathcal{P}}^{\tilde{\pi}^n}} \left[ \overline{Q}_{h,i}^n(\tau, \boldsymbol{a}) - \max_{\omega \in \Omega_i} Q_{h,i}^{\omega \circ \pi^n}(\tau, \boldsymbol{a}) \right]$$

$$= \mathbb{E}_{(\tau, \boldsymbol{a}) \sim d_{\mathcal{P}}^{\tilde{\pi}^n}} \left[ \hat{b}_{n,h}(\tau, \boldsymbol{a}) + \left( \hat{\mathcal{P}}_h \overline{V}_{h+1,i}^n \right)(\tau, \boldsymbol{a}) - \left( \mathcal{P}_h \max_{\omega \in \Omega_i} V_{h+1,i}^{\omega \circ \pi^n} \right)(\tau, \boldsymbol{a}) \right]$$

$$= \mathbb{E}_{(\tau_h, \boldsymbol{a}_h) \sim d_{\widehat{\mathcal{P}}_{n,h}}^{\tilde{\pi}^n}} \Big[ \hat{b}_{n,h}(\tau_h, \boldsymbol{a}_h)$$

$$+ \left( \widehat{\mathcal{P}}_{n,h} \left( \overline{V}_{h+1,i}^n - V_{h+1,i}^{\omega \circ \pi^n} \right) \right)(\tau_h, \boldsymbol{a}_h) + \left( \left( \widehat{\mathcal{P}}_{n,h} - \mathcal{P}_h \right) V_{h+1,i}^{\omega \circ \pi^n} \right)(\tau_h, \boldsymbol{a}_h) \Big]$$

$$= \mathbb{E}_{(\tau_h, \boldsymbol{a}_h) \sim d_{\widehat{\mathcal{P}}_{n,h}}^{\tilde{\pi}^n}} \left[ \hat{b}_{n,h}(\tau_h, \boldsymbol{a}_h) + \left( \left( \widehat{\mathcal{P}}_{n,h} - \mathcal{P}_h \right) V_{h+1,i}^{\omega \circ \pi^n} \right)(\tau_h, \boldsymbol{a}_h) \right]$$

$$+ \mathbb{E}_{\tau_{h+1} \sim d_{\widehat{\mathcal{P}}_{n,h+1}}^{\tilde{\pi}^n}} \left[ \overline{V}_{h+1,i}^n(\tau_{h+1}) - V_{h+1,i}^{\omega \circ \pi^n}(\tau_{h+1}) \right]$$

$$\geqslant \mathbb{E}_{(\tau_h, \boldsymbol{a}_h) \sim d_{\widehat{\mathcal{P}}_{n,h}}^{\tilde{\pi}^n}} \left[ \hat{b}_{n,h}(\tau_h, \boldsymbol{a}_h) - H \min \left\{ f_h^n(\tau_h, \boldsymbol{a}_h), 1 \right\} \right]$$

$$+ \mathbb{E}_{\tau_{h+1} \sim d_{\widehat{\mathcal{P}}_{n,h+1}}^{\tilde{\pi}^n}} \left[ \overline{V}_{h+1,i}^n(\tau_{h+1}) - V_{h+1,i}^{\omega \circ \pi^n}(\tau_{h+1}) \right]$$

$$\geqslant \sum_{h'=h}^{H} \mathbb{E}_{(\tau_h, \boldsymbol{a}_h) \sim d_{\widehat{\mathcal{P}}_{n,h'}}^{\tilde{\pi}^n}} \left[ \hat{b}_{n,h'}(\tau_h, \boldsymbol{a}_h) - H \min \left\{ f_{h'}^n(\tau_h, \boldsymbol{a}_h), 1 \right\} \right],$$

where we use the fact

$$\left| \left( \widehat{\mathcal{P}}_{n,h} - \mathcal{P}_h \right) V_{h+1,i}^{\omega \circ \pi^n} \right|(\tau, \boldsymbol{a}) \leqslant \min \left\{ H, \left\| \widehat{\mathcal{P}}_{n,h}(\cdot|\tau, \boldsymbol{a}) - \mathcal{P}_h(\cdot|\tau, \boldsymbol{a}) \right\|_1 \left\| V_{h+1,i}^{\omega \circ \pi^n} \right\|_\infty \right\}$$

$$\leqslant H \min \left\{ 1, \left\| \widehat{\mathcal{P}}_{n,h}(\cdot|\tau, \boldsymbol{a}) - \mathcal{P}_h(\cdot|\tau, \boldsymbol{a}) \right\|_1 \right\}$$

$$= H \min \left\{ 1, f_{h'}^n(\tau, \boldsymbol{a}) \right\}$$

and the last row uses the induction assumption.

Therefore, we have proved equation 9. We then apply $h = 0$ to equation 9, and get

$$\mathbb{E}_{\tau \sim d_0} \left[ \overline{V}_{0,i}^n(\tau) - V_{0,i}^{\omega \circ \pi^n}(\tau) \right]$$

$$= \mathbb{E}_{\tau \sim d_{\widehat{\mathcal{P}}_{n,0}}^{\tilde{\pi}^n}} \left[ \overline{V}_{0,i}^n(\tau) - V_{0,i}^{\omega \circ \pi^n}(\tau) \right]$$

$$\geqslant \sum_{h=0}^{H} \mathbb{E}_{(\tau, \boldsymbol{a}) \sim d_{\widehat{\mathcal{P}}_{n,h}}^{\tilde{\pi}^n}} \left[ \hat{b}_{n,h}(\tau, \boldsymbol{a}) - H \min \left\{ f_h^n(\tau, \boldsymbol{a}), 1 \right\} \right]$$

$$= \sum_{h=0}^{H} \mathbb{E}_{(\tau, \boldsymbol{a}) \sim d_{\widehat{\mathcal{P}}_{n,h}}^{\tilde{\pi}^n}} \left[ \hat{b}_{n,h}(\tau, \boldsymbol{a}) \right] - H \sum_{h=0}^{H} \mathbb{E}_{(\tau, \boldsymbol{a}) \sim d_{\widehat{\mathcal{P}}_{n,h}}^{\tilde{\pi}^n}} \left[ \min \left\{ f_h^n(\tau, \boldsymbol{a}), 1 \right\} \right].$$

Next we are going to bound the second term. Applying Lemma 16 to $g_h(\tau, \boldsymbol{a}) = \min\{f_h^n(\tau, \boldsymbol{a}), 1\}$, we have

$$\sum_{k \in [n]} \sum_{h=0}^{H} \mathbb{E}_{(\tau_h, \boldsymbol{a}_h) \sim \rho_{k,h}} \left[ \min\{f_h^k(\tau_h, \boldsymbol{a}_h), 1\} \right]$$

$$\leqslant \sum_{k \in [n]} \sum_{h=0}^{H} \mathbb{E}_{(\tau_{h-1}, \boldsymbol{a}_{h-1}) \sim \rho_{k,h-1}} \left[ \left\| \hat{\phi}_{k,h-1}(\tau_{h-1}, \boldsymbol{a}_{h-1}) \right\|_{\Sigma_{\rho_{k,h-1}, \hat{\phi}}^{-1}} \right]$$

$$\cdot \sqrt{n|\mathcal{A}| \cdot \mathbb{E}_{(\tilde{\tau}_h, \tilde{\boldsymbol{a}}_h) \sim \rho_{k,h-1}} \left[ \min\{f_h^k(\tilde{\tau}_h, \tilde{\boldsymbol{a}}_h), 1\}^2 \right] + \lambda d + n\zeta_n}$$

$$\leqslant \sum_{k \in [n]} \sum_{h=0}^{H} \mathbb{E}_{(\tau_{h-1}, \boldsymbol{a}_{h-1}) \sim \rho_{k,h-1}} \left[ \left\| \alpha_k \hat{\phi}_{k,h-1}(\tau_{h-1}, \boldsymbol{a}_{h-1}) \right\|_{\Sigma_{\rho_{k,h-1}, \hat{\phi}}^{-1}} \right].$$

Note that we here use the fact $\min\{f_h^n(\tau, \boldsymbol{a}), 1\} \leqslant 1$, $\mathbb{E}_{(\tilde{\tau}_h, \tilde{\boldsymbol{a}}_h) \sim \rho_{n,h-1}} \left[ \min\{f_h^n(\tilde{\tau}_h, \tilde{\boldsymbol{a}}_h), 1\}^2 \right] \leqslant \zeta_n$ and our choice of $\alpha_n$.

Combining all things together,

$$\sum_{k \in [n]} \overline{v}_i^k - \max_{\omega \in \Omega_i} v_i^{\omega \circ \pi^k} = \sum_{k \in [n]} \mathbb{E}_{\tau \sim \rho_{k,h}} \left[ \overline{V}_{0,i}^k(\tau) - \max_{\omega \in \Omega_i} V_{0,i}^{\omega \circ \pi^k}(\tau) \right]$$

$$\geqslant \sum_{k \in [n]} \sum_{h=1}^{H} \mathbb{E}_{(\tau_h, \boldsymbol{a}_h) \sim \rho_{k,h}} \left[ \hat{b}_{k,h}(\tau_h, \boldsymbol{a}_h) \right] - H \sum_{k \in [n]} \sum_{h=1}^{H} \mathbb{E}_{(\tau_h, \boldsymbol{a}_h) \sim \rho_{k,h}} \left[ \min\{f_h^k(\tau_h, \boldsymbol{a}_h), 1\} \right]$$

$$\geqslant 0,$$

Since the inequality holds for all $n$, we have that $\overline{v}_i^n - \max_{\omega \in \Omega_i} v_i^{\omega \circ \pi^n}$ for all $n$. $\qquad \square$

**Lemma 19** (Pessimism). *For episode $n \in [N]$, set*

$$\hat{b}_{n,h} = \min\left\{ \alpha_n \|\hat{\phi}_{n,h-1}(\tau_{h-1}, \boldsymbol{a}_{h-1})\|_{L_2(\mu), \hat{\Sigma}_{n,h,\hat{\phi}_n}^{-1}}, H \right\},$$

*with $\alpha_n = \Theta(H\sqrt{\lambda d + nA\zeta_n})$, $\lambda = \Theta(d \log(nH|\mathcal{M}|/\delta))$,*

$$\hat{\Sigma}_{n,h,\hat{\phi}_n} : L_2(\mu) \to L_2(\mu), \quad \hat{\Sigma}_{n,h,\hat{\phi}_n} := \sum_{(\tau_h, \boldsymbol{a}_h) \in \mathcal{D}_{n,h}} \left[ \hat{\phi}_{n,h}(\tau_h, \boldsymbol{a}_h) \hat{\phi}_{n,h}(\tau_h, \boldsymbol{a}_h)^\top \right] + \lambda I_d.$$

*Then with probability at least $1 - \delta$, $\forall n \in [N], i \in [M]$ we have*

$$\underline{v}_i^n - v_i^{\pi^n} \leqslant 0, \quad \forall n \in [N], i \in [M].$$

*Proof.* Let $f_h^n(\tau, \boldsymbol{a}) = \left\| \widehat{\mathcal{P}}_{n,h}(\cdot | \tau, \boldsymbol{a}) - P_h(\cdot | \tau, \boldsymbol{a}) \right\|_1$, then according to lemma 51 and lemma 56, we have that using the chosen $\lambda$, with probability at least $1 - \delta$, $\forall n \in [N], h \in [H], \widehat{\mathcal{P}} \in \mathcal{M}$,

$$\mathbb{E}_{(\tau, \boldsymbol{a}) \sim \hat{\rho}_{n,h}} \left[ (f_h^n(\tau, \boldsymbol{a}))^2 \right] \leqslant \zeta_n, \quad \mathbb{E}_{(\tau, \boldsymbol{a}) \sim \tilde{\rho}_{n,h}} \left[ (f_h^n(\tau, \boldsymbol{a}))^2 \right] \leqslant \zeta_n,$$

$$\|\hat{\phi}_{n,h}(\tau_{h-1}, \boldsymbol{a}_{h-1})\|_{\hat{\Sigma}_{n,h-1,\phi}^{-1}} = \Theta\left( \|\hat{\phi}_{n,h}(\tau_{h-1}, \boldsymbol{a}_{h-1})\|_{\hat{\Sigma}_{\rho_{n,h-1},\phi}^{-1}} \right).$$

A direct conclusion is we can find an absolute constant $c$, such that

$$\hat{b}_{n,h}(\tau_h, \boldsymbol{a}_h) = \min\left\{ \alpha_n \left\| \hat{\phi}_{n,h}(\tau_{h-1}, \boldsymbol{a}_{h-1}) \right\|_{\Sigma_{n,h-1,\hat{\phi}}^{-1}}, H \right\}$$

$$\geqslant \min\left\{ c\alpha_n \left\| \hat{\phi}_{n,h}(\tau_{h-1}, \boldsymbol{a}_{h-1}) \right\|_{\Sigma_{n,h-1,\hat{\phi}}^{-1}}, H \right\}, \quad \forall n \in [N], h \in [H].$$

Again, we prove by induction that $\forall h \in [H]$,

$$\mathbb{E}_{\tau \sim d^n_{\tilde{\mathcal{P}}_n}} \left[ \underline{V}^n_{h,i}(\tau) - V^{\pi^n}_{h,i}(\tau) \right] \leqslant \sum_{h'=h}^{H} \mathbb{E}_{(\tau_{h'}, \boldsymbol{a}_{h'}) \sim d^{\tilde{\pi}^n}_{\widehat{\mathcal{P}}_{n,h'}}} \left[ -\hat{b}_{n,h'}(\tau_{h'}, \boldsymbol{a}_{h'}) + H \min\{f^n_{h'}(\tau_{h'}, \boldsymbol{a}_{h'}), 1\} \right].$$

$$(10)$$

- When $h = H$, we have

$$\mathbb{E}_{\tau \sim d^{\tilde{\pi}^n}_{\widehat{\mathcal{P}}_n}} \left[ \underline{V}^n_{H,i}(\tau) - V^{\pi^n}_{H,i}(\tau) \right] = \mathbb{E}_{(\tau, \boldsymbol{a}) \sim d^{\tilde{\pi}^n}_{\widehat{\mathcal{P}}_n}} \left[ \underline{Q}^n_{H,i}(\tau, \boldsymbol{a}) - Q^{\pi^n}_{H,i}(\tau, \boldsymbol{a}) \right]$$

$$= \mathbb{E}_{(\tau, \boldsymbol{a}) \sim d^{\tilde{\pi}^n}_{\widehat{\mathcal{P}}_n}} \left[ -\hat{b}_{n,H}(\tau, \boldsymbol{a}) + H \min\{f^n_H(\tau, \boldsymbol{a}), 1\} \right].$$

- Suppose the statement is true for step $h + 1$, then for step $h$, we have

$$\mathbb{E}_{\tau \sim d^{\tilde{\pi}^n}_{\widehat{\mathcal{P}}_n}} \left[ \underline{V}^n_{h,i}(\tau) - V^{\pi^n}_{h,i}(\tau) \right]$$

$$= \mathbb{E}_{(\tau, \boldsymbol{a}) \sim d^{\tilde{\pi}^n}_{\widehat{\mathcal{P}}_n}} \left[ \underline{Q}^n_{h,i}(\tau, \boldsymbol{a}) - Q^{\pi^n}_{h,i}(\tau, \boldsymbol{a}) \right]$$

$$= \mathbb{E}_{(\tau_h, \boldsymbol{a}_h) \sim d^{\tilde{\pi}^n}_{\widehat{\mathcal{P}}_{n,h}}} \left[ -\hat{b}_{n,h}(\tau, \boldsymbol{a}) + \left( \widehat{\mathcal{P}}_{n,h} \underline{V}^n_{h+1,i} \right)(\tau_h, \boldsymbol{a}_h) - \left( \mathcal{P}_h V^{\pi^n}_{h+1,i} \right)(\tau_h, \boldsymbol{a}_h) \right]$$

$$= \mathbb{E}_{(\tau_h, \boldsymbol{a}_h) \sim d^{\tilde{\pi}^n}_{\widehat{\mathcal{P}}_{n,h}}} \left[ -\hat{b}_{n,h}(\tau_h, \boldsymbol{a}_h) \right.$$

$$\left. + \left( \widehat{\mathcal{P}}_{n,h} \left( \underline{V}^n_{h+1,i} - V^{\pi^n}_{h+1,i} \right) \right)(\tau_h, \boldsymbol{a}_h) + \left( \left( \widehat{\mathcal{P}}_{n,h} - \mathcal{P}_h \right) V^{\pi^n}_{h+1,i} \right)(\tau_h, \boldsymbol{a}_h) \right]$$

$$= \mathbb{E}_{(\tau_h, \boldsymbol{a}_h) \sim d^{\tilde{\pi}^n}_{\widehat{\mathcal{P}}_{n,h}}} \left[ -\hat{b}_{n,h}(\tau_h, \boldsymbol{a}_h) + \left( \left( \widehat{\mathcal{P}}_{n,h} - \mathcal{P}_h \right) V^{\pi^n}_{h+1,i} \right)(\tau_h, \boldsymbol{a}_h) \right]$$

$$+ \mathbb{E}_{\tau_{h+1} \sim d^{\tilde{\pi}^n}_{\widehat{\mathcal{P}}_{n,h+1}}} \left[ \underline{V}^n_{h+1,i}(\tau_{h+1}) - V^{\pi^n}_{h+1,i}(\tau_{h+1}) \right]$$

$$\leqslant \mathbb{E}_{(\tau_h, \boldsymbol{a}_h) \sim d^{\tilde{\pi}^n}_{\widehat{\mathcal{P}}_{n,h}}} \left[ -\hat{b}_{n,h}(\tau_h, \boldsymbol{a}_h) + H \min\{f^n_h(\tau_h, \boldsymbol{a}_h), 1\} \right]$$

$$+ \mathbb{E}_{\tau_{h+1} \sim d^{\tilde{\pi}^n}_{\widehat{\mathcal{P}}_{n,h+1}}} \left[ \underline{V}^n_{h+1,i}(\tau_{h+1}) - V^{\pi^n}_{h+1,i}(\tau_{h+1}) \right]$$

$$\leqslant \sum_{h'=h}^{H} \mathbb{E}_{(\tau_h, \boldsymbol{a}_h) \sim d^{\tilde{\pi}^n}_{\widehat{\mathcal{P}}_{n,h'}}} \left[ \hat{b}_{n,h'}(\tau_h, \boldsymbol{a}_h) - H \min\{f^n_{h'}(\tau_h, \boldsymbol{a}_h), 1\} \right],$$

where we use the fact

$$\left| \left( \widehat{\mathcal{P}}_{n,h} - \mathcal{P}_h \right) V^{\pi^n}_{h+1,i} \right| (\tau, \boldsymbol{a}) \leqslant \min \left\{ H, \left\| \widehat{\mathcal{P}}_{n,h}(\cdot | \tau, \boldsymbol{a}) - P_h(\cdot | \tau, \boldsymbol{a}) \right\|_1 \left\| V^{\pi^n}_{h+1,i} \right\|_\infty \right\}$$

$$\leqslant H \min \left\{ 1, \left\| \widehat{\mathcal{P}}_{n,h}(\cdot | \tau, \boldsymbol{a}) - P_h(\cdot | \tau, \boldsymbol{a}) \right\|_1 \right\}$$

$$= H \min \left\{ 1, f^n_{h'}(\tau, \boldsymbol{a}) \right\}$$

and the last row uses the induction assumption.

The remaining steps are exactly the same as the proof in Lemma 17 or Lemma 18, we get

$$\sum_{k \in [n]} \sum_{h=0}^{H} \mathbb{E}_{(\tau_h, \boldsymbol{a}_h) \sim \rho_{k,h}} \left[ \min \left\{ f^k_h(\tau_h, \boldsymbol{a}_h), 1 \right\} \right]$$

$$\leqslant \sum_{k \in [n]} \sum_{h=0}^{H} \mathbb{E}_{(\tau_{h-1}, \boldsymbol{a}_{h-1}) \sim \rho_{k,h-1}} \left[ \left\| \alpha_k \hat{\phi}_{k,h}(\tau_{h-1}, \boldsymbol{a}_{h-1}) \right\|_{\Sigma^{-1}_{\rho_{k,h-1}, \hat{\phi}}} \right]$$

Combining all things together, We have

$$
\sum_{k\in[n]} \underline{v}_i^k - v_i^{\pi^k} = \sum_{k\in[n]} \mathbb{E}_{\tau\sim d_0}\left[\overline{V}_{0,i}^k(\tau) - V_{0,i}^{\dagger,\pi_{-i}^k}(\tau)\right]
$$

$$
\leqslant \sum_{k\in[n]}\sum_{h=1}^{H} \mathbb{E}_{(\tau_h,\boldsymbol{a}_h)\sim\rho_{k,h}}\left[-\hat{b}_{k,h}(\tau_h,\boldsymbol{a}_h)\right] + H\sum_{k\in[n]}\sum_{h=1}^{H}\mathbb{E}_{(\tau_h,\boldsymbol{a}_h)\sim\rho_{k,h}}\left[\min\left\{f_h^k(\tau_h,\boldsymbol{a}_h),1\right\}\right]
$$

$$
\leqslant 0,
$$

which has finished the proof. $\qquad\square$

**Lemma 20.** *For episode* $n\in[N]$*, set*

$$
\hat{b}_{n,h} = \min\left\{\alpha_n\|\hat{\phi}_{n,h-1}(\tau_{h-1},\boldsymbol{a}_{h-1})\|_{L_2(\mu),\hat{\Sigma}_{n,h,\hat{\phi}_n}^{-1}}, H\right\},
$$

*with* $\alpha_n = \Theta(H\sqrt{\lambda d + nA\zeta_n})$, $\lambda = \Theta(d\log(nH|\mathcal{M}|/\delta))$,

$$
\hat{\Sigma}_{n,h,\hat{\phi}_n} : L_2(\mu) \to L_2(\mu), \quad \hat{\Sigma}_{n,h,\hat{\phi}_n} := \sum_{(\tau_h,\boldsymbol{a}_h)\in\mathcal{D}_{n,h}}\left[\hat{\phi}_{n,h}(\tau_h,\boldsymbol{a}_h)\hat{\phi}_{n,h}(\tau_h,\boldsymbol{a}_h)^\top\right] + \lambda I_d.
$$

*Then with probability at least* $1-\delta$*,* $\forall n\in[N], i\in[M]$ *we have*

$$
\sum_{n=1}^{N}\Delta^n \lesssim \mathcal{O}\left(H^3 d^2 N^{\frac{1}{2}} A\log(\frac{HN|\mathcal{M}|}{\delta})\right)
$$

*Proof.* Let $f_h^n(\tau,\boldsymbol{a}) = \left\|\widehat{\mathcal{P}}_{n,h}(\cdot|\tau,\boldsymbol{a}) - P_h(\cdot|\tau,\boldsymbol{a})\right\|_1$, then according to lemma 51 and lemma 56, we have that using the chosen $\lambda$, with probability at least $1-\delta$, $\forall n\in[N], h\in[H], \widehat{\mathcal{P}}\in\mathcal{M}$,

$$
\mathbb{E}_{(\tau,\boldsymbol{a})\sim\hat{\rho}_{n,h}}\left[(f_h^n(\tau,\boldsymbol{a}))^2\right] \leqslant \zeta_n, \quad \mathbb{E}_{(\tau,\boldsymbol{a})\sim\tilde{\rho}_{n,h}}\left[(f_h^n(\tau,\boldsymbol{a}))^2\right] \leqslant \zeta_n,
$$

$$
\|\phi_h(\tau_{h-1},\boldsymbol{a}_{h-1})\|_{\hat{\Sigma}_{n,h-1,\phi}^{-1}} = \Theta\left(\|\phi_h(\tau_{h-1},\boldsymbol{a}_{h-1})\|_{\hat{\Sigma}_{\rho_{n,h-1},\phi}^{-1}}\right).
$$

By definition, we have

$$
\Delta^n = \max_{i\in[M]}\left\{\overline{v}_i^n - \underline{v}_i^n\right\}.
$$

For each fixed $i\in[M], h\in[H]$ and $n\in[N]$, we have

$$
\mathbb{E}_{\tau\sim d_{\mathcal{P},h}^{\pi^n}}\left[\overline{V}_{h,i}^n(\tau) - \underline{V}_{h,i}^n(\tau)\right]
$$

$$
=\mathbb{E}_{\tau\sim d_{\mathcal{P},h}^{\pi^n}}\left[\left(\mathbb{D}_{\pi_h^n}\overline{Q}_{h,i}^n\right)(\tau) - \left(\mathbb{D}_{\pi_h^n}\underline{Q}_{h,i}^n\right)(\tau)\right]
$$

$$
=\mathbb{E}_{(\tau,\boldsymbol{a})\sim d_{\mathcal{P},h}^{\pi^n}}\left[\overline{Q}_{h,i}^n(\tau,\boldsymbol{a}) - \underline{Q}_{h,i}^n(\tau,\boldsymbol{a})\right]
$$

$$
=\mathbb{E}_{(\tau_h,\boldsymbol{a}_h)\sim d_{\mathcal{P},h}^{\pi^n}}\left[2\hat{b}_{n,h}(\tau_h,\boldsymbol{a}_h) + \widehat{\mathcal{P}}_{n,h}\left(\overline{V}_{h+1,i}^n - \underline{V}_{h+1,i}^n\right)(\tau_h,\boldsymbol{a}_h)\right]
$$

$$
=\mathbb{E}_{(\tau_h,\boldsymbol{a}_h)\sim d_{\mathcal{P},h}^{\pi^n}}\left[2\hat{b}_{n,h}(\tau_h,\boldsymbol{a}_h) + \left(\left(\widehat{\mathcal{P}}_{n,h} - \mathcal{P}_h\right)\left(\overline{V}_{h+1,i}^n - \underline{V}_{h+1,i}^n\right)\right)(\tau_h,\boldsymbol{a}_h)\right]
$$

$$
+ \mathbb{E}_{\tau_{h+1}\sim d_{\mathcal{P},h+1}^{\pi^n}}\left[\overline{V}_{h+1,i}^n(\tau_{h+1}) - \underline{V}_{h+1,i}^n(\tau_{h+1})\right]
$$

$$
\leqslant\mathbb{E}_{(\tau_h,\boldsymbol{a}_h)\sim d_{\mathcal{P},h}^{\pi^n}}\left[2\hat{b}_{n,h}(\tau_h,\boldsymbol{a}_h) + 2H^2 f_h^n(\tau_h,\boldsymbol{a}_h)\right] + \mathbb{E}_{\tau_{h+1}\sim d_{\mathcal{P},h+1}^{\pi^n}}\left[\overline{V}_{h+1,i}^n(\tau_{h+1}) - \underline{V}_{h+1,i}^n(\tau_{h+1})\right]
$$

Note that we use the fact $\overline{V}_{h+1,i}^n(\tau) - \underline{V}_{h+1,i}^n(\tau)$ is upper bounded by $2H^2$, which can be proved easily using induction using the fact that $\hat{b}_h^n(\tau,\boldsymbol{a}) \leqslant H$. Applying the above formula recursively to

$\mathbb{E}_{\tau \sim d_{\mathcal{P},h+1}^{\pi^n}}\left[\overline{V}_{h+1,i}^n(\tau) - \underline{V}_{h+1,i}^n(\tau)\right]$, one gets the following result (or more formally, one can prove by induction, just like what we did in Lemma 36, Lemma 37 and Lemma 38):

$$\mathbb{E}_{\tau \sim d_{\mathcal{P},0}^{\pi^n}}\left[\overline{V}_{0,i}^n(\tau) - \underline{V}_{0,i}^n(\tau)\right] \leqslant 2\underbrace{\sum_{h=0}^H \mathbb{E}_{(\tau_h,\boldsymbol{a}_h)\sim d_{P,h}^{\pi^n}}\left[\hat{b}_{n,h}(\tau_h,\boldsymbol{a}_h)\right]}_{(a)} + 2H^2\underbrace{\sum_{h=0}^H \mathbb{E}_{(\tau_h,\boldsymbol{a}_h)\sim d_{P,h}^{\pi^n}}\left[f_h^n(\tau_h,\boldsymbol{a}_h)\right]}_{(b)}.$$

(11)

First, we calculate the first term (a) in Inequality equation 11. Following Lemma 15 and noting the bonus $\hat{b}_{n,h}$ is $O(H)$, we have

$$\sum_{k\in[n]}\sum_{h=0}^H \mathbb{E}_{(\tau_h,\boldsymbol{a}_h)\sim\rho_{k,h}}\left[\hat{b}_{k,h}(\tau_h,\boldsymbol{a}_h)\right]$$

$$\lesssim \sum_{k\in[n]}\sum_{h=0}^H \mathbb{E}_{(\tau_h,\boldsymbol{a}_h)\sim\rho_{k,h}}\left[\min\left\{\alpha_k \left\|\hat{\phi}_{k,h}(\tau_{h-1},\boldsymbol{a}_{h-1})\right\|_{\Sigma_{k,\hat{\rho}_{k,h},\hat{\phi}}^{-1}}, H\right\}\right]$$

$$\lesssim \sum_{k\in[n]}\sum_{h=0}^H \mathbb{E}_{(\tau_{h-1},\boldsymbol{a}_{h-1})\sim\rho_{k,h-1}}\left[\|\phi_h(\tau_{h-1},\boldsymbol{a}_{h-1})\|_{\Sigma_{k,\hat{\rho}_{k,h},\hat{\phi}}^{-1}}\right]$$

$$\cdot \sqrt{n|\mathcal{A}|(\alpha_k)^2 \cdot \mathbb{E}_{(\tilde{\tau}_h,\tilde{\boldsymbol{a}}_h)\sim\rho_{k,h}}\left[\|\hat{\phi}_{k,h+1}(\tilde{\tau}_h,\tilde{\boldsymbol{a}}_h)\|_{\Sigma_{n,\hat{\rho}_{k,h},\hat{\phi}}^{-1}}^2\right] + \lambda H^2 d}.$$

Note that we use the fact that $B = H$ when applying Lemma 15. In addition, we have that for all $n$,

$$n\mathbb{E}_{(\tilde{\tau}_h,\tilde{\boldsymbol{a}}_h)\sim\rho_{n,h}}\left[\|\hat{\phi}_{n,h+1}(\tilde{\tau}_h,\tilde{\boldsymbol{a}}_h)\|_{\Sigma_{n,\hat{\rho}_{n,h},\hat{\phi}}^{-1}}^2\right]$$

$$=n\mathrm{Tr}\left(\mathbb{E}_{(\tilde{\tau}_h,\tilde{\boldsymbol{a}}_h)\sim\rho_{n,h}}\left[\hat{\phi}_{n,h+1}(\tilde{\tau}_h,\tilde{\boldsymbol{a}}_h)\hat{\phi}_{n,h+1}(\tilde{\tau}_h,\tilde{\boldsymbol{a}}_h)^\top\right]\left(n\mathbb{E}_{(\tilde{\tau}_h,\tilde{\boldsymbol{a}}_h)\sim\rho_{n,h}}\left[\hat{\phi}_{n,h+1}(\tilde{\tau}_h,\tilde{\boldsymbol{a}}_h)\hat{\phi}_{n,h+1}(\tilde{\tau}_h,\tilde{\boldsymbol{a}}_h)^\top\right] + \lambda I_d\right)^{-1}\right)$$

$$\leqslant d.$$

Then,

$$\sum_{k\in[n]}\sum_{h=0}^H \mathbb{E}_{(\tau_h,\boldsymbol{a}_h)\sim\rho_{k,h}}\left[\hat{b}_{k,h}(\tau_h,\boldsymbol{a}_h)\right] \leqslant \sum_{k\in[n]}\sum_{h=0}^H \mathbb{E}_{(\tau_{h-1},\boldsymbol{a}_{h-1})\sim\rho_{k,h-1}}\left[\|\phi_h(\tau_{h-1},\boldsymbol{a}_{h-1})\|_{\Sigma_{k,\hat{\rho}_{k,h},\hat{\phi}}^{-1}}\right]\sqrt{dA(\alpha_k)^2 + H^2 d\lambda}.$$

Second, we calculate the term (b) in inequality equation 11. Following Lemma 15, we have

$$\sum_{k\in[n]}\sum_{h=0}^H \mathbb{E}_{(\tau_h,\boldsymbol{a}_h)\sim\rho_{k,h}}[f_h^k(\tau_h,\boldsymbol{a}_h)]$$

$$\leqslant \sum_{k\in[n]}\sum_{h=0}^{H-1} \mathbb{E}_{(\tau_{h-1},\boldsymbol{a}_{h-1})\sim\rho_{k,h-1}}\left[\|\phi_h(\tau_{h-1},\boldsymbol{a}_{h-1})\|_{\Sigma_{\rho_{k,h-1},\phi}^{-1}}\right]$$

$$\cdot \sqrt{n|A|\mathbb{E}_{(\tilde{\tau}_h,\tilde{\boldsymbol{a}}_h)\sim\rho_{k,h-1}}\left[\left(f_h^k(\tilde{\tau}_h,\tilde{\boldsymbol{a}}_h)\right)^2\right] + d\lambda}$$

$$\leqslant \sum_{k\in[n]}\sum_{h=0}^{H-1} \mathbb{E}_{(\tau_{h-1},\boldsymbol{a}_{h-1})\sim\rho_{k,h-1}}\left[\|\phi_h(\tau_{h-1},\boldsymbol{a}_{h-1})\|_{\Sigma_{\rho_{k,h-1},\phi}^{-1}}\right]\sqrt{n|A|\zeta_k + d\lambda}$$

$$\lesssim \sum_{k\in[n]}\sum_{h=0}^{H-1} \mathbb{E}_{(\tau_{h-1},\boldsymbol{a}_{h-1})\sim\rho_{k,h-1}}\left[\frac{\alpha_k}{H}\|\phi_h(\tau_{h-1},\boldsymbol{a}_{h-1})\|_{\Sigma_{\rho_{k,h-1},\phi}^{-1}}\right].$$

Then, by combining the above calculation of the term (a) and term (b) in inequality equation 11, we have:

$$\sum_{k\in[n]} \overline{v}_i^{(k)} - \underline{v}_i^{(k)} = \sum_{k\in[n]} \mathbb{E}_{s\sim\rho_{k,0}}\left[\overline{V}_{0,i}^{(k)}(s) - \underline{V}_{0,i}^{(k)}(s)\right]$$

$$\lesssim \sum_{k\in[n]}\sum_{h=1}^{H}\left(\mathbb{E}_{(\tau_{h-1},\boldsymbol{a}_{h-1})\sim\rho_{k,h-1}}\left[\|\phi_h(\tau_{h-1},\boldsymbol{a}_{h-1})\|_{\Sigma_{k,\hat{\rho}_{k,h},\phi}^{-1}}\right]\sqrt{dA\left(\alpha_k\right)^2 + H^2 d\lambda}\right)$$

$$+ H^2\sum_{k\in[n]}\sum_{h=0}^{H-1}\mathbb{E}_{(\tau_{h-1},\boldsymbol{a}_{h-1})\sim\rho_{k,h-1}}\left[\frac{\alpha_k}{H}\|\phi_h(\tau_{h-1},\boldsymbol{a}_{h-1})\|_{\Sigma_{\rho_{k,h-1},\phi}^{-1}}\right].$$

Note that

$$\sum_{n=1}^{N}\mathbb{E}_{(\tau_{h-1},\boldsymbol{a}_{h-1})\sim d_{\mathcal{P},h-1}^{\pi^n}}\left[\|\phi_h(\tau_{h-1},\boldsymbol{a}_{h-1})\|_{\Sigma_{n,\hat{\rho}_{n,h},\phi}^{-1}}\right]$$

$$\leqslant \sqrt{N\sum_{n=1}^{N}\mathbb{E}_{(\tau_{h-1},\boldsymbol{a}_{h-1})\sim d_{\mathcal{P},h-1}^{\pi^n}}\left[\phi_h(\tau_{h-1},\boldsymbol{a}_{h-1})^\top\Sigma_{n,\gamma_h^{(n)},\phi_h^\star}^{-1}\phi_h(\tau_{h-1},\boldsymbol{a}_{h-1})\right]} \quad \text{(CS inequality)}$$

$$\lesssim \sqrt{N\left(\log\det\left(\sum_{n=1}^{N}\mathbb{E}_{(\tau_{h-1},\boldsymbol{a}_{h-1})\sim d_{\mathcal{P},h-1}^{\pi^n}}[\phi_h(\tau_{h-1},\boldsymbol{a}_{h-1})\phi_h(\tau_{h-1},\boldsymbol{a}_{h-1})^\top]\right) - \log\det(\lambda I_d)\right)}$$

$$\text{(Lemma 53)}$$

$$\leqslant \sqrt{dN\log\left(1 + \frac{N}{d\lambda}\right)}.$$

(Potential function bound, Lemma 54 noting $\|\phi_h^\star(s,\boldsymbol{a})\|_2 \leqslant 1$ for any $(s,\boldsymbol{a})$.)

Taking maximum over $i$ on both sides, we get

$$\sum_{n=1}^{N}\Delta^{(n)} = \sum_{n=1}^{N}\max_{i\in[M]}\left\{\overline{v}_i^{(n)} - \underline{v}_i^{(n)}\right\}$$

$$\lesssim H\sqrt{dN\log\left(1 + \frac{N}{d\lambda}\right)}\sqrt{dA\left(\alpha_N\right)^2 + H^2 d\lambda}$$

$$+ H^3\left(\frac{1}{H}\alpha_N\sqrt{dN\log\left(1 + \frac{N}{d\lambda}\right)}\right)$$

$$\lesssim H^2 d\alpha_N\sqrt{NA\log(1 + \frac{N}{d\lambda})}$$

$$\lesssim H^3 d^2 N^{\frac{1}{2}} A\log(\frac{HN|\mathcal{M}|}{\delta})$$

$$\square$$

**Theorem 21** (PAC guarantee of OFOVI-MLE). *When OFOVI-MLE is applied with parameters* $\zeta_n = \Theta\left(\log(Hn|\mathcal{M}|/\delta)/n\right)$, $\lambda = \Theta(d\log(NH|\mathcal{M}|/\delta))$, $\hat{b}_{n,h} = \min\left\{\alpha_n\|\hat{\phi}_{n,h-1}(\tau_{h-1},\boldsymbol{a}_{h-1})\|_{L_2(\mu),\hat{\Sigma}_{n,h,\hat{\phi}_n}^{-1}}, H\right\}$ *and* $\alpha_n = \Theta(\sqrt{\lambda d + n|\mathcal{A}|\zeta_n})$ *by setting the number of episodes* $N = \tilde{\mathcal{O}}(H^6 d^4 |\mathcal{A}|^2 \epsilon^{-2}\log(Hd|\mathcal{A}||\mathcal{M}|/\delta\varepsilon))$ *with probability* $1 - \delta$, *the output policy* $\hat{\pi}$ *is an* $\varepsilon$-*approximate* $\{\text{NE}, \text{CCE}, \text{CE}\}$.

*Proof.* For any fixed episode $n$ and agent $i$, by Lemma 17, Lemma 18 and Lemma 19, we have

$$v_i^{\dagger,\pi_{-i}^n} - v_i^{\pi^n}\left(\text{or }\max_{\omega\in\Omega_i} v_i^{\omega\circ\pi^n} - v_i^{\pi^n}\right) \leqslant \overline{v}_i^n - \hat{\underline{v}}_i^n \leqslant \Delta^n.$$

Taking maximum over $i$ on both sides, we have

$$\max_{i \in [M]} \left\{ v_i^{\dagger, \pi_{-i}^n} - v_i^{\pi^n} \right\} \left( \text{or } \max_{i \in [M]} \left\{ \max_{\omega \in \Omega_i} v_i^{\omega \circ \pi^n} - v_i^{\pi^n} \right\} \right) \leqslant \Delta^n. \tag{12}$$

From Lemma 20, with probability $1 - \delta$, we can ensure

$$\sum_{k=1}^{N} \Delta^n \lesssim \mathcal{O}\left( H^3 d^2 N^{\frac{1}{2}} A \log(\frac{HN|\mathcal{M}|}{\delta}) \right)$$

Therefore, according to Lemma 54, when we pick $N$ to be

$$\tilde{\mathcal{O}}\left( \frac{H^6 d^4 A^2}{\varepsilon^2} \log \frac{HdA|\mathcal{M}|}{\delta \varepsilon} \right)$$

we have

$$\frac{1}{N} \sum_{n=1}^{N} \Delta^n \leqslant \varepsilon.$$

On the other hand, we have

$$\max_{i \in [M]} \left\{ v_i^{\dagger, \hat{\pi}_{-i}} - v_i^{\hat{\pi}} \right\} \left( \text{or } \max_{i \in [M]} \left\{ \max_{\omega \in \Omega_i} v_i^{\omega \circ \hat{\pi}} - v_i^{\hat{\pi}} \right\} \right)$$

$$= \max_{i \in [M]} \left\{ v_i^{\dagger, \pi_{-i}^{n^\star}} - v_i^{\pi^{n^\star}} \right\} \left( \text{or } \max_{i \in [M]} \left\{ \max_{\omega \in \Omega_i} v_i^{\omega \circ \pi^{n^\star}} - v_i^{\pi^{n^\star}} \right\} \right)$$

$$\leqslant \Delta^{n^\star} = \min_{n \in [N]} \Delta^n \leqslant \frac{1}{N} \sum_{n=1}^{N} \Delta^n \leqslant \varepsilon,$$

which has finished the proof. $\square$

### B.3 Proof of Sec. 4.2

We will provide the proof of Theorem 7 in this subsection. We first introduce the following additional assumptions on the representation and the reward.

**Assumption 1.** $\int_{\mathcal{T}} (\int_{\mathcal{A}} \|\phi(\tau, \boldsymbol{a})\|_2 d\boldsymbol{a})^2 d\tau \leqslant d$ for all $\phi \in \Phi$ and $\int_{\mathcal{T}} (\int_{\mathcal{A}} r(\tau, \boldsymbol{a}) d\boldsymbol{a})^2 d\tau \leqslant d$.

**Lemma 22** ($L_2$ norm of value function). $\forall i \in [N], h \in [H]$, for any policy $\pi$, we have that

$$\|V_{h,i}^\pi\|_2 \leqslant 2d + 2H^2 d^2 \lesssim H^2 d^2.$$

*Proof.* From the proper of low-rank POMG, we know that there exists $\omega^\pi$, such that $\|\omega^\pi\|_2 \leqslant \sqrt{d} H$ and $Q_{h,i}^\pi(\tau, \boldsymbol{a}) = \phi(\tau, \boldsymbol{a})^\top \omega^\pi$ for all $h \in [H], i \in [N]$. Then, we have

$\|V_{h,i}^\pi\|_2^2$

$= \int_{\mathcal{T}} V_{h,i}^\pi(\tau_h)^2 d\tau_h$

$= \int_{\mathcal{T}} \left( \int_{\mathcal{A}} \pi(\boldsymbol{a}_h|\tau_h) \left( r(\tau_h, \boldsymbol{a}_h) + \mathcal{P}(\tau_{h+1}|\tau_h, \boldsymbol{a}_h) \pi(\boldsymbol{a}_{h+1}|\tau_{h+1}) Q_{h+1,i}^\pi(\tau_{h+1}, \boldsymbol{a}_{h+1}) \right) d\boldsymbol{a}_h \right)^2 d\mathcal{T}_h$

$\leqslant \int_{\mathcal{T}} \left( \int_{\mathcal{A}} \left( r(\tau_h, \boldsymbol{a}_h) + \mathcal{P}(\tau_{h+1}|\tau_h, \boldsymbol{a}_h) \pi(\boldsymbol{a}_{h+1}|\tau_{h+1}) Q_{h+1,i}^\pi(\tau_{h+1}, \boldsymbol{a}_{h+1}) \right) d\boldsymbol{a}_h \right)^2 d\mathcal{T}_h$

$\leqslant 2 \int_{\mathcal{T}} \left( \int_{\mathcal{A}} r(\tau_h, \boldsymbol{a}_h) d\boldsymbol{a}_h \right)^2 d\mathcal{T}_h + 2H^2 d \int_{\mathcal{T}} \left( \int_{\mathcal{A}} \|\phi(\tau_h, \boldsymbol{a}_h)\|_2 d\boldsymbol{a}_h \right)^2 d\mathcal{T}_h$

$\leqslant 2d + 2H^2 d^2$

$\lesssim H^2 d^2$

$\square$

**Theorem 23** (PAC guarantee of OBOVI-SDR). *When OBOVI-SDR is applied with parameters $\zeta_n = \Theta\left(\log(Hn|\mathcal{M}|/\delta)/n\right)$, $\lambda = \Theta(d\log(NH|\mathcal{M}|/\delta))$, and $\alpha_n = \Theta(Hd\sqrt{\lambda d + n|\mathcal{A}|\zeta_n})$ by setting the number of episodes $N = \tilde{O}(\varepsilon^{-2}d^2\log(H|\mathcal{M}|/\delta\varepsilon))$ with probability $1 - \delta$, the output policy $\hat{\pi}$ is an $\varepsilon$-approximate $\{\text{NE}, \text{CCE}, \text{CE}\}$.*

*Proof.* Recall that the estimated transition satisfies

$$\mathbb{E}_{(x_h, \boldsymbol{a}_h) \sim \mathcal{D}_{h,n}} \left\| \mathbb{P}_h^{\mathcal{P}}(\cdot|x_h, \boldsymbol{a}_h) - \mathbb{P}_h^{\widehat{\mathcal{P}}_n}(\cdot|x_h, \boldsymbol{a}_h) \right\|_2^2 \leqslant \zeta_n.$$

Denote by $V_P^i(\pi)$ the value function of player $i$ under policy $\pi$ and transition $P$. Since the returned policy $\hat{\pi}$ is an equilibrium with respect to $\hat{\mathcal{P}}$, we have for all $i \in [N]$: $V_{\hat{\mathcal{P}}}^i(\hat{\pi}) = \max_{\tilde{\pi}^i} V_{\hat{\mathcal{P}}}^i(\tilde{\pi}^i, \hat{\pi}^{-i}) := V_{\hat{\mathcal{P}}}^{i,\dagger}(\hat{\pi}^i)$.

Note that

$$|V_{\hat{\mathcal{P}}}^{i,\dagger}(\hat{\pi}^i) - V_{\mathcal{P}}^{i,\dagger}(\hat{\pi}^i)| = |\max_{\tilde{\pi}^i} V_{\hat{\mathcal{P}}}^i(\tilde{\pi}^i, \pi^{-i}) - \max_{\tilde{\pi}^i} V_{\mathcal{P}}^i(\tilde{\pi}^i, \pi^{-i})|$$

$$\leqslant \max_{\tilde{\pi}^i} |V_{\hat{\mathcal{P}}}^i(\tilde{\pi}^i, \pi^{-i}) - V_{\mathcal{P}}^i(\tilde{\pi}^i, \pi^{-i})|$$

$$\leqslant d\sqrt{\zeta_n}$$

Thus, we have

$$V_{\mathcal{P}}^i(\hat{\pi}) \geqslant V_{\hat{\mathcal{P}}}^i(\hat{\pi}) - d\sqrt{\zeta_n}$$

$$= V_{\hat{\mathcal{P}}}^{i,\dagger}(\hat{\pi}^{-i}) - d\sqrt{\zeta_n}$$

$$\geqslant V_{\mathcal{P}}^{i,\dagger}(\hat{\pi}^{-i}) - 2d\sqrt{\zeta_n}$$

Hence, $\hat{\pi}$ is an $2d\sqrt{\zeta_n}$-approximate equilibrium.

To guarantee an $\varepsilon$-approximate equilibrium, we require $2d\sqrt{\zeta_n} \leqslant \varepsilon$, which leads to $N = \tilde{O}(\varepsilon^{-2}d^2\log(H|\mathcal{M}|/\delta\varepsilon))$. □

**Theorem 24** (PAC guarantee of OFOVI-SDR). *When OFOVI-SDR is applied with parameters $\zeta_n = \Theta\left(\log(Hn|\mathcal{M}|/\delta)/n\right)$, $\lambda = \Theta(d\log(NH|\mathcal{M}|/\delta))$, and $\alpha_n = \Theta(Hd\sqrt{\lambda d + n|\mathcal{A}|\zeta_n})$, by setting the number of episodes $N = \tilde{\mathcal{O}}(H^6 d^4 |\mathcal{A}|^2 \epsilon^{-2} \log(Hd|\mathcal{A}||\mathcal{M}|/\delta\varepsilon))$ with probability $1 - \delta$, the output policy $\hat{\pi}$ is an $\varepsilon$-approximate $\{\text{NE}, \text{CCE}, \text{CE}\}$.*

*Proof.* Similar to the proof of Theorem 21, with Lemma 22, we have that

$$\overline{v}_i^{(n)} - \underline{v}_i^{(n)} = \sum_{h=1}^{H} \left( \mathbb{E}_{(\tau_{h-1}, \boldsymbol{a}_{h-1}) \sim d_{\mathcal{P},h-1}^{\pi^n}} \left[ \|\phi_h(\tau_{h-1}, \boldsymbol{a}_{h-1})\|_{\Sigma_{n,\hat{\rho}_{n,h},\phi}^{-1}} \right] \sqrt{dA\left(\alpha_n\right)^2 + H^2 d\lambda} \right)$$

$$+ H^2 \sum_{h=0}^{H-1} \mathbb{E}_{(\tau_{h-1}, \boldsymbol{a}_{h-1}) \sim d_{\mathcal{P},h-1}^{\pi^n}} \left[ \frac{\alpha_n}{H} \|\phi_h(\tau_{h-1}, \boldsymbol{a}_{h-1})\|_{\Sigma_{\rho_{n,h-1},\phi}^{-1}} \right].$$

Taking maximum over $i$ and taking dominating term out, we have

$$\sum_{k=1}^{N} \Delta^n \lesssim \mathcal{O}\left( H^3 d^2 N^{\frac{1}{2}} A \log(\frac{HN|\mathcal{M}|}{\delta}) \right)$$

The remaining steps of the proof follow similarly to the proof of Theorem 21. □

## C BELIEF-BASED MG AND DERIVATION OF LLVR

### C.1 EQUIVALENT BELIEF-BASED MG CONSTRUCTION

We show that a POMG can be converted to an equivalent belief-based MG. Recall that the belief function is initialized as $f_{belief}(s_1|\boldsymbol{o}_0) = \mathbb{P}(s_0|\boldsymbol{o}_0)$, with recursive updates: $f_{belief}(s_{h+1}|\tau_{h+1}) \propto$

$\int_{\mathcal{S}} f_{belief}(s_h|\tau_h)P(s_{h+1}|s_h,\boldsymbol{a}_h)\mathbb{O}(\boldsymbol{o}_{h+1}|s_{h+1})\,ds_h$. This enables a transformation from a POMG to an equivalent MG over beliefs, denoted as $\mathcal{M}_b = (\mathcal{B}, \{\mathcal{A}_i\}_{i=1}^M, \{\mathcal{R}_{i,b}\}_{i=1}^M, H, \mu_b, \mathbb{P}_b)$, where $\mathcal{B} \subset \Delta(\mathcal{S})$ represents the set of possible beliefs, $\mu_b(\beta_1) = \int \mathbf{1}_{\beta_0 = f_{belief}(\cdot|\boldsymbol{o}_0)}\mu_0(s_1)\mathbb{O}(\boldsymbol{o}_0|s_0)\,ds_0$, and $\mathbb{P}_b(\beta_{h+1}|\beta_h,\boldsymbol{a}_h) = \int \mathbf{1}_{\beta_{h+1} = f_{belief}(\tau_h,\boldsymbol{a}_h,\boldsymbol{o}_{h+1})}\mathbb{P}(\boldsymbol{o}_{h+1}|\beta_h,\boldsymbol{a}_h)\,d\boldsymbol{o}_{h+1}$. Any joint policy $\pi(\cdot|\tau)$ of the original POMG uniquely maps to a belief-based policy $\pi_b(\cdot|f_{belief}(\tau))$ in the associated MG.

For a given belief-based policy $\pi_b$, the state value function $V_h^{\pi_b}(\beta_h)$ and state-action value function $Q_h^{\pi_b}(\beta_h,\boldsymbol{a}_h)$ for the belief Markov game can be defined as: $V_h^{\pi_b}(\beta_h) = \mathbb{E}\left[\sum_{t=h}^H r(\boldsymbol{o}_t,\boldsymbol{a}_t)|\beta_h\right]$, $\quad Q_h^{\pi_b}(\beta_h,\boldsymbol{a}_h) = \mathbb{E}\left[\sum_{t=h}^H r(\boldsymbol{o}_t,\boldsymbol{a}_t)|\beta_h,\boldsymbol{a}_h\right]$. Therefore, the Bellman equation can be expressed as

$$V_h^{\pi_b}(\beta_h) = \mathbb{E}_{\pi_b}\left[Q_h^{\pi_b}(\beta_h,\boldsymbol{a}_h)\right], \quad Q_h^{\pi_b}(\beta_h,\boldsymbol{a}_h) = r(\boldsymbol{o}_h,\boldsymbol{a}_h) + \mathbb{E}_{\mathbb{P}_b}\left[V_{h+1}^{\pi_b}(\beta_{h+1})\right]. \tag{13}$$

Note that the equivalent MG is based on beliefs, which are not directly observed. More importantly, these beliefs rely on the entire history, including all players' observations and actions. Consequently, the joint distribution is supported on a space with exponentially growing dimensionality. This exponential representation complexity leads to infeasible computational and statistical demands, highlighting the inherent limitations of directly applying MG-based RL algorithms to POMGs. Consequently, several special structures, such as $L$-decodability, have been investigated to reduce the statistical complexity of learning in a POMG, motivating our work.

## C.2 DERIVATION OF LLVR

We now derive LLVR that leverages the underlying structure of $L$-decodability to support exact and tractable linear representation of the value functions over the latent space in POMGs without full history dependence.

As mentioned above, though an equivalent belief-based MG can provide a Markovian Bellman recursion (cf. equation 13), operating within the belief space tends to be computationally challenging. We derive $L$-step Latent Variable Representation (LLVR) for $L$-decodable POMGs that leverages the underlying structure of $L$-decodability to remove the need for explicit belief calculations.

By Definition 9, an $L$-step memory state $\tau_h^L$ contains sufficient information. Therefore, we obtain the simplification $Q_h^{\pi_b}(f_{belief}(\tau_h),\boldsymbol{a}_h) = Q_h^{\pi_b}(p^*(\tau_h^L),\boldsymbol{a}_h)$. Since any belief-based policy $\pi_b$ has a corresponding joint policy $\pi$, we will henceforth make no distinction between them and uniformly denote both as $\pi$. To simplify notation, we redefine $Q_h^{\pi}(\tau_h^L,\boldsymbol{a_h}) = Q_h^{\pi}(p^*(\tau_h^L),\boldsymbol{a}_h)$, leading to the simplifie Bellman equation:

$$Q_h^{\pi}(\tau_h^L,\boldsymbol{a}_h) = r(\boldsymbol{o}_h,\boldsymbol{a}_h) + \mathbb{E}_{\mathbb{P}^{\pi}(\boldsymbol{o}_{h+1}|\tau_h^L,\boldsymbol{a}_h)}\left[V_{h+1}^{\pi}(\tau_{h+1}^L)\right], \tag{14}$$

where $\mathbb{P}^{\pi}(\cdot)$ denotes the probability distribution under policy $\pi$.

Note that in equation 14, there is an additional dependence of $V_{h+1}^{\pi}(\tau_{h+1}^L)$ on $(\tau_h^L,\boldsymbol{a}_h)$ since $\tau_{h+1}^L$ shares overlapping components with $(\tau_h^L,\boldsymbol{a}_h)$. Specifically, $\tau_{h+1}^L$ includes $(\boldsymbol{o}_{h-L+2},\boldsymbol{a}_{h-L+2},\cdots,\boldsymbol{o}_h,\boldsymbol{a}_h)$ from $(\tau_h^L,\boldsymbol{a}_h)$. Consequently, we turn to the following $L$-step Bellman equation to avoid this overlapping.

$$Q_h^{\pi}(\tau_h^L,\boldsymbol{a}_h) = \mathbb{E}_{\mathbb{P}^{\pi}(\boldsymbol{o}_{h+1:h+L-1}|\tau_h^L,\boldsymbol{a}_h)}\left[\left(\sum_{i=h}^{h+L-1} r(\boldsymbol{o}_i,\boldsymbol{a}_i)\right) + V_{h+L}^{\pi}(\tau_{h+L}^L)|\tau_h^L,\boldsymbol{a}_h\right]. \tag{15}$$

We note that by $L$-decodability, there exists a moment-matching policy $\chi_{\pi}$ for arbitrary policy $\pi$, which is conditioned on a latent variable to generate the same expected observation dynamics while being independent of history older than $L$ steps (14). We defer the detailed construction of $\chi_{\pi}$ to Appendix D for brevity. Using such a correspondent moment-matching policy $\chi_{\pi}$ of $\pi$, one can write

$$\mathbb{P}^{\pi}(\tau_{h+L}^L|\tau_h^L,\boldsymbol{a}_h) = \int p(z_{h+1}|\tau_h^L,\boldsymbol{a}_h)\mathbb{P}^{\chi_{\pi}}(\tau_{h+L}^L|z_{h+1})\,dz_{h+1} = \langle p(\cdot|\tau_h^L,\boldsymbol{a}_h), \mathbb{P}^{\chi_{\pi}}(\tau_{h+L}^L|\cdot)\rangle \tag{16}$$

where $z$ denotes the latent variable and the first equality follows from the construction of $\chi_{\pi}$.

Substituting equation 16 back into equation 15 enables a reformulation of $Q_h^\pi(\tau_h^L, \boldsymbol{a}_h)$ in linear form. Each reward and value term in equation 15 becomes an inner product of $p\left(z_{h+1} \mid \tau_h^L, \boldsymbol{a}_h\right)$ with the corresponding integrals. Specifically, for the first term in (15), for all $k \in \{0, \cdots, L-1\}$, we have

$$
\mathbb{E}_{\boldsymbol{o}_{h+k}|\tau_h^L, \boldsymbol{a}_h}^\pi[r(\boldsymbol{o}_{h+k}, \boldsymbol{a}_{h+k})] = \left\langle p(\cdot|\tau_h^L, \boldsymbol{a}_h), \underbrace{\int \mathbb{P}^{\chi_\pi}(\boldsymbol{o}_{h+k}, \boldsymbol{a}_{h+k}|\cdot) r(\boldsymbol{o}_{h+k}, \boldsymbol{a}_{h+k}) \, d\boldsymbol{o}_{h+k} d\boldsymbol{a}_{h+k}}_{\omega_{h+k}^\pi(\cdot)} \right\rangle
$$

Similarly, for the second term in (15), we have

$$
\mathbb{E}^\pi[V_{h+L}^\pi(\tau_{h+L}^L)] = \left\langle p(\cdot|\tau_h^L, \boldsymbol{a}_h), \underbrace{\int \mathbb{P}^{\chi_\pi}(\tau_{h+L}^L|\cdot) V(\tau_{h+L}^L) \, d\tau_{h+L}}_{\omega_{h+L}^\pi(\cdot)} \right\rangle \tag{17}
$$

Altogether, we conclude that in an $L$-decodable POMG, both the reward function $r$ and the value function $Q_h^\pi(\tau_h^L, \boldsymbol{a}_h)$ can be linearly represented with $p(z_{h+1}|\tau_h^L, \boldsymbol{a}_h)$. Specifically, defining $\tilde{\omega}_h^\pi(\cdot) = \sum_{k=0}^L \omega_{h+k}^\pi(\cdot)$, the Q-function can be represented as $Q_h^\pi(\tau_h^L, \boldsymbol{a}) = \langle p(\cdot|\tau_h^L, \boldsymbol{a}_h), \tilde{\omega}_h^\pi(\cdot) \rangle$. With this linear representation for $Q_h^\pi$, the backup step of the Bellman recursion can be replaced by a least squares regression in the space spanned by $p(\cdot|\tau_h^L, \boldsymbol{a})$. Specifically, at step $h$, the estimate of $Q_h^\pi(\tau_h, \boldsymbol{a}_h)$ can be obtained by the optimization:

$$
\min_{\tilde{\omega}_h^\pi} \mathbb{E}_{\tau_{h:h+L}^L, \boldsymbol{a}_h}^\pi \left[ \left( \langle p(\cdot|\tau_h^L, \boldsymbol{a}_h), \tilde{\omega}_h^\pi(\cdot) \rangle - \left( \left( \sum_{i=h}^{h+L-1} r(\boldsymbol{o}_i, \boldsymbol{a}_i) \right) + \langle p(\cdot|\tau_{h+L}^L, \boldsymbol{a}_{h+L}), \omega_{h+L}^\pi(\cdot) \rangle \right) \right)^2 \right].
$$

$$\tag{18}$$

Since $p(z_{h+1}|\tau_h^L, \boldsymbol{a}_h)$ is typically unknown a priori that must be learned from data, we can estimate it via MLE for conditional density estimation,

$$
\max_{p, \mathbb{P}^{\chi_\pi}} \quad \log \mathbb{P}^{\chi_\pi}(\boldsymbol{o}_{h+1:h+l}|\tau_h^L, \boldsymbol{a}_h) = \log \left\langle p(\cdot|\tau_h^L, \boldsymbol{a}_h), \mathbb{P}^{\chi_\pi}(\boldsymbol{o}_{h+1:h+l}|\cdot) \right\rangle
$$

Note that solving this MLE problem is generally intractable and the following evidence lower bound (ELBO) can be constructed as a tractable surrogate objective for MLE (47):

$$
\max_{q \in \Delta(\mathcal{Z})} \mathbb{E}_{q(\cdot|\tau_h^L, \boldsymbol{a}_h, \boldsymbol{o}_{h+1:h+l})}[\log \mathbb{P}^{\chi_\pi}(\boldsymbol{o}_{h+1:h+l}|z_h)] - KL(q(\cdot|\tau_h^L, \boldsymbol{a}_h, \boldsymbol{o}_{h+1:h+l})||p(z_h|\tau_h^L, \boldsymbol{a}_h)).
$$

$$\tag{19}$$

We provide the complete mathematical derivation of the ELBO in Appendix F. This derivation establishes a computational friendly variational ELBO and the methods for solving this ELBO have been extensively explored within the variational inference community (48) (see Appendix F for detailed analysis). We remark that under Assumption 4, the estimator obtained by maximizing the ELBO is identical to the estimator obtained by MLE and the ELBO can be efficiently optimized using variational inference techniques. We can parameterize the solution to the ELBO with a variational distribution class $\mathcal{Q} = \{\{q_h(z|\tau_h^L, \boldsymbol{a}_h, \boldsymbol{o}_{h+1:h+l})\}_{h\in[H]}\}$ and model class $\mathcal{M} = \{\{(p_h(z|\tau_h^L, \boldsymbol{a}_h), p_h(\boldsymbol{o}_{h+1:h+l}|z))\}_{h\in[H]}\}$. Practically, both $\mathcal{Q}$ and $\mathcal{M}$ can be implemented as neural networks, yielding approximate solutions $\hat{q}(z|\tau_h^L, \boldsymbol{a}_h, \boldsymbol{o}_{h+1:h+l})$, $\hat{p}_{h,n}(\boldsymbol{o}_{h+1:h+l}|z)$ and $\hat{p}_{n,h}(z_h|\tau_h^L, \boldsymbol{a}_h)$ and approximated transition $\hat{\mathcal{P}}_n = \{(\hat{p}_{h,n}(z_h|\tau_h^L, \boldsymbol{a}_h), \hat{p}_{h,n}(\boldsymbol{o}_{h+1}|z_h))\}_{h\in[H]}$.

Once $\hat{p}_{n,h}(z|\tau_h^L, \boldsymbol{a}_h)$ is obtained, the Q-function can be approximated as $Q_h^\pi(\tau_h^L, \boldsymbol{a}_h) = \langle \hat{p}(z|\tau_h^L, \boldsymbol{a}), \omega(z) \rangle$ and can be obtained by a least square regression (18). However, if $z$ is continuous, then $\omega(z)$ is infinite-dimensional. To deal with the infinite-dimensional $\omega(z)$, we follow the trick in (41) that forms $Q^\pi(\tau_h^L, \boldsymbol{a}_h)$ as an expectation $Q^\pi(\tau_h^L, \boldsymbol{a}_h) = \langle p(z|\tau_h^L, \boldsymbol{a}_h), w^\pi(z) \rangle = \mathbb{E}_{p(z|\tau_h^L, \boldsymbol{a}_h)}[w^\pi(z)]$ and then approximate it with random feature quadrature. Specifically, we consider $\omega(z)$ lying in certain RKHS with $\varphi$ as its random feature basis, i.e., $\omega(z) = \mathbb{E}_{P(\xi)}[\varphi(\xi, z)]$. As a result, $Q^\pi(\tau_h^L, \boldsymbol{a}_h) \approx \frac{1}{K} \sum_{i=1}^K \omega^\pi(\xi_i) \varphi(z_i, \xi_i)$ where the latent variables $z_i \sim \hat{p}(z|\tau_h^L, \boldsymbol{a}_h)$ and random features $\xi_i \sim P(\xi)$. If the random feature $\varphi$ is specified, then $\omega$ can be implemented by a neural network $\omega_\theta$. We defer the detailed derivation to Appendix E.1.

## D    MOMENT MATCHING POLICY

We provide a formal definition of the moment matching policy below.

**Definition 25** (Moment Matching Policy (14)). *With the $L$-decodability assumption, for $h \in [H]$, $h' \in [h - L + 1, h]$ and $l = h' - h + L - 1$, we can define the moment matching policy $\chi^{\pi,h} = \{\chi^{h,\pi}_{h'} : \mathcal{S}^l \times \mathcal{O}^l \times \mathcal{A}^{l-1} \to \Delta(\mathcal{A})\}^h_{h'=h-L+1}$ introduced by (14) , such that*

$$\chi^{\pi,h}_{h'}(\boldsymbol{a}_{h'}|(s_{h-L+1:h'}, \boldsymbol{o}_{h-L+1:h'}, \boldsymbol{a}_{h-L+1:h'-1}))$$
$$:= \mathbb{E}^{\mathcal{P}}_\pi[\pi_{h'}(\boldsymbol{a}_{h'}|x_{h'})|(s_{h-L+1:h'}, \boldsymbol{o}_{h-L+1:h'}, \boldsymbol{a}_{h-L+1:h'-1})], \quad \forall h' \leqslant h - 1,$$

*and $\chi^{\pi,h}_h = \pi_h$. We further define $\tilde{\pi}^h$, which takes first $h - L$ actions from $\pi$ and the remaining $L$ actions from $\chi^{\pi,h}$.*

The primary motivation for defining the moment matching policy is to construct a policy that is conditionally independent of the past history for theoretical analysis while remaining indistinguishable from the history-dependent policy to align with practical algorithms. By Lemma B.2 in (14), under the $L$-decodability assumption, for a fixed $h \in [H]$, we have $d^\pi_{P,h}(x_h) = d^{\tilde{\pi}_h}_{P,h}(x_h)$, for all $L$-step policy $\pi$ and $x_h \in \mathcal{X}_h$. As $\chi^{\pi,h}_h = \pi_h$, we have $d^\pi_{P,h}(x_h, a_h) = d^{\tilde{\pi}_h}_{P,h}(x_h, a_h)$. This enables the factorization in equation 17 without the dependency of the overlap observation trajectory.

## E    THEORETICAL ANALYSIS FOR METHODS FOR $L$-DECODABLE POMGS

This section presents the theoretical guarantees for our algorithms for $L$-decodable POMGs, including key technical background and assumptions, and proof for online and offline setting. For notational simplicity, we denote $x$ and $\mathcal{X}$ by $\tau^L$ and $\mathcal{T}^L$, respectively, in this section.

### E.1    TECHNICAL BACKGROUND ABOUT KERNEL METHOD

In this subsection, we revisit several important concepts from functional analysis that will be repeatedly used in our theoretical analysis. We start from the concept of the $L_2(\mu)$ space. For a complete introduction, we refer the reader to (41).

**Definition 26** ($L_2(\mu)$ space). *Let $(\mathcal{X}, \mathcal{A}, \mu)$ be a measure space. The $L_2(\mu)$ space is defined as the Hilbert space consists of square-integrable function with respect to $\mu$, with inner product*

$$\langle f, g \rangle_{L_2(\mu)} := \int_\mathcal{X} fg d\mu,$$

*and the norm*

$$\|f\|_{L_2(\mu)} := \left( \int_\mathcal{X} f^2 d\mu \right)^{1/2}.$$

*Throughout the paper, $\mu$ is specified as the Lebesgue measure for continuous $\mathcal{X}$ and the counting measure for discrete $\mathcal{X}$. Specifically, when $\mathcal{X}$ is discrete, we can represent $f$ as a sequence $[f(x)]_{x \in \mathcal{X}}$, and the corresponding $L_2(\mu)$ inner product and $L_2(\mu)$ norm is identical to the $\ell^2$ inner product and $\ell^2$ norm, which is defined as*

$$\langle f, g \rangle_{l^2} = \sum_{x \in \mathcal{X}} f(x)g(x), \quad \|f\|_{l^2} = \left( \sum_{x \in \mathcal{X}} f^2(x) \right)^{1/2},$$

*that is closely related to the inner product and norm of the Euclidean space.*

Then we introduce several concepts of the kernel and the reproducing kernel Hilbert space (RKHS).

**Definition 27** ((Positive-Definite) Kernel (21)). *A symmetric function $k : \mathcal{X} \times \mathcal{X} \to \mathbb{R}$ is said to be a positive definite kernel if for any $\{x_1, \ldots, x_m\} \subset \mathcal{X}$, the matrix $\mathbf{K} = [k(x_i, x_j)]_{ij} \in \mathbb{R}^{m \times m}$ is symmetric positive-definite.*

**Definition 28** (Reproducing Kernel Hilbert Space (RKHS) (49))**.** *Let $k : \mathcal{X} \times \mathcal{X} \to \mathbb{R}$ be a Positive-Definite kernel. Then, there exists a Hilbert space $\mathcal{H}_k$ and a mapping $\phi : \mathcal{X} \to \mathcal{H}_k$ such that:*

$$\forall x, x' \in \mathcal{X}, \quad k(x, x') = \langle \phi(x), \phi(x') \rangle.$$

*Furthermore, $\mathcal{H}_k$ has the following property known as the reproducing property:*

$$\forall h \in \mathcal{H}_k, \forall x \in \mathcal{X} \quad f(x) = \langle f, k(x, \cdot) \rangle.$$

*$\mathcal{H}_k$ is called a reproducing kernel Hilbert space (RKHS) associated to $k$.*

**Theorem 29** (Mercer's Theorem (50))**.** *Let $k(x, x')$ be a bounded continuous positive definite kernel. Then, $k(x, x')$ admits **Mercer decomposition**, i.e. there exists a countable orthonormal basis $\{e_i\}_{i=1}^{\infty}$ of $\mathcal{L}_2(\mu)$ with corresponding eigenvalues $\{\nu_i\}_{i=1}^{\infty}$, such that*

$$k(x, x') = \sum_{i=1}^{\infty} \nu_i e_i(x) e_i(x'), \tag{20}$$

*where the convergence is absolute and uniform for all $(x, x') \in \mathcal{X} \times \mathcal{X}$. Without loss of generality, we assume $\nu_1 \geqslant \nu_2 \geqslant \cdots > 0$.*

**Definition 30** (Random Feature)**.** *The kernel $k : \mathcal{X} \times \mathcal{X} \to \mathbb{R}$ has a random feature representation if there exists a function $\psi : \mathcal{X} \times \Xi \to \mathbb{R}$ and a probability measure $P$ over $\Xi$ such that*

$$k(x, x') = \int_{\Xi} \psi(x; \xi) \psi(x'; \xi) dP(\xi).$$

**Remark (random feature quadrature):** We here justify the random feature quadrature (41) for completeness.

We can represent $Q_h^\pi$ as an expectation,

$$Q_h^\pi(x_h, \boldsymbol{a}_h) = \langle p(z|x_h), w_h^\pi(z) \rangle = \mathbb{E}_{p(z|x_h)} [w_h^\pi(z)]_{L_2(\mu)}$$

Under the assumption that $w_h^\pi(\cdot) \in \mathcal{H}_k$, where $\mathcal{H}_k$ denoting some RKHS with some kernel $k(\cdot, \cdot)$. When $k(\cdot, \cdot)$ can be represented through random feature, *i.e.*,

$$k(x, y) = \mathbb{E}_{P(\xi)} [\psi(x; \xi) \psi(y; \xi)],$$

the $w_h^\pi(z)$ admits a representation as

$$w_h^\pi(z) = \mathbb{E}_{P(\xi)} [\tilde{w}_h^\pi(\xi) \psi(z; \xi)].$$

Therefore, we plug this random feature representation of $w_h^\pi(z)$ to $Q_h^\pi(x_h, a_h)$, we obtain

$$Q_h^\pi(x_h, \boldsymbol{a}_h) = \mathbb{E}_{p(z|x_h), P(\xi)} [\tilde{w}_h^\pi(\xi) \psi(z; \xi)]. \tag{21}$$

Applying Monte-Carlo approximation to equation 21, we obtain the random feature quadrature.

### E.2 TECHNICAL CONDITIONS

We adopt the following assumptions for the reproducing kernel, which have been previously used in (41; 15) in the single-agent setting.

**Assumption 2** (Regularity Conditions)**.** *Let $\mathcal{Z}$ be a compact metric space with respect to the Lebesgue measure $\nu$ when $\mathcal{Z}$ is continuous. Additionally, we assume that $\int_{\mathcal{Z}} k(z, z) d\nu \leqslant 1$.*

**Remark 31.** *Assumption 2 is mainly for the ease of presentation. The assumption $\int_{\mathcal{Z}} k(z, z) d\nu \leqslant 1$ can be relaxed to $\int_{\mathcal{Z}} k(z, z) d\nu \leqslant c$ with some positive constant $c$, at the cost of additional terms at most $poly(c)$ in the sample complexity.*

**Assumption 3** (Eigendecay Conditions)**.** *Assume that the sequence $\{\nu_i\}_{i \in I}$ defined in Theorem 29 satisfies one of the following conditions:*

- *$\beta$-finite spectrum: for some positive integer $\beta$, we have $\nu_i = 0$, $\forall i > \beta$.*
- *$\beta$-polynomial decay: $\nu_i \leqslant C_0 i^{-\beta}$ with absolute constant $C_0$ and $\beta > 1$.*
- *$\beta$-exponential decay: $\nu_i \leqslant C_1 \exp(-C_2 i^\beta)$, with absolute constants $C_1$, $C_2$ and $\beta > 0$.*

*We will use $C_{\text{poly}}$ to denote constants in the analysis of $\beta$-polynomial decay, which depend only on $C_0$ and $\beta$, and $C_{\text{exp}}$ to denote constants in the analysis of $\beta$-exponential decay, which depend only on $C_1$, $C_2$, and $\beta$. This simplifies the dependency on the constant terms. Both $C_{\text{poly}}$ and $C_{\text{exp}}$ may vary step by step.*

**Remark 32.** *Most existing kernels satisfy one of these eigendecay conditions. For example, the linear kernel and the polynomial kernel satisfy the $\beta$-finite spectrum condition, the Matern kernel satisfies the $\beta$-polynomial decay and the Gaussian kernel satisfies the $\beta$-exponential decay.*

### E.3 ALGORITHM AND GUARANTEE FOR LLVR WITH EXACT VALUE ORACLE

In this subsection, we provide PAC guarantee of OBOVI-LLVR.

**Theorem 33** (PAC guarantee of OBOVI-LLVR). *When OBOVI-LLVR is applied with parameters $\zeta_n = \Theta\left(\log(Hn|\mathcal{M}|/\delta)/n\right), \hat{b}_{n,h} = \min\left\{\alpha_n\|\hat{p}_n(\cdot|x_{h-L}, a_{h-L})\|_{L_2(\mu), \hat{\Sigma}_{n,h,\hat{p}_n}^{-1}}, H\right\}$ with $\alpha_n = \Theta\sqrt{\lambda C + nL|\mathcal{A}|^L \zeta_n}$ and*

- *$\beta$-finite spectrum: $\lambda = \Theta(\beta \log N + \log(N|\mathcal{M}|/\delta))$;*

- *$\beta$-polynomial decay: $\lambda = \Theta(C_{\text{poly}} N^{1/(1+\beta)} + \log(N|\mathcal{M}|/\delta))$;*

- *$\beta$-exponential decay: $\lambda = \Theta(C_{\text{exp}}(\log N)^{1/\beta} + \log(N|\mathcal{M}|/\delta))$;*

*by setting the number of episodes $N$ to be at $N = \tilde{O}(\varepsilon^{-2}\log(H|\mathcal{M}|/\delta\varepsilon))$, with probability $1 - \delta$, the output policy $\hat{\pi}$ is an $\varepsilon$-approximate $\{\text{NE}, \text{CCE}, \text{CE}\}$.*

*Proof.* According to Lemma 56, the estimated transition satisfies

$$\mathbb{E}_{(x_h, \boldsymbol{a}_h) \sim \mathcal{D}_{h,n}} \left\|\mathbb{P}_h^{\mathcal{P}}(\cdot|x_h, \boldsymbol{a}_h) - \mathbb{P}_h^{\widehat{\mathcal{P}}_n}(\cdot|x_h, \boldsymbol{a}_h)\right\|_1^2 \leqslant \zeta_n.$$

Denote by $V_P^i(\pi)$ the value function of player $i$ under policy $\pi$ and transition $P$. Since the returned policy $\hat{\pi}$ is an equilibrium with respect to $\hat{\mathcal{P}}$, we have for all $i \in [N]$: $V_{\hat{\mathcal{P}}}^i(\hat{\pi}) = \max_{\tilde{\pi}^i} V_{\hat{\mathcal{P}}}^i(\tilde{\pi}^i, \hat{\pi}^{-i}) := V_{\hat{\mathcal{P}}}^{i,\dagger}(\hat{\pi}^i)$.

Note that

$$|V_{\hat{\mathcal{P}}}^{i,\dagger}(\hat{\pi}^i) - V_{\mathcal{P}}^{i,\dagger}(\hat{\pi}^i)| = |\max_{\tilde{\pi}^i} V_{\hat{\mathcal{P}}}^i(\tilde{\pi}^i, \pi^{-i}) - \max_{\tilde{\pi}^i} V_{\mathcal{P}}^i(\tilde{\pi}^i, \pi^{-i})|$$
$$\leqslant \max_{\tilde{\pi}^i} |V_{\hat{\mathcal{P}}}^i(\tilde{\pi}^i, \pi^{-i}) - V_{\mathcal{P}}^i(\tilde{\pi}^i, \pi^{-i})|$$
$$\leqslant \sqrt{\zeta_n}$$

Thus, we have

$$V_{\mathcal{P}}^i(\hat{\pi}) \geqslant V_{\hat{\mathcal{P}}}^i(\hat{\pi}) - \sqrt{\zeta_n}$$
$$= V_{\hat{\mathcal{P}}}^{i,\dagger}(\hat{\pi}^{-i}) - \sqrt{\zeta_n}$$
$$\geqslant V_{\mathcal{P}}^{i,\dagger}(\hat{\pi}^{-i}) - 2\sqrt{\zeta_n}$$

Hence, $\hat{\pi}$ is an $2\sqrt{\zeta_n}$-approximate equilibrium.

To guarantee an $\varepsilon$-approximate equilibrium, we require $2\sqrt{\zeta_n} \leqslant \varepsilon$, which leads to $N = \tilde{O}(\varepsilon^{-2}\log(H|\mathcal{M}|/\delta\varepsilon))$. $\qquad\square$

### E.4 FORMAL PROOF FOR ONLINE SETTING

In this subsection, we provide analysis for OFOVI, establishing key technical lemmas that culminate in the convergence theorem. We start from the following assumptions, that are commonly used in the literature (15; 41; 37; 30).

**Assumption 4** (Realizability). *Assume $\{(p_h(z|\tau_h^L, \boldsymbol{a}_h), \mathbb{P}_h^\pi(\boldsymbol{o}_{h+1:h+l}|z))\}_{h \in [H]} \in \mathcal{M}$ and $p_h(z|\tau_h^L, \boldsymbol{a}_h, \boldsymbol{o}_{h+1:h+l}) \in \mathcal{Q}$ for all $(p_h(z|\tau_h^L, \boldsymbol{a}_h), p_h(\boldsymbol{o}_{h+1:h+l}|z)) \in \mathcal{M}$.*

**Assumption 5** (Normalization Conditions). $\forall \hat{\mathcal{P}} = \{(\hat{p}_h(z|\tau_h^L, \boldsymbol{a}_h), \hat{p}_h(\boldsymbol{o}_{h+1}|z))\}_{h \in [H]} \in \mathcal{M}$, $h \in [H]$, $(\tau_h^L, \boldsymbol{a}_h) \in \mathcal{T}^L \times \mathcal{A}$, $\|\hat{p}_h(\cdot|\tau_h^L, \boldsymbol{a}_h)\|_{\mathcal{H}_K} \leqslant 1$ *for some kernel K. Furthermore,* $\forall g : \mathcal{T}^L \to \mathbb{R}$ *such that* $\|g\|_{\infty} \leqslant 1$*, we have* $\left\| \int_{\mathcal{T}^L} p(\tau_{h+L}^L|\cdot)g(\tau_{h+L}^L)d\tau_{h+L}^L \right\|_{\mathcal{H}_K} \leqslant C$.

We remark that under Assumption 4, the estimator obtained by maximizing the ELBO is identical to the estimator obtained by MLE and the ELBO can be efficiently optimized using variational inference techniques. In Appendix F, we further explain why ELBO optimization offers superior computational tractability compared to direct MLE optimization.

**Lemma 34** (*L*-step back inequality for the true model). *Given a set of functions* $[g_h]_{h \in [H]}$, *where* $g_h : \mathcal{X} \times \mathcal{A} \to \mathbb{R}$, $\|g_h\|_{\infty} \leqslant B$, $\forall h \in [H]$, *we have that* $\forall \pi$,

$$\sum_{h \in [H]} \mathbb{E}_{(x_h, \boldsymbol{a}_h) \sim d_{\mathcal{P},h}^{\pi}}[g_h(x_h, \boldsymbol{a}_h)] \leqslant \sum_{h \in [H]} \mathbb{E}_{(x_{h-L}, \boldsymbol{a}_{h-L}) \sim d_{\mathcal{P},h-L}^{\pi}} \left[ \|p(\cdot|x_{h-L}, \boldsymbol{a}_{h-L})\|_{L_2(\mu), \Sigma_{\rho_{n,h-L},p}^{-1}} \right]$$

$$\cdot \sqrt{n|\mathcal{A}|^L \cdot \mathbb{E}_{(\tilde{x}_h, \tilde{\boldsymbol{a}}_h) \sim \rho_{n,h-L} \circ^L \mathcal{U}(\mathcal{A})}[g_h(\tilde{x}_h, \tilde{\boldsymbol{a}}_h)^2] + \lambda B^2 C}$$

*Proof.* The proof can be adapted from the proof of Lemma 6 in (41), and we include it for the completeness. Recall the moment matching policy $\chi_{\pi}$ Since $\chi_{\pi,h}$ does not depend on $(x_{h-L}, \boldsymbol{a}_{h-L})$, we can make the following decomposition:

$$\mathbb{E}_{(x_h, \boldsymbol{a}_h) \sim d_{P,h}^{\pi}} g_h(x_h, \boldsymbol{a}_h)$$

$$= \mathbb{E}_{(x_{h-L}, \boldsymbol{a}_{h-L}) \sim d_{P,h-L}^{\pi}} \left[ \int_{x_h} \langle p(\cdot|x_{h-L}, \boldsymbol{a}_{h-L}), \mathbb{P}^{\chi_{\pi}}(x_h|\cdot) \rangle_{L_2(\mu)} \cdot \mathbb{E}_{\boldsymbol{a}_h \sim \chi_{\pi,h}}[g_h(x_h, \boldsymbol{a}_h)] dx_h \right]$$

$$\leqslant \mathbb{E}_{(x_{h-L}, \boldsymbol{a}_{h-L}) \sim d_{P,h-L}^{\pi}} \|p(\cdot|x_{h-L}, \boldsymbol{a}_{h-L})\|_{L_2(\mu), \Sigma_{\rho_{n,h-L},p}^{-1}}$$

$$\cdot \left\| \int_{x_h} \mathbb{P}^{\chi_{\pi}}(x_h|\cdot) \mathbb{E}_{\boldsymbol{a}_h \sim \chi_{\pi,h}}[g_h(x_h, \boldsymbol{a}_h)] dx_h \right\|_{L_2(\mu), \Sigma_{\rho_{n,h-L},p}} .$$

Direct computation shows that

$$\left\| \int_{x_h} \mathbb{P}^{\chi_{\pi}}(x_h|\cdot) \mathbb{E}_{\boldsymbol{a}_h \sim \chi_{\pi,h}(\cdot|x_h)}[g_h(x_h, \boldsymbol{a}_h)] dx_h \right\|_{L_2(\mu), \Sigma_{\rho_{n,h-L},p}}^2$$

$$= n \mathbb{E}_{(\tilde{x}_{h-L}, \tilde{\boldsymbol{a}}_{h-L}) \sim \rho_{n,h-L}} \left[ \mathbb{E}_{x_h \sim \mathbb{P}^{\chi_{\pi}}(\cdot|x_{h-L}, \boldsymbol{a}_{h-L}), \boldsymbol{a}_h \sim \chi_{\pi,h}(\cdot|x_h)}[g_h(x_h, \boldsymbol{a}_h)] \right]^2$$

$$+ \lambda \left\| \int_{x_h} \mathbb{P}^{\chi_{\pi}}(x_h|\cdot) \cdot \mathbb{E}_{\boldsymbol{a}_h \sim \chi_{\pi,h}(\cdot|x_h)}[g_h(x_h, \boldsymbol{a}_h)] dx_h \right\|_{\mathcal{H}}^2$$

$$\leqslant n \mathbb{E}_{(\tilde{x}_{h-L}, \tilde{\boldsymbol{a}}_{h-L}) \sim \rho_{n,h-L}} \mathbb{E}_{x_h \sim \mathbb{P}^{\chi_{\pi}}(\cdot|x_{h-L}, \boldsymbol{a}_{h-L}), \boldsymbol{a}_h \sim \chi_{\pi,h}(\cdot|x_h)}[g_h(x_h, \boldsymbol{a}_h)]^2 + \lambda B^2 C$$

$$\leqslant n|\mathcal{A}|^L \mathbb{E}_{(\tilde{x}_h, \tilde{\boldsymbol{a}}_h) \sim \rho_{n,h-L} \circ^L \mathcal{U}(\mathcal{A})}[g_h(\tilde{x}_h, \tilde{\boldsymbol{a}}_h)]^2 + \lambda B^2 C,$$

which finishes the proof. □

**Lemma 35** (*L*-step back inequality for the learned model). *Assume we have a set of functions* $[g_h]_{h \in [H]}$, *where* $g_h : \mathcal{X} \times \mathcal{A} \to \mathbb{R}$, $\|g_h\|_{\infty} \leqslant B$, $\forall h \in [H]$. *Given Lemma 51, we have that* $\forall \pi$,

$$\sum_{h \in [H]} \mathbb{E}_{(x_h, \boldsymbol{a}_h) \sim d_{\hat{\mathcal{P}}_n,h}^{\pi}}[g_h(x_h, \boldsymbol{a}_h)]$$

$$\leqslant \sum_{h \in [H]} \mathbb{E}_{(x_{h-L}, \boldsymbol{a}_{h-L}) \sim d_{\hat{\mathcal{P}}_n,h-L}^{\pi}} \left[ \|\hat{p}(\cdot|x_{h-L}, \boldsymbol{a}_{h-L})\|_{L_2(\mu), \Sigma_{\rho_{n,h-2L} \circ^L \mathcal{U}(\mathcal{A}), \hat{p}}^{-1}} \right]$$

$$\cdot \sqrt{n|\mathcal{A}|^L \cdot \mathbb{E}_{(\tilde{x}_h, \tilde{\boldsymbol{a}}_h) \sim \rho_{n,h-2L} \circ^{2L} \mathcal{U}(\mathcal{A})}[g_h(\tilde{x}_h, \tilde{\boldsymbol{a}}_h)^2] + \lambda B^2 C + nL|\mathcal{A}|^{L-1} B^2 \zeta_n}$$

*Proof.* The proof can be adapted from the proof of Lemma 5 in (41), and we include it for the completeness. We define a similar moment matching policy and make the following decomposition:

$$\mathbb{E}_{(x_h, \boldsymbol{a}_h) \sim d^\pi_{\widehat{\mathcal{P}}_{n,h}}} [g_h(x_h, \boldsymbol{a}_h)]$$

$$= \mathbb{E}_{(x_{h-L}, \boldsymbol{a}_{h-L}) \sim d^\pi_{\widehat{\mathcal{P}}_{n,h-L}}} \left[ \int_{x_h} \langle \hat{p}(\cdot|x_{h-L}, \boldsymbol{a}_{h-L}), \hat{p}(x_h|\cdot) \rangle_{L_2(\mu)} \cdot \mathbb{E}_{\boldsymbol{a}_h \sim \chi_{\pi,h}} [g_h(x_h, \boldsymbol{a}_h)] dx_h \right]$$

$$\leqslant \mathbb{E}_{(x_{h-L}, \boldsymbol{a}_{h-L}) \sim d^\pi_{\widehat{\mathcal{P}}_{n,h-L}}} \| \hat{p}(\cdot|x_{h-L}, \boldsymbol{a}_{h-L}) \|_{L_2(\mu), \Sigma^{-1}_{\rho_{n,h-2L} \circ {}^L \mathcal{U}(\mathcal{A}), \hat{p}}}$$

$$\cdot \left\| \int_{x_h} \hat{p}(x_h|\cdot) \mathbb{E}_{\boldsymbol{a}_h \sim \chi_{\pi,h}(\cdot|x_h)} [g_h(x_h, \boldsymbol{a}_h)] dx_h \right\|_{L_2(\mu), \Sigma_{\rho_{n,h-2L} \circ {}^L \mathcal{U}(\mathcal{A}), \hat{p}}}.$$

Direct computation shows that

$$\left\| \int_{x_h} \hat{p}(x_h|\cdot) \mathbb{E}_{\boldsymbol{a}_h \sim \chi_{\pi,h}(\cdot|x_h)} [g_h(x_h, \boldsymbol{a}_h)] dx_h \right\|^2_{L_2(\mu), \Sigma_{\rho_{n,h-2L} \circ {}^L \mathcal{U}(\mathcal{A}), \hat{p}}}$$

$$= n \mathbb{E}_{(\tilde{x}_{h-L}, \tilde{\boldsymbol{a}}_{h-L}) \sim \rho_{n,h-2L} \circ {}^L \mathcal{U}(\mathcal{A})} \left[ \mathbb{E}_{x_h \sim \widehat{\mathcal{P}}_n(\cdot|\tilde{x}_{h-L}, \tilde{\boldsymbol{a}}_{h-L}), \boldsymbol{a}_h \sim \chi_{\pi,h}(\cdot|x_h)} [g_h(x_h, \boldsymbol{a}_h)] \right]^2$$

$$+ \lambda \left\| \int_{x_h} \hat{p}(x_h|\cdot) \mathbb{E}_{\boldsymbol{a}_h \sim \chi_{\pi,h}(\cdot|x_h)} [g_h(x_h, \boldsymbol{a}_h)] dx_h \right\|^2_{\mathcal{H}}$$

$$\leqslant n \mathbb{E}_{(\tilde{x}_{h-L}, \tilde{\boldsymbol{a}}_{h-L}) \sim \rho_{n,h-2L} \circ {}^L \mathcal{U}(\mathcal{A})} \mathbb{E}_{x_h \sim \widehat{\mathcal{P}}_n(\cdot|\tilde{x}_{h-L}, \tilde{\boldsymbol{a}}_{h-L}), \boldsymbol{a}_h \sim \chi_{\pi,h}(\cdot|x_h)} [g_h(x_h, \boldsymbol{a}_h)]^2 + \lambda B^2 C$$

$$\leqslant n |\mathcal{A}|^L \mathbb{E}_{(\tilde{x}_h, \tilde{\boldsymbol{a}}_h) \sim \rho_{n,h-2L} \circ {}^{2L} \mathcal{U}(\mathcal{A})} [g_h(\tilde{x}_h, \tilde{\boldsymbol{a}}_h)]^2 + n L |\mathcal{A}|^{L-1} B^2 \zeta_n + \lambda B^2 C,$$

where we use the MLE guarantee for each individual step to obtain the last inequality. This finishes the proof. $\square$

**Lemma 36** (Optimism for NE and CCE). *For episode* $n \in [N]$, *set*

$$\hat{b}_{n,h} = \min \left\{ \alpha_n \| \hat{p}_{n,h-1}(\cdot|x_{h-L}, \boldsymbol{a}_{h-L}) \|_{L_2(\mu), \hat{\Sigma}^{-1}_{n,h,\hat{p}_n}}, H \right\},$$

*with* $\alpha_n = \Theta \sqrt{\lambda C + n L |\mathcal{A}|^L \zeta_n}$,

$$\hat{\Sigma}_{n,h,\hat{p}_n} : L_2(\mu) \to L_2(\mu), \quad \hat{\Sigma}_{n,h,\hat{p}_n} := \sum_{(x_h, \boldsymbol{a}_h) \in \mathcal{D}_{n,h}} \left[ \hat{p}_n(z|x_h, \boldsymbol{a}_h) \hat{p}_n(z|x_h, \boldsymbol{a}_h)^\top \right] + \lambda T_K^{-1}$$

*where* $T_K$ *is the integral operator associated with* $K$ *(i.e.* $T_K f = \int f(x) K(x, \cdot) dx$*) and* $\lambda$ *is set for different eigendecay of* $K$ *as follows:*

- $\beta$-*finite spectrum:* $\lambda = \Theta(\beta \log N + \log(N|\mathcal{M}|/\delta))$;

- $\beta$-*polynomial decay:* $\lambda = \Theta(C_{\text{poly}} N^{1/(1+\beta)} + \log(N|\mathcal{M}|/\delta))$;

- $\beta$-*exponential decay:* $\lambda = \Theta(C_{\text{exp}} (\log N)^{1/\beta} + \log(N|\mathcal{M}|/\delta))$;

$c$ *is an absolute constant.* $\pi^n$ *is computed by solving NE or CCE. Then with probability at least* $1 - \delta$, $\forall n \in [N], i \in [M]$ *we have*

$$\overline{v}^n_i(x) - v^{\dagger, \pi^n_{-i}}_i(x) \geqslant 0.$$

*Proof.* Define $\tilde{\mu}^n_{h,i}(\cdot|x) := \arg\max_\mu \left( \mathbb{D}_{\mu, \pi^n_{h,-i}} Q^{\dagger, \pi^n_{-i}}_{h,i} \right)(x)$ as the best response policy for player $i$ at step $h$, and let $\tilde{\pi}^n_h = \tilde{\mu}^n_{h,i} \times \pi^n_{h,-i}$. Let $f^n_h(x, \boldsymbol{a}) = \left\| \widehat{\mathcal{P}}_{n,h}(\cdot|x, \boldsymbol{a}) - P_h(\cdot|x, \boldsymbol{a}) \right\|_1$, then according to lemma 51 and lemma 56, we have that using the chosen $\lambda$, with probability at least $1 - \delta$, $\forall n \in [N], h \in [H], \widehat{\mathcal{P}} \in \mathcal{M}$,

$$\mathbb{E}_{(x,\boldsymbol{a})\sim\hat{\rho}_{n,h}}\left[\left(f_h^n(x,\boldsymbol{a})\right)^2\right] \leqslant \zeta_n, \quad \mathbb{E}_{(x,\boldsymbol{a})\sim\tilde{\rho}_{n,h}}\left[\left(f_h^n(x,\boldsymbol{a})\right)^2\right] \leqslant \zeta_n,$$

$$\|p_h(\cdot|x_{h-L},\boldsymbol{a}_{h-L})\|_{\hat{\Sigma}_{n,h-L,\hat{p}_n}^{-1}} = \Theta\left(\|p_h(\cdot|x_{h-L},\boldsymbol{a}_{h-L})\|_{\hat{\Sigma}_{\rho_{n,h-L},\hat{p}_n}^{-1}}\right).$$

A direct conclusion is we can find an absolute constant $c$, such that

$$\hat{b}_{n,h}(x_h,\boldsymbol{a}_h) = \min\left\{\alpha_n \|\hat{p}_{n,h}(\cdot|x_{h-L},\boldsymbol{a}_{h-L})\|_{\Sigma_{n,h-L,\hat{p}_n}^{-1}}, H\right\}$$

$$\geqslant \min\left\{c\alpha_n \|\hat{p}_{n,h}(\cdot|x_{h-L},\boldsymbol{a}_{h-L})\|_{\Sigma_{n,h-L,\hat{p}_n}^{-1}}, H\right\}, \quad \forall n \in [N], h \in [H].$$

Next, we prove by induction that $\forall h \in [H]$,

$$\mathbb{E}_{x\sim d_{\tilde{\mathcal{P}}_{n,h}}^{\tilde{\pi}^n}}\left[\overline{V}_{h,i}^n(x) - V_{h,i}^{\dagger,\pi_{-i}^n}(x)\right] \geqslant \sum_{h'=h}^{H} \mathbb{E}_{(x,\boldsymbol{a})\sim d_{\tilde{\mathcal{P}}_{n,h'}}^{\tilde{\pi}^n}}\left[\hat{b}_{n,h'}(x,\boldsymbol{a}) - H\min\left\{f_{h'}^n(x,\boldsymbol{a}),1\right\}\right]. \tag{22}$$

First, notice that $\forall h \in [H]$,

$$\mathbb{E}_{x\sim d_{\tilde{\mathcal{P}}_{n,h}}^{\tilde{\pi}^n}}\left[\overline{V}_{h,i}^n(x) - V_{h,i}^{\dagger,\pi_{-i}^n}(x)\right] = \mathbb{E}_{x\sim d_{\tilde{\mathcal{P}}_{n,h}}^{\tilde{\pi}^n}}\left[\left(\mathbb{D}_{\pi_h^n}\overline{Q}_{h,i}^n\right)(x) - \left(\mathbb{D}_{\tilde{\pi}_h^n}Q_{h,i}^{\dagger,\pi_{-i}^n}\right)(x)\right]$$

$$\geqslant \mathbb{E}_{x\sim d_{\tilde{\mathcal{P}}_{n,h}}^{\tilde{\pi}^n}}\left[\left(\mathbb{D}_{\tilde{\pi}_h^n}\overline{Q}_{h,i}^n\right)(x) - \left(\mathbb{D}_{\tilde{\pi}_h^n}Q_{h,i}^{\dagger,\pi_{-i}^n}\right)(x)\right]$$

$$= \mathbb{E}_{(x,\boldsymbol{a})\sim d_{\tilde{\mathcal{P}}_{n,h}}^{\tilde{\pi}^n}}\left[\overline{Q}_{h,i}^n(x,\boldsymbol{a}) - Q_{h,i}^{\dagger,\pi_{-i}^n}(x,\boldsymbol{a})\right],$$

where the inequality uses the fact that $\pi_h^n$ is the NE (or CCE) solution for $\left\{\overline{Q}_{h,i}^n\right\}_{i=1}^{M}$. Now we are ready to prove equation 22:

- When $h = H$, we have

$$\mathbb{E}_{x\sim d_{\tilde{\mathcal{P}}_{n,H}}^{\tilde{\pi}^n}}\left[\overline{V}_{H,i}^n(x) - V_{H,i}^{\dagger,\pi_{-i}^n}(x)\right] \geqslant \mathbb{E}_{(x,\boldsymbol{a})\sim d_{\tilde{\mathcal{P}}_{n,H}}^{\tilde{\pi}^n}}\left[\overline{Q}_{H,i}^n(x,\boldsymbol{a}) - Q_{H,i}^{\dagger,\pi_{-i}^n}(x,\boldsymbol{a})\right]$$

$$= \mathbb{E}_{(x,\boldsymbol{a})\sim d_{\tilde{\mathcal{P}}_{n,H}}^{\tilde{\pi}^n}}\left[\hat{b}_{n,H}(x,\boldsymbol{a})\right]$$

$$\geqslant \mathbb{E}_{(x,\boldsymbol{a})\sim d_{\tilde{\mathcal{P}}_{n,H}}^{\tilde{\pi}^n}}\left[\hat{b}_{n,H}(x,\boldsymbol{a}) - H\min\left\{f_H^n(x,\boldsymbol{a}),1\right\}\right].$$

- Suppose the statement is true for step $h+1$, then for step $h$, we have

$$\mathbb{E}_{x\sim d^{\tilde{\pi}^n}_{\widehat{\mathcal{P}}_{n,h}}}\left[\overline{V}^n_{h,i}(x)-V^{\dagger,\pi^n_{-i}}_{h,i}(x)\right]$$

$$\geqslant\mathbb{E}_{(x,\boldsymbol{a})\sim d^{\tilde{\pi}^n}_{\widehat{\mathcal{P}}_{n,h}}}\left[\overline{Q}^n_{h,i}(x,\boldsymbol{a})-Q^{\dagger,\pi^n_{-i}}_{h,i}(x,\boldsymbol{a})\right]$$

$$=\mathbb{E}_{(x,\boldsymbol{a})\sim d^{\tilde{\pi}^n}_{\widehat{\mathcal{P}}_{n,h}}}\left[\hat{b}_{n,h}(x,\boldsymbol{a})+\left(\widehat{\mathcal{P}}_{n,h}\overline{V}^n_{h+1,i}\right)(x,\boldsymbol{a})-\left(P_hV^{\dagger,\pi^n_{-i}}_{h+1,i}\right)(x,\boldsymbol{a})\right]$$

$$=\mathbb{E}_{(x,\boldsymbol{a})\sim d^{\tilde{\pi}^n}_{\widehat{\mathcal{P}}_{n,h}}}\left[\hat{b}_{n,h}(x,\boldsymbol{a})\right.$$
$$\left.+\left(\widehat{\mathcal{P}}_{n,h}\left(\overline{V}^n_{h+1,i}-V^{\dagger,\pi^n_{-i}}_{h+1,i}\right)\right)(x,\boldsymbol{a})+\left(\left(\widehat{\mathcal{P}}_{n,h}-P_h\right)V^{\dagger,\pi^n_{-i}}_{h+1,i}\right)(x,\boldsymbol{a})\right]$$

$$=\mathbb{E}_{(x,\boldsymbol{a})\sim d^{\tilde{\pi}^n}_{\widehat{\mathcal{P}}_{n,h}}}\left[\hat{b}_{n,h}(x,\boldsymbol{a})+\left(\left(\widehat{\mathcal{P}}_{n,h}-P_h\right)V^{\dagger,\pi^n_{-i}}_{h+1,i}\right)(x,\boldsymbol{a})\right]$$
$$+\mathbb{E}_{x\sim d^{\tilde{\pi}^n}_{\widehat{\mathcal{P}}_{n,h+1}}}\left[\overline{V}^n_{h+1,i}(x)-V^{\dagger,\pi^n_{-i}}_{h+1,i}(x)\right]$$

$$\geqslant\mathbb{E}_{(x,\boldsymbol{a})\sim d^{\tilde{\pi}^n}_{\widehat{\mathcal{P}}_{n,h}}}\left[\hat{b}_{n,h}(x,\boldsymbol{a})-H\min\left\{f^n_h(x,\boldsymbol{a}),1\right\}\right]$$
$$+\mathbb{E}_{x\sim d^{\tilde{\pi}^n}_{\widehat{\mathcal{P}}_{n,h+1}}}\left[\overline{V}^n_{h+1,i}(x)-V^{\dagger,\pi^n_{-i}}_{h+1,i}(x)\right]$$

$$\geqslant\sum_{h'=h}^{H}\mathbb{E}_{(x,\boldsymbol{a})\sim d^{\tilde{\pi}^n}_{\widehat{\mathcal{P}}_{n,h'}}}\left[\hat{b}_{n,h'}(x,\boldsymbol{a})-H\min\left\{f^n_{h'}(x,\boldsymbol{a}),1\right\}\right],$$

where we use the fact

$$\left|\left(\widehat{\mathcal{P}}_{n,h}-P_h\right)V^{\dagger,\pi^n_{-i}}_{h+1,i}\right|(x,\boldsymbol{a})\leqslant\min\left\{H,\left\|\widehat{\mathcal{P}}_{n,h}(\cdot|x,\boldsymbol{a})-P_h(\cdot|x,\boldsymbol{a})\right\|_1\left\|V^{\dagger,\pi^n_{-i}}_{h+1,i}\right\|_\infty\right\}$$

$$\leqslant H\min\left\{1,\left\|\widehat{\mathcal{P}}_{n,h}(\cdot|x,\boldsymbol{a})-P_h(\cdot|x,\boldsymbol{a})\right\|_1\right\}$$

$$=H\min\left\{1,f^n_{h'}(x,\boldsymbol{a})\right\}$$

and the last row uses the induction assumption.

Therefore, we have proved equation 22. We then apply $h=0$ to equation 22, and get

$$\mathbb{E}_{x\sim d_0}\left[\overline{V}^n_{0,i}(x)-V^{\dagger,\pi^n_{-i}}_{0,i}(x)\right]$$

$$=\mathbb{E}_{x\sim d^{\tilde{\pi}^n}_{\widehat{\mathcal{P}}_{n,0}}}\left[\overline{V}^n_{0,i}(x)-V^{\dagger,\pi^n_{-i}}_{0,i}(x)\right]$$

$$\geqslant\sum_{h=0}^{H}\mathbb{E}_{(x,\boldsymbol{a})\sim d^{\tilde{\pi}^n}_{\widehat{\mathcal{P}}_{n,h}}}\left[\hat{b}_{n,h}(x,\boldsymbol{a})-H\min\left\{f^n_h(x,\boldsymbol{a}),1\right\}\right]$$

$$=\sum_{h=0}^{H}\mathbb{E}_{(x,\boldsymbol{a})\sim d^{\tilde{\pi}^n}_{\widehat{\mathcal{P}}_{n,h}}}\left[\hat{b}_{n,h}(x,\boldsymbol{a})\right]-H\sum_{h=0}^{H}\mathbb{E}_{(x,\boldsymbol{a})\sim d^{\tilde{\pi}^n}_{\widehat{\mathcal{P}}_{n,h}}}\left[\min\left\{f^n_h(x,\boldsymbol{a}),1\right\}\right].$$

Next we are going to bound the second term. Applying Lemma 35 to $g_h(x,\boldsymbol{a})=\min\left\{f^n_h(x,\boldsymbol{a}),1\right\}$, we have

$$\sum_{h=0}^{H}\mathbb{E}_{(x,\boldsymbol{a})\sim d^{\tilde{\pi}^n}_{\widehat{\mathcal{P}}_{n,h}}}\left[\min\left\{f^n_h(x,\boldsymbol{a}),1\right\}\right]$$

$$\leqslant\sum_{h=0}^{H}\mathbb{E}_{(x_{h-L},\boldsymbol{a}_{h-L})\sim d^{\tilde{\pi}^n}_{\widehat{\mathcal{P}}_{n,h-L}}}\left[\left\|\hat{p}_{n,h-1}(\cdot|x_{h-L},\boldsymbol{a}_{h-L})\right\|_{\Sigma^{-1}_{\rho_{n,h-2L}\circ^L\mathcal{U}(\mathcal{A}),\hat{p}}}\right]$$
$$\cdot\sqrt{n|\mathcal{A}|^L\cdot\mathbb{E}_{(\tilde{x}_h,\tilde{\boldsymbol{a}}_h)\sim\rho_{n,h-2L}\circ^{2L}\mathcal{U}(\mathcal{A})}\left[\min\left\{f^n_h(\tilde{x}_h,\tilde{\boldsymbol{a}}_h),1\right\}^2\right]+\lambda C}+nL|\mathcal{A}|^{L-1}\zeta_n$$

$$\leqslant\sum_{h=0}^{H}\mathbb{E}_{(x_{h-L},\boldsymbol{a}_{h-L})\sim d^{\tilde{\pi}^n}_{\widehat{\mathcal{P}}_{n,h-L}}}\left[\left\|\alpha_n\hat{p}_{n,h-1}(\cdot|x_{h-L},\boldsymbol{a}_{h-L})\right\|_{\Sigma^{-1}_{\rho_{n,h-2L}\circ^L\mathcal{U}(\mathcal{A}),\hat{p}}}\right]$$

Note that we here use the fact $\min\{f_h^n(x,\boldsymbol{a}),1\} \leqslant 1$, $\mathbb{E}_{(\tilde{x}_h,\tilde{\boldsymbol{a}}_h)\sim\rho_{n,h-2L}\circ^{2L}\mathcal{U}(\mathcal{A})}\left[\min\{f_h^n(\tilde{x}_h,\tilde{\boldsymbol{a}}_h),1\}^2\right] \leqslant$ $\zeta_n$ and our choice of $\alpha_n$.

Combining all things together,

$$
\begin{aligned}
\overline{v}_i^n - v_i^{\dagger,\pi_{-i}^n} &= \mathbb{E}_{x\sim d_0}\left[\overline{V}_{0,i}^n(x) - V_{0,i}^{\dagger,\pi_{-i}^n}(x)\right] \\
&\geqslant \sum_{h=1}^{H} \mathbb{E}_{(x,\boldsymbol{a})\sim d_{\hat{\mathcal{P}}_{n,h}}^{\tilde{\pi}^n}}\left[\hat{b}_{n,h}(x,\boldsymbol{a})\right] - H\sum_{h=1}^{H}\mathbb{E}_{(x,\boldsymbol{a})\sim d_{\hat{\mathcal{P}}_{n,h}}^{\tilde{\pi}^n}}\left[\min\{f_h^n(x,\boldsymbol{a}),1\}\right] \\
&\geqslant 0,
\end{aligned}
$$

which proves the inequality. $\qquad\square$

**Lemma 37** (Optimism for CE). *For episode $n \in [N]$, set*

$$
\hat{b}_{n,h} = \min\left\{\alpha_n\|\hat{p}_{n,h-1}(\cdot|x_{h-L},\boldsymbol{a}_{h-L})\|_{L_2(\mu),\hat{\Sigma}_{n,h,\hat{p}_n}^{-1}}, H\right\},
$$

*with $\alpha_n = \Theta\sqrt{\lambda C + nL|\mathcal{A}|^L\zeta_n}$,*

$$
\hat{\Sigma}_{n,h,\hat{p}_n} : L_2(\mu) \to L_2(\mu), \quad \hat{\Sigma}_{n,h,\hat{p}_n} := \sum_{(x_h,\boldsymbol{a}_h)\in\mathcal{D}_{n,h}}\left[\hat{p}_n(z|x_h,\boldsymbol{a}_h)\hat{p}_n(z|x_h,\boldsymbol{a}_h)^\top\right] + \lambda T_K^{-1}
$$

*where $T_K$ is the integral operator associated with $K$ (i.e. $T_K f = \int f(x)K(x,\cdot)dx$) and $\lambda$ is set for different eigendecay of $K$ as follows:*

- *$\beta$-finite spectrum: $\lambda = \Theta(\beta\log N + \log(N|\mathcal{M}|/\delta))$*

- *$\beta$-polynomial decay: $\lambda = \Theta(C_{\text{poly}}N^{1/(1+\beta)} + \log(N|\mathcal{M}|/\delta))$;*

- *$\beta$-exponential decay: $\lambda = \Theta(C_{\exp}(\log N)^{1/\beta} + \log(N|\mathcal{M}|/\delta))$;*

*$c$ is an absolute constant. $\pi^n$ is computed by solving CE. Then with probability at least $1 - \delta$, $\forall n \in [N], i \in [M]$ we have*

$$
\overline{v}_i^n(x) - \max_{\omega\in\Omega_i} v_i^{\omega\circ\pi^n}(x) \geqslant 0, \quad \forall n \in [N], i \in [M].
$$

*Proof.* Denote $\tilde{\omega}_{h,i}^{(n)} = \arg\max_{\omega_h\in\Omega_{h,i}}\left(\mathbb{D}_{\omega_h\circ\pi_h^{(n)}}\max_{\omega\in\Omega_i}Q_{h,i}^{\omega\circ\pi^{(n)}}\right)(s)$ and let $\tilde{\pi}_h^{(n)} = \tilde{\omega}_{h,i}^{(n)}\circ\pi_h^{(n)}$. Let $f_h^n(x,\boldsymbol{a}) = \left\|\widehat{\mathcal{P}}_{n,h}(\cdot|x,\boldsymbol{a}) - P_h(\cdot|x,\boldsymbol{a})\right\|_1$, then according to lemma 51 and lemma 56, we have that using the chosen $\lambda$, with probability at least $1 - \delta$, $\forall n \in [N], h \in [H], \widehat{\mathcal{P}} \in \mathcal{M}$,

$$
\mathbb{E}_{(x,\boldsymbol{a})\sim\hat{\rho}_{n,h}}\left[(f_h^n(x,\boldsymbol{a}))^2\right] \leqslant \zeta_n, \quad \mathbb{E}_{(x,\boldsymbol{a})\sim\tilde{\rho}_{n,h}}\left[(f_h^n(x,\boldsymbol{a}))^2\right] \leqslant \zeta_n,
$$

$$
\|p_h(\cdot|x_{h-L},\boldsymbol{a}_{h-L})\|_{\hat{\Sigma}_{n,h-L,\hat{p}_n}^{-1}} = \Theta\left(\|p_h(\cdot|x_{h-L},\boldsymbol{a}_{h-L})\|_{\hat{\Sigma}_{\rho_{n,h-L},\hat{p}_n}^{-1}}\right).
$$

A direct conclusion is we can find an absolute constant $c$, such that

$$
\begin{aligned}
\hat{b}_{n,h}(x_h,\boldsymbol{a}_h) &= \min\left\{\alpha_n\|\hat{p}_{n,h}(\cdot|x_{h-L},\boldsymbol{a}_{h-L})\|_{\Sigma_{n,h-L,\hat{p}_n}^{-1}}, H\right\} \\
&\geqslant \min\left\{c\alpha_n\|\hat{p}_{n,h}(\cdot|x_{h-L},\boldsymbol{a}_{h-L})\|_{\Sigma_{n,h-L,\hat{p}_n}^{-1}}, H\right\}, \quad \forall n \in [N], h \in [H].
\end{aligned}
$$

Next, we prove by induction that $\forall h \in [H]$,

$$
\mathbb{E}_{x\sim d_{\hat{\mathcal{P}}_{n,h}}^{\tilde{\pi}^n}}\left[\overline{V}_{h,i}^n(x) - \max_{\omega\in\Omega_i}V_{h,i}^{\omega\circ\pi^n}(x)\right] \geqslant \sum_{h'=h}^{H}\mathbb{E}_{(x,\boldsymbol{a})\sim d_{\hat{\mathcal{P}}_{n,h'}}^{\tilde{\pi}^n}}\left[\hat{b}_{n,h'}(x,\boldsymbol{a}) - H\min\{f_{h'}^n(x,\boldsymbol{a}),1\}\right].
\tag{23}
$$

First, notice that $\forall h \in [H]$,

$$
\begin{aligned}
\mathbb{E}_{x \sim d_{\widehat{\mathcal{P}}_{n,h}}^{\tilde{\pi}^n}} \left[ \overline{V}_{h,i}^n(x) - \max_{\omega \in \Omega_i} V_{h,i}^{\omega \circ \pi^n}(x) \right] &= \mathbb{E}_{x \sim d_{\widehat{\mathcal{P}}_{n,h}}^{\tilde{\pi}^n}} \left[ \left( \mathbb{D}_{\pi_h^n} \overline{Q}_{h,i}^n \right)(x) - \left( \mathbb{D}_{\tilde{\pi}_h^n} \max_{\omega \in \Omega_i} Q_{h,i}^{\omega \circ \pi^n} \right)(x) \right] \\
&\geqslant \mathbb{E}_{x \sim d_{\widehat{\mathcal{P}}_{n,h}}^{\tilde{\pi}^n}} \left[ \left( \mathbb{D}_{\tilde{\pi}_h^n} \overline{Q}_{h,i}^n \right)(x) - \left( \mathbb{D}_{\tilde{\pi}_h^n} \max_{\omega \in \Omega_i} Q_{h,i}^{\omega \circ \pi^n} \right)(x) \right] \\
&= \mathbb{E}_{(x,\boldsymbol{a}) \sim d_{\widehat{\mathcal{P}}_{n,h}}^{\tilde{\pi}^n}} \left[ \overline{Q}_{h,i}^n(x,\boldsymbol{a}) - \max_{\omega \in \Omega_i} Q_{h,i}^{\omega \circ \pi^n}(x,\boldsymbol{a}) \right],
\end{aligned}
$$

where the inequality uses the fact that $\pi_h^n$ is the CE solution for $\left\{ \overline{Q}_{h,i}^n \right\}_{i=1}^M$. Now we are ready to prove equation 23:

- When $h = H$, we have

$$
\begin{aligned}
&\mathbb{E}_{x \sim d_{\widehat{\mathcal{P}}_{n,H}}^{\tilde{\pi}^n}} \left[ \overline{V}_{H,i}^n(x) - \max_{\omega \in \Omega_i} V_{H,i}^{\omega \circ \pi^n}(x) \right] \\
&\geqslant \mathbb{E}_{(x,\boldsymbol{a}) \sim d_{\widehat{\mathcal{P}}_{n,H}}^{\tilde{\pi}^n}} \left[ \overline{Q}_{H,i}^n(x,\boldsymbol{a}) - \max_{\omega \in \Omega_i} Q_{H,i}^{\omega \circ \pi^n}(x,\boldsymbol{a}) \right] \\
&= \mathbb{E}_{(x,\boldsymbol{a}) \sim d_{\widehat{\mathcal{P}}_{n,H}}^{\tilde{\pi}^n}} \left[ \hat{b}_{n,H}(x,\boldsymbol{a}) \right] \\
&\geqslant \mathbb{E}_{(x,\boldsymbol{a}) \sim d_{\widehat{\mathcal{P}}_{n,H}}^{\tilde{\pi}^n}} \left[ \hat{b}_{n,H}(x,\boldsymbol{a}) - H \min \left\{ f_H^n(x,\boldsymbol{a}), 1 \right\} \right].
\end{aligned}
$$

- Suppose the statement is true for step $h + 1$, then for step $h$, we have

$$
\begin{aligned}
&\mathbb{E}_{x \sim d_{\widehat{\mathcal{P}}_{n,h}}^{\tilde{\pi}^n}} \left[ \overline{V}_{h,i}^n(x) - \max_{\omega \in \Omega_i} V_{h,i}^{\omega \circ \pi^n}(x) \right] \\
&\geqslant \mathbb{E}_{(x,\boldsymbol{a}) \sim d_{\widehat{\mathcal{P}}_{n,h}}^{\tilde{\pi}^n}} \left[ \overline{Q}_{h,i}^n(x,\boldsymbol{a}) - \max_{\omega \in \Omega_i} Q_{h,i}^{\omega \circ \pi^n}(x,\boldsymbol{a}) \right] \\
&= \mathbb{E}_{(x,\boldsymbol{a}) \sim d_{\widehat{\mathcal{P}}_{n,h}}^{\tilde{\pi}^n}} \left[ \hat{b}_{n,h}(x,\boldsymbol{a}) + \left( \widehat{\mathcal{P}}_{n,h} \overline{V}_{h+1,i}^n \right)(x,\boldsymbol{a}) - \left( \mathcal{P}_h \max_{\omega \in \Omega_i} V_{h+1,i}^{\omega \circ \pi^n} \right)(x,\boldsymbol{a}) \right] \\
&= \mathbb{E}_{(x,\boldsymbol{a}) \sim d_{\widehat{\mathcal{P}}_{n,h}}^{\tilde{\pi}^n}} \left[ \hat{b}_{n,h}(x,\boldsymbol{a}) \right. \\
&\qquad \left. + \left( \widehat{\mathcal{P}}_{n,h} \left( \overline{V}_{h+1,i}^n - \max_{\omega \in \Omega_i} V_{h+1,i}^{\omega \circ \pi^n} \right) \right)(x,\boldsymbol{a}) + \left( \left( \widehat{\mathcal{P}}_{n,h} - \mathcal{P}_h \right) \max_{\omega \in \Omega_i} V_{h+1,i}^{\omega \circ \pi^n} \right)(x,\boldsymbol{a}) \right] \\
&= \mathbb{E}_{(x,\boldsymbol{a}) \sim d_{\widehat{\mathcal{P}}_{n,h}}^{\tilde{\pi}^n}} \left[ \hat{b}_{n,h}(x,\boldsymbol{a}) + \left( \left( \widehat{\mathcal{P}}_{n,h} - \mathcal{P}_h \right) \max_{\omega \in \Omega_i} V_{h+1,i}^{\omega \circ \pi^n} \right)(x,\boldsymbol{a}) \right] \\
&\qquad + \mathbb{E}_{x \sim d_{\widehat{\mathcal{P}}_{n,h+1}}^{\tilde{\pi}^n}} \left[ \overline{V}_{h+1,i}^n(x) - \max_{\omega \in \Omega_i} V_{h+1,i}^{\omega \circ \pi^n}(x) \right] \\
&\geqslant \mathbb{E}_{(x,\boldsymbol{a}) \sim d_{\widehat{\mathcal{P}}_{n,h}}^{\tilde{\pi}^n}} \left[ \hat{b}_{n,h}(x,\boldsymbol{a}) - H \min \left\{ f_h^n(x,\boldsymbol{a}), 1 \right\} \right] \\
&\qquad + \mathbb{E}_{x \sim d_{\widehat{\mathcal{P}}_{n,h+1}}^{\tilde{\pi}^n}} \left[ \overline{V}_{h+1,i}^n(x) - \max_{\omega \in \Omega_i} V_{h+1,i}^{\omega \circ \pi^n}(x) \right] \\
&\geqslant \sum_{h'=h}^H \mathbb{E}_{(x,\boldsymbol{a}) \sim d_{\widehat{\mathcal{P}}_{n,h'}}^{\tilde{\pi}^n}} \left[ \hat{b}_{n,h'}(x,\boldsymbol{a}) - H \min \left\{ f_{h'}^n(x,\boldsymbol{a}), 1 \right\} \right],
\end{aligned}
$$

where we use the fact

$$\left|\left(\widehat{\mathcal{P}}_{n,h} - \mathcal{P}_h\right)\max_{\omega \in \Omega_i} V_{h+1,i}^{\omega \circ \pi^n}\right|(x, \boldsymbol{a})$$

$$\leqslant \min\left\{H, \left\|\widehat{\mathcal{P}}_{n,h}(\cdot|x, \boldsymbol{a}) - \mathcal{P}_h(\cdot|x, \boldsymbol{a})\right\|_1 \left\|\max_{\omega \in \Omega_i} V_{h+1,i}^{\omega \circ \pi^n}\right\|_\infty\right\}$$

$$\leqslant H \min\left\{1, \left\|\widehat{\mathcal{P}}_{n,h}(\cdot|x, \boldsymbol{a}) - \mathcal{P}_h(\cdot|x, \boldsymbol{a})\right\|_1\right\}$$

$$= H \min\left\{1, f_{h'}^n(x, \boldsymbol{a})\right\}$$

and the last row uses the induction assumption.

Therefore, we have proved equation 23. We then apply $h = 0$ to equation 23, and get

$$\mathbb{E}_{x \sim d_0}\left[\overline{V}_{0,i}^n(x) - \max_{\omega \in \Omega_i} V_{0,i}^{\omega \circ \pi^n}(x)\right]$$

$$= \mathbb{E}_{x \sim d_{\widehat{\mathcal{P}}_{n,0}}^{\tilde{\pi}^n}}\left[\overline{V}_{0,i}^n(x) - \max_{\omega \in \Omega_i} V_{0,i}^{\omega \circ \pi^n}(x)\right]$$

$$\geqslant \sum_{h=0}^{H} \mathbb{E}_{(x, \boldsymbol{a}) \sim d_{\widehat{\mathcal{P}}_{n,h}}^{\tilde{\pi}^n}}\left[\hat{b}_{n,h}(x, \boldsymbol{a}) - H \min\left\{f_h^n(x, \boldsymbol{a}), 1\right\}\right]$$

$$= \sum_{h=0}^{H} \mathbb{E}_{(x, \boldsymbol{a}) \sim d_{\widehat{\mathcal{P}}_{n,h}}^{\tilde{\pi}^n}}\left[\hat{b}_{n,h}(x, \boldsymbol{a})\right] - H \sum_{h=0}^{H} \mathbb{E}_{(x, \boldsymbol{a}) \sim d_{\widehat{\mathcal{P}}_{n,h}}^{\tilde{\pi}^n}}\left[\min\left\{f_h^n(x, \boldsymbol{a}), 1\right\}\right].$$

Next we are going to bound the second term. Applying Lemma 35 to $g_h(x, \boldsymbol{a}) = \min\left\{f_h^n(x, \boldsymbol{a}), 1\right\}$, we have

$$\sum_{h=0}^{H} \mathbb{E}_{(s, \boldsymbol{a}) \sim d_{\widehat{\mathcal{P}}_{n,h}}^{\tilde{\pi}^n}}\left[\min\left\{f_h^n(s, \boldsymbol{a}), 1\right\}\right]$$

$$\leqslant \sum_{h=0}^{H} \mathbb{E}_{(x_{h-L}, \boldsymbol{a}_{h-L}) \sim d_{\widehat{\mathcal{P}}_{n,h-L}}^{\tilde{\pi}^n}}\left[\|\hat{p}_{n,h-1}(\cdot|x_{h-L}, \boldsymbol{a}_{h-L})\|_{\Sigma_{\rho_{n,h-2L} \circ^L \mathcal{U}(\mathcal{A}), \hat{p}}^{-1}}\right]$$

$$\cdot \sqrt{n|\mathcal{A}|^L \cdot \mathbb{E}_{(\tilde{x}_h, \tilde{\boldsymbol{a}}_h) \sim \rho_{n,h-2L} \circ^{2L} \mathcal{U}(\mathcal{A})}\left[\min\left\{f_h^n(\tilde{x}_h, \tilde{\boldsymbol{a}}_h), 1\right\}^2\right] + \lambda C + nL|\mathcal{A}|^{L-1}\zeta_n}$$

$$\leqslant \sum_{h=0}^{H} \mathbb{E}_{(x_{h-L}, \boldsymbol{a}_{h-L}) \sim d_{\widehat{\mathcal{P}}_{n,h-L}}^{\tilde{\pi}^n}}\left[\|\alpha_n \hat{p}_{n,h-1}(\cdot|x_{h-L}, \boldsymbol{a}_{h-L})\|_{\Sigma_{\rho_{n,h-2L} \circ^L \mathcal{U}(\mathcal{A}), \hat{p}}^{-1}}\right]$$

Note that we here use the fact $\min\left\{f_h^n(x, \boldsymbol{a}), 1\right\} \leqslant 1$, $\mathbb{E}_{(\tilde{x}_h, \tilde{\boldsymbol{a}}_h) \sim \rho_{n,h-2L} \circ^{2L} \mathcal{U}(\mathcal{A})}\left[\min\left\{f_h^n(\tilde{x}_h, \tilde{\boldsymbol{a}}_h), 1\right\}^2\right] \leqslant \zeta_n$ and our choice of $\alpha_n$.

Combining all things together,

$$\overline{v}_i^n(x) - \max_{\omega \in \Omega_i} v_i^{\omega \circ \pi^n}(x) = \mathbb{E}_{x \sim d_0}\left[\overline{V}_{0,i}^n(x) - \max_{\omega \in \Omega_i} V_{0,i}^{\omega \circ \pi^n}(x)\right]$$

$$\geqslant \sum_{h=0}^{H} \mathbb{E}_{(x, \boldsymbol{a}) \sim d_{\widehat{\mathcal{P}}_{n,h}}^{\tilde{\pi}^n}}\left[\hat{b}_{n,h}(x, \boldsymbol{a})\right] - H \sum_{h=0}^{H} \mathbb{E}_{(x, \boldsymbol{a}) \sim d_{\widehat{\mathcal{P}}_{n,h}}^{\tilde{\pi}^n}}\left[\min\left\{f_h^n(x, \boldsymbol{a}), 1\right\}\right]$$

$$\geqslant 0,$$

which proves the inequality. $\qquad\square$

**Lemma 38** (Pessimism). *For episode $n \in [N]$, set*

$$\hat{b}_{n,h} = \min\left\{\alpha_n \|\hat{p}_{n,h-1}(\cdot|x_{h-L}, \boldsymbol{a}_{h-L})\|_{L_2(\mu), \hat{\Sigma}_{n,h,\hat{p}_n}^{-1}}, H\right\},$$

with $\alpha_n = \Theta\sqrt{\lambda C + nL|\mathcal{A}|^L \zeta_n}$,

$$\hat{\Sigma}_{n,h,\hat{p}_n} : L_2(\mu) \to L_2(\mu), \quad \hat{\Sigma}_{n,h,\hat{p}_n} := \sum_{(x_h, \boldsymbol{a}_h) \in \mathcal{D}_{n,h}} \left[\hat{p}_n(z|x_h, \boldsymbol{a}_{h,i})\hat{p}_n(z|x_h, \boldsymbol{a}_{h,i})^\top\right] + \lambda T_K^{-1}$$

where $T_K$ is the integral operator associated with $K$ (i.e. $T_K f = \int f(x)K(x, \cdot)dx$) and $\lambda$ is set for different eigendecay of $K$ as follows:

- $\beta$-*finite spectrum:* $\lambda = \Theta(\beta \log N + \log(N|\mathcal{M}|/\delta))$

- $\beta$-*polynomial decay:* $\lambda = \Theta(C_{\mathrm{poly}} N^{1/(1+\beta)} + \log(N|\mathcal{M}|/\delta))$;

- $\beta$-*exponential decay:* $\lambda = \Theta(C_{\exp}(\log N)^{1/\beta} + \log(N|\mathcal{M}|/\delta))$;

$c$ *is an absolute constant. Then with probability at least* $1 - \delta$, $\forall n \in [N], i \in [M]$ *we have*

$$\underline{v}_i^n(x) - v_i^{\pi^n}(x) \leqslant 0, \quad \forall n \in [N], i \in [M].$$

*Proof.* Let $f_h^n(x, \boldsymbol{a}) = \left\|\widehat{\mathcal{P}}_{n,h}(\cdot|x, \boldsymbol{a}) - \mathcal{P}_h(\cdot|x, \boldsymbol{a})\right\|_1$, then according to lemma 51 and lemma 56, we have that using the chosen $\lambda$, with probability at least $1 - \delta$,

$$\mathbb{E}_{(x,\boldsymbol{a}) \sim \hat{\rho}_{n,h}}\left[(f_h^n(x, \boldsymbol{a}))^2\right] \leqslant \zeta_n, \quad \mathbb{E}_{(x,\boldsymbol{a}) \sim \tilde{\rho}_{n,h}}\left[(f_h^n(x, \boldsymbol{a}))^2\right] \leqslant \zeta_n, \quad \forall n \in [N], h \in [H],$$

$$\|p_h(\cdot|x, \boldsymbol{a})\|_{\hat{\Sigma}_{n,h-L,\hat{p}_n}^{-1}} = \Theta\left(\|p_h(\cdot|x, \boldsymbol{a})\|_{\hat{\Sigma}_{\rho_{n,h-L},\hat{p}_n}^{-1}}\right), \quad \forall n \in [N], h \in [H], \widehat{\mathcal{P}} \in \mathcal{M}.$$

A direct conclusion is we can find an absolute constant $c$, such that

$$\hat{b}_{n,h}(x_h, \boldsymbol{a}_h) = \min\left\{\alpha_n \|\hat{p}_{n,h-1}(\cdot|x_{h-L}, \boldsymbol{a}_{h-L})\|_{\Sigma_{n,h-L,\hat{p}_n}^{-1}}, H\right\}$$

$$\geqslant \min\left\{c\alpha_n \|\hat{p}_{n,h-1}(\cdot|x_{h-L}, \boldsymbol{a}_{h-L})\|_{\Sigma_{n,h-L,\hat{p}_n}^{-1}}, H\right\}, \quad \forall n \in [N], h \in [H].$$

Again, we prove by induction that $\forall h \in [H]$,

$$\mathbb{E}_{x \sim d_{\widehat{\mathcal{P}}_{n,h}}^{\tilde{\pi}^n}}\left[\underline{V}_{h,i}^n(x) - V_{h,i}^{\pi^n}(x)\right] \leqslant \sum_{h'=h}^H \mathbb{E}_{(x,\boldsymbol{a}) \sim d_{\widehat{\mathcal{P}}_{n,h}'}^{\tilde{\pi}^n}}\left[-\hat{b}_{n,h'}(x, \boldsymbol{a}) + H \min\{f_{h'}^n(x, \boldsymbol{a}), 1\}\right].$$

(24)

- When $h = H$, we have

$$\mathbb{E}_{x \sim d_{\widehat{\mathcal{P}}_{n,H}}^{\tilde{\pi}^n}}\left[\underline{V}_{H,i}^n(x) - V_{H,i}^{\pi^n}(x)\right] = \mathbb{E}_{(x,\boldsymbol{a}) \sim d_{\widehat{\mathcal{P}}_{n,H}}^{\tilde{\pi}^n}}\left[\underline{Q}_{H,i}^n(x, \boldsymbol{a}) - Q_{H,i}^{\pi^n}(x, \boldsymbol{a})\right]$$

$$= \mathbb{E}_{(x,\boldsymbol{a}) \sim d_{\widehat{\mathcal{P}}_{n,H}}^{\tilde{\pi}^n}}\left[-\hat{b}_{n,h}(x, \boldsymbol{a})\right]$$

$$\leqslant \mathbb{E}_{(x,\boldsymbol{a}) \sim d_{\widehat{\mathcal{P}}_{n,H}}^{\tilde{\pi}^n}}\left[-\hat{b}_{n,h}(x, \boldsymbol{a}) + H \min\{f_H^n(x, \boldsymbol{a}), 1\}\right].$$

- Suppose the statement is true for step $h+1$, then for step $h$, we have

$$\mathbb{E}_{x \sim d^{\tilde{\pi}^n}_{\widehat{\mathcal{P}}_{n,h}}} \left[ \underline{V}^n_{h,i}(x) - V^{\pi^n}_{h,i}(x) \right]$$

$$= \mathbb{E}_{(x,\boldsymbol{a}) \sim d^{\tilde{\pi}^n}_{\widehat{\mathcal{P}}_{n,h}}} \left[ \underline{Q}^n_{h,i}(x,\boldsymbol{a}) - Q^{\pi^n}_{h,i}(x,\boldsymbol{a}) \right]$$

$$= \mathbb{E}_{(x,\boldsymbol{a}) \sim d^{\tilde{\pi}^n}_{\widehat{\mathcal{P}}_{n,h}}} \left[ -\hat{b}_{n,h}(x,\boldsymbol{a}) + \left( \widehat{\mathcal{P}}_{n,h} \underline{V}^n_{h+1,i} \right)(x,\boldsymbol{a}) - \left( \mathcal{P}_h V^{\pi^n}_{h+1,i} \right)(x,\boldsymbol{a}) \right]$$

$$= \mathbb{E}_{(x,\boldsymbol{a}) \sim d^{\tilde{\pi}^n}_{\widehat{\mathcal{P}}_{n,h}}} \left[ -\hat{b}_{n,h}(x,\boldsymbol{a}) \right.$$

$$\left. + \left( \widehat{\mathcal{P}}_{n,h} \left( \underline{V}^n_{h+1,i} - V^{\pi^n}_{h+1,i} \right) \right)(x,\boldsymbol{a}) + \left( \left( \widehat{\mathcal{P}}_{n,h} - \mathcal{P}_h \right) V^{\pi^n}_{h+1,i} \right)(x,\boldsymbol{a}) \right]$$

$$= \mathbb{E}_{(x,\boldsymbol{a}) \sim d^{\tilde{\pi}^n}_{\widehat{\mathcal{P}}_{n,h}}} \left[ -\hat{b}_{n,h}(x,\boldsymbol{a}) + \left( \left( \widehat{\mathcal{P}}_{n,h} - \mathcal{P}_h \right) V^{\pi^n}_{h+1,i} \right)(x,\boldsymbol{a}) \right]$$

$$+ \mathbb{E}_{x \sim d^{\tilde{\pi}^n}_{\widehat{\mathcal{P}}_{n,h+1}}} \left[ \underline{V}^n_{h+1,i}(x) - V^{\pi^n}_{h+1,i}(x) \right]$$

$$\geqslant \mathbb{E}_{(x,\boldsymbol{a}) \sim d^{\tilde{\pi}^n}_{\widehat{\mathcal{P}}_{n,h}}} \left[ -\hat{b}_{n,h}(x,\boldsymbol{a}) - H \min \left\{ f^n_h(x,\boldsymbol{a}), 1 \right\} \right]$$

$$+ \mathbb{E}_{x \sim d^{\tilde{\pi}^n}_{\widehat{\mathcal{P}}_{n,h+1}}} \left[ \underline{V}^n_{h+1,i}(x) - V^{\pi^n}_{h+1,i}(x) \right]$$

$$\geqslant \sum_{h'=h}^{H} \mathbb{E}_{(x,\boldsymbol{a}) \sim d^{\tilde{\pi}^n}_{\widehat{\mathcal{P}}_{n,h'}}} \left[ -\hat{b}_{n,h'}(x,\boldsymbol{a}) + H \min \left\{ f^n_{h'}(x,\boldsymbol{a}), 1 \right\} \right],$$

where we use the fact

$$\left| \left( \widehat{\mathcal{P}}_{n,h} - \mathcal{P}_h \right) V^{\pi^n}_{h+1,i} \right|(x,\boldsymbol{a}) \leqslant \min \left\{ H, \left\| \widehat{\mathcal{P}}_{n,h}(\cdot|x,\boldsymbol{a}) - \mathcal{P}_h(\cdot|x,\boldsymbol{a}) \right\|_1 \left\| V^{\pi^n}_{h+1,i} \right\|_\infty \right\}$$

$$\leqslant H \min \left\{ 1, \left\| \widehat{\mathcal{P}}_{n,h}(\cdot|x,\boldsymbol{a}) - \mathcal{P}_h(\cdot|x,\boldsymbol{a}) \right\|_1 \right\}$$

$$= H \min \left\{ 1, f^n_{h'}(x,\boldsymbol{a}) \right\}$$

and the last row uses the induction assumption.

Therefore, we have proved equation 24. We then apply $h = 0$ to equation 24, and get

$$\mathbb{E}_{x \sim d_0} \left[ \overline{V}^n_{0,i}(x) - V^{\dagger,\pi^n_{-i}}_{0,i}(x) \right]$$

$$= \mathbb{E}_{x \sim d^{\tilde{\pi}^n}_{\widehat{\mathcal{P}}_{n,0}}} \left[ \overline{V}^n_{0,i}(x) - V^{\dagger,\pi^n_{-i}}_{0,i}(x) \right]$$

$$\geqslant \sum_{h=0}^{H} \mathbb{E}_{(x,\boldsymbol{a}) \sim d^{\tilde{\pi}^n}_{\widehat{\mathcal{P}}_{n,h}}} \left[ \hat{b}_{n,h}(x,\boldsymbol{a}) - H \min \left\{ f^n_h(x,\boldsymbol{a}), 1 \right\} \right]$$

$$= \sum_{h=0}^{H} \mathbb{E}_{(x,\boldsymbol{a}) \sim d^{\tilde{\pi}^n}_{\widehat{\mathcal{P}}_{n,h}}} \left[ \hat{b}_{n,h}(x,\boldsymbol{a}) \right] - H \sum_{h=0}^{H} \mathbb{E}_{(x,\boldsymbol{a}) \sim d^{\tilde{\pi}^n}_{\widehat{\mathcal{P}}_{n,h}}} \left[ \min \left\{ f^n_h(x,\boldsymbol{a}), 1 \right\} \right].$$

Next we are going to bound the second term. Applying Lemma 35 to $g_h(x,\boldsymbol{a}) = \min \left\{ f^n_h(x,\boldsymbol{a}), 1 \right\}$, we have

$$\sum_{h=0}^{H} \mathbb{E}_{(x,\boldsymbol{a}) \sim d^{\tilde{\pi}^n}_{\widehat{\mathcal{P}}_{n,h}}} \left[ \min \left\{ f^n_h(x,\boldsymbol{a}), 1 \right\} \right]$$

$$\leqslant \sum_{h=0}^{H} \mathbb{E}_{(x_{h-L},\boldsymbol{a}_{h-L}) \sim d^{\tilde{\pi}^n}_{\widehat{\mathcal{P}}_{n,h-L}}} \left[ \left\| \hat{p}_{n,h-1}(\cdot|x_{h-L},\boldsymbol{a}_{h-L}) \right\|_{\Sigma^{-1}_{\rho_{n,h-2L} \circ {}^L \mathcal{U}(\mathcal{A}), \hat{p}}} \right]$$

$$\cdot \sqrt{n|\mathcal{A}|^L \cdot \mathbb{E}_{(\tilde{x}_h,\tilde{\boldsymbol{a}}_h) \sim \rho_{n,h-2L} \circ {}^{2L} \mathcal{U}(\mathcal{A})} \left[ \min \left\{ f^n_h(\tilde{x}_h,\tilde{\boldsymbol{a}}_h), 1 \right\}^2 \right] + \lambda C + nL|\mathcal{A}|^{L-1} \zeta_n}$$

$$\leqslant \sum_{h=0}^{H} \mathbb{E}_{(x_{h-L},\boldsymbol{a}_{h-L}) \sim d^{\tilde{\pi}^n}_{\widehat{\mathcal{P}}_{n,h-L}}} \left[ \left\| \alpha_n \hat{p}_{n,h-1}(\cdot|x_{h-L},\boldsymbol{a}_{h-L}) \right\|_{\Sigma^{-1}_{\rho_{n,h-2L} \circ {}^L \mathcal{U}(\mathcal{A}), \hat{p}}} \right]$$

Note that we here use the fact $\min\{f_h^n(x,\boldsymbol{a}),1\} \leqslant 1$, $\mathbb{E}_{(\tilde{x}_h,\tilde{\boldsymbol{a}}_h)\sim\rho_{n,h-2L}\circ^{2L}\mathcal{U}(\mathcal{A})}\left[\min\{f_h^n(\tilde{x}_h,\tilde{\boldsymbol{a}}_h),1\}^2\right] \leqslant$ $\zeta_n$ and our choice of $\alpha_n$.

Combining all things together,

$$
\begin{aligned}
\underline{v}_i^n - v_i^{\pi^n} =& \mathbb{E}_{x\sim d_0}\left[\underline{V}_{0,i}^n(x) - V_{0,i}^{\pi^n}(x)\right] \\
\leqslant& \sum_{h=0}^H \mathbb{E}_{(x,\boldsymbol{a})\sim d_{\hat{\mathcal{P}}_{n,h}}^{\tilde{\pi}^n}}\left[-\hat{b}_{n,h}(x,\boldsymbol{a})\right] + H\sum_{h=0}^H \mathbb{E}_{(x,\boldsymbol{a})\sim d_{\hat{\mathcal{P}}_{n,h}}^{\tilde{\pi}^n}}\left[\min\{f_h^n(x,\boldsymbol{a}),1\}\right] \\
\leqslant& 0,
\end{aligned}
$$

which has finished the proof. $\qquad\square$

**Lemma 39.** *For episode $n \in [N]$, set*

$$
\hat{b}_{n,h} = \min\left\{\alpha_n\|\hat{p}_{n,h-1}(\cdot|x_{h-L},\boldsymbol{a}_{h-L})\|_{L_2(\mu),\hat{\Sigma}_{n,h,\hat{p}_n}^{-1}},H\right\},
$$

*with $\alpha_n = \Theta\sqrt{\lambda C + nL|\mathcal{A}|^L\zeta_n}$,*

$$
\hat{\Sigma}_{n,h,\hat{p}_n}:L_2(\mu)\to L_2(\mu),\quad \hat{\Sigma}_{n,h,\hat{p}_n} := \sum_{(x_h,\boldsymbol{a}_h)\in\mathcal{D}_{n,h}}\left[\hat{p}_{n,h}(z|x_h,\boldsymbol{a}_h)\hat{p}_{n,h}(z|x_h,\boldsymbol{a}_h)^\top\right] + \lambda T_K^{-1}
$$

*where $T_K$ is the integral operator associated with $K$ (i.e. $T_K f = \int f(x)K(x,\cdot)dx$) and $\lambda$ is set for different eigendecay of $K$ as follows:*

- *$\beta$-finite spectrum: $\lambda = \Theta(\beta\log N + \log(NH|\mathcal{M}|/\delta))$*

- *$\beta$-polynomial decay: $\lambda = \Theta(C_{\text{poly}}N^{1/(1+\beta)} + \log(NH|\mathcal{M}|/\delta))$;*

- *$\beta$-exponential decay: $\lambda = \Theta(C_{\text{exp}}(\log N)^{1/\beta} + \log(NH|\mathcal{M}|/\delta))$;*

*$c$ is an absolute constant. Then with probability at least $1 - \delta$, $\forall n \in [N], i \in [M]$ we have*

- *for $\beta$-finite spectrum,*

$$
\sum_{k=1}^N \Delta^n \lesssim \mathcal{O}\left(H^3\beta\log N\sqrt{N|A|^LC\log\frac{NH|\mathcal{M}|}{\delta}}\right)
$$

- *for $\beta$-polynomial decay,*

$$
\sum_{n=1}^N \Delta^n \lesssim \mathcal{O}\left(H^3 C_{\text{poly}}N^{\frac{1}{2(1+\beta)}}\log N\sqrt{N|A|^LC\log\frac{NH|\mathcal{M}|}{\delta}}\right)
$$

- *for $\beta$-exponential decay,*

$$
\sum_{n=1}^N \Delta^n \lesssim \mathcal{O}\left(H^3 C_{\text{exp}}(\log N)^{1+1/\beta}\sqrt{N|A|^LC\log\frac{NH|\mathcal{M}|}{\delta}}\right)
$$

*Proof.* Let $f_h^n(x,\boldsymbol{a}) = \left\|\widehat{\mathcal{P}}_{n,h}(\cdot|s,\boldsymbol{a}) - P_h(\cdot|x,\boldsymbol{a})\right\|_1$. With our choice of $\lambda$ and $\zeta_n$, according to Lemma 56, we have $\forall n \in [N], h \in [H], \widehat{\mathcal{P}} \in \mathcal{M}$,

$$
\mathbb{E}_{x\sim\hat{\rho}_{n,h}}\left[(f_h^n(x,\boldsymbol{a}))^2\right] \leqslant \zeta_n, \|p_h(\cdot|x,\boldsymbol{a})\|_{\left(\hat{\Sigma}_{h,\phi_h}^n\right)^{-1}} = \Theta\left(\|p_h(\cdot|x,\boldsymbol{a})\|_{\Sigma_{n,\hat{\rho}_{n,h},\phi_h}^{-1}}\right). \tag{25}
$$

By definition, we have

$$
\Delta^n = \max_{i\in[M]}\left\{\overline{v}_i^n - \underline{v}_i^n\right\}.
$$

For each fixed $i \in [M], h \in [H]$ and $n \in [N]$, we have

$$\mathbb{E}_{x \sim d_{\mathcal{P},h}^{\pi^n}} \left[ \overline{V}_{h,i}^n(x) - \underline{V}_{h,i}^n(x) \right]$$

$$= \mathbb{E}_{x \sim d_{\mathcal{P},h}^{\pi^n}} \left[ \left( \mathbb{D}_{\pi_h^n} \overline{Q}_{h,i}^n \right)(x) - \left( \mathbb{D}_{\pi_h^n} \underline{Q}_{h,i}^n \right)(x) \right]$$

$$= \mathbb{E}_{(x,\boldsymbol{a}) \sim d_{\mathcal{P},h}^{\pi^n}} \left[ \overline{Q}_{h,i}^n(x,\boldsymbol{a}) - \underline{Q}_{h,i}^n(x,\boldsymbol{a}) \right]$$

$$= \mathbb{E}_{(x,\boldsymbol{a}) \sim d_{\mathcal{P},h}^{\pi^n}} \left[ 2\hat{b}_{n,h}(x,\boldsymbol{a}) + \left( \widehat{\mathcal{P}}_{n,h} \left( \overline{V}_{h+1,i}^n - \underline{V}_{h+1,i}^n \right) \right)(x,\boldsymbol{a}) \right]$$

$$= \mathbb{E}_{(x,\boldsymbol{a}) \sim d_{\mathcal{P},h}^{\pi^n}} \left[ 2\hat{b}_{n,h}(x,\boldsymbol{a}) + \left( \left( \widehat{\mathcal{P}}_{n,h} - \mathcal{P}_h \right) \left( \overline{V}_{h+1,i}^n - \underline{V}_{h+1,i}^n \right) \right)(x,\boldsymbol{a}) \right]$$

$$+ \mathbb{E}_{x \sim d_{\mathcal{P},h+1}^{\pi^n}} \left[ \overline{V}_{h+1,i}^n(x) - \underline{V}_{h+1,i}^n(x) \right]$$

$$\leqslant \mathbb{E}_{(x,\boldsymbol{a}) \sim d_{\mathcal{P},h}^{\pi^n}} \left[ 2\hat{b}_{n,h}(x,\boldsymbol{a}) + 2H^2 f_h^n(x,\boldsymbol{a}) \right] + \mathbb{E}_{x \sim d_{\mathcal{P},h+1}^{\pi^n}} \left[ \overline{V}_{h+1,i}^n(x) - \underline{V}_{h+1,i}^n(x) \right].$$

Note that we use the fact $\overline{V}_{h+1,i}^n(x) - \underline{V}_{h+1,i}^n(x)$ is upper bounded by $2H^2$, which can be proved easily using induction using the fact that $\hat{b}_{n,h}(x,\boldsymbol{a}) \leqslant H$. Applying the above formula recursively to $\mathbb{E}_{x \sim d_{\mathcal{P},h+1}^{\pi^n}} \left[ \overline{V}_{h+1,i}^n(x) - \underline{V}_{h+1,i}^n(x) \right]$, one gets the following result (or more formally, one can prove by induction, just like what we did in Lemma 36, Lemma 37 and Lemma 38):

$$\mathbb{E}_{x \sim d_{P,0}^{\pi^n}} \left[ \overline{V}_{0,i}^n(x) - \underline{V}_{0,i}^n(x) \right] \leqslant 2 \underbrace{\sum_{h=0}^{H} \mathbb{E}_{(x,\boldsymbol{a}) \sim d_{P,h}^{\pi^n}} \left[ \hat{b}_{n,h}(x,\boldsymbol{a}) \right]}_{(a)} + 2H^2 \underbrace{\sum_{h=0}^{H} \mathbb{E}_{(x,\boldsymbol{a}) \sim d_{P,h}^{\pi^n}} \left[ f_h^n(x,\boldsymbol{a}) \right]}_{(b)}.$$

(26)

First, we calculate the first term (a) in Inequality equation 26. Following Lemma 34 and noting the bonus $\hat{b}_h^n$ is $O(H)$, we have

$$\sum_{h=0}^{H} \mathbb{E}_{(x,\boldsymbol{a}) \sim d_{\mathcal{P},h}^{\pi^n}} \left[ \hat{b}_{n,h}(x,\boldsymbol{a}) \right]$$

$$\lesssim \sum_{h=0}^{H} \mathbb{E}_{(x,\boldsymbol{a}) \sim d_{\mathcal{P},h}^{\pi^n}} \left[ \min \left\{ \alpha_n \left\| \hat{b}_{n,h}(\cdot | x_{h-L}, \boldsymbol{a}_{h-L}) \right\|_{\Sigma_{n,\hat{\rho}_{n,h},\hat{p}_n}^{-1}}, H \right\} \right] \qquad \text{(From equation 25)}$$

$$\lesssim \sum_{h=0}^{H-1} \mathbb{E}_{(x_{h-L}, \boldsymbol{a}_{h-L}) \sim d_{\mathcal{P},h-L}^{\pi^n}} \left[ \left\| p_h(\cdot | x_{h-L}, \boldsymbol{a}_{h-L}) \right\|_{\Sigma_{\rho_{n,h-L},p}^{-1}} \right]$$

$$\cdot \sqrt{n |\mathcal{A}|^L \cdot \mathbb{E}_{(\tilde{x}_h, \tilde{\boldsymbol{a}}_h) \sim \rho_{n,h-L} \circ^L \mathcal{U}(\mathcal{A})} \left[ (\hat{b}_{n,h}(\tilde{x}_h, \tilde{\boldsymbol{a}}_h))^2 \right] + \lambda H^2 C}.$$

Note that we use the fact that $B = H$ when applying Lemma 34. In addition, following the proof of Lemma 8 in (41), we have that

- for $\beta$-finite spectrum,

$$n \mathbb{E}_{(\tilde{x}_h, \tilde{\boldsymbol{a}}_h) \sim \rho_{n,h-L} \circ^L \mathcal{U}(\mathcal{A})} \left[ \widehat{b}_{n,h}(\tilde{x}_h, \tilde{\boldsymbol{a}}_h)^2 \right] = O(\beta \log N);$$

$$\sum_{n \in [N]} \mathbb{E}_{(x_{h-L}, \boldsymbol{a}_{h-L}) \sim d_{P,h}^{\pi^n}} \left[ \left\| p_h(\cdot | x_{h-L}, \boldsymbol{a}_{h-L}) \right\|_{L_2(\mu), \Sigma_{\rho_{n,h-L},p}^{-1}}^2 \right] = O(\beta \log N);$$

- for $\beta$-polynomial decay,

$$n \mathbb{E}_{(\tilde{x}_h, \tilde{\boldsymbol{a}}_h) \sim \rho_{n,h-L} \circ^L \mathcal{U}(\mathcal{A})} \left[ \widehat{b}_{n,h}(\tilde{x}_h, \tilde{\boldsymbol{a}}_h)^2 \right] = O \left( C_{\text{poly}} N^{\frac{1}{2(1+\beta)}} \log N \right);$$

$$\sum_{n \in [N]} \mathbb{E}_{(x_{h-L}, \boldsymbol{a}_{h-L}) \sim d_{P,h}^{\pi^n}} \left[ \left\| p_h(\cdot | x_{h-L}, \boldsymbol{a}_{h-L}) \right\|_{L_2(\mu), \Sigma_{\rho_{n,h-L},p}^{-1}}^2 \right] = O \left( C_{\text{poly}} N^{\frac{1}{2(1+\beta)}} \log N \right);$$

- for $\beta$-exponential decay,

$$n\mathbb{E}_{(\tilde{x}_h, \tilde{\boldsymbol{a}}_h) \sim \rho_{n,h-L} \circ^L \mathcal{U}(\mathcal{A})} \left[ \widehat{b}_{n,h}(\tilde{x}_h, \tilde{\boldsymbol{a}}_h)^2 \right] = O\left( C_{\exp}(\log N)^{1+1/\beta} \right).$$

$$\sum_{n \in [N]} \mathbb{E}_{(x_{h-L}, \boldsymbol{a}_{h-L}) \sim d_{P,h}^{\pi_n}} \left[ \|p_h(\cdot | x_{h-L}, \boldsymbol{a}_{h-L})\|_{L_2(\mu), \Sigma_{\rho_{n,h-L}, P}^{-1}}^2 \right] = O\left( C_{\exp}(\log N)^{1+1/\beta} \right).$$

Second, we calculate the term (b) in inequality equation 26. Following Lemma 34 and noting that $f_h^n(x, \boldsymbol{a})$ is upper-bounded by 2 (i.e., $B = 2$ in Lemma 34), we have

$$\sum_{h=0}^{H} \mathbb{E}_{(x, \boldsymbol{a}) \sim d_{\mathcal{P},h}^{\pi_n}} [f_h^n(x, \boldsymbol{a})]$$

$$\leqslant \sum_{h=0}^{H-1} \mathbb{E}_{(x_{h-L}, \boldsymbol{a}_{h-L}) \sim d_{\mathcal{P},h-L}^{\pi_n}} \left[ \|p_h(\cdot | x_{h-L}, \boldsymbol{a}_{h-L})\|_{\Sigma_{\rho_{n,h-L}, P}^{-1}} \right]$$

$$\cdot \sqrt{n|A|^L \mathbb{E}_{(\tilde{x}_h, \tilde{\boldsymbol{a}}_h) \sim \rho_{n,h-L} \circ^L \mathcal{U}(\mathcal{A})} \left[ (f_h^n(\tilde{x}_h, \tilde{\boldsymbol{a}}_h))^2 \right] + 4C\lambda}$$

$$\leqslant \sum_{h=0}^{H-1} \mathbb{E}_{(x_{h-L}, \boldsymbol{a}_{h-L}) \sim d_{\mathcal{P},h-L}^{\pi_n}} \left[ \|p_h(\cdot | x_{h-L}, \boldsymbol{a}_{h-L})\|_{\Sigma_{\rho_{n,h-L}, P}^{-1}} \right] \sqrt{n|A|^L \zeta_n + 4C\lambda},$$

where in the second inequality, we use $\mathbb{E}_{(\tilde{x}_h, \tilde{\boldsymbol{a}}_h) \sim \rho_{n,h-L} \circ^L \mathcal{U}(\mathcal{A})} \left[ (f_h^n(\tilde{x}_h, \tilde{\boldsymbol{a}}_h))^2 \right] \leqslant \zeta_n$.

Then, by combining the above calculation of the term (a) and term (b) in inequality equation 26, we have:

- for $\beta$-finite spectrum,

$$\sum_{n=1}^{N} \Delta^n \lesssim \mathcal{O}\left( \sqrt{\beta \log N} \cdot \left( H\sqrt{|\mathcal{A}|^L \beta \log N + \lambda H^2 C} + H^3 \sqrt{N|\mathcal{A}|^L \zeta_N + 4C\lambda} \right) \right)$$

- for $\beta$-polynomial decay,

$$\sum_{n=1}^{N} \Delta^n \lesssim \mathcal{O}\left( \sqrt{C_{\text{poly}} N^{\frac{1}{2(1+\beta)}} \log N} \right.$$
$$\left. \cdot \left( H\sqrt{|\mathcal{A}|^L C_{\text{poly}} N^{\frac{1}{2(1+\beta)}} \log N + \lambda H^2 C} + H^3 \sqrt{N|\mathcal{A}|^L \zeta_N + 4C\lambda} \right) \right)$$

- for $\beta$-exponential decay,

$$\sum_{n=1}^{N} \Delta^n \lesssim \mathcal{O}\left( \sqrt{C_{\exp}(\log N)^{1+1/\beta}} \right.$$
$$\left. \cdot \left( H\sqrt{|\mathcal{A}|^L C_{\exp}(\log N)^{1+1/\beta} + \lambda H^2 C} + H^3 \sqrt{N|\mathcal{A}|^L \zeta_N + 4C\lambda} \right) \right)$$

By substituting $\lambda$ into the results, we obtain:

- for $\beta$-finite spectrum,

$$\sum_{k=1}^{N} \Delta^n \lesssim \mathcal{O}\left( H^3 \beta \log N \sqrt{N|A|^L C \log \frac{NH|\mathcal{M}|}{\delta}} \right)$$

- for $\beta$-polynomial decay,

$$\sum_{n=1}^{N} \Delta^n \lesssim \mathcal{O}\left( H^3 C_{\text{poly}} N^{\frac{1}{2(1+\beta)}} \log N \sqrt{N|A|^L C \log \frac{NH|\mathcal{M}|}{\delta}} \right)$$

- for $\beta$-exponential decay,

$$\sum_{n=1}^{N} \Delta^n \lesssim \mathcal{O}\left(H^3 C_{\exp}(\log N)^{1+1/\beta}\sqrt{N|A|^L C \log \frac{NH|\mathcal{M}|}{\delta}}\right)$$

This concludes the proof. $\qquad\square$

**Theorem 40** (PAC guarantee of OFOVI-LLVR). *Assume Assumption 4,5 in Appendix E.4 hold and the kernel $K$ satisfies the regularity conditions in Appendix E.2. When OFOVI-LLVR is applied with parameters $\zeta_n = \Theta\left(\log(Hn|\mathcal{M}|/\delta)/n\right), \hat{b}_{n,h} = \min\left\{\alpha_n\|\hat{p}_n(\cdot|x_{h-L}, a_{h-L})\|_{L_2(\mu),\hat{\Sigma}_{n,h,\hat{p}_n}^{-1}}, H\right\}$ with $\alpha_n = \Theta\sqrt{\lambda C + nL|\mathcal{A}|^L\zeta_n}$ and*

- *$\beta$-finite spectrum: $\lambda = \Theta(\beta \log N + \log(NH|\mathcal{M}|/\delta))$;*

- *$\beta$-polynomial decay: $\lambda = \Theta(C_{\text{poly}}N^{1/(1+\beta)} + \log(NH|\mathcal{M}|/\delta))$;*

- *$\beta$-exponential decay: $\lambda = \Theta(C_{\exp}(\log N)^{1/\beta} + \log(NH|\mathcal{M}|/\delta))$;*

*by setting the number of episodes $N$ to be at most*

- *for $\beta$-finite spectrum,*

$$\tilde{\mathcal{O}}\left(\frac{H^3\beta|\mathcal{A}|^{L/2}C^{1/2}\log\frac{H|\mathcal{A}|^{L/2}|\mathcal{M}|}{\delta\epsilon}}{\epsilon}\right)$$

- *for $\beta$-polynomial decay,*

$$\tilde{\mathcal{O}}\left(\left(\frac{H^3 C_{\text{poly}}|\mathcal{A}|^{L/2}C^{1/2}\log\frac{H|\mathcal{A}|^{L/2}|\mathcal{M}|}{\delta\epsilon}}{\epsilon}\right)^{2+\frac{2}{\beta}}\right)$$

- *for $\beta$-exponential decay,*

$$\tilde{\mathcal{O}}\left(\left(\frac{H^3 C_{\exp}|A|^{L/2}C^{1/2}\log\frac{H|\mathcal{A}|^{L/2}|\mathcal{M}|}{\delta\epsilon}}{\epsilon}\right)^{2}\right)$$

*with probability $1 - \delta$, the output policy $\hat{\pi}$ is an $\varepsilon$-approximate $\{\text{NE}, \text{CCE}, \text{CE}\}$.*

*Proof.* For any fixed episode $n$ and agent $i$, by Lemma 36, Lemma 37 and Lemma 38, we have

$$v_i^{\dagger,\pi_{-i}^n} - v_i^{\pi^n}\left(\text{or } \max_{\omega\in\Omega_i} v_i^{\omega\circ\pi^n} - v_i^{\pi^n}\right) \leqslant \overline{v}_i^n - \hat{\underline{v}}_i^n \leqslant \Delta^n.$$

Taking maximum over $i$ on both sides, we have

$$\max_{i\in[M]}\left\{v_i^{\dagger,\pi_{-i}^n} - v_i^{\pi^n}\right\}\left(\text{or } \max_{i\in[M]}\left\{\max_{\omega\in\Omega_i} v_i^{\omega\circ\pi^n} - v_i^{\pi^n}\right\}\right) \leqslant \Delta^n. \qquad (27)$$

From Lemma 39, with probability $1 - \delta$, we can ensure

- for $\beta$-finite spectrum,

$$\sum_{k=1}^{N} \Delta^n \lesssim \mathcal{O}\left(H^3\beta\log N\sqrt{N|A|^L C \log \frac{NH|\mathcal{M}|}{\delta}}\right)$$

- for $\beta$-polynomial decay,

$$\sum_{n=1}^{N} \Delta^n \lesssim \mathcal{O}\left(H^3 C_{\text{poly}} N^{\frac{1}{2(1+\beta)}}\log N\sqrt{N|A|^L C \log \frac{NH|\mathcal{M}|}{\delta}}\right)$$

- for $\beta$-exponential decay,

$$\sum_{n=1}^{N} \Delta^n \lesssim \mathcal{O}\left(H^3 C_{\exp}(\log N)^{1+1/\beta}\sqrt{N|A|^L C \log \frac{NH|\mathcal{M}|}{\delta}}\right)$$

Therefore, when we pick $N$ to be

- for $\beta$-finite spectrum,

$$\tilde{\mathcal{O}}\left(\frac{H^3 \beta |\mathcal{A}|^{L/2} C^{1/2} \log \frac{H||\mathcal{A}|^{L/2}|\mathcal{M}|}{\delta \epsilon}}{\epsilon}\right)$$

- for $\beta$-polynomial decay,

$$\tilde{\mathcal{O}}\left(\left(\frac{H^3 C_{\text{poly}} |\mathcal{A}|^{L/2} C^{1/2} \log \frac{H||\mathcal{A}|^{L/2}|\mathcal{M}|}{\delta \epsilon}}{\epsilon}\right)^{2+\frac{2}{\beta}}\right)$$

- for $\beta$-exponential decay,

$$\tilde{\mathcal{O}}\left(\left(\frac{H^3 C_{\exp} |A|^{L/2} C^{1/2} \log \frac{H|\mathcal{A}|^{L/2}|\mathcal{M}|}{\delta \epsilon}}{\epsilon}\right)^{2}\right)$$

we have

$$\frac{1}{N}\sum_{n=1}^{N} \Delta^n \leqslant \varepsilon.$$

On the other hand, from equation 27, we have

$$\max_{i \in [M]} \left\{v_i^{\dagger, \hat{\pi}_{-i}} - v_i^{\hat{\pi}}\right\} \left(\text{or } \max_{i \in [M]} \left\{\max_{\omega \in \Omega_i} v_i^{\omega \circ \hat{\pi}} - v_i^{\hat{\pi}}\right\}\right)$$

$$= \max_{i \in [M]} \left\{v_i^{\dagger, \pi_{-i}^{n^\star}} - v_i^{\pi^{n^\star}}\right\} \left(\text{or } \max_{i \in [M]} \left\{\max_{\omega \in \Omega_i} v_i^{\omega \circ \pi^{n^\star}} - v_i^{\pi^{n^\star}}\right\}\right)$$

$$\leqslant \Delta^{n^\star} = \min_{n \in [N]} \Delta^n \leqslant \frac{1}{N}\sum_{n=1}^{N} \Delta^n \leqslant \varepsilon,$$

which has finished the proof. $\qquad\square$

### E.5 OFFLINE SETTING

In this subsection, we show the theoretical analysis for offline exploitation. For offline exploitation, we have the access to a offline dataset, which we assume is collected from the stationary distribution of the fixed behavior policy set $\pi_b$, which we will denote as $\rho$. And we are not allowed to interact with the environments to collect new data. The only difference between the algorithms for offline exploitation and online exploration is that, as we do not have access to the new data from the environment, we cannot further explore the state-action pair that the offline dataset do not cover. Hence, we need to penalize the visitation to the unseen state action pair to avoid the risky behavior.

Similar to the online setting, we can obtain the upper bound of the statistical error for $\hat{\pi}$, which is stated in the following:

**Theorem 41** (PAC Guarantee for Offline Exploitation). *Define* $\omega := \max_{x,a} \pi_b^{-1}(a|x)$, *and*

$$C_\pi^* := \sup_{y \in L_2(\mu)} \frac{\mathbb{E}_{(x,a)\sim d_P^\pi}\left[\langle p(\cdot|x,a), y\rangle_{L_2(\mu)}\right]^2}{\mathbb{E}_{(x,a)\sim\rho}\left[\langle p(\cdot|x,a), y\rangle_{L_2(\mu)}\right]^2}.$$

*When Alg. 5 is applied with parameters* $\zeta = \Theta\left(\log(H|\mathcal{M}|/\delta)/n\right), \hat{b}_h = \min\left\{\alpha\|\hat{p}(\cdot|x_{h-L}, a_{h-L})\|_{L_2(\mu), \hat{\Sigma}_{n,\hat{p}}^{-1}}, H\right\}$ *with* $\alpha = \Theta\sqrt{\lambda C + nL\omega^{L-1}\zeta}$ *and*

---

**Algorithm 5** Offline OFOVI-LLVR for $L-$decodable POMGs

---

1: **Input:** Variational Distribution Class $\mathcal{Q} = \{\{q_h(z|x_h, \boldsymbol{a}_h, \boldsymbol{o}_{h+1})\}_{h \in [H]}\}$, Model Class $\mathcal{M} = \{\{(p_h(z|x_h, \boldsymbol{a}_h), p_h(\boldsymbol{o}_{h+1}|z))\}_{h \in [H]}\}$, offline dataset $\mathcal{D} = \{\mathcal{D}_h\}_{h \in [H]}$, Regularizer $\lambda$, parameter $\alpha, \zeta$.

2: **for** step $h = H, H - 1 \ldots, 1$ **do**

3:     Learn the latent variable model $\hat{p}(z|x_h, \boldsymbol{a}_h)$ with $\mathcal{D}_h$ via maximizing the ELBO, and obtain the learned model $\hat{\mathcal{P}} = \{(\hat{p}_h(z|x_h, \boldsymbol{a}_h), \hat{p}_h(\boldsymbol{o}_{h+1}|z))\}_{h \in [H]}$.

4: **end for**

5: Compute $\hat{b}_h$ from equation 7. For each $(x, \boldsymbol{a}) \in \mathcal{X} \times \mathcal{A}, i \in [M]$, set

$$\overline{Q}_{h,i}(x, \boldsymbol{a}) = r_{h,i}(x, \boldsymbol{a}) + \left(\hat{P}_h \overline{V}_{h+1,i}\right)(x, \boldsymbol{a}) + \hat{b}_h(x, \boldsymbol{a})$$

$$\underline{Q}_{h,i}(x, \boldsymbol{a}) = r_{h,i}(x, \boldsymbol{a}) + \left(\hat{P}_h \underline{V}_{h+1,i}\right)(x, \boldsymbol{a}) - \hat{b}_h(x, \boldsymbol{a})$$

6: Compute $\hat{\pi}_h$ from equation equation 1 or equation equation 2 or equation equation 3. For each $x \in \mathcal{X}, i \in [M]$, set

$$\overline{V}_{h,i} = \left(\mathbb{D}_{\pi_h} \overline{Q}_{h,i}\right)(x), \quad \underline{V}_{h,i} = \left(\mathbb{D}_{\pi_h} \underline{Q}_{h,i}\right)(x).$$

7: Compute $\Delta = \max_{i \in [M]} \{\overline{v}_i - \underline{v}_i\}$, where $\overline{v}_i = \int_{\mathcal{X}} \overline{V}_{0,i}(x) \mu_0(x) \, dx$ and $\underline{v}_i = \int_{\mathcal{X}} \underline{V}_{0,i}(x) \mu_0(x) \, dx$.

8: **Return** $\hat{\pi}$.

---

- *$\beta$-finite spectrum:* $\lambda = \Theta(\beta \log n + \log(|\mathcal{M}|/\delta))$;

- *$\beta$-polynomial decay:* $\lambda = \Theta(C_{\text{poly}} n^{1/(1+\beta)} + \log(|\mathcal{M}|/\delta))$;

- *$\beta$-exponential decay:* $\lambda = \Theta(C_{\exp}(\log n)^{1/\beta} + \log(|\mathcal{M}|/\delta))$;

*with probability $1 - \delta$, the output policy $\hat{\pi}$ is an $\varepsilon$-approximate $\{\text{NE}, \text{CCE}, \text{CE}\}$ with*

- *for $\beta$-finite spectrum,*

$$\varepsilon = \mathcal{O}\left(H^3 \beta \log n \sqrt{C_{\hat{\pi}}^{\star} n \omega^L C \zeta \log \frac{|\mathcal{M}|}{\delta}}\right)$$

- *for $\beta$-polynomial decay,*

$$\varepsilon = \mathcal{O}\left(H^3 C_{\text{poly}} n^{\frac{1}{2(1+\beta)}} \log n \sqrt{C_{\hat{\pi}}^{\star} n \omega^L C \zeta \log \frac{|\mathcal{M}|}{\delta}}\right)$$

- *for $\beta$-exponential decay,*

$$\varepsilon = \mathcal{O}\left(H^3 C_{\exp}(\log n)^{1+1/\beta} \sqrt{C_{\hat{\pi}}^{\star} n \omega^L C \zeta \log \frac{|\mathcal{M}|}{\delta}}\right)$$

We start by showing that $C_{\pi}^*$ can be viewed as a measure of the offline data quality, which can be demonstrated by the following lemma, that was first introduced in (51):

**Lemma 42** (Distribution Shift Lemma). *For any positive definite operator $\Lambda : L_2(\mu) \to L_2(\mu)$, we have that*

$$\mathbb{E}_{(x, \boldsymbol{a}) \sim d_{\mathcal{P}}^{\pi}} \langle p(\cdot|x, \boldsymbol{a}), \Lambda p(\cdot|x, \boldsymbol{a}) \rangle_{L_2(\mu)} \leqslant C_{\pi}^* \mathbb{E}_{(x, \boldsymbol{a}) \sim \rho} \langle p(\cdot|x, \boldsymbol{a}), \Lambda p(\cdot|x, \boldsymbol{a}) \rangle_{L_2(\mu)}.$$

*Proof.* We denote the eigendecomposition of $\Lambda$ as $\Lambda = U\Sigma U$ where $\{\sigma_i, u_i\}$ is the eigensystem of $\Lambda$. Then we have

$$\mathbb{E}_{(x,\boldsymbol{a})\sim d_P^\pi}\langle p(\cdot|x,\boldsymbol{a}), \Lambda p(\cdot|x,\boldsymbol{a})\rangle_{L_2(\mu)}$$

$$=\sum_{i\in I}\sigma_i\mathbb{E}_{(x,\boldsymbol{a})\sim d_P^\pi}\langle u_i, p(\cdot|x,\boldsymbol{a})^\top\rangle_{L_2(\mu)}^2$$

$$\leqslant C_\pi\sum_{i\in I}\sigma_i\mathbb{E}_{(x,\boldsymbol{a})\sim\rho}\langle u_i, p(\cdot|x,\boldsymbol{a})^\top\rangle_{L_2(\mu)}^2$$

$$=C_\pi\mathbb{E}_{(x,\boldsymbol{a})\sim\rho}\langle p(\cdot|x,\boldsymbol{a}), \Lambda p(\cdot|x,\boldsymbol{a})\rangle_{L_2(\mu)},$$

which finishes the proof. $\qquad\square$

We also define the $\Sigma_{\rho,\phi} : L_2(\mu) \to L_2(\mu)$:

$$\Sigma_{\rho,\phi} := n\mathbb{E}_{(x,\boldsymbol{a})\sim\rho}\left[\phi(x,\boldsymbol{a})\phi^\top(x,\boldsymbol{a})\right] + \lambda T_k^{-1},$$

where $\rho$ is the stationary distribution of $\pi_b$.

**Lemma 43** ($L$-step back inequality for the true model). *Given a set of functions* $[g_h]_{h\in[H]}$, *where* $g_h : \mathcal{X} \times \mathcal{A} \to \mathbb{R}$, $\|g_h\|_\infty \leqslant B$, $\forall h \in [H]$, *we have that* $\forall\pi$,

$$\sum_{h\in[H]}\mathbb{E}_{(x_h,\boldsymbol{a}_h)\sim d_{P,h}^\pi}[g_h(x_h,\boldsymbol{a}_h)] \leqslant \sum_{h\in[H]}\mathbb{E}_{(x_{h-L},\boldsymbol{a}_{h-L})\sim d_{P_{h-L}}^\pi}\left[\|p(\cdot|x_{h-L},\boldsymbol{a}_{h-L})\|_{L_2(\mu),\Sigma_{\rho,p}^{-1}}\right]$$

$$\cdot\sqrt{n\omega^L\cdot\mathbb{E}_{(\tilde{x}_h,\tilde{\boldsymbol{a}}_h)\sim\rho\circ^L\pi_b(\cdot|x)}[g_h(\tilde{x}_h,\tilde{\boldsymbol{a}}_h)^2] + \lambda B^2 C}$$

*Proof.* The proof can be adapted from the proof of Lemma 6 in (41), and we include it for the completeness. Recall the moment matching policy $\chi_\pi$ Since $\chi_{\pi,h}$ does not depend on $(x_{h-L}, a_{h-L})$, we can make the following decomposition:

$$\mathbb{E}_{(x_h,\boldsymbol{a}_h)\sim d_{P,h}^\pi}[g_h(x_h,\boldsymbol{a}_h)]$$

$$=\mathbb{E}_{(x_{h-L},\boldsymbol{a}_{h-L})\sim d_{P,h-L}^\pi}\left[\int_{x_h}\langle p(\cdot|x_{h-L},\boldsymbol{a}_{h-L}), \mathbb{P}^{\chi_\pi}(x_h|\cdot)\rangle_{L_2(\mu)}\cdot\mathbb{E}_{\boldsymbol{a}_h\sim\chi_{\pi,h}}[g_h(x_h,\boldsymbol{a}_h)]dx_h\right]$$

$$\leqslant\mathbb{E}_{(x_{h-L},\boldsymbol{a}_{h-L})\sim d_{P,h-L}^\pi}\|p(\cdot|x_{h-L},\boldsymbol{a}_{h-L})\|_{L_2(\mu),\Sigma_{\rho,p}^{-1}}$$

$$\cdot\left\|\int_{x_h}\mathbb{P}^{\chi_\pi}(x_h|\cdot)\mathbb{E}_{\boldsymbol{a}_h\sim\chi_{\pi,h}}[g_h(x_h,\boldsymbol{a}_h)]dx_h\right\|_{L_2(\mu),\Sigma_{\rho,p}}.$$

Direct computation shows that

$$\left\|\int_{x_h}\mathbb{P}^{\chi_\pi}(x_h|\cdot)\mathbb{E}_{\boldsymbol{a}_h\sim\chi_{\pi,h}(\cdot|x_h)}[g_h(x_h,\boldsymbol{a}_h)]dx_h\right\|_{L_2(\mu),\Sigma_{\rho,p}}^2$$

$$=n\mathbb{E}_{(\tilde{x}_{h-L},\tilde{\boldsymbol{a}}_{h-L})\sim\rho}\left[\mathbb{E}_{x_h\sim\mathbb{P}^{\chi_\pi}(\cdot|x_{h-L},\boldsymbol{a}_{h-L}),\boldsymbol{a}_h\sim\chi_{\pi,h}(\cdot|x_h)}[g_h(x_h,\boldsymbol{a}_h)]\right]^2$$

$$+\lambda\left\|\int_{x_h}\mathbb{P}^{\chi_\pi}(x_h|\cdot)\cdot\mathbb{E}_{\boldsymbol{a}_h\sim\chi_{\pi,h}(\cdot|x_h)}[g_h(x_h,\boldsymbol{a}_h)]dx_h\right\|_{\mathcal{H}}^2$$

$$\leqslant n\mathbb{E}_{(\tilde{x}_{h-L},\tilde{\boldsymbol{a}}_{h-L})\sim\rho}\mathbb{E}_{x_h\sim\mathbb{P}^{\chi_\pi}(\cdot|x_{h-L},\boldsymbol{a}_{h-L}),\boldsymbol{a}_h\sim\chi_{\pi,h}(\cdot|x_h)}[g_h(x_h,\boldsymbol{a}_h)]^2 + \lambda B^2 C$$

$$\leqslant n\omega^L\mathbb{E}_{(\tilde{x}_h,\tilde{\boldsymbol{a}}_h)\sim\rho\circ^L\pi_b(\cdot|x)}[g_h(\tilde{x}_h,\tilde{\boldsymbol{a}}_h)]^2 + \lambda B^2 C,$$

which finishes the proof. $\qquad\square$

**Lemma 44** ($L$-step back inequality for the learned model). *Assume we have a set of functions* $[g_h]_{h\in[H]}$, *where* $g_h : \mathcal{X} \times \mathcal{A} \to \mathbb{R}$, $\|g_h\|_\infty \leqslant B$, $\forall h \in [H]$. *Given Lemma 51, we have that* $\forall\pi$,

$$\sum_{h\in[H]}\mathbb{E}_{(x_h,\boldsymbol{a}_h)\sim d_{\hat{\mathcal{P}}_h}^\pi}[g_h(x_h,\boldsymbol{a}_h)] \leqslant \sum_{h\in[H]}\mathbb{E}_{(x_{h-L},\boldsymbol{a}_{h-L})\sim d_{\hat{\mathcal{P}}_{h-L}}^\pi}\left[\|\hat{p}(\cdot|x_{h-L},\boldsymbol{a}_{h-L})\|_{L_2(\mu),\Sigma_{\rho,\hat{p}}^{-1}}\right]$$

$$\cdot\sqrt{n\omega^L\cdot\mathbb{E}_{(\tilde{x}_h,\tilde{\boldsymbol{a}}_h)\sim\rho}[g_h(\tilde{x}_h,\tilde{\boldsymbol{a}}_h)^2] + \lambda B^2 C + nL\omega^{L-1}B^2\zeta}$$

*Proof.* The proof can be adapted from the proof of Lemma 5 in (41), and we include it for the completeness. We define a similar moment matching policy and make the following decomposition:

$$\mathbb{E}_{(x_h, \boldsymbol{a}_h) \sim d^\pi_{\widehat{\mathcal{P}}, h}} [g_h(x_h, \boldsymbol{a}_h)]$$

$$= \mathbb{E}_{(x_{h-L}, \boldsymbol{a}_{h-L}) \sim d^\pi_{\widehat{\mathcal{P}}, h-L}} \left[ \int_{x_h} \langle \hat{p}(\cdot | x_{h-L}, \boldsymbol{a}_{h-L}), \hat{p}(x_h | \cdot) \rangle_{L_2(\mu)} \cdot \mathbb{E}_{\boldsymbol{a}_h \sim \chi_{\pi, h}} [g_h(x_h, \boldsymbol{a}_h)] dx_h \right]$$

$$\leqslant \mathbb{E}_{(x_{h-L}, \boldsymbol{a}_{h-L}) \sim d^\pi_{\widehat{\mathcal{P}}, h-L}} \|\hat{p}(\cdot | x_{h-L}, \boldsymbol{a}_{h-L})\|_{L_2(\mu), \Sigma^{-1}_{\rho, \hat{p}}}$$

$$\cdot \left\| \int_{x_h} \hat{p}(x_h | \cdot) \mathbb{E}_{\boldsymbol{a}_h \sim \chi_{\pi, h}(\cdot | x_h)} [g_h(x_h, \boldsymbol{a}_h)] dx_h \right\|_{L_2(\mu), \Sigma_{\rho, \hat{p}}}.$$

Direct computation shows that

$$\left\| \int_{x_h} \hat{p}(x_h | \cdot) \mathbb{E}_{\boldsymbol{a}_h \sim \chi_{\pi, h}(\cdot | x_h)} [g_h(x_h, \boldsymbol{a}_h)] dx_h \right\|^2_{L_2(\mu), \Sigma_{\rho, \hat{p}}}$$

$$= n \mathbb{E}_{(\tilde{x}_{h-L}, \tilde{\boldsymbol{a}}_{h-L}) \sim \rho} \left[ \mathbb{E}_{x_h \sim \widehat{\mathcal{P}}(\cdot | \tilde{x}_{h-L}, \tilde{\boldsymbol{a}}_{h-L}), \boldsymbol{a}_h \sim \chi_{\pi, h}(\cdot | x_h)} [g_h(x_h, \boldsymbol{a}_h)] \right]^2$$

$$+ \lambda \left\| \int_{x_h} \hat{p}(x_h | \cdot) \mathbb{E}_{\boldsymbol{a}_h \sim \chi_{\pi, h}(\cdot | x_h)} [g_h(x_h, \boldsymbol{a}_h)] dx_h \right\|^2_{\mathcal{H}}$$

$$\leqslant n \mathbb{E}_{(\tilde{x}_{h-L}, \tilde{\boldsymbol{a}}_{h-L}) \sim \rho} \mathbb{E}_{x_h \sim \widehat{\mathcal{P}}(\cdot | \tilde{x}_{h-L}, \tilde{\boldsymbol{a}}_{h-L}), \boldsymbol{a}_h \sim \chi_{\pi, h}(\cdot | x_h)} [g_h(x_h, \boldsymbol{a}_h)]^2 + \lambda B^2 C$$

$$\leqslant n \omega^L \mathbb{E}_{(\tilde{x}_h, \tilde{\boldsymbol{a}}_h) \sim \rho} [g_h(\tilde{x}_h, \tilde{\boldsymbol{a}}_h)]^2 + n L \omega^{L-1} B^2 \zeta + \lambda B^2 C,$$

where we use the MLE guarantee for each individual step to obtain the last inequality. This finishes the proof. $\qquad \square$

**Lemma 45** (Optimism for NE and CCE). *Set*

$$\hat{b}_h = \min \left\{ \alpha \|\hat{p}(\cdot | x_{h-L}, \boldsymbol{a}_{h-L})\|_{L_2(\mu), \hat{\Sigma}^{-1}_{h, \hat{p}}}, H \right\},$$

*with* $\alpha = \Theta \sqrt{\lambda C + n L \omega^L \zeta}$, $\zeta = O(\log(H|\mathcal{M}|/\delta)/n)$

$$\hat{\Sigma}_{h, \hat{p}} : L_2(\mu) \to L_2(\mu), \quad \hat{\Sigma}_{n, \hat{p}} := \sum_{(x_i, \boldsymbol{a}_i) \in \mathcal{D}} \left[ \hat{p}(z | x_i, \boldsymbol{a}_i) \hat{p}(z | x_i, \boldsymbol{a}_i)^\top \right] + \lambda T_K^{-1}$$

*where* $T_K$ *is the integral operator associated with* $K$ *(i.e.* $T_K f = \int f(x) K(x, \cdot) dx$*) and* $\lambda$ *is set for different eigendecay of* $K$ *as follows:*

- *$\beta$-finite spectrum:* $\lambda = \Theta(\beta \log n + \log(|\mathcal{M}|/\delta))$

- *$\beta$-polynomial decay:* $\lambda = \Theta(C_{\text{poly}} n^{1/(1+\beta)} + \log(|\mathcal{M}|/\delta))$;

- *$\beta$-exponential decay:* $\lambda = \Theta(C_{\exp}(\log n)^{1/\beta} + \log(|\mathcal{M}|/\delta))$;

*$c$ is an absolute constant. $\pi$ is computed by solving NE or CCE. Then with probability at least $1 - \delta$, $\forall i \in [M]$ we have*

$$\overline{v}_i(x) - v_i^{\dagger, \pi_{-i}}(x) \geqslant 0.$$

*Proof.* Define $\tilde{\mu}_{h,i}(\cdot | x) := \arg\max_\mu \left( \mathbb{D}_{\mu, \pi_{h,-i}} Q_{h,i}^{\dagger, \pi_{-i}} \right)(x)$ as the best response policy for player $i$ at step $h$, and let $\tilde{\pi}_h = \tilde{\mu}_{h,i} \times \pi_{h,-i}$. Let $f_h(x, \boldsymbol{a}) = \left\| \widehat{\mathcal{P}}_h(\cdot | x, \boldsymbol{a}) - P_h(\cdot | x, \boldsymbol{a}) \right\|_1$, then according to lemma 51 and lemma 56, we have that using the chosen $\lambda$, with probability at least $1 - \delta$,

$$\mathbb{E}_{(x, \boldsymbol{a}) \sim \rho} \left[ (f_h(x, \boldsymbol{a}))^2 \right] \leqslant \zeta, \quad \forall h \in [H],$$

$$\|p_h(\cdot | x_{h-L}, \boldsymbol{a}_{h-L})\|_{\hat{\Sigma}^{-1}_{n, \hat{p}}} = \Theta \left( \|p_h(\cdot | x_{h-L}, \boldsymbol{a}_{h-L})\|_{\hat{\Sigma}^{-1}_{\rho, \hat{p}}} \right), \quad \forall h \in [H], \widehat{\mathcal{P}} \in \mathcal{M}.$$

A direct conclusion is we can find an absolute constant $c$, such that

$$\hat{b}_h(x_h, \boldsymbol{a}_h) = \min\left\{\alpha \left\|\hat{p}_h(\cdot | x_{h-L}, \boldsymbol{a}_{h-L})\right\|_{\Sigma_{n,\hat{p}}^{-1}}, H\right\}$$

$$\geqslant \min\left\{c\alpha \left\|\hat{p}_h(\cdot | x_{h-L}, \boldsymbol{a}_{h-L})\right\|_{\Sigma_{n,\hat{p}_n}^{-1}}, H\right\}, \quad \forall h \in [H].$$

Next, we prove by induction that

$$\mathbb{E}_{x \sim d_{\widehat{\mathcal{P}},h}^{\tilde{\pi}}}\left[\overline{V}_{h,i}(x) - V_{h,i}^{\dagger,\pi_{-i}}(x)\right] \geqslant \sum_{h'=h}^{H} \mathbb{E}_{(x,\boldsymbol{a}) \sim d_{\widehat{\mathcal{P}},h'}^{\tilde{\pi}}}\left[\hat{b}_{h'}(x, \boldsymbol{a}) - H\min\{f_{h'}(x, \boldsymbol{a}), 1\}\right], \forall h \in [H]. \tag{28}$$

First, notice that $\forall h \in [H]$,

$$\mathbb{E}_{x \sim d_{\widehat{\mathcal{P}},h}^{\tilde{\pi}}}\left[\overline{V}_{h,i}(x) - V_{h,i}^{\dagger,\pi_{-i}}(x)\right] = \mathbb{E}_{x \sim d_{\widehat{\mathcal{P}},h}^{\tilde{\pi}}}\left[\left(\mathbb{D}_{\pi_h}\overline{Q}_{h,i}\right)(x) - \left(\mathbb{D}_{\tilde{\pi}_h}Q_{h,i}^{\dagger,\pi_{-i}}\right)(x)\right]$$

$$\geqslant \mathbb{E}_{x \sim d_{\widehat{\mathcal{P}},h}^{\tilde{\pi}}}\left[\left(\mathbb{D}_{\tilde{\pi}_h}\overline{Q}_{h,i}\right)(x) - \left(\mathbb{D}_{\tilde{\pi}_h}Q_{h,i}^{\dagger,\pi_{-i}}\right)(x)\right]$$

$$= \mathbb{E}_{(x,\boldsymbol{a}) \sim d_{\widehat{\mathcal{P}},h}^{\tilde{\pi}}}\left[\overline{Q}_{h,i}(x, \boldsymbol{a}) - Q_{h,i}^{\dagger,\pi_{-i}}(x, \boldsymbol{a})\right],$$

where the inequality uses the fact that $\pi_h$ is the NE (or CCE) solution for $\{\overline{Q}_{h,i}\}_{i=1}^{M}$. Now we are ready to prove equation 28:

- When $h = H$, we have

$$\mathbb{E}_{x \sim d_{\widehat{\mathcal{P}},H}^{\tilde{\pi}}}\left[\overline{V}_{H,i}(x) - V_{H,i}^{\dagger,\pi_{-i}}(x)\right] \geqslant \mathbb{E}_{(x,\boldsymbol{a}) \sim d_{\widehat{\mathcal{P}},H}^{\tilde{\pi}}}\left[\overline{Q}_{H,i}(x, \boldsymbol{a}) - Q_{H,i}^{\dagger,\pi_{-i}}(x, \boldsymbol{a})\right]$$

$$= \mathbb{E}_{(x,\boldsymbol{a}) \sim d_{\widehat{\mathcal{P}},H}^{\tilde{\pi}}}\left[\hat{b}_H(x, \boldsymbol{a})\right]$$

$$\geqslant \mathbb{E}_{(x,\boldsymbol{a}) \sim d_{\widehat{\mathcal{P}},H}^{\tilde{\pi}}}\left[\hat{b}_H(x, \boldsymbol{a}) - H\min\{f_H(x, \boldsymbol{a}), 1\}\right].$$

- Suppose the statement is true for step $h + 1$, then for step $h$, we have

$$\mathbb{E}_{x \sim d_{\widehat{\mathcal{P}},h}^{\tilde{\pi}}}\left[\overline{V}_{h,i}(x) - V_{h,i}^{\dagger,\pi_{-i}}(x)\right]$$

$$\geqslant \mathbb{E}_{(x,\boldsymbol{a}) \sim d_{\widehat{\mathcal{P}},h}^{\tilde{\pi}}}\left[\overline{Q}_{h,i}(x, \boldsymbol{a}) - Q_{h,i}^{\dagger,\pi_{-i}}(x, \boldsymbol{a})\right]$$

$$= \mathbb{E}_{(x,\boldsymbol{a}) \sim d_{\widehat{\mathcal{P}},h}^{\tilde{\pi}}}\left[\hat{b}_h(x, \boldsymbol{a}) + \left(\widehat{\mathcal{P}}_h \overline{V}_{h+1,i}\right)(x, \boldsymbol{a}) - \left(\mathcal{P}_h V_{h+1,i}^{\dagger,\pi_{-i}}\right)(x, \boldsymbol{a})\right]$$

$$= \mathbb{E}_{(x,\boldsymbol{a}) \sim d_{\widehat{\mathcal{P}},h}^{\tilde{\pi}}}\Big[\hat{b}_h(x, \boldsymbol{a})$$

$$+ \left(\widehat{\mathcal{P}}_h \left(\overline{V}_{h+1,i} - V_{h+1,i}^{\dagger,\pi_{-i}}\right)\right)(x, \boldsymbol{a}) + \left(\left(\widehat{\mathcal{P}}_h - \mathcal{P}_h\right)V_{h+1,i}^{\dagger,\pi_{-i}}\right)(x, \boldsymbol{a})\Big]$$

$$= \mathbb{E}_{(x,\boldsymbol{a}) \sim d_{\widehat{\mathcal{P}},h}^{\tilde{\pi}}}\left[\hat{b}_h(x, \boldsymbol{a}) + \left(\left(\widehat{\mathcal{P}}_h - \mathcal{P}_h\right)V_{h+1,i}^{\dagger,\pi_{-i}}\right)(x, \boldsymbol{a})\right]$$

$$+ \mathbb{E}_{x \sim d_{\widehat{\mathcal{P}},h+1}^{\tilde{\pi}}}\left[\overline{V}_{h+1,i}(x) - V_{h+1,i}^{\dagger,\pi_{-i}}(x)\right]$$

$$\geqslant \mathbb{E}_{(x,\boldsymbol{a}) \sim d_{\widehat{\mathcal{P}},h}^{\tilde{\pi}}}\left[\hat{b}_h(x, \boldsymbol{a}) - H\min\{f_h(x, \boldsymbol{a}), 1\}\right] + \mathbb{E}_{x \sim d_{\widehat{\mathcal{P}},h+1}^{\tilde{\pi}}}\left[\overline{V}_{h+1,i}(x) - V_{h+1,i}^{\dagger,\pi_{-i}}(x)\right]$$

$$\geqslant \sum_{h'=h}^{H} \mathbb{E}_{(x,\boldsymbol{a}) \sim d_{\widehat{\mathcal{P}},h'}^{\tilde{\pi}}}\left[\hat{b}_{h'}(x, \boldsymbol{a}) - H\min\{f_{h'}(x, \boldsymbol{a}), 1\}\right],$$

where we use the fact

$$\left|\left(\widehat{\mathcal{P}}_h - \mathcal{P}_h\right)V_{h+1,i}^{\dagger,\pi_{-i}}\right|(x, \boldsymbol{a}) \leqslant \min\left\{H, \left\|\widehat{\mathcal{P}}_h(\cdot | x, \boldsymbol{a}) - \mathcal{P}_h(\cdot | x, \boldsymbol{a})\right\|_1 \left\|V_{h+1,i}^{\dagger,\pi_{-i}}\right\|_\infty\right\}$$

$$\leqslant H\min\left\{1, \left\|\widehat{\mathcal{P}}_h(\cdot | x, \boldsymbol{a}) - \mathcal{P}_h(\cdot | x, \boldsymbol{a})\right\|_1\right\}$$

$$= H\min\{1, f_{h'}(x, \boldsymbol{a})\}$$

and the last row uses the induction assumption.

Therefore, we have proved equation 28. We then apply $h = 0$ to equation 28, and get

$$\mathbb{E}_{x \sim d_0} \left[ \overline{V}_{0,i}(x) - V_{0,i}^{\dagger,\pi_{-i}}(x) \right]$$

$$= \mathbb{E}_{x \sim d_{\hat{\mathcal{P}},0}^{\tilde{\pi}}} \left[ \overline{V}_{0,i}(x) - V_{0,i}^{\dagger,\pi_{-i}}(x) \right]$$

$$\geqslant \sum_{h=0}^{H} \mathbb{E}_{(x,\boldsymbol{a}) \sim d_{\hat{\mathcal{P}}_h}^{\tilde{\pi}}} \left[ \hat{b}_h(x, \boldsymbol{a}) - H \min\left\{ f_h(x, \boldsymbol{a}), 1 \right\} \right]$$

$$= \sum_{h=0}^{H} \mathbb{E}_{(x,\boldsymbol{a}) \sim d_{\hat{\mathcal{P}}_h}^{\tilde{\pi}}} \left[ \hat{b}_h(x, \boldsymbol{a}) \right] - H \sum_{h=0}^{H} \mathbb{E}_{(x,\boldsymbol{a}) \sim d_{\hat{\mathcal{P}}_h}^{\tilde{\pi}}} \left[ \min\left\{ f_h(x, \boldsymbol{a}), 1 \right\} \right].$$

Next we are going to bound the second term. Applying Lemma 44 to $g_h(x, \boldsymbol{a}) = \min\left\{ f_h(x, \boldsymbol{a}), 1 \right\}$, we have

$$\sum_{h=0}^{H} \mathbb{E}_{(x,\boldsymbol{a}) \sim d_{\hat{\mathcal{P}},h}^{\tilde{\pi}}} \left[ \min\left\{ f_h(x, \boldsymbol{a}), 1 \right\} \right]$$

$$\leqslant \sum_{h=0}^{H} \mathbb{E}_{(x_{h-L},\boldsymbol{a}_{h-L}) \sim d_{\hat{\mathcal{P}}_{h-L}}^{\tilde{\pi}}} \left[ \left\| \hat{p}_{h-1}(\cdot | x_{h-L}, \boldsymbol{a}_{h-L}) \right\|_{\Sigma_{\rho,\hat{p}}^{-1}} \right]$$

$$\cdot \sqrt{n\omega^L \cdot \mathbb{E}_{(\tilde{x}_h, \tilde{\boldsymbol{a}}_h) \sim \rho} \left[ \min\left\{ f_h(\tilde{x}_h, \tilde{\boldsymbol{a}}_h), 1 \right\}^2 \right] + \lambda C + nL\omega^{L-1}\zeta}$$

$$\leqslant \sum_{h=0}^{H} \mathbb{E}_{(x_{h-L},\boldsymbol{a}_{h-L}) \sim d_{\hat{\mathcal{P}}_{h-L}}^{\tilde{\pi}}} \left[ \left\| \alpha \hat{p}_{h-1}(\cdot | x_{h-L}, \boldsymbol{a}_{h-L}) \right\|_{\Sigma_{\rho,\hat{p}}^{-1}} \right]$$

Note that we here use the fact $\min\left\{ f_h(x, \boldsymbol{a}), 1 \right\} \leqslant 1$, $\mathbb{E}_{(\tilde{x}_h, \tilde{\boldsymbol{a}}_h) \sim \rho} \left[ \min\left\{ f_h(\tilde{x}_h, \tilde{\boldsymbol{a}}_h), 1 \right\}^2 \right] \leqslant \zeta$ and our choice of $\alpha$.

Combining all things together,

$$\overline{v}_i - v_i^{\dagger,\pi_{-i}} = \mathbb{E}_{x \sim d_0} \left[ \overline{V}_{0,i}(x) - V_{0,i}^{\dagger,\pi_{-i}}(x) \right]$$

$$\geqslant \sum_{h=1}^{H} \mathbb{E}_{(x,\boldsymbol{a}) \sim d_{\hat{\mathcal{P}},h}^{\tilde{\pi}}} \left[ \hat{b}_h(x, \boldsymbol{a}) \right] - H \sum_{h=1}^{H} \mathbb{E}_{(x,\boldsymbol{a}) \sim d_{\hat{\mathcal{P}}_h}^{\tilde{\pi}}} \left[ \min\left\{ f_h(x, \boldsymbol{a}), 1 \right\} \right]$$

$$\geqslant 0,$$

which proves the inequality. $\qquad\square$

**Lemma 46** (Optimism for CE). *Set*

$$\hat{b}_h = \min\left\{ \alpha \| \hat{p}(\cdot | x_{h-L}, \boldsymbol{a}_{h-L}) \|_{L_2(\mu), \hat{\Sigma}_{n,\hat{p}}^{-1}}, H \right\},$$

*with $\alpha = \Theta\sqrt{\lambda C + nL\omega^L \zeta}$,*

$$\hat{\Sigma}_{n,\hat{p}} : L_2(\mu) \to L_2(\mu), \quad \hat{\Sigma}_{n,\hat{p}} := \sum_{(x_i,\boldsymbol{a}_i) \in \mathcal{D}} \left[ \hat{p}_n(z|x_i, \boldsymbol{a}_i)\hat{p}_n(z|x_i, \boldsymbol{a}_i)^\top \right] + \lambda T_K^{-1}$$

*where $T_K$ is the integral operator associated with $K$ (i.e. $T_K f = \int f(x)K(x, \cdot)dx$) and $\lambda$ is set for different eigendecay of $K$ as follows:*

- *$\beta$-finite spectrum: $\lambda = \Theta(\beta \log n + \log(|\mathcal{M}|/\delta))$*

- *$\beta$-polynomial decay: $\lambda = \Theta(C_{\text{poly}} n^{1/(1+\beta)} + \log(|\mathcal{M}|/\delta))$;*

- *$\beta$-exponential decay: $\lambda = \Theta(C_{\exp}(\log n)^{1/\beta} + \log(|\mathcal{M}|/\delta))$;*

*c is an absolute constant. $\pi$ is computed by solving CE. Then with probability at least $1 - \delta$, $\forall i \in [M]$ we have*

$$\overline{v}_i(x) - \max_{\omega \in \Omega_i} v_i^{\omega \circ \pi}(x) \geqslant 0, \quad \forall i \in [M].$$

*Proof.* Denote $\tilde{\omega}_{h,i} = \arg\max_{\omega_h \in \Omega_{h,i}} \left( \mathbb{D}_{\omega_h \circ \pi_h} \max_{\omega \in \Omega_i} Q_{h,i}^{\omega \circ \pi} \right)(s)$ and let $\tilde{\pi}_h = \tilde{\omega}_{h,i} \circ \pi_h$. Let $f_h(x, \boldsymbol{a}) = \left\| \widehat{\mathcal{P}}_h(\cdot | x, \boldsymbol{a}) - P_h(\cdot | x, \boldsymbol{a}) \right\|_1$, then according to lemma 51 and lemma 56, we have that using the chosen $\lambda$, with probability at least $1 - \delta$,

$$\mathbb{E}_{(x,\boldsymbol{a}) \sim \rho} \left[ (f_h(x, \boldsymbol{a}))^2 \right] \leqslant \zeta, \quad \forall h \in [H],$$

$$\|p_h(\cdot | x_{h-L}, \boldsymbol{a}_{h-L})\|_{\hat{\Sigma}_{n,\hat{p}_n}^{-1}} = \Theta \left( \|p_h(\cdot | x_{h-L}, \boldsymbol{a}_{h-L})\|_{\hat{\Sigma}_{\rho,\hat{p}}^{-1}} \right), \quad \forall h \in [H], \widehat{\mathcal{P}} \in \mathcal{M}.$$

A direct conclusion is we can find an absolute constant $c$, such that

$$\hat{b}_h(x_h, \boldsymbol{a}_h) = \min \left\{ \alpha \|\hat{p}_h(\cdot | x_{h-L}, \boldsymbol{a}_{h-L})\|_{\Sigma_{n,\hat{p}}^{-1}}, H \right\}$$

$$\geqslant \min \left\{ c\alpha \|\hat{p}_h(\cdot | x_{h-L}, \boldsymbol{a}_{h-L})\|_{\Sigma_{n,\hat{p}}^{-1}}, H \right\}, \quad \forall h \in [H].$$

Next, we prove by induction that $\forall h \in [H]$,

$$\mathbb{E}_{x \sim d_{\widehat{\mathcal{P}}_h}^{\tilde{\pi}}} \left[ \overline{V}_{h,i}(x) - \max_{\omega \in \Omega_i} V_{h,i}^{\omega \circ \pi}(x) \right] \geqslant \sum_{h'=h}^{H} \mathbb{E}_{(x,\boldsymbol{a}) \sim d_{\widehat{\mathcal{P}},h'}^{\tilde{\pi}}} \left[ \hat{b}_{h'}(x, \boldsymbol{a}) - H \min \{ f_{h'}(x, \boldsymbol{a}), 1 \} \right].$$

$$(29)$$

First, notice that $\forall h \in [H]$,

$$\mathbb{E}_{x \sim d_{\widehat{\mathcal{P}}_h}^{\tilde{\pi}}} \left[ \overline{V}_{h,i}(x) - \max_{\omega \in \Omega_i} V_{h,i}^{\omega \circ \pi}(x) \right] = \mathbb{E}_{x \sim d_{\widehat{\mathcal{P}}_h}^{\tilde{\pi}}} \left[ \left( \mathbb{D}_{\pi_h} \overline{Q}_{h,i} \right)(x) - \left( \mathbb{D}_{\tilde{\pi}_h} \max_{\omega \in \Omega_i} Q_{h,i}^{\omega \circ \pi} \right)(x) \right]$$

$$\geqslant \mathbb{E}_{x \sim d_{\widehat{\mathcal{P}}_h}^{\tilde{\pi}}} \left[ \left( \mathbb{D}_{\tilde{\pi}_h} \overline{Q}_{h,i} \right)(x) - \left( \mathbb{D}_{\tilde{\pi}_h} \max_{\omega \in \Omega_i} Q_{h,i}^{\omega \circ \pi} \right)(x) \right]$$

$$= \mathbb{E}_{(x,\boldsymbol{a}) \sim d_{\widehat{\mathcal{P}}_h}^{\tilde{\pi}}} \left[ \overline{Q}_{h,i}(x, \boldsymbol{a}) - \max_{\omega \in \Omega_i} Q_{h,i}^{\omega \circ \pi}(x, \boldsymbol{a}) \right],$$

where the inequality uses the fact that $\pi_h$ is the CE solution for $\{\overline{Q}_{h,i}\}_{i=1}^{M}$. Now we are ready to prove equation 29:

- When $h = H$, we have

$$\mathbb{E}_{x \sim d_{\widehat{\mathcal{P}},H}^{\tilde{\pi}}} \left[ \overline{V}_{H,i}(x) - \max_{\omega \in \Omega_i} V_{H,i}^{\omega \circ \pi}(x) \right] \geqslant \mathbb{E}_{(x,\boldsymbol{a}) \sim d_{\widehat{\mathcal{P}},H}^{\tilde{\pi}}} \left[ \overline{Q}_{H,i}(x, \boldsymbol{a}) - \max_{\omega \in \Omega_i} Q_{H,i}^{\omega \circ \pi}(x, \boldsymbol{a}) \right]$$

$$= \mathbb{E}_{(x,\boldsymbol{a}) \sim d_{\widehat{\mathcal{P}},H}^{\tilde{\pi}}} \left[ \hat{b}_H(x, \boldsymbol{a}) \right]$$

$$\geqslant \mathbb{E}_{(x,\boldsymbol{a}) \sim d_{\widehat{\mathcal{P}},H}^{\tilde{\pi}}} \left[ \hat{b}_H(x, \boldsymbol{a}) - H \min \{ f_H(x, \boldsymbol{a}), 1 \} \right].$$

- Suppose the statement is true for step $h + 1$, then for step $h$, we have

$$\mathbb{E}_{x \sim d_{\widehat{\mathcal{P}}, h}^{\tilde{\pi}}} \left[ \overline{V}_{h,i}(x) - \max_{\omega \in \Omega_i} V_{h,i}^{\omega \circ \pi}(x) \right]$$

$$\geqslant \mathbb{E}_{(x,\boldsymbol{a}) \sim d_{\widehat{\mathcal{P}}_h}^{\tilde{\pi}}} \left[ \overline{Q}_{h,i}(x, \boldsymbol{a}) - \max_{\omega \in \Omega_i} Q_{h,i}^{\omega \circ \pi}(x, \boldsymbol{a}) \right]$$

$$= \mathbb{E}_{(x,\boldsymbol{a}) \sim d_{\widehat{\mathcal{P}}_h}^{\tilde{\pi}}} \left[ \hat{b}_h(x, \boldsymbol{a}) + \left( \widehat{\mathcal{P}}_h \overline{V}_{h+1,i} \right)(x, \boldsymbol{a}) - \left( \mathcal{P}_h \max_{\omega \in \Omega_i} V_{h+1,i}^{\omega \circ \pi} \right)(x, \boldsymbol{a}) \right]$$

$$= \mathbb{E}_{(x,\boldsymbol{a}) \sim d_{\widehat{\mathcal{P}}_h}^{\tilde{\pi}}} \left[ \hat{b}_h(x, \boldsymbol{a}) \right.$$

$$\left. + \left( \widehat{\mathcal{P}}_h \left( \overline{V}_{h+1,i} - \max_{\omega \in \Omega_i} V_{h+1,i}^{\omega \circ \pi} \right) \right)(x, \boldsymbol{a}) + \left( \left( \widehat{\mathcal{P}}_h - \mathcal{P}_h \right) \max_{\omega \in \Omega_i} V_{h+1,i}^{\omega \circ \pi} \right)(x, \boldsymbol{a}) \right]$$

$$= \mathbb{E}_{(x,\boldsymbol{a}) \sim d_{\widehat{\mathcal{P}}_h}^{\tilde{\pi}}} \left[ \hat{b}_h(x, \boldsymbol{a}) + \left( \left( \widehat{\mathcal{P}}_h - \mathcal{P}_h \right) \max_{\omega \in \Omega_i} V_{h+1,i}^{\omega \circ \pi} \right)(x, \boldsymbol{a}) \right]$$

$$+ \mathbb{E}_{x \sim d_{\widehat{\mathcal{P}}, h+1}^{\tilde{\pi}}} \left[ \overline{V}_{h+1,i}(x) - \max_{\omega \in \Omega_i} V_{h+1,i}^{\omega \circ \pi}(x) \right]$$

$$\geqslant \mathbb{E}_{(x,\boldsymbol{a}) \sim d_{\widehat{\mathcal{P}}_h}^{\tilde{\pi}}} \left[ \hat{b}_h(x, \boldsymbol{a}) - H \min \left\{ f_h(x, \boldsymbol{a}), 1 \right\} \right]$$

$$+ \mathbb{E}_{x \sim d_{\widehat{\mathcal{P}}, h+1}^{\tilde{\pi}}} \left[ \overline{V}_{h+1,i}(x) - \max_{\omega \in \Omega_i} V_{h+1,i}^{\omega \circ \pi}(x) \right]$$

$$\geqslant \sum_{h'=h}^{H} \mathbb{E}_{(x,\boldsymbol{a}) \sim d_{\widehat{\mathcal{P}}_h'}^{\tilde{\pi}}} \left[ \hat{b}_{h'}(x, \boldsymbol{a}) - H \min \left\{ f_{h'}(x, \boldsymbol{a}), 1 \right\} \right],$$

where we use the fact

$$\left| \left( \widehat{\mathcal{P}}_h - \mathcal{P}_h \right) \max_{\omega \in \Omega_i} V_{h+1,i}^{\omega \circ \pi} \right| (x, \boldsymbol{a})$$

$$\leqslant \min \left\{ H, \left\| \widehat{\mathcal{P}}_h(\cdot | x, \boldsymbol{a}) - \mathcal{P}_h(\cdot | x, \boldsymbol{a}) \right\|_1 \left\| \max_{\omega \in \Omega_i} V_{h+1,i}^{\omega \circ \pi} \right\|_\infty \right\}$$

$$\leqslant H \min \left\{ 1, \left\| \widehat{\mathcal{P}}_h(\cdot | x, \boldsymbol{a}) - \mathcal{P}_h(\cdot | x, \boldsymbol{a}) \right\|_1 \right\}$$

$$= H \min \left\{ 1, f_{h'}(x, \boldsymbol{a}) \right\}$$

and the last row uses the induction assumption.

Therefore, we have proved equation 29. We then apply $h = 0$ to equation 29, and get

$$\mathbb{E}_{x \sim d_0} \left[ \overline{V}_{0,i}(x) - \max_{\omega \in \Omega_i} V_{0,i}^{\omega \circ \pi}(x) \right]$$

$$= \mathbb{E}_{x \sim d_{\widehat{\mathcal{P}}, 0}^{\tilde{\pi}}} \left[ \overline{V}_{0,i}(x) - \max_{\omega \in \Omega_i} V_{0,i}^{\omega \circ \pi}(x) \right]$$

$$\geqslant \sum_{h=0}^{H} \mathbb{E}_{(x,\boldsymbol{a}) \sim d_{\widehat{\mathcal{P}}_h}^{\tilde{\pi}}} \left[ \hat{b}_h(x, \boldsymbol{a}) - H \min \left\{ f_h(x, \boldsymbol{a}), 1 \right\} \right]$$

$$= \sum_{h=0}^{H} \mathbb{E}_{(x,\boldsymbol{a}) \sim d_{\widehat{\mathcal{P}}_h}^{\tilde{\pi}}} \left[ \hat{b}_h(x, \boldsymbol{a}) \right] - H \sum_{h=0}^{H} \mathbb{E}_{(x,\boldsymbol{a}) \sim d_{\widehat{\mathcal{P}}_h}^{\tilde{\pi}}} \left[ \min \left\{ f_h(x, \boldsymbol{a}), 1 \right\} \right].$$

Next we are going to bound the second term. Applying Lemma 44 to $g_h(x, \boldsymbol{a}) = \min\{f_h(x, \boldsymbol{a}), 1\}$, we have

$$\sum_{h=0}^{H} \mathbb{E}_{(x,\boldsymbol{a})\sim d_{\widehat{\mathcal{P}}_h}^{\tilde{\pi}}} \left[\min\{f_h(x, \boldsymbol{a}), 1\}\right]$$

$$\leqslant \sum_{h=0}^{H} \mathbb{E}_{(x_{h-L}, \boldsymbol{a}_{h-L})\sim d_{\widehat{\mathcal{P}}_{h-L}}^{\tilde{\pi}}} \left[\|\hat{p}_{h-1}(\cdot|x_{h-L}, \boldsymbol{a}_{h-L})\|_{\Sigma_{\rho,\hat{p}}^{-1}}\right]$$

$$\cdot \sqrt{n\omega^L \cdot \mathbb{E}_{(\tilde{x}_h, \tilde{\boldsymbol{a}}_h)\sim\rho} \left[\min\{f_h(\tilde{x}_h, \tilde{\boldsymbol{a}}_h), 1\}^2\right] + \lambda C + nL\omega^{L-1}\zeta}$$

$$\leqslant \sum_{h=0}^{H} \mathbb{E}_{(x_{h-L}, \boldsymbol{a}_{h-L})\sim d_{\widehat{\mathcal{P}}_{h-L}}^{\tilde{\pi}}} \left[\|\alpha\hat{p}_{h-1}(\cdot|x_{h-L}, \boldsymbol{a}_{h-L})\|_{\Sigma_{\rho,\hat{p}}^{-1}}\right]$$

Note that we here use the fact $\min\{f_h(x, \boldsymbol{a}), 1\} \leqslant 1$, $\mathbb{E}_{(\tilde{x}_h, \tilde{\boldsymbol{a}}_h)\sim\rho} \left[\min\{f_h(\tilde{x}_h, \tilde{\boldsymbol{a}}_h), 1\}^2\right] \leqslant \zeta$ and our choice of $\alpha$.

Combining all things together,

$$\overline{v}_i(x) - \max_{\omega\in\Omega_i} v_i^{\omega\circ\pi}(x) = \mathbb{E}_{x\sim d_0} \left[\overline{V}_{0,i}(x) - \max_{\omega\in\Omega_i} V_{0,i}^{\omega\circ\pi}(x)\right]$$

$$\geqslant \sum_{h=0}^{H} \mathbb{E}_{(x,\boldsymbol{a})\sim d_{\widehat{\mathcal{P}},h}^{\tilde{\pi}}} \left[\hat{b}_h(x, \boldsymbol{a})\right] - H\sum_{h=0}^{H} \mathbb{E}_{(x,\boldsymbol{a})\sim d_{\widehat{\mathcal{P}},h}^{\tilde{\pi}}} \left[\min\{f_h(x, \boldsymbol{a}), 1\}\right]$$

$$\geqslant 0,$$

which proves the inequality. $\qquad\square$

**Lemma 47** (Pessimism). *Set*

$$\hat{b}_h = \min\left\{\alpha\|\hat{p}_n(\cdot|x_{h-L}, \boldsymbol{a}_{h-L})\|_{L_2(\mu), \hat{\Sigma}_{n,h,\hat{p}}^{-1}}, H\right\},$$

*with $\alpha = \Theta\sqrt{\lambda C + nL\omega^L\zeta}$,*

$$\hat{\Sigma}_{n,\hat{p}} : L_2(\mu) \to L_2(\mu), \quad \hat{\Sigma}_{n,\hat{p}} := \sum_{(x_i, \boldsymbol{a}_i)\in\mathcal{D}} \left[\hat{p}(z|x_i, \boldsymbol{a}_i)\hat{p}(z|x_i, \boldsymbol{a}_i)^\top\right] + \lambda T_K^{-1}$$

*where $T_K$ is the integral operator associated with $K$ (i.e. $T_K f = \int f(x)K(x,\cdot)dx$) and $\lambda$ is set for different eigendecay of $K$ as follows:*

- *$\beta$-finite spectrum: $\lambda = \Theta(\beta\log n + \log(|\mathcal{M}|/\delta))$*

- *$\beta$-polynomial decay: $\lambda = \Theta(C_{\text{poly}} n^{1/(1+\beta)} + \log(|\mathcal{M}|/\delta))$;*

- *$\beta$-exponential decay: $\lambda = \Theta(C_{\exp}(\log n)^{1/\beta} + \log(|\mathcal{M}|/\delta))$;*

*$c$ is an absolute constant. Then with probability at least $1 - \delta$, $\forall i \in [M]$ we have*

$$\underline{v}_i(x) - v_i^\pi(x) \leqslant 0, \quad \forall i \in [M].$$

*Proof.* Let $f_h(x, \boldsymbol{a}) = \left\|\widehat{\mathcal{P}}_h(\cdot|x, \boldsymbol{a}) - P_h(\cdot|x, \boldsymbol{a})\right\|_1$, then according to lemma 51 and lemma 56, we have that using the chosen $\lambda$, with probability at least $1 - \delta$,

$$\mathbb{E}_{(x,\boldsymbol{a})\sim\rho} \left[(f_h(x, \boldsymbol{a}))^2\right] \leqslant \zeta, \quad \forall h \in [H],$$

$$\|p_h(\cdot|x, \boldsymbol{a})\|_{\hat{\Sigma}_{n,\hat{p}}^{-1}} = \Theta\left(\|p_h(\cdot|x, \boldsymbol{a})\|_{\hat{\Sigma}_{\rho,\hat{p}}^{-1}}\right), \quad \forall h \in [H], \widehat{\mathcal{P}} \in \mathcal{M}.$$

A direct conclusion is we can find an absolute constant $c$, such that

$$\hat{b}_h(x_h, \boldsymbol{a}_h) = \min\left\{\alpha \left\|\hat{p}_h(\cdot|x_{h-L}, \boldsymbol{a}_{h-L})\right\|_{\Sigma_{n,\hat{p}}^{-1}}, H\right\}$$

$$\geqslant \min\left\{c\alpha \left\|\hat{p}_h(\cdot|x_{h-L}, \boldsymbol{a}_{h-L})\right\|_{\Sigma_{n,\hat{p}}^{-1}}, H\right\}, \quad \forall h \in [H].$$

Again, we prove by induction that

$$\mathbb{E}_{x \sim d_{\widehat{\mathcal{P}},h}^{\tilde{\pi}}}\left[\underline{V}_{h,i}(x) - V_{h,i}^\pi(x)\right] \leqslant \sum_{h'=h}^{H} \mathbb{E}_{(x,\boldsymbol{a}) \sim d_{\widehat{\mathcal{P}},h'}^{\tilde{\pi}}}\left[-\hat{b}_{h'}(x, \boldsymbol{a}) + H\min\left\{f_{h'}(x, \boldsymbol{a}), 1\right\}\right], \forall h \in [H].$$
$$(30)$$

- When $h = H$, we have

$$\mathbb{E}_{x \sim d_{\widehat{\mathcal{P}},H}^{\tilde{\pi}}}\left[\underline{V}_{H,i}(x) - V_{H,i}^\pi(x)\right] = \mathbb{E}_{(x,\boldsymbol{a}) \sim d_{\widehat{\mathcal{P}},H}^{\tilde{\pi}}}\left[\underline{Q}_{H,i}(x, \boldsymbol{a}) - Q_{H,i}^\pi(x, \boldsymbol{a})\right]$$

$$= \mathbb{E}_{(x,\boldsymbol{a}) \sim d_{\widehat{\mathcal{P}},H}^{\tilde{\pi}}}\left[-\hat{b}_h(x, \boldsymbol{a})\right]$$

$$\leqslant \mathbb{E}_{(x,\boldsymbol{a}) \sim d_{\widehat{\mathcal{P}},H}^{\tilde{\pi}}}\left[-\hat{b}_h(x, \boldsymbol{a}) + H\min\left\{f_H(x, \boldsymbol{a}), 1\right\}\right].$$

- Suppose the statement is true for step $h + 1$, then for step $h$, we have

$$\mathbb{E}_{x \sim d_{\widehat{\mathcal{P}},h}^{\tilde{\pi}}}\left[\underline{V}_{h,i}(x) - V_{h,i}^\pi(x)\right]$$

$$= \mathbb{E}_{(x,\boldsymbol{a}) \sim d_{\widehat{\mathcal{P}},h}^{\tilde{\pi}}}\left[\underline{Q}_{h,i}(x, \boldsymbol{a}) - Q_{h,i}^\pi(x, \boldsymbol{a})\right]$$

$$= \mathbb{E}_{(x,\boldsymbol{a}) \sim d_{\widehat{\mathcal{P}},h}^{\tilde{\pi}}}\left[-\hat{b}_h(x, \boldsymbol{a}) + \left(\widehat{\mathcal{P}}_h \underline{V}_{h+1,i}\right)(x, \boldsymbol{a}) - \left(\mathcal{P}_h V_{h+1,i}^\pi\right)(x, \boldsymbol{a})\right]$$

$$= \mathbb{E}_{(x,\boldsymbol{a}) \sim d_{\widehat{\mathcal{P}},h}^{\tilde{\pi}}}\left[-\hat{b}_h(x, \boldsymbol{a}) + \left(\widehat{\mathcal{P}}_h \left(\underline{V}_{h+1,i} - V_{h+1,i}^\pi\right)\right)(x, \boldsymbol{a}) + \left(\left(\widehat{\mathcal{P}}_h - \mathcal{P}_h\right) V_{h+1,i}^\pi\right)(x, \boldsymbol{a})\right]$$

$$= \mathbb{E}_{(x,\boldsymbol{a}) \sim d_{\widehat{\mathcal{P}},h}^{\tilde{\pi}}}\left[-\hat{b}_h(x, \boldsymbol{a}) + \left(\left(\widehat{\mathcal{P}}_h - \mathcal{P}_h\right) V_{h+1,i}^\pi\right)(x, \boldsymbol{a})\right]$$

$$+ \mathbb{E}_{x \sim d_{\widehat{\mathcal{P}},h+1}^{\tilde{\pi}}}\left[\underline{V}_{h+1,i}(x) - V_{h+1,i}^\pi(x)\right]$$

$$\geqslant \mathbb{E}_{(x,\boldsymbol{a}) \sim d_{\widehat{\mathcal{P}},h}^{\tilde{\pi}}}\left[-\hat{b}_h(x, \boldsymbol{a}) - H\min\left\{f_h(x, \boldsymbol{a}), 1\right\}\right] + \mathbb{E}_{x \sim d_{\widehat{\mathcal{P}},h+1}^{\tilde{\pi}}}\left[\underline{V}_{h+1,i}(x) - V_{h+1,i}^\pi(x)\right]$$

$$\geqslant \sum_{h'=h}^{H} \mathbb{E}_{(x,\boldsymbol{a}) \sim d_{\widehat{\mathcal{P}},h'}^{\tilde{\pi}}}\left[-\hat{b}_{h'}(x, \boldsymbol{a}) + H\min\left\{f_{h'}(x, \boldsymbol{a}), 1\right\}\right],$$

where we use the fact

$$\left|\left(\widehat{\mathcal{P}}_h - \mathcal{P}_h\right) V_{h+1,i}^\pi\right|(x, \boldsymbol{a}) \leqslant \min\left\{H, \left\|\widehat{\mathcal{P}}_h(\cdot|x, \boldsymbol{a}) - \mathcal{P}_h(\cdot|x, \boldsymbol{a})\right\|_1 \left\|V_{h+1,i}^\pi\right\|_\infty\right\}$$

$$\leqslant H\min\left\{1, \left\|\widehat{\mathcal{P}}_h(\cdot|x, \boldsymbol{a}) - \mathcal{P}_h(\cdot|x, \boldsymbol{a})\right\|_1\right\}$$

$$= H\min\left\{1, f_{h'}(x, \boldsymbol{a})\right\}$$

and the last row uses the induction assumption.

Therefore, we have proved equation 30. We then apply $h = 0$ to equation 30, and get

$$\mathbb{E}_{x \sim d_0}\left[\overline{V}_{0,i}^n(x) - V_{0,i}^{\dagger,\pi-i}(x)\right]$$

$$= \mathbb{E}_{x \sim d_{\widehat{\mathcal{P}},0}^{\tilde{\pi}}}\left[\overline{V}_{0,i}(x) - V_{0,i}^{\dagger,\pi-i}(x)\right]$$

$$\geqslant \sum_{h=0}^{H} \mathbb{E}_{(x,\boldsymbol{a}) \sim d_{\widehat{\mathcal{P}}_h}^{\tilde{\pi}}}\left[\hat{b}_h(x, \boldsymbol{a}) - H\min\left\{f_h(x, \boldsymbol{a}), 1\right\}\right]$$

$$= \sum_{h=0}^{H} \mathbb{E}_{(x,\boldsymbol{a}) \sim d_{\widehat{\mathcal{P}},h}^{\tilde{\pi}}}\left[\hat{b}_h(x, \boldsymbol{a})\right] - H\sum_{h=0}^{H} \mathbb{E}_{(x,\boldsymbol{a}) \sim d_{\widehat{\mathcal{P}}_h}^{\tilde{\pi}}}\left[\min\left\{f_h(x, \boldsymbol{a}), 1\right\}\right].$$

Next we are going to bound the second term. Applying Lemma 44 to $g_h(x, \boldsymbol{a}) = \min\{f_h(x, \boldsymbol{a}), 1\}$, we have

$$\sum_{h=0}^{H} \mathbb{E}_{(x,\boldsymbol{a}) \sim d_{\hat{\mathcal{P}}_h}^{\tilde{\pi}}} \left[\min\{f_h(x, \boldsymbol{a}), 1\}\right]$$

$$\leqslant \sum_{h=0}^{H} \mathbb{E}_{(x_{h-L}, \boldsymbol{a}_{h-L}) \sim d_{\hat{\mathcal{P}}_{h-L}}^{\tilde{\pi}}} \left[\|\hat{p}_{h-1}(\cdot|x_{h-L}, \boldsymbol{a}_{h-L})\|_{\Sigma_{\rho,\hat{p}}^{-1}}\right]$$

$$\cdot \sqrt{n\omega^L \cdot \mathbb{E}_{(\tilde{x}_h, \tilde{\boldsymbol{a}}_h) \sim \rho}\left[\min\{f_h(\tilde{x}_h, \tilde{\boldsymbol{a}}_h), 1\}^2\right] + \lambda C + nL\omega^{L-1}\zeta}$$

$$\leqslant \sum_{h=0}^{H} \mathbb{E}_{(x_{h-L}, \boldsymbol{a}_{h-L}) \sim d_{\hat{\mathcal{P}}_{h-L}}^{\tilde{\pi}}} \left[\|\alpha \hat{p}_{h-1}(\cdot|x_{h-L}, \boldsymbol{a}_{h-L})\|_{\Sigma_{\rho,\hat{p}}^{-1}}\right]$$

Note that we here use the fact $\min\{f_h(x, \boldsymbol{a}), 1\} \leqslant 1$, $\mathbb{E}_{(\tilde{x}_h, \tilde{\boldsymbol{a}}_h) \sim \rho}\left[\min\{f_h(\tilde{x}_h, \tilde{\boldsymbol{a}}_h), 1\}^2\right] \leqslant \zeta$ and our choice of $\alpha$.

Combining all things together,

$$\begin{aligned}
\underline{v}_i - v_i^{\pi} &= \mathbb{E}_{x \sim d_0}\left[\underline{V}_{0,i}(x) - V_{0,i}^{\pi}(x)\right] \\
&\leqslant \sum_{h=0}^{H} \mathbb{E}_{(x,\boldsymbol{a}) \sim d_{\hat{\mathcal{P}}_h}^{\tilde{\pi}}} \left[-\hat{b}_h(x, \boldsymbol{a})\right] + H \sum_{h=0}^{H} \mathbb{E}_{(x,\boldsymbol{a}) \sim d_{\hat{\mathcal{P}}_h}^{\tilde{\pi}}} \left[\min\{f_h(x, \boldsymbol{a}), 1\}\right] \\
&\leqslant 0,
\end{aligned}$$

which has finished the proof. $\qquad\square$

**Lemma 48.** *Set*

$$\hat{b}_h = \min\left\{\alpha\|\hat{p}(\cdot|x_{h-L}, \boldsymbol{a}_{h-L})\|_{L_2(\mu), \hat{\Sigma}_{n,\hat{p}}^{-1}}, H\right\},$$

*with $\alpha = \Theta\sqrt{\lambda C + nL\omega^L\zeta}$,*

$$\hat{\Sigma}_{n,\hat{p}} : L_2(\mu) \to L_2(\mu), \quad \hat{\Sigma}_{n,\hat{p}} := \sum_{(x_i, \boldsymbol{a}_i) \in \mathcal{D}} \left[\hat{p}(z|x_i, \boldsymbol{a}_i)\hat{p}(z|x_i, \boldsymbol{a}_i)^{\top}\right] + \lambda T_K^{-1}$$

*where $T_K$ is the integral operator associated with $K$ (i.e. $T_K f = \int f(x)K(x, \cdot)dx$) and $\lambda$ is set for different eigendecay of $K$ as follows:*

- *$\beta$-finite spectrum: $\lambda = \Theta(\beta \log n + \log(|\mathcal{M}|/\delta))$*

- *$\beta$-polynomial decay: $\lambda = \Theta(C_{\text{poly}} n^{1/(1+\beta)} + \log(|\mathcal{M}|/\delta))$;*

- *$\beta$-exponential decay: $\lambda = \Theta(C_{\text{exp}}(\log n)^{1/\beta} + \log(|\mathcal{M}|/\delta))$;*

*$c$ is an absolute constant. Then with probability at least $1 - \delta$, $\forall i \in [M]$ we have*

- *for $\beta$-finite spectrum,*

$$\Delta \lesssim \mathcal{O}\left(H^3 \beta \log n \sqrt{C_{\tilde{\pi}}^{\star} n\omega^L C\zeta \log \frac{|\mathcal{M}|}{\delta}}\right)$$

- *for $\beta$-polynomial decay,*

$$\Delta \lesssim \mathcal{O}\left(H^3 C_{\text{poly}} n^{\frac{1}{2(1+\beta)}} \log n \sqrt{C_{\tilde{\pi}}^{\star} n\omega^L C\zeta \log \frac{|\mathcal{M}|}{\delta}}\right)$$

- *for $\beta$-exponential decay,*

$$\Delta \lesssim \mathcal{O}\left(H^3 C_{\text{exp}}(\log n)^{1+1/\beta} \sqrt{C_{\tilde{\pi}}^{\star} n\omega^L C\zeta \log \frac{|\mathcal{M}|}{\delta}}\right)$$

*Proof.* Let $f_h(x, \boldsymbol{a}) = \left\| \widehat{\mathcal{P}}_h(\cdot|s, \boldsymbol{a}) - P_h(\cdot|x, \boldsymbol{a}) \right\|_1$. With our choice of $\lambda$ and $\zeta$, according to Lemma 56, we have

$$\mathbb{E}_{x \sim \rho} \left[ (f_h(x, \boldsymbol{a}))^2 \right] \leqslant \zeta, \|p_h(\cdot|x, \boldsymbol{a})\|_{\hat{\Sigma}_{n, \phi_h}^{-1}} = \Theta \left( \|p_h(\cdot|x, \boldsymbol{a})\|_{\Sigma_{\rho, \phi_h}^{-1}} \right), \quad \forall h \in [H], \widehat{\mathcal{P}} \in \mathcal{M}. \tag{31}$$

By definition, we have

$$\Delta = \max_{i \in [M]} \{ \overline{v}_i - \underline{v}_i \}.$$

For each fixed $i \in [M], h \in [H]$, we have

$$\mathbb{E}_{x \sim d_{\mathcal{P}, h}^\pi} \left[ \overline{V}_{h,i}(x) - \underline{V}_{h,i}(x) \right]$$
$$= \mathbb{E}_{x \sim d_{\mathcal{P}, h}^\pi} \left[ \left( \mathbb{D}_{\pi_h} \overline{Q}_{h,i} \right)(x) - \left( \mathbb{D}_{\pi_h} \underline{Q}_{h,i} \right)(x) \right]$$
$$= \mathbb{E}_{(x, \boldsymbol{a}) \sim d_{\mathcal{P}, h}^\pi} \left[ \overline{Q}_{h,i}(x, \boldsymbol{a}) - \underline{Q}_{h,i}(x, \boldsymbol{a}) \right]$$
$$= \mathbb{E}_{(x, \boldsymbol{a}) \sim d_{\mathcal{P}, h}^\pi} \left[ 2\hat{b}_h(x, \boldsymbol{a}) + \left( \widehat{\mathcal{P}}_h \left( \overline{V}_{h+1,i} - \underline{V}_{h+1,i} \right) \right)(x, \boldsymbol{a}) \right]$$
$$= \mathbb{E}_{(x, \boldsymbol{a}) \sim d_{\mathcal{P}, h}^\pi} \left[ 2\hat{b}_h(x, \boldsymbol{a}) + \left( \left( \widehat{\mathcal{P}}_h - \mathcal{P}_h \right) \left( \overline{V}_{h+1,i} - \underline{V}_{h+1,i} \right) \right)(x, \boldsymbol{a}) \right]$$
$$+ \mathbb{E}_{x \sim d_{\mathcal{P}, h+1}^\pi} \left[ \overline{V}_{h+1,i}(x) - \underline{V}_{h+1,i}(x) \right]$$
$$\leqslant \mathbb{E}_{(x, \boldsymbol{a}) \sim d_{\mathcal{P}, h}^\pi} \left[ 2\hat{b}_h(x, \boldsymbol{a}) + 2H^2 f_h(x, \boldsymbol{a}) \right] + \mathbb{E}_{x \sim d_{\mathcal{P}, h+1}^\pi} \left[ \overline{V}_{h+1,i}(x) - \underline{V}_{h+1,i}(x) \right].$$

Note that we use the fact $\overline{V}_{h+1,i}(x) - \underline{V}_{h+1,i}(x)$ is upper bounded by $2H^2$, which can be proved easily using induction using the fact that $\hat{b}_h(x, \boldsymbol{a}) \leqslant H$. Applying the above formula recursively to $\mathbb{E}_{x \sim d_{\mathcal{P}, h+1}^\pi} \left[ \overline{V}_{h+1,i}(x) - \underline{V}_{h+1,i}(x) \right]$, one gets the following result (or more formally, one can prove by induction, just like what we did in Lemma 45, Lemma 46 and Lemma 47):

$$\mathbb{E}_{x \sim d_{\mathcal{P}, 0}^\pi} \left[ \overline{V}_{0,i}(x) - \underline{V}_{0,i}(x) \right] \leqslant 2 \underbrace{\sum_{h=0}^H \mathbb{E}_{(x, \boldsymbol{a}) \sim d_{\mathcal{P}, h}^\pi} \left[ \hat{b}_h(x, \boldsymbol{a}) \right]}_{(a)} + 2H^2 \underbrace{\sum_{h=0}^H \mathbb{E}_{(x, \boldsymbol{a}) \sim d_{\mathcal{P}, h}^\pi} \left[ f_h(x, \boldsymbol{a}) \right]}_{(b)}. \tag{32}$$

First, we calculate the first term (a) in Inequality equation 32. Following Lemma 43 and noting the bonus $\hat{b}_h$ is $O(H)$, we have

$$\sum_{h=0}^H \mathbb{E}_{(x, \boldsymbol{a}) \sim d_{\mathcal{P}, h}^\pi} \left[ \hat{b}_h(x, \boldsymbol{a}) \right]$$
$$\lesssim \sum_{h=0}^H \mathbb{E}_{(x, \boldsymbol{a}) \sim d_{\mathcal{P}, h}^\pi} \left[ \min \left\{ \alpha \left\| \hat{b}_h(\cdot|x_{h-L}, \boldsymbol{a}_{h-L}) \right\|_{\Sigma_{\rho, \hat{p}}^{-1}}, H \right\} \right] \qquad \text{(From equation 31)}$$
$$\lesssim \sum_{h=0}^{H-1} \mathbb{E}_{(x_{h-L}, \boldsymbol{a}_{h-L}) \sim d_{\mathcal{P}, h-L}^\pi} \left[ \|p_h(\cdot|x_{h-L}, \boldsymbol{a}_{h-L})\|_{\Sigma_{\rho, p}^{-1}} \right]$$
$$\cdot \sqrt{n\omega^L \cdot \mathbb{E}_{(\tilde{x}_h, \tilde{\boldsymbol{a}}_h) \sim \rho} \left[ (\hat{b}_h(\tilde{x}_h, \tilde{\boldsymbol{a}}_h))^2 \right]} + \lambda H^2 C.$$

Note that we use the fact that $B = H$ when applying Lemma 43. In addition, following the proof of Lemma 8 in (41), we have that

- for $\beta$-finite spectrum,

$$n\mathbb{E}_{(\tilde{x}_h,\tilde{\boldsymbol{a}}_h)\sim\rho}\left[\widehat{b}_h(\tilde{x}_h,\tilde{\boldsymbol{a}}_h)^2\right] = O(\beta\log n);$$

$$\mathbb{E}_{(x_{h-L},\boldsymbol{a}_{h-L})\sim\rho}\left[\|p_h(\cdot|x_{h-L},\boldsymbol{a}_{h-L})\|_{L_2(\mu),\Sigma_{\rho,p}^{-1}}^2\right] = O(\beta\log n);$$

- for $\beta$-polynomial decay,

$$n\mathbb{E}_{(\tilde{x}_h,\tilde{\boldsymbol{a}}_h)\sim\rho}\left[\widehat{b}_h(\tilde{x}_h,\tilde{\boldsymbol{a}}_h)^2\right] = O\left(C_{\text{poly}}n^{\frac{1}{2(1+\beta)}}\log n\right);$$

$$\mathbb{E}_{(x_{h-L},\boldsymbol{a}_{h-L})\sim\rho}\left[\|p_h(\cdot|x_{h-L},\boldsymbol{a}_{h-L})\|_{L_2(\mu),\Sigma_{\rho,p}^{-1}}^2\right] = O\left(C_{\text{poly}}n^{\frac{1}{2(1+\beta)}}\log n\right);$$

- for $\beta$-exponential decay,

$$n\mathbb{E}_{(\tilde{x}_h,\tilde{\boldsymbol{a}}_h)\sim\rho}\left[\widehat{b}_h(\tilde{x}_h,\tilde{\boldsymbol{a}}_h)^2\right] = O\left(C_{\text{exp}}(\log n)^{1+1/\beta}\right).$$

$$\mathbb{E}_{(x_{h-L},\boldsymbol{a}_{h-L})\sim\rho}\left[\|p_h(\cdot|x_{h-L},\boldsymbol{a}_{h-L})\|_{L_2(\mu),\Sigma_{\rho,p}^{-1}}^2\right] = O\left(C_{\text{exp}}(\log n)^{1+1/\beta}\right).$$

Moreover, according to lemma 42, we know

$$\mathbb{E}_{(x_{h-L},\boldsymbol{a}_{h-L})\sim d_{P,h-L}^\pi}\left[\|p_h(\cdot|x_{h-L},\boldsymbol{a}_{h-L})\|_{L_2(\mu),\Sigma_{\rho,p}^{-1}}\right]$$

$$\leqslant\sqrt{\mathbb{E}_{(x_{h-L},\boldsymbol{a}_{h-L})\sim d_{P,h-L}^\pi}\left[\|p_h(\cdot|x_{h-L},\boldsymbol{a}_{h-L})\|_{L_2(\mu),\Sigma_{\rho,p}^{-1}}^2\right]}$$

$$\leqslant\sqrt{C_\pi^\star\mathbb{E}_{(x_{h-L},\boldsymbol{a}_{h-L})\sim\rho}\left[\|p_h(\cdot|x_{h-L},\boldsymbol{a}_{h-L})\|_{L_2(\mu),\Sigma_{\rho,p}^{-1}}^2\right]}.$$

Second, we calculate the term (b) in inequality equation 32. Following Lemma 43 and noting that $f_h(x,\boldsymbol{a})$ is upper-bounded by 2 (i.e., $B=2$ in Lemma 43), we have

$$\sum_{h=0}^{H}\mathbb{E}_{(x,\boldsymbol{a})\sim d_{P,h}^\pi}[f_h(x,\boldsymbol{a})]$$

$$\leqslant\sum_{h=0}^{H-1}\mathbb{E}_{(x_{h-L},\boldsymbol{a}_{h-L})\sim d_{P,h-L}^\pi}\left[\|p_h(\cdot|x_{h-L},\boldsymbol{a}_{h-L})\|_{\Sigma_{\rho,p}^{-1}}\right]\sqrt{n\omega^L\mathbb{E}_{(\tilde{x}_h,\tilde{\boldsymbol{a}}_h)\sim\rho}\left[(f_h(\tilde{x}_h,\tilde{\boldsymbol{a}}_h))^2\right]+4C\lambda}$$

$$\leqslant\sum_{h=0}^{H-1}\mathbb{E}_{(x_{h-L},\boldsymbol{a}_{h-L})\sim d_{P,h-L}^\pi}\left[\|p_h(\cdot|x_{h-L},\boldsymbol{a}_{h-L})\|_{\Sigma_{\rho,p}^{-1}}\right]\sqrt{n\omega^L\zeta+4C\lambda},$$

where in the second inequality, we use $\mathbb{E}_{(\tilde{x}_h,\tilde{\boldsymbol{a}}_h)\sim\rho}\left[(f_h(\tilde{x}_h,\tilde{\boldsymbol{a}}_h))^2\right]\leqslant\zeta$.

Then, by combining the above calculation of the term (a) and term (b) in inequality equation 32, we have:

- for $\beta$-finite spectrum,

$$\Delta\lesssim\mathcal{O}\left(\sqrt{C_\pi^\star\beta\log n}\cdot(H\sqrt{\omega^L\beta\log n}+\lambda H^2 C+H^3\sqrt{n\omega^L\zeta+4C\lambda})\right)$$

- for $\beta$-polynomial decay,

$$\Delta\lesssim\mathcal{O}\left(\sqrt{C_\pi^\star C_{\text{poly}}n^{\frac{1}{2(1+\beta)}}\log n}\right.$$

$$\left.\cdot\left(H\sqrt{\omega^L C_{\text{poly}}n^{\frac{1}{2(1+\beta)}}\log n}+\lambda H^2 C+H^3\sqrt{n\omega^L\zeta+4C\lambda}\right)\right)$$

- for $\beta$-exponential decay,

$$\Delta \lesssim \mathcal{O}\left(\sqrt{C_{\hat{\pi}}^{\star} C_{\exp}(\log n)^{1+1/\beta}}\right.$$
$$\left. \cdot \left(H\sqrt{\omega^L C_{\exp}(\log n)^{1+1/\beta} + \lambda H^2 C} + H^3\sqrt{n\omega^L \zeta + 4C\lambda}\right)\right)$$

By substituting $\lambda$ into the results, we obtain:

- for $\beta$-finite spectrum,

$$\Delta \lesssim \mathcal{O}\left(H^3 \beta \log n \sqrt{C_{\hat{\pi}}^{\star} n\omega^L C\zeta \log \frac{|\mathcal{M}|}{\delta}}\right)$$

- for $\beta$-polynomial decay,

$$\Delta \lesssim \mathcal{O}\left(H^3 C_{\text{poly}} n^{\frac{1}{2(1+\beta)}} \log n \sqrt{C_{\hat{\pi}}^{\star} n\omega^L C\zeta \log \frac{|\mathcal{M}|}{\delta}}\right)$$

- for $\beta$-exponential decay,

$$\Delta \lesssim \mathcal{O}\left(H^3 C_{\exp}(\log n)^{1+1/\beta} \sqrt{C_{\hat{\pi}}^{\star} n\omega^L C\zeta \log \frac{|\mathcal{M}|}{\delta}}\right)$$

This concludes the proof. $\qquad\square$

**Theorem 49** (PAC guarantee of Algorithm 5). *When Alg. 5 is applied with parameters $\zeta = \Theta\left(\log(H|\mathcal{M}|/\delta)/n\right), \hat{b}_h = \min\left\{\alpha\|\hat{p}(\cdot|x_{h-L}, a_{h-L})\|_{L_2(\mu), \hat{\Sigma}_{n,\hat{p}}^{-1}}, H\right\}$ with $\alpha = \Theta\sqrt{\lambda C + nL\omega^{L-1}\zeta}$ and*

- *$\beta$-finite spectrum: $\lambda = \Theta(\beta \log n + \log(|\mathcal{M}|/\delta))$;*

- *$\beta$-polynomial decay: $\lambda = \Theta(C_{\text{poly}} n^{1/(1+\beta)} + \log(|\mathcal{M}|/\delta))$;*

- *$\beta$-exponential decay: $\lambda = \Theta(C_{\exp}(\log n)^{1/\beta} + \log(|\mathcal{M}|/\delta))$;*

*with probability $1 - \delta$, the output policy $\hat{\pi}$ is an $\varepsilon$-approximate $\{\text{NE}, \text{CCE}, \text{CE}\}$ with*

- *for $\beta$-finite spectrum,*

$$\varepsilon = \mathcal{O}\left(H^3 \beta \log n \sqrt{C_{\hat{\pi}}^{\star} n\omega^L C\zeta \log \frac{|\mathcal{M}|}{\delta}}\right)$$

- *for $\beta$-polynomial decay,*

$$\varepsilon = \mathcal{O}\left(H^3 C_{\text{poly}} n^{\frac{1}{2(1+\beta)}} \log n \sqrt{C_{\hat{\pi}}^{\star} n\omega^L C\zeta \log \frac{|\mathcal{M}|}{\delta}}\right)$$

- *for $\beta$-exponential decay,*

$$\varepsilon = \mathcal{O}\left(H^3 C_{\exp}(\log n)^{1+1/\beta} \sqrt{C_{\hat{\pi}}^{\star} n\omega^L C\zeta \log \frac{|\mathcal{M}|}{\delta}}\right)$$

*Proof.* For any agent $i$, by Lemma 45, Lemma 46 and Lemma 47, we have

$$v_i^{\dagger, \pi_{-i}} - v_i^{\pi} \left(\text{or } \max_{\omega \in \Omega_i} v_i^{\omega \circ \pi} - v_i^{\pi}\right) \leqslant \overline{v}_i - \underline{v}_i \leqslant \Delta.$$

Taking maximum over $i$ on both sides, we have

$$\max_{i \in [M]} \left\{ v_i^{\dagger, \pi_{-i}} - v_i^\pi \right\} \left( \text{or } \max_{i \in [M]} \left\{ \max_{\omega \in \Omega_i} v_i^{\omega \circ \pi} - v_i^\pi \right\} \right) \leqslant \Delta. \tag{33}$$

From Lemma 48, with probability $1 - \delta$, we can ensure the output policy $\hat{\pi}$ is an $\varepsilon$-approximate $\{\text{NE}, \text{CCE}, \text{CE}\}$ with

- for $\beta$-finite spectrum,

$$\varepsilon = \mathcal{O}\left( H^3 \beta \log n \sqrt{C_{\hat{\pi}}^\star n \omega^L C \zeta \log \frac{|\mathcal{M}|}{\delta}} \right)$$

- for $\beta$-polynomial decay,

$$\varepsilon = \mathcal{O}\left( H^3 C_{\text{poly}} n^{\frac{1}{2(1+\beta)}} \log n \sqrt{C_{\hat{\pi}}^\star n \omega^L C \zeta \log \frac{|\mathcal{M}|}{\delta}} \right)$$

- for $\beta$-exponential decay,

$$\varepsilon = \mathcal{O}\left( H^3 C_{\text{exp}} (\log n)^{1+1/\beta} \sqrt{C_{\hat{\pi}}^\star n \omega^L C \zeta \log \frac{|\mathcal{M}|}{\delta}} \right)$$

which has finished the proof. $\qquad\qquad\qquad\qquad\qquad\qquad\qquad\qquad\qquad\qquad\qquad\qquad \square$

## F   DERIVATION AND OPTIMIZATION OF ELBO

This section presents the derivation of the Evidence Lower Bound (ELBO) as a tractable surrogate objective for maximum likelihood estimation (MLE), followed by an analysis of its computational advantages over direct MLE optimization.

We begin with the ELBO derivation through variational calculus:

$$\log \mathbb{P}^{\chi_\pi}(\boldsymbol{o}_{h+1:h+l} | \tau_h^L, \boldsymbol{a}_h)$$

$$= \log \int_{\mathcal{Z}} p(z_h | \tau_h^L, \boldsymbol{a}_h) \mathbb{P}^{\chi_\pi}(\boldsymbol{o}_{h+1:h+l} | z_h) \, dz_h$$

$$= \log \int_{\mathcal{Z}} \frac{p(z_h | \tau_h^L, \boldsymbol{a}_h) \mathbb{P}^{\chi_\pi}(\boldsymbol{o}_{h+1:h+l} | z_h)}{q(z | \tau_h^L, \boldsymbol{a}_h, \boldsymbol{o}_{h+1:h+l})} \cdot q(z | \tau_h^L, \boldsymbol{a}_h, \boldsymbol{o}_{h+1:h+l}) \, dz_h \tag{34}$$

$$= \max_{q \in \Delta(\mathcal{Z})} \mathbb{E}_{q(\cdot | \tau_h^L, \boldsymbol{a}_h, \boldsymbol{o}_{h+1:h+l})} \left[ \log \mathbb{P}^{\chi_\pi}(\boldsymbol{o}_{h+1:h+l} | z_h) \right] - KL(q(z_h | \tau_h^L, \boldsymbol{a}_h, \boldsymbol{o}_{h+1:h+l}) || p(z_h | \tau_h^L, \boldsymbol{a}_h))$$

where the last equation comes from Jensen's inequality, with equality holding when $q(z | \tau_h^L, \boldsymbol{a}_h, \boldsymbol{o}_{h+1:h+l}) \propto p(z_h | \tau_h^L, \boldsymbol{a}_h) \mathbb{P}^{\chi_\pi}(\boldsymbol{o}_{h+1:h+l} | z_h)$. Notably, under Assumption 4, the $q(z | \tau_h^L, \boldsymbol{a}_h, \boldsymbol{o}_{h+1:h+l}) \in \mathcal{Q}$ for all $(p(z_h | \tau_h^L, \boldsymbol{a}_h), \mathbb{P}^{\chi_\pi}(\boldsymbol{o}_{h+1:h+l} | z_h)) \in \mathcal{M}$, so the equality always holds and the estimator obtained by maximizing the ELBO is identical to the estimator obtained by MLE.

Compared to the standard MLE objective, maximizing the ELBO is computationally efficient because it avoids the need to compute integrals explicitly. Instead, the ELBO only requires evaluating an expectation and a KL divergence term, both of which can be approximated efficiently via sampling.

Note that the ELBO objective

$$\mathbb{E}_{q(z | \tau_h^L, a_h, o_{h+1:h+l})} [\log \mathbb{P}^{\chi_\pi}(o_{h+1:h+l} | z_h)] - KL((q(z | \tau_h^L, a_h, o_{h+1:h+l}) || p(z_h | \tau_h^L, a_h))$$

$$= - KL(q(z | \tau_h^L, a_h, o_{h+1:h+l}) || \mathbb{P}^{\chi_\pi}(o_{h+1:h+l} | z_h) p(z_h | \tau_h^L, a_h)).$$

---

**Algorithm 6** OVI-OF with Generalized PSRs

---

1: **Input:** Regularizer $\lambda$, iteration $N$, parameter $\{\alpha_n\}_{n=1}^N$, $\{\zeta_n\}_{n=1}^N$, and model class $\mathcal{M} = \{\{(p_h(\cdot|x_h, \boldsymbol{a}_h), p_h(\boldsymbol{o}_{h+1}|\cdot))\}_{h\in[H]}\}$.

2: Initialize $\pi^0$ to be uniform; set datasets $\mathcal{D}_h^0 = \emptyset, \tilde{\mathcal{D}}_h^0 = \emptyset, \forall h \in [H]$.

3: **for** episode $n = 1, 2, \cdots, N$ **do**

4:    Set $\overline{V}_{H+1,i}^n = 0, \underline{V}_{H+1,i}^n = 0$ for all $i \in [M]$.

5:    Sample trajectory and learn representation like Algorithm 2 in (12).

6:    **for** step $h = H, H-1 \ldots, 1$ **do**

7:       Compute $\hat{b}_h^n$ from equation 7 and update $\overline{Q}, \underline{Q}$ as following:

$$\overline{Q}_{h,i}^n(\tau_h, \boldsymbol{a}) = r_{h,i}(\tau_h, \boldsymbol{a}) + \mathbb{E}_{\hat{\mathcal{P}}_h^n}\left[\overline{V}_{h+1,i}^n(\tau_{h+1})|\tau_h, \boldsymbol{a}_h\right] + \hat{b}_h^n(\tau_h, \boldsymbol{a})$$

$$\underline{Q}_{h,i}^n(\tau_h, \boldsymbol{a}) = r_{h,i}(\tau_h, \boldsymbol{a}) + \mathbb{E}_{\hat{\mathcal{P}}_h^n}\left[\underline{V}_{h+1,i}^n(\tau_{h+1})|\tau_h, \boldsymbol{a}_h\right] - \hat{b}_h^n(\tau_h, \boldsymbol{a})$$

8:       Compute the NE/CE/CCE solution $\pi_h^n$ according to equation equation 1/equation 2/equation 3 and update value function as following:

$$\overline{V}_{h,i}^n(\tau_h) = \mathbb{E}_{\boldsymbol{a}\sim\pi_h^n(\cdot|\tau_h)}[\overline{Q}_{h,i}^n(\tau_h, \boldsymbol{a})], \quad \underline{V}_{h,i}^n(\tau_h) = \mathbb{E}_{\boldsymbol{a}\sim\pi_h^n(\cdot|\tau_h)}[\underline{Q}_{h,i}^n(\tau_h, \boldsymbol{a})].$$

9:    **end for**

10:   Compute $\Delta^n = \max_{i\in[M]}\{\overline{v}_i^n - \underline{v}_i^n\}$ with $\overline{v}_i^n = \int_{\mathcal{X}}\overline{V}_{0,i}^n(x)\mu_0(x)dx, \underline{v}_i^n = \int_{\mathcal{X}}\underline{V}_{0,i}^n(x)\mu_0(x)dx$.

11: **end for**

12: **Return** $\hat{\pi} = \pi^{n^\star}$ where $n^\star = \arg\min_{n\in[N]}\Delta^n$.

---

This is a variational inference problem. When Assumption 4 holds, then $q(z|\tau_h^L, a_h, o_{h+1:h+l})$ and $\mathbb{P}^\pi(o_{h+1:h+l}|z_h)p(z_h|\tau_h^L, a_h)$ is a conjugate family and this problem admits a closed-form solution. Otherwise, we can solve this problem via black box variational inference (48). In particular, we consider a parameterized family $q_\theta$ and its derivative w.r.t. $\theta$ can be calculate as follows.

$$\nabla_\theta \mathbb{E}_{q_\theta(z|\tau_h^L, a_h, o_{h+1:h+l})}[\log \mathbb{P}^{\chi_\pi}(o_{h+1:h+l}|z_h)] - KL((q_\theta(z|\tau_h^L, a_h, o_{h+1:h+l})||p(z_h|\tau_h^L, a_h))$$

$$= \int q_\theta(z|\tau_h^L, a_h, o_{h+1:h+l})\nabla_\theta \log q_\theta(z|\tau_h^L, a_h, o_{h+1:h+l})\cdot$$

$$(\log q_\theta(z|\tau_h^L, a_h, o_{h+1:h+l}) - \log \mathbb{P}^{\chi_\pi}(o_{h+1:h+l}|z_h)p(z_h|\tau_h^L, a_h))dz$$

As a result, we can find an approximate solution of the ELBO maximization by stochastic gradient ascent type methods.

# G    COMPARISON WITH (12)

(12) construct a generalized PSR representation for $\gamma$-well-conditioned POMGs. Note that if the rank of the core test set is uniform across all time steps $h$, i.e. $d_h = d$ for all $h$, the POMG satisfies the assumptions in (12) is a special subclass of low-rank POMGs and this representation can be integrated into our framework.

Now consider a POMG that satisfies the assumptions in (12), where the set of generalized PSR representation is $\mathcal{M}$ and the rank of the core test set is uniform across all time steps $h$, i.e. $d_h = d$ for all $h$.

As shown in (12), learning generalized PSRs via MLE and conduct self-play UCB algorithm with an access to an oracle for the exact value function, the algorithm terminates with a sample complexity of $\tilde{O}((d + \frac{A^2 H}{\gamma^2})\frac{d^2 H^3 |\mathcal{S}|^2 A^4 \log(|\mathcal{M}_\epsilon|)}{\gamma^4 \epsilon^2})$, where $\mathcal{M}_\epsilon$ is an optimistic $\epsilon$-cover of $\mathcal{M}$, as defined in (12). We extend their method to oracle-free setting.

**Theorem 50** (PAC guarantee of Algorithm 6). *Assume assumptions in (12) holds and the rank of the core test set is uniform across all time steps $h$. Suppose $\alpha_n = \max\{\frac{A\sqrt{H}d\lambda}{\gamma^2} + \frac{|\mathcal{S}|A\sqrt{\beta}}{\gamma}, d\lambda +$*

$H\sqrt{nAt_N}\}$, $\lambda = \frac{\gamma|\mathcal{S}|^2 Q\beta \max\{\sqrt{r}_{PSR}, A\sqrt{H}/\gamma\}}{\sqrt{dH}}$ and $\beta = O(\log(|\mathcal{M}_\epsilon|))$. *Then, by setting the number of episodes $N$ to be* $\tilde{O}\left(\frac{H^{10} d^4 A^6 |\mathcal{S}|^2 \log(|\mathcal{M}_\epsilon|)}{\varepsilon^2 \gamma^6}\right)$, *with probability at least* $1 - \delta$, *the returned policy* $\hat{\pi}$ *is an $\varepsilon$-approximate equilibrium.*

*Proof.* Denote $t_N = (r + \frac{Q_A^2 H}{\gamma^2}) \frac{r d H^3 |\mathcal{S}|^2 Q_A^4 \beta}{\gamma^4 N}$. Analogous to the previous proof, we obtain:

$$\mathbb{E}_{(\tau_h, \boldsymbol{a}_h) \sim d_{\mathcal{P}}^{\hat{\pi}}} \|\mathcal{P}(\cdot|\tau_h, \boldsymbol{a}_h) - \hat{\mathcal{P}}(\cdot|\tau_h, \boldsymbol{a}_h)\|_1^2 \leqslant O(t_N)$$

and

$$\sum_{n=1}^{N} \Delta^{(n)} \lesssim H\left(\sqrt{dN \log\left(1 + \frac{N}{d}\right)}\sqrt{dA\left(\alpha_N\right)^2 + H^2 d\lambda} + \sum_{n=1}^{N} \sqrt{\frac{dA\left(\alpha_n\right)^2}{n}}\right)$$

$$+ H^3 \left(\frac{1}{H}\sqrt{dN \log\left(1 + \frac{N}{d\lambda}\right)}\alpha_N + N\sqrt{At_N}\right)$$

$$\lesssim \frac{H^5 d^2 N^{\frac{1}{2}} A^3 |\mathcal{S}| \log^{1/2}(|\mathcal{M}_\epsilon|)}{\gamma^3}.$$

when we pick $N$ to be

$$\tilde{O}\left(\frac{H^{10} d^4 A^6 |\mathcal{S}|^2 \log(|\mathcal{M}_\epsilon|)}{\varepsilon^2 \gamma^6}\right),$$

we have

$$\frac{1}{N}\sum_{n=1}^{N} \Delta^{(n)} \leqslant \varepsilon.$$

Then, we have

$$\max_{i \in [M]} \left\{v_i^{\dagger, \hat{\pi}_{-i}} - v_i^{\hat{\pi}}\right\} \left(\text{or } \max_{i \in [M]} \left\{\max_{\omega \in \Omega_i} v_i^{\omega \circ \hat{\pi}} - v_i^{\hat{\pi}}\right\}\right)$$

$$= \max_{i \in [M]} \left\{v_i^{\dagger, \pi_{-i}^{n^\star}} - v_i^{\pi^{n^\star}}\right\} \left(\text{or } \max_{i \in [M]} \left\{\max_{\omega \in \Omega_i} v_i^{\omega \circ \pi^{n^\star}} - v_i^{\pi^{n^\star}}\right\}\right)$$

$$\leqslant \Delta^{n^\star} = \min_{n \in [N]} \Delta^n \leqslant \frac{1}{N}\sum_{n=1}^{N} \Delta^n \leqslant \varepsilon,$$

which has finished the proof. $\qquad\square$

## H    TECHNICAL LEMMA

In this section, we present some technical lemmas used in the proof.

**Lemma 51** (MLE Guarantee). *For any episode $n \in [N]$, step $h \in [H]$, define $\rho_h$ as the joint distribution of $(x_h, a_h)$ in the dataset $\mathcal{D}_{h,n}$ at episode $n$. Then with probability at least $1 - \delta$, we have that*

$$\mathbb{E}_{(x_h, a_h) \sim \mathcal{D}_{h,n}} \left\|\mathbb{P}_h^{\mathcal{P}}(\cdot|x_h, a_h) - \mathbb{P}_h^{\widehat{\mathcal{P}}_n}(\cdot|x_h, a_h)\right\|_1^2 \leqslant \zeta_n,$$

*where $\zeta_n = O(\log(Hn|\mathcal{M}|/\delta)/n)$.*

For the proof, see (36).

**Lemma 52** ($l_2$ Guarantee). *For any episode $n \in [N]$, step $h \in [H]$, with probability at least $1 - \delta$, we have that*

$$\mathbb{E}_{(\tau_h, \boldsymbol{a}_h) \sim \mathcal{D}_{h,n}} \left\|\mathbb{P}_h^{\mathcal{P}}(\cdot|\tau_h, \boldsymbol{a}_h) - \mathbb{P}_h^{\widehat{\mathcal{P}}_n}(\cdot|\tau_h, \boldsymbol{a}_h)\right\|_2^2 \leqslant \zeta_n,$$

*where $\zeta_n = O(\log(Hn|\mathcal{M}|/\delta)/n)$.*

For the proof, see (40).

**Lemma 53** ((52), Lemma G.2). *Consider the following process. For $n = 1, \ldots, N$, $M_n = M_{n-1} + G_n$ with $M_0 = \lambda_0 I$ and $G_n$ being a positive semidefinite matrix with eigenvalues upper bounded by 1. We have*

$$2 \log \det(M_N) - 2 \log \det(\lambda_0 I) \geqslant \sum_{n=1}^{N} \operatorname{Tr}(G_n M_{n-1}^{-1}).$$

**Lemma 54** (Potential function lemma). *Suppose $\operatorname{Tr}(G_n) \leqslant B^2$.*

$$2 \log \det(M_N) - 2 \log \det(\lambda_0 I) \leqslant d \log \left( 1 + \frac{NB^2}{d\lambda_0} \right)$$

*Proof.* Let $\sigma_1, \cdots, \sigma_d$ be the set of singular values of $M_N$ recalling $M_N$ is a positive semidefinite matrix. Then, by the AM-GM inequality,

$$\log \det(M_N) / \det(\lambda_0 I) = \log \prod_{i=1}^{d} (\sigma_i / \lambda_0) \leqslant \log d \left( \frac{1}{d} \sum_{i=1}^{d} (\sigma_i / \lambda_0) \right)$$

Since we have $\sum_i \sigma_i = \operatorname{Tr}(M_N) \leqslant d\lambda_0 + NB^2$, the statement is concluded. $\square$

**Lemma 55.** *For parameters $A, B, \varepsilon$ such that $\frac{A^2 B}{\varepsilon^2}$ is larger than some absolute constant, when we pick $N = \frac{A^2}{\varepsilon^2} \log^2 \frac{A^4 B^2}{\varepsilon^4} = O\left( \frac{A^2}{\varepsilon^2} \log^2 \frac{AB}{\varepsilon} \right)$, we have*

$$\frac{A}{\sqrt{N}} \log(BN) \leqslant \varepsilon.$$

*Proof.* We have

$$\frac{A}{\sqrt{N}} \log(BN) = \varepsilon \frac{\log\left( \frac{A^2 B}{\varepsilon^2} \log^2 \frac{A^4 B^2}{\varepsilon^4} \right)}{\log \frac{A^4 B^2}{\varepsilon^4}}$$

Note that

$$\frac{A^2 B}{\varepsilon^2} \log^2 \frac{A^4 B^2}{\varepsilon^4} \leqslant \frac{A^4 B^2}{\varepsilon^4} \Leftrightarrow \log^2 \frac{A^4 B^2}{\varepsilon^4} \leqslant \frac{A^2 B}{\varepsilon^2}$$

where the right hand side is always true whenever $\frac{A^2 B}{\varepsilon^2}$ is larger than some given constant. Therefore, we get

$$\frac{A}{\sqrt{N}} \log(BN) \leqslant \varepsilon.$$

$\square$

**Lemma 56** (Concentration of the Bonuses). *Let $\mu_i$ be the conditional distribution of $\phi$ given the sampled $\phi_1, \cdots, \phi_{i-1}$, define $\Sigma : L_2(\mu) \to L_2(\mu)$, $\Sigma_n := \frac{1}{n} \sum_{i \in [n]} \mathbb{E}_{\phi \sim \mu_i} \phi \phi^\top$. Assume $\|\phi\|_{\mathcal{H}_k} \leqslant 1$ for any realization of $\phi$. If $\lambda$ satisfies the following conditions for each eigendecay condition:*

- *$\beta$-finite spectrum: $\lambda = \Theta(\beta \log N + \log(N/\delta))$;*

- *$\beta$-polynomial decay: $\lambda = \Theta(C_{\mathrm{poly}} N^{1/(1+\beta)} + \log(N/\delta))$;*

- *$\beta$-exponential decay: $\lambda = \Theta(C_{\exp}(\log N)^{1/\beta} + \log(N/\delta))$, where $C_3$ is a constant depends on $C_1$ and $C_2$;*

Table 2: Average exploitability of the final policy of DQN and OVI-OF-LLVR over 5 trails. Note that lower exploitability implies that the policy is closer to the equilibrium.

| Exploitability($\downarrow$) | DQN | OVI-OF-LLVR |
|---|---|---|
| H=3 | 1.1101($\pm$0.0033) | **0.2566**($\pm$0.0063) |
| H=10 | 3.1124 ($\pm$0.0875) | **0.7290**($\pm$0.0138) |

*then there exists absolute constant $c_1$ and $c_2$, such that $\forall x \in \mathcal{H}_k$, the following event holds with probability at least $1 - \delta$:*

$$\forall n \in [N], \quad c_1 \left\langle x, \left(n\Sigma_n + \lambda T_k^{-1}\right) x\right\rangle_{L_2(\mu)} \leqslant \left\langle x, \left(\sum_{i\in[n]} \phi_i\phi_i^\top + \lambda T_k^{-1}\right) x\right\rangle_{L_2(\mu)},$$

$$\text{and} \quad \left\langle x, \left(\sum_{i\in[n]} \phi_i\phi_i^\top + \lambda T_k^{-1}\right) x\right\rangle_{L_2(\mu)} \leqslant c_2 \left\langle x, \left(n\Sigma_n + \lambda T_k^{-1}\right) x\right\rangle_{L_2(\mu)}.$$

*In the same event above, the following event must hold as well:*

$$\forall n \in [N], \quad \frac{1}{c_2} \left\langle x, \left(n\Sigma_n + \lambda T_k^{-1}\right)^{-1} x\right\rangle_{L_2(\mu)} \leqslant \left\langle x, \left(\sum_{i\in[n]} \phi_i\phi_i^\top + \lambda T_k^{-1}\right)^{-1} x\right\rangle_{L_2(\mu)}$$

$$\text{and} \quad \left\langle x, \left(\sum_{i\in[n]} \phi_i\phi_i^\top + \lambda T_k^{-1}\right)^{-1} x\right\rangle_{L_2(\mu)} \leqslant \frac{1}{c_1} \left\langle x, \left(n\Sigma_n + \lambda T_k^{-1}\right)^{-1} x\right\rangle_{L_2(\mu)}$$

For the proof, see (41).

# I EXPERIMENTS

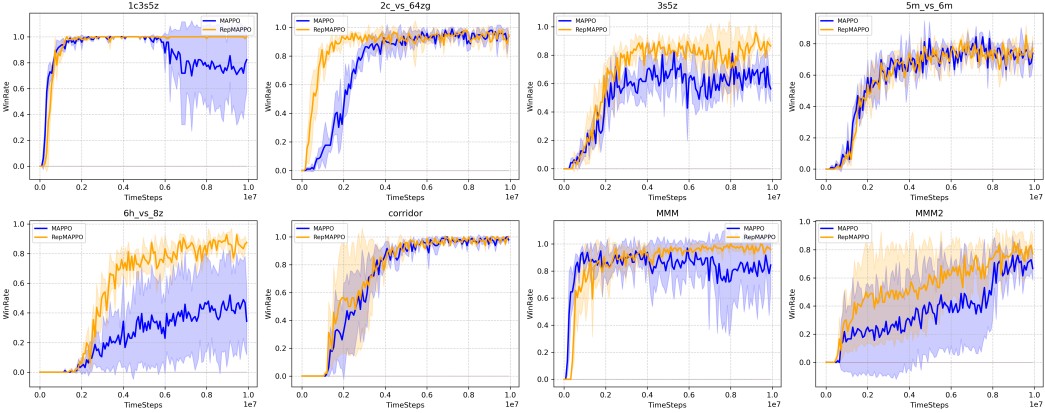

Figure 1: Comparison of win rate between REPMAPPO and MAPPO in SMAC. Y axis denotes the win rate and X axis denotes the number of steps taken in the environment.

In this section, we present two experiments to evaluate our methods. The first experiment focuses on a simple POMG with random latent transitions and rewards, aiming to validate the convergence of OVI-OF-LLVR. Our second experiment evaluate LLVR on the StarCraft Multi-Agent Challenge (SMAC) environments (53), a widely used benchmark for Dec-POMDPs, to assess its effectiveness. Note that, in Dec-POMDPs, all agents cooperate and share observations during training, making this setting well-suited for our method.

## I.1 EXPERIMENT ON SIMPLE POMG

Our first experiment aims to verify whether OVI-OF-LLVR can reliably converge under $L$-decodability assumption with random dynamics and rewards. To achieve this, the algorithm must not only decode the latent structure accurately but also solve the POMG to produce NE/CE/CCE policies. Accordingly, we construct a Block Partially Observable Markov Games (BPOMGs) defined as follows:

**Definition 57** (BPOMG). *(30) For any $h \in [H]$, a BPOMG has an emission distribution $e_h(\cdot|z) \in \Delta_{\mathcal{S}}$ and a latent state space transition $T_h(z'|z, \boldsymbol{a})$, such that for any $s \in \mathcal{S}, e_h(s|z) > 0$ for a unique latent state $z \in \mathcal{Z}$, denoted as $\psi_h^\star(s)$. Together with the ground truth decoder $\psi_h^\star$, it defines the transitions $P_h(s'|s, \boldsymbol{a}) = \sum_{z' \in \mathcal{Z}} e_h(s'|z') T_h(z'|\psi_h^\star(s), \boldsymbol{a})$.*

It is straightforward to see that BPOMG is a special case of the 1-decodable POMG when we define the latent state z to be exactly equivalent to the current observation (i.e., $z = o_h$).

We construct two two-player zero-sum BPOMG variants, with horizons $H = 3$ (short) and $H = 10$ (long). Each BPOMG is randomly generated with $H$ horizon, 3 states, 2 players each with 3 actions, random reward matrix $r_h \in (0, 1)^{3 \times 9 \times H}$ and random latent transition matrix $T_h$. The dimension of the observation space is $2^{\lceil \log(H+|\mathcal{S}|+1) \rceil}$. Note that similar mechanism has also been adopted in (32) to construct a Block MDP. See Appendix J for details.

We adopt DQN (54) together with fictitious self-play (55) as baseline, and measure the policy exploitability to assess the performance. For each variant, we run 5 trials and report the mean exploitability and its variance. As shown in Table 2, OVI-OF-LLVR obtains policy closer to the equilibrium, with significantly lower exploitability.

## I.2 EXPERIMENT ON SMAC

Our second experiment verify the effectiveness of our proposed representation in OVI-OF-LLVR on the SMAC benchmark environments. In this experiment, we learn the latent representations by predicting future outcomes from a history of length $L$. Specifically, we employ a continuous latent variable model similar to (15), approximating probability distributions with Gaussians parameterized by their mean and variance. The learned representations can be integrated with various MARL methods by feeding them into the value function. In our experiments, we select MAPPO (56) as baseline and compare it against its representational variant, REPMAPPO. Detailed implementation information, including hyperparameters, is provided in Appendix J. We apply $L = 1$ across all domains.

In Figure 1, we report the results from 8 selected SMAC scenarios —4 Hard and 4 Super Hard—out of 23 scenarios. It is shown that REPMAPPO achieves significantly better empirical results in 5 scenarios and marginally outperforms MAPPO in other scenarios. Moreover, it can also be observed that REPMAPPO demonstrates greater stability, with smaller variance and a smoother training curve during evaluation, on most scenarios.

# J EXPERIMENT DETAILS

## J.1 DETAILED EXPERIMENT SETUP

In this section, we provide the detailed setups for the two experiments conducted to evaluate our methods. For completeness we repeat certain details already introduced in the main text.

Firstly, we introduce the details of the environment construction of the BPOMGs. We designed a BPOMG by randomly generating a tabular POMG with horizon $H$, 3 states, 2 players each with 3 actions, and random reward matrix $r_h \in (0, 1)^{3 \times 9 \times H}$ and random latent transition matrix $T_h$. The dimension of the observation space is $2^{\lceil \log(H+|\mathcal{S}|+1) \rceil}$, in line with the design of (32).

For the implementation of LLVR, we break down the introduction into two parts: the implementation of feature learning and the implementation of game solving algorithm using the current features. For the implementation of feature learning, we assume that the features follow a Gaussian distribution. To model the mean and log standard deviation of this distribution, we adopt a three-layer neural network

with ReLU non-linearity. This approach allows us to effectively capture and represent the underlying feature distributions necessary for solving the game.

For solving the POMGs, in addition to following OVI-OF-LLVR, we implement the NE/CCE solvers based on the public repository: https://github.com/quantumiracle/MARS.

For the SMAC experiment, we implement MAPPO and REPMAPPO based on the public repository: https://github.com/marlbenchmark/on-policy. We employ a continuous latent variable model similar to (15), using Gaussian distributions parameterized by their mean and variance. To enhance training stability, we utilize a target network for feature updates, applying a soft target network update mechanism. All parameters are set to their default values.

## J.2 HYPERPARAMETERS

In this subsection, we include the hyperparameters for LLVR and REPMAPPO in Table 3.

Table 3: Hyperparameters for LLVR and REPMAPPO in the experiment

| LLVR | Value | REPMAPPO | Value |
|---|---|---|---|
| | | GAE $\lambda$ | 0.95 |
| Buffer size | 100000 | $\gamma$ | 0.99 |
| Batch size | 256 | Feature dimension | 64 |
| Feature dimension | 32 | Feature Target Update Tau | 0.01 |
| Hidden dimension | 32 | Hidden dimension | 64 |
| Optimizer | sgd | Optimizer | Adam |
| Learning rate | 0.01 | Actor learning rate | 5e-4 |
| LSVI bonus coefficient $\alpha$ | 0.1 | Critic learning rate | 5e-4 |
| LSVI regularization coefficient $\lambda$ | 1 | Feature Learning rate | 5e-4 |
| Warm up number | 10 | GAE $\lambda$ | 0.95 |
| | | $\gamma$ | 0.99 |

