# OpenReview forum: "Optimistic Value Iteration with Representation Learning for Low-Rank POMGs"
_ICLR.cc/2026/Conference — Submitted to ICLR 2026_

### Official Review · Reviewer_SUf7 · 2025-10-16

**Soundness:** 4
**Presentation:** 4
**Contribution:** 4
**Rating:** 6
**Confidence:** 4

**Summary:**

This paper studies low-rank partially observable Markov games in the context of multi-agent reinforcement learning with partial observability and representation learning.

The authors propose an optimistic value iteration type algorithm and present two specific instantiations based on maximum likelihood estimation (MLE) and spectral decomposition–based representation learning.

The paper also provides solid theoretical results supporting the proposed methods.

**Strengths:**

The proposed setting is both novel and important, addressing a crucial gap in multi-agent reinforcement learning under partial observability.

The paper presents solid theoretical results.

Assumptions are clearly stated and analyses are well-structured.

Overall, the paper is clearly written and easy to follow, with assumptions and main results stated explicitly and intuitively.

Theoretical bounds are well-highlighted and effectively summarized in Table 1, which helps the reader grasp the key takeaways at a glance.

**Weaknesses:**

While the proposed Optimistic Value Iteration (OVI) framework may not be entirely new, and similar representation learning approaches have appeared in prior work, extending it to the low-rank partially observable Markov game setting represents a meaningful and valuable contribution. This extension broadens the applicability of OVI and strengthens its theoretical foundation in a more complex environment.

The paper lacks empirical results and concrete motivating examples, which would help illustrate the practical relevance and intuition behind the theoretical findings. That said, the theoretical development alone provides substantial contribution and novelty, making the paper strong even in the absence of experiments.

**Questions:**

Two relevant prior works could be incorporated and discussed for a more complete positioning of this paper within the literature:

Model-Free Representation Learning and Exploration in Low-Rank MDPs, and

Reinforcement Learning in Low-Rank MDPs with Density Features.

A discussion of how the proposed approach relates to or differs from these studies would help clarify the paper’s novelty and theoretical advancement.

It would also be interesting to consider whether a model-free representation learning approach could be developed under this framework. While the Partially Observable Markov Game (POMG) setting certainly introduces additional layers of complexity, an analysis or discussion of the challenges and potential feasibility of extending the method to the model-free case would add further depth.

The definition of the D hat dataset appears to be missing or unclear—clarifying what it represents and how it differs from the original dataset D would improve readability and precision.

It would strengthen the paper to explicitly highlight the key technical challenges and sources of novelty compared to prior work, perhaps in the main text or in a dedicated discussion section. Such clarification would make the unique contributions of the paper more visible.

Could the proposed setting reduce to the single-agent case? If so, it would be valuable to provide the corresponding theoretical bounds in this simplified scenario and to compare them against existing single-agent results in the literature. This would help situate the contribution relative to known benchmarks and demonstrate consistency with established results.

---

> ### Author Response · Authors · 2025-11-17
> **Response to Reviewer SUf7**
>
> We sincerely thank the reviewer for their thoughtful assessment. Your questions are insightful and allow us to better position our work and clarify technical details.
>
> [Empirical Results]
>
> While our paper primarily focuses on establishing a general theoretical framework and rigorous statistical guarantees, we have indeed provided numerical experiments in Appendix I of the submission. These experiments empirically evaluate our methods (specifically the OVI-OF-LLVR instantiation) and confirm the convergence properties and the effectiveness of the LLVR representation.
>
> [Comparison with Prior Works]
>
> We thank the reviewer for pointing out the papers. These two papers provide foundational work on efficient learning within the Low-Rank MDP setting (single-agent, fully observable). Our work tackles a fundamentally more complex environment: a Partially Observable Markov Game.  We have included a detailed discussion in related work section of the revised manuscript to clearly delineate our technical contributions and differences from these low-rank MDP works.
>
> [Model-Free Representation Learning]
>
> We agree that developing a model-free representation-learning instantiation is a promising direction. For example, the model-free representation-learning method in [2] can be interpreted as a candidate instantiation of the representation module within our OVI framework. Concretely, if a model-free learner (such as the one in [2]) produces representations for which one can establish explicit bounds on the representation error, then those error bounds can be plugged into our theoretical analysis and the same end-to-end guarantees follow.
>
>
>
>
> [Clarification of $\hat{D}$ Dataset]
>
> We thank the reviewer for pointing this out. The dataset $\hat{D}$ in Algorithm 1 is defined as $\hat{D}=$ {$\tau_{h+1},a_{h+1},o_{h+2}$} for all $h$, i.e., the collection of tuples consisting of the next-step history, the corresponding joint action, and the subsequent observation at step $h+2$. This dataset is used to ensure proper sampling for representation learning and is critical for bounding the representation error. In the revised manuscript, we have explicitly defined $\hat{D}$ and clarified its role to improve readability and precision.
>
> [Highlight Novelty]
>
> Our proposal is a general OVI framework designed for low-rank POMGs that explicitly accommodates various low-rank representation learning methods. We provide the first statistical complexity guarantees for an OVI-based planning algorithm in POMGs under this general framework, proving that the representation error and the planning error interact well to achieve a bounded sample complexity. We have restated and emphasized these key technical challenges and sources of novelty in the introduction of the revised manuscript.
>
> [Reduction to Single-Agent Setting]
>
> Our approach naturally reduces to the single-agent setting when we eliminate the game aspect. In this reduction, the NE/CE/CCE oracle used in the multi-agent setting is simply replaced by an argmax over the value-function estimator, exactly as in standard optimistic RL for low-rank MDPs. Under this specialization, all components of our OVI framework—optimistic planning, representation learning, and the resulting error propagation analysis—carry over in a fully analogous manner.
> Furthermore, it is worth noting that while several papers have studied low-rank MDPs, there is no single unified framework for representation learning in this setting; our work, when reduced to the single-agent case, serves as a comprehensive framework-level synthesis that integrates optimistic planning with various representation learning strategies.
>
> Importantly, the theoretical guarantees obtained in this single-agent reduction are consistent with existing results in the low-rank MDP literature. For example, when instantiated with OVI-OB-MLE and OVI-OB-SDR, the reduction yields an $\epsilon$-optimal policy sample complexity of $O(\frac{d^4|A|^2\ln(|M|/\delta)}{(1-\gamma)^5\epsilon^2})$ and $O(\frac{d^4|A|^2\ln(|M|/\delta)}{(1-\gamma)^6\epsilon^2})$, respectively, matching the bounds established in [1] and [3].
>
> [1]M. Uehara, X. Zhang, and W. Sun, Representation learning for online and offline rl in low-rank mdps.
>
> [2]A.Modi, J.Chen, A.Krishnamurthy, N.Jiang, and A.Agarwal, Model-free representation learning and exploration in low-rank mdps.
>
> [3]T.Ren, T.Zhang, L.Lee, J.E.Gonzalez, D.Schuurmans, and B.Dai, Spectral decomposition representation for reinforcement learning.

---

> ### Author Response · Authors · 2025-11-26
> **Kindly Checking In as the Discussion Period Nears Its End**
>
> Dear Reviewer,
>
> I hope this message finds you well. As the discussion period is nearing its end with less than one week remaining, l wanted to ensure we have addressed all your concerns satisfactorily. If there are any additional points or feedback you'd like us to consider, please let us know.Your insights are invaluable to us, and we'e eager to address any remaining issues to improve our work.
>
> Thank you for your time and effort in reviewing our paper.

---

> ### Comment · Reviewer_SUf7 · 2025-11-27
>
> I read the response and other reviews in detail.
>
> There seems be scores on both sides. I plan to keep my rating, but I may not champion this paper.

---

### Official Review · Reviewer_hehM · 2025-10-28

**Soundness:** 2
**Presentation:** 2
**Contribution:** 2
**Rating:** 4
**Confidence:** 3

**Summary:**

This paper focuses on partially observable Markov games (POMGs) with low-rank structures. This paper proposes two types of optimistic value iteration (VI) algorithms to compute NE, CCE, and CE in the games, one is oracle-based and the other is oracle-free. To learn the low-rank representations, this paper introduces both an MLE-based approach and a spectral decomposition representation method. Finally, POMGs with infinite-dimensional latent spaces are discussed.

**Strengths:**

(+) This paper is technically solid and presents comprehensive results.

(+) The low-rank assumption seems to be novel and promising to address the partial observation.

**Weaknesses:**

(-) The algorithm operates on the joint action space $\mathcal{A}$ of all the players, which could be computationally costly.

(-) The feasibility of the low-rank assumption (Def 4) needs further justification. See Q1 below.

**Questions:**

Typo: In Definition 4, should it be $\tau\in\mathcal{T}\_h$ and $\tau'\in \mathcal{T}\_{h+1}$?

Q1. Are there any concrete examples of low-rank representations (Def 4)? Of course, POMGs with a discrete state-action space are finite-rank, but the embedding dimension could be exponential to $H$.

Q2. In the OVI-OF algorithm, the computation in Line 15 could be intractable, i.e., integrals over all possible observations. Are there any solutions?

Q3. What is the relationship between Section 5 and the previous sections? The LLVR algorithm deviates from the low-rank assumption and switches to a neural network-based learning method.

Q4. Can you explain how Algorithm 1 is different from [1]?

[1] Ni et al. Representation learning for low-rank general-sum Markov games. ICLR 2023.

---

> ### Author Response · Authors · 2025-11-17
> **Response to Reviewer hehM**
>
> We thank the reviewer for your review! Below we provide a detailed response to each of your feedback and we hope our response could address your concerns.
>
> [Computation on Joint Action Space]
>
> Due to the inherent difficulty of POMGs, almost all existing theoretical works on this topic rely on the joint action space [6-8]]; avoiding it makes the problem significantly harder. For our proposed method, we note that when a value function oracle is available, OVI-OB’s dependence on the joint action space is limited—it only requires an $\arg\max$ over joint actions when selecting the optimistic policy. As the problem setting becomes more complex and such a value oracle is no longer accessible, this dependence naturally increases. In particular OVI-OF must learn and update a Q-function defined over the joint action space, which is unavoidable in the absence of a value-function oracle.
>
> We agree developing decentralized algorithms to avoid operating on the full joint action space is a crucial direction, but extending our framework is nontrivial since computing NE/CCE/CE typically requires joint planning.
> A similar challenge arises across essentially all existing POMG algorithms [7–9], highlighting that avoiding the joint action space is a fundamental difficulty for the entire field.
>
> [Feasibility of Low-Rank Assumption]
>
> The low-rank assumption is widely used in the literature [1-5] to enable efficient learning and planning. Specifically, Low-Rank MDPs have been shown to be more expressive than other classes like Block MDPs or low Bellman-rank MDPs [2]. Analogous results hold for Low-Rank POMGs, allowing them to capture a broader range of complex, partially observable dynamics than simpler block-structured models.
>
> To give a concrete example, consider a simplex feature space: let the feature vectors $\{\phi_h(\tau, a)\}$ lie in a $d$-dimensional simplex, i.e., {$\phi_h| \sum_{i=1}^d \phi^{[i]}_h= 1$ and $\phi^{[i]}_h \geq 0$ for all $i$}.
>
> A low-rank POMG can then be instantiated by defining $\mu_h(\cdot)=\sum_{i=1}^d \mu_h^{[i]}(\cdot)e_i$ where $\mu_h^{[i]}$ is a probability measure over $\mathcal{T}_{h+1}$.
>
> In this case, the transition kernel $P_h(\tau'|\tau,a)=\sum_{i=1}^d\phi_h^{[i]}(\tau,a)\mu^{[i]}_h(\tau')$ is simply a convex combination of the base distributions $\mu_h^{[i]}$, and hence is a valid probability distribution.
> This example illustrates that non-trivial, tractable low-rank structures exist.
> Thus, the low-rank assumption identifies a meaningful subclass of POMGs for which provable sample-efficient learning is possible.
>
> [Computation in Line 15]
>
> The computation in Line 15 involves calculating the expectation of the estimated value function to evaluate the expected performance gap between the optimistic and pessimistic estimators over the initial history distribution $\mu_0$. In practice, this integral can be tractably approximated:
>
> 1.	In many practical scenarios, the support of the initial distribution $\mu_0$ is often small or fixed to a single initial state.
>
> 2.	For continuous history distributions, the expectation can be efficiently approximated via Monte Carlo (MC) sampling. The error introduced by MC sampling converges at a rate of $O(1/\sqrt{n})$, and as long as the sample size $n$ is sufficiently large, the MC error is controllable and does not meaningfully affect the overall algorithm. Importantly, this approximation error only affects the evaluation step used to select the best optimistic policy, and with sufficiently large $n$, this approximation will not change the identity of the selected policy.
> Thus, this step does not pose a fundamental barrier to the algorithm's feasibility.
>
> [Relationship between Sec 5 and Previous Sections]
>
> Sec 5 is a crucial extension of our OVI framework. The preceding sections establish guarantees for the low-rank POMG setting,leading naturally to the question of whether our OVI framework can handle scenarios where the rank is too large to be practically assumed finite.
>
> The motivation behind Sec 5 is precisely to address this question. By introducing the $L$-decodability assumption, we extend our OVI framework to encompass POMGs with possible infinite-dimensional latent spaces. This essentially "mitigates" the finite-rank restriction, allowing the framework to utilize expressive neural network-based representations that are more flexible and applicable to complex observation spaces encountered in practice.
>
> [Difference between Alg 1 and [1]]
>
> While [1] focuses on fully observable Markov games, our work addresses the harder partially observable setting. Our Alg 1 provides a general OVI framework that abstracts the representation-learning module. In fact, [1] can be seen as a special case of our OVI-OF framework with fully observable states (i.e., $o=s$) and MLE-based representation learning.
>
> [Typographical Correction]
>
> Thank you for spotting the typo in Def 4, we have corrected this in the revised manuscript.

---

> > ### Author Response · Authors · 2025-11-17
> > **References used in the Response**
> >
> > [1]C.Ni, Y.Song, X.Zhang,C.Jin, and M.Wang, Representation learning for general-sum low-rank markov games.
> >
> > [2]A.Agarwal, S.Kakade, A.Krishnamurthy, and W.Sun, Flambe: Structural complexity and representation learning of low rank mdps.
> >
> > [3]M. Uehara, X. Zhang, and W. Sun, Representation learning for online and offline rl in low-rank mdps.
> >
> > [4]T.Ren, T.Zhang, L.Lee, J.E.Gonzalez, D.Schuurmans, and B.Dai, Spectral decomposition representation for reinforcement learning.
> >
> > [5]A.Modi, J.Chen, A.Krishnamurthy, N.Jiang, and A.Agarwal, Model-free representation learning and exploration in low-rank mdps.
> >
> > [6]Q.Liu, C.Szepesv´ari, and C.Jin, Sample-efficient reinforcement learning of partially observable markov games.
> >
> > [7]S.Qiu, Z.Dai, H.Zhong, Z.Wang, Z.Yang, and T.Zhang, Posterior sampling for competitive rl: function approximation and partial observation.
> >
> > [8]A.Altabaa and Z.Yang, On the role of information structure in reinforcement learning for partially observable sequential teams and games.

---

> ### Author Response · Authors · 2025-11-26
> **Kindly Checking In as the Discussion Period Nears Its End**
>
> Dear Reviewer,
>
> I hope this message finds you well. As the discussion period is nearing its end with less than one week remaining, l wanted to ensure we have addressed all your concerns satisfactorily. If there are any additional points or feedback you'd like us to consider, please let us know.Your insights are invaluable to us, and we'e eager to address any remaining issues to improve our work.
>
> Thank you for your time and effort in reviewing our paper.

---

### Official Review · Reviewer_wsqY · 2025-10-31

**Soundness:** 2
**Presentation:** 3
**Contribution:** 2
**Rating:** 6
**Confidence:** 2

**Summary:**

This work proposes a unified optimistic value iteration (OVI) framework for solving low-rank POMGs. Combined with MLE, SDR and L-step latent variable representation (LLVR), the OVI algorithm has shown to converge to the approximate NE, CE and CCE, provided the representation error is bounded.

**Strengths:**

The proposed OVI framework accommodates different low-rank representation learning methods for learning approximate equilibria in low-rank POMGs with finite sample complexity guarantee. The L-step variable representation extends OVI to infinite-dimensional latent spaces under L-decodability assumption may be of independent interest.

**Weaknesses:**

The OVI framework is intuitive, and there is limited novel design tailored to POMGs. Further, the analysis is largely based on the existing analysis in low-rank MDPs and POMGs.

**Questions:**

1 See Weaknesses. The main concern is the novelty of this work. It seems the analysis for MLE and SDR based OVI are rather standard. The novelty of the proposed OVI algorithm and the novel techniques are not clearly stated.

2 While the LLVR representation is interest, the L-decodability assumption seems quite strong and hard to verity. a) For practical problems, are there simple tests for this property? b) Is it possible to relax the L-decodability assumption? Say, if the L-decodability is approximately true.

3 It is known that learning NE in Markov games is much harder than learning CE and CCE. This works assumes that we have access to an oracle for computing NE,CE,CCE. What is the computational complexity for learning NE,CE,CCE in POMGs?

Minor issues:

1* In background, in many places $r(s,a)$ is adopted and should be replaced with $r(o,a)$.

2* In Lemma 13, $\alpha_N$  as a function of $\zeta$ is not defined.

---

> ### Author Response · Authors · 2025-11-17
> **Response to Reviewer wsqY**
>
> Thank you for your valuable feedback and deep engagement with the technical aspects of our paper. We address your comments below.
>
> [Novelty of the OVI Framework]
>
> We emphasize that the main contribution of our work lies in establishing a unified OVI framework for low-rank POMGs together with the first statistical complexity guarantees for such a framework. As you noted, components such as MLE and SDR are well-known tools. However, the challenge—and the key novelty of our work—is how to leverage these components jointly in the POMG setting to obtain a theoretically sound algorithm with provable convergence to an approximate equilibrium.
>
> Though the individual ingredients may be standard, integrating them in a way that ensures correct behavior in POMGs is far from trivial. In particular, our analysis must carefully characterize (1) how the representation error propagates into the optimistic value iteration step and accumulates over the horizon under partial observability, and (2) how the learned representation interacts with the equilibrium computation to produce valid sample-complexity guarantees.
> These interactions have not been previously systymatically investigated in the low-rank POMG literature, where prior works typically focus on specific fixed representation models or assume access to accurate value oracles.
>
> In contrast, our framework explicitly accommodates multiple representation learners and provides end-to-end guarantees for the  OVI method. Specifically, we developed the bound of the propagation of representation error under partial observability in Lemma 16, and addressed the interaction between the learned representation and the equilibrium computation in Lemma 17-19. When the value-function oracle is unavailable, the algorithm must operate entirely on learned Q-function estimators. Importantly, Our regret and equilibrium guarantees required a new analysis that does not appear in prior POMG works.
>
> Thus, while the components themselves are known, the way we combine them and the theoretical analysis required to show their compatibility under the POMG structure constitute the central novelty of the paper.
>
> [L-Decodability Assumption]
>
> The $L$-decodability assumption is a common tool for provable efficiency in partially observable settings [3-6]. We acknowledge that the $L$-decodability assumption can be difficult to verify in practice. However, its motivation aligns with a widely observed phenomenon in long-horizon games: the most recent few steps typically have the strongest influence on the current latent state, while information from distant history often becomes negligible. The condition formalizes this intuition by assuming that a short window of recent observations and actions contains sufficient information to decode the latent state relevant for planning.
> Regarding practical verification, simple tests are indeed challenging, and developing such diagnostics remains an open problem. The assumption should be viewed as a structural prior rather than a directly testable condition.
>
> Relaxing the $L$-decodability requirement is an interesting direction for future work. To the best of our knowledge, no existing work has explored approximate or softened versions of this assumption in the context of POMDPs or POMGs.
>
> [Computational Complexity for Equilibrium Learning]
>
> Thank you for raising this important point. Our work primarily focuses on the statistical complexity, and we did not explicitly discuss the computational complexity in the manuscript.
> It is worth emphasizing that in our framework, we only require NE/CE/CCE oracles for the matrix normal-form game that arises within each step of the OVI procedure. This is significantly simpler than computing an NE/CE/CCE of general POMGs.
> 1. For CE and CCE, the matrix normal-form game subproblem can be formulated as a linear program, which is solvable in polynomial time using standard methods [1]. Thus, for CE and CCE, the computational burden introduced by this oracle is polynomial.
>
> 2. For NE, computing a solution of a normal-form game is indeed PPAD-hard [7]. However, this computational difficulty is fully aligned with the nature of the problem: computing an NE in a POMG is inherently more challenging than computing a CE/CCE, and even for fully observed Markov games, NE-based methods typically rely on NE subsolvers as components of planning algorithms [8-10]. Hence, it is natural that solving such a harder equilibrium requires a correspondingly stronger subroutine. In this sense, using an NE solver for the induced normal-form game is a reasonable and consistent choice.
>
> Our analysis therefore isolates the statistical sample complexity of the OVI procedure, while treating the normal-form game solver as a computational oracle, following common practice in the literature.
>
> [typo issue]
>
> Thank you for your meticulous review. The typo issue have been corrected in the revised version.

---

> > ### Author Response · Authors · 2025-11-17
> > **References used in the Response**
> >
> > [1]Papadimitriou, C. H. and Roughgarden, T.,  Computing correlated equilibria in multi-player games.
> >
> > [2]X.Zhang, Y.Song, M.Uehara, M.Wang, A.Agarwal, and W.Sun, Efficient reinforcement learning in block mdps: A model-free representation learning approach.
> >
> > [3]H.Zhang, T.Ren, C.Xiao, D.Schuurmans, and B.Dai, Provable representation with efficient planning for partially observable reinforcement learning.
> >
> > [4]Efroni, Y., Jin, C., Krishnamurthy, A., and Miryoosefi, S., Provable reinforcement learning with a short-term memory. In International Conference on Machine Learning, pp. 5832–5850. PMLR, 2022.
> >
> > [5]Guo, J., Li, Z., Wang, H., Wang, M., Yang, Z., and Zhang, X., Provably efficient representation learning with tractable planning in low-rank pomdp.
> >
> > [6]Qiu,S., Dai,Z., Zhong,H., Wang,Z., Yang,Z., and Zhang, T., Posterior sampling for competitive rl: function approximation and partial observation.
> >
> > [7]Chen X, Deng X, Teng S H., Settling the complexity of computing two-player Nash equilibria.
> >
> > [8]]C.Ni, Y.Song, X.Zhang,C.Jin, and M.Wang, Representation learning for general-sum low-rank markov games.
> >
> > [9] Bai Y, Jin C., Provable self-play algorithms for competitive reinforcement learning.
> >
> > [10]Hu J, Wellman M P., Nash Q-learning for general-sum stochastic games.

---

> ### Author Response · Authors · 2025-11-26
> **Kindly Checking In as the Discussion Period Nears Its End**
>
> Dear Reviewer,
>
> I hope this message finds you well. As the discussion period is nearing its end with less than one week remaining, l wanted to ensure we have addressed all your concerns satisfactorily. If there are any additional points or feedback you'd like us to consider, please let us know.Your insights are invaluable to us, and we'e eager to address any remaining issues to improve our work.
>
> Thank you for your time and effort in reviewing our paper.

---

### Official Review · Reviewer_L12U · 2025-11-01

**Soundness:** 2
**Presentation:** 3
**Contribution:** 2
**Rating:** 2
**Confidence:** 3

**Summary:**

This paper studies low-rank partially observable markov games (POMG). It proposed a value iteration method with optimism design that incorporates a bonus function using learned representations. It then proposed a representation learning algorithm for infinite-dimension latent space with additional L-decodability assumption. The paper proved a polynomial sample complexity under this low-rank POMG setting.

**Strengths:**

1. Some techniques proposed or adopted in this paper are new and interesting. For example, the adoption of Spectral Decomposition Representation (SDR) enable the algorithm to remove unrealistic oracles.

2. The paper also proposed way to handle infinite-dimension latent space by using variational methods, which is also novel to some extend.

**Weaknesses:**

1. One weakness or question is about the low-rank structure for POMG. This is also the primary reason that I recommend reject at the current stage.

In definition 4, the low-rank structure takes the form $P(\tau'|\tau,a)=\phi(\tau,a)^\top\mu(\tau')$, while this is acceptable for MDP, in POMDP, we have if $(\tau, a)$ is not the history of $\tau'$, then $P(\tau'|\tau, a) = 0$. This means that if we want to define low-rank structure in this way, we have to enforce $\mu(\tau')$ to be orthogonal to all $\phi(\tau,a)$ as long as $(\tau,a)$ is not the history of $\tau'$.

This implies that $d\geq (OA)^{h}$ since for each $(\tau,a)$, we need distinct direction of $\phi(\tau,a)$, which seems unrealistic.

**Questions:**

Please see weakness part.

---

> ### Author Response · Authors · 2025-11-17
> **Response to Reviewer L12U**
>
> We thank the reviewer for this insightful observation. To clarify, our intended low-rank assumption for POMGs applies the form $P(\tau'|\tau,a) = \phi(\tau, a)^\top \mu(\tau')$ only to those triplets $(\tau, a, \tau')$ where $(\tau, a)$ is indeed the immediate history of $\tau'$. For all other pairs where $(\tau,a)$ is not the history of $\tau'$, the transition probability is zero by definition of the history state, and the low-rank form is not imposed in those cases.
>
> This has been our intended meaning, and the subsequent analyses throughout the paper—including all bounds involving the feature dimension $d$—are already carried out under this interpretation. The rank $d$ therefore does not need to scale with $(OA)^h$, and remains small and realistic in our setting. We have updated the wording of Definition 4 in the revised manuscript to make this intended condition explicit, and added a brief discussion below the definition on why the required rank remains small and does not scale with $(OA)^h$.

---

> > ### Author Response · Authors · 2025-11-26
> > **Kindly Checking In as the Discussion Period Nears Its End**
> >
> > Dear Reviewer,
> >
> > I hope this message finds you well. As the discussion period is nearing its end with less than one week remaining, l wanted to ensure we have addressed all your concerns satisfactorily. If there are any additional points or feedback you'd like us to consider, please let us know.Your insights are invaluable to us, and we'e eager to address any remaining issues to improve our work.
> >
> > Thank you for your time and effort in reviewing our paper.

---

> ### Comment · Reviewer_L12U · 2025-11-28
>
> Thanks for the response. If this is the case, then I have a concern about the validity of one-step back inequality.
>
> Specifically, in the proof of lemma 16, line ~815, the integral is not w.r.t. $\tau\_h$, but $o\_h$. Then, after applying Cauchy's inequality, and starting to deal with the second term, (line ~826), the integral cannot be written as the expectation of $g(\tau\_h,a\_h)$. Because $\tau_h$ in $g$ does **not** match with the $\tau\_h$ in the subscript of the expection, where the latter is the **combination** of $o\_h$ mentioned before (or the $o\_h$ in $g$) and the history $\tau_{h-1}$ in **dataset**.
>
> Moreover, this issue cannot be simply solved by replacing $\tau\_h$ by $o\_h$. Because again, $g$ function is defined on a specific trajectory $\tau_h$, whose history(one-step back) is highly likely not in the dataset, while one-step technique back is trying to do exactly the opposite thing: using history to recalculate the expectation. This is not an issue in low-rank MDP, since any state action pair (history) can transit to a specific next state through low-rank structure. However, in POMDP, only a specific history can transit to $\tau\_h$.
>
> This low-rank assumption in POMDP is my only concern and I am willing to have further discussion about it.

---

> > ### Author Response · Authors · 2025-12-01
> > **Response to Reviewer L12U**
> >
> > Thank you for the detailed comments. We would first like to clarify that the concern arises from a misunderstanding of how the expectations are taken in the proof. Our original notation in Lemma 16 did not explicitly distinguish between $o_h$ and $\tau_h$ when conditioning on $\tau_{h-1}$, which may have caused confusion.
> > To address this, we have revised lines 812–833 in the updated version to explicitly state the conditioning measure for each expectation and to separate the roles of $o_h$ and $\tau_h$.
> > More specifically, every $\tau_h$ in Lemma 16 should be interpreted as the combination of the current observation $o_h$ and the history $\tau_{h-1}$. After applying Cauchy’s inequality and beginning the analysis of the second term, the $\tau_h$ appearing inside the $g$-function and the $\tau_h$ in the subscript of the expectation are both generated from a history $\tau_{h-1}$ that is sampled from the same distribution. Once the conditioning on $\tau_{h-1}$ is made explicit, these two $\tau_h$ indeed follow the same distribution. Therefore, the expression can be correctly written as an expectation of $g(\tau_h,a_h)$, and the one-step-back inequality remains fully valid.
> >
> > Regarding the low-rank assumption in partially observed systems, we emphasize that this assumption is standard and widely adopted in the literature (e.g., [1–4]). Its practicality comes from the fact that many real-world POMDPs possess low intrinsic dimensionality in their latent dynamics, even when observations or histories are high-dimensional. The low-rank structure enables identifiability of the latent transition model, generalization beyond the specific histories observed in the dataset, and sample-efficient learning through spectral or model-based methods. Thus, the low-rank assumption is both reasonable and essential for enabling the theoretical guarantees we derive.
> >
> > We appreciate the reviewer’s interest and have revised the proof in the updated manuscript.
> >
> >
> >
> > [1]Wang, L., Cai, Q., Yang, Z., and Wang, Z, Represent to control partially observed systems: Representation learning with provable sample efficiency.
> >
> > [2]J.Guo, Z.Li, H.Wang, M.Wang, Z.Yang, and X.Zhang, Provably efficient representation learning with tractable planning in low-rank pomdp.
> >
> > [3]W.Zhan, M.Uehara, W.Sun, and J.D.Lee, Pac reinforcement learning for predictive state representations.
> >
> > [4]Uehara, M., Sekhari, A., Lee, J. D., Kallus, N., and Sun, W, Provably efficient reinforcement learning in partially observable dynamical systems.

---

### Author Response · Authors · 2025-12-01
**Global Response**

We thank all reviewers for their thoughtful and constructive feedback. This paper addresses a core challenge in multi-agent RL: achieving sample-efficient learning in low-rank POMGs, a setting that is fundamentally hard due to partial observability, latent representations, and the need to compute equilibria from non-Markovian data. Our main contribution is to develop the first unified framework that integrates low-rank representation learning, optimistic value iteration, and equilibrium computation into a single analyzable pipeline. The key difficulty lies in controlling how representation error propagates through value iteration and equilibrium computation under partial observability—something that prior methods for MGs or structured POMGs do not handle. By systematically combining techniques that were previously studied only in isolated contexts (optimistic VI, spectral/MLE latent modeling, and variational methods for infinite-dimensional latent spaces), we obtain end-to-end statistical guarantees for approximate equilibria in both oracle-free and oracle-based settings, and even for infinite-rank POMGs via LLVR. We appreciate that multiple reviewers recognize the importance of building such a general theoretical foundation. Below we address the key concerns shared across reviews.

[Clarifying the Low-Rank Assumption and Lemma 16]
A central point of confusion raised by Reviewer L12U concerns both the interpretation of our low-rank transition assumption and the one-step-back inequality in Lemma 16. We have clarified in the revision that the factorization $P(\tau' \mid \tau, a) = \phi(\tau, a)^\top \mu(\tau')$ is only applied when $(\tau, a)$ is the immediate history of $\tau'$; all other transitions are structurally zero. Thus no rank constraint is imposed on arbitrary history pairs, avoiding the scaling concerns with $(OA)^h$ and eliminating the need for orthogonality among all feature vectors. This matches standard formulations in low-rank POMGs and captures settings where latent dynamics are low-dimensional even with large observation spaces.

For Lemma 16, the reviewer’s concern arose from a notational ambiguity: all expectations are in fact conditioned on $\tau_{h-1}$ and the $\tau_h$ appearing inside the g-function and the $\tau_h$ in the expectation are generated from $\tau_{h-1}$ drawn from the same distribution. Once this is made explicit, the equality used in the proof follows directly, and the one-step-back inequality remains valid. The revised manuscript now makes these conditioning steps clear.

[Novelty of the OVI Framework and Technical Contributions]
Our work develops a unified OVI framework for low-rank POMGs and provides the first end-to-end statistical guarantees for this setting.
While existing components like MLE, SDR, or value iteration are individually familiar, our framework is novel in how it integrates low-rank representation learning with optimistic value iteration to control representation error and ensure convergence. This framework generalizes prior methods (e.g., [1] becomes a special case  with fully observable states and MLE-based representation learning) and also covers settings not addressed in earlier work, including oracle-based method with representations in [2]. We have revised the introduction and related-work sections to more prominently highlight these conceptual and analytical contributions.

[L-Decodability and the LLVR Extension]
Several reviewers asked about the L-decodability assumption used in Sec 5. This commonly used assumption formalizes that a short recent window captures the relevant latent information. It enables us to extend the OVI framework to POMGs with very large or infinite latent dimension. LLVR then provides a tractable representation with an exact linear value-function form under this assumption, "mitigating" the finite-rank restriction and accommodating expressive neural representations.

[Computational Feasibility: Joint Actions and Equilibrium Oracles]
Concerns about dependence on the joint action space and equilibrium oracles reflect inherent challenges of POMGs: nearly all theoretical works require joint-action reasoning, and exact NE computation is PPAD-hard. In our framework, CE and CCE oracles reduce to polynomial-time linear programs, while the PPAD-hardness of NE computation is a fundamental difficulty of game theory rather than our method. When a value-function oracle is available (OVI-OB), computation is significantly lighter. We also clarify that integrals in Line 15 can be evaluated exactly for small-support initial distributions or approximated efficiently using Monte Carlo sampling.

We thank all reviewers again for their feedback. We believe the revisions substantially improve clarity and address all concerns.

[1]Ni et al., Representation learning for low-rank general-sum Markov games.

[2]Altabaa and Yang, On the role of information structure in reinforcement learning for partially observable sequential teams and games.

---

### Meta-Review · Area_Chair_C9Ht · 2026-01-12

**Summary:**

This work proposes a unified optimistic value iteration framework for solving low-rank POMGs. The optimistic value iteration algorithm has shown to converge to the approximate NE, CE and CCE for POMGs with low-rank dynamics.

The reviewers have concerns including (i) validity of the low-rank assumptions, (ii) computational feasibility of the algorithms, (iii) novelty of the technical contributions and (iv) correctness of the proofs.

**Reviewer Concerns:**

Although the authors provided responses trying to address concerns on all the above aspects, it is unclear if all the above concerns, especially the ones regarding the proof correctness and technical novelty, have been fully addressed. At this point, it seems to me that another around of peer-review is necessary. Given the extremely high bar of ICLR, I would suggest rejection.

**Reviewer Scores:**

As mentioned above, it is unclear to me if the two reviewers with negative scores would change their scores given a full discussion.

---

### Decision · Program_Chairs · 2026-01-26

Reject